# A Statistical Theory of Overfitting for Imbalanced Classification

**Jingyang Lyu[1], Kangjie Zhou[2], Yiqiao Zhong[1]**
[1] Department of Statistics, University of Wisconsin–Madison, WI 53706, USA
[2] Department of Statistics, Columbia University, New York, NY 10027, USA
{jlyu55, yiqiao.zhong}@wisc.edu, kz2326@columbia.edu

## Abstract

Classification with imbalanced data is a common challenge in machine learning, where minority classes form only a small fraction of the training samples. Classical theory, relying on large-sample asymptotics and finite-sample corrections, is often ineffective in high dimensions, leaving many overfitting phenomena unexplained. In this paper, we develop a statistical theory for high-dimensional imbalanced linear classification, showing that dimensionality induces truncation or skewing effects on the logit distribution, which we characterize via a variational problem. For linearly separable Gaussian mixtures, logits follow $\mathsf{N}(0, 1)$ on the test set but converge to $\max\{\kappa, \mathsf{N}(0, 1)\}$ on the training set—a pervasive phenomenon we confirm on tabular, image, and text data. This phenomenon explains why the minority class is more severely affected by overfitting. We further show that margin rebalancing mitigates minority accuracy drop and provide theoretical insights into calibration and uncertainty quantification.

## 1 Introduction

Classification tasks are ubiquitous in statistics and machine learning. In many practical applications, training data are often imbalanced, meaning that some classes (minority classes) contain substantially fewer samples than others. In binary classification, particularly, we observe training data $\{(\boldsymbol{x}_i, y_i)\}_{i=1}^n \overset{\text{i.i.d.}}{\sim} P_{\boldsymbol{x},y}$ with features $\boldsymbol{x}_i \in \mathbb{R}^d$ and binary labels $y_i \in \{\pm 1\}$. Denote by $P_{\boldsymbol{x}}$ (resp. $P_y$) the marginal distribution of $\boldsymbol{x}$ (resp. $y$), and the expected fractions of the two classes by

$$\pi_+ := \mathbb{P}(y_i = +1), \qquad \pi_- := \mathbb{P}(y_i = -1).$$

We say that the data set is imbalanced if $\pi_+ < \pi_-$[1]. Imbalanced classification is common in applications where the minority class represents rare diseases, rare events, anomalies, or underrepresented groups (Kubat et al., 1998; King & Zeng, 2001; Weiss & Provost, 2003; Chandola et al., 2009; Ngai et al., 2011; Litjens et al., 2017; Tschandl et al., 2018; Buolamwini & Gebru, 2018)

We focus on classifiers which take the form of $\boldsymbol{x} \mapsto 2\mathbb{1}\{f(\boldsymbol{x}) > 0\} - 1$, with $f(\boldsymbol{x}) = \langle \boldsymbol{x}, \boldsymbol{\beta} \rangle + \beta_0$. This simple form is widely used in statistics and deep learning. Particularly for image and text data, the last classification layer of a deep neural network (DNN) usually takes this form (e.g., softmax), where $\boldsymbol{x}_i$ is the extracted feature in high dimensions.

**Challenge 1: High-dimensional features from pretrained neural networks** In low dimensions ($d \ll n$), prediction and estimation are well understood. But when $d$ is comparable to $n$, classical theory becomes inaccurate, motivating refined asymptotic analyses of high-dimensional learning.

Regarding parameter estimation, a line of work (Dobriban & Wager, 2018; Sur et al., 2019; Sur & Candès, 2019; Candès & Sur, 2020; Montanari et al., 2023) has studied logistic regression under the proportional regime $n/d \to \delta$. As $d$ increases (i.e., $\delta$ decreases), the estimation error of the maximum likelihood estimator (MLE) $\widehat{\boldsymbol{\beta}}$ grows, and classical likelihood tests require modification.

Regarding generalization, high dimensionality usually leads to a gap between the training and test errors. Recent work on *double descent* (Belkin et al., 2019) shows that in overparameterized models, gradient descent induces implicit regularization, leading to benign overfitting (Bartlett et al., 2020).

---

[1]Without loss of generality, we assume that the minority class is assigned the label $+1$.

Finally, the distribution of logits is highly valuable for feature visualization and interpretation, yet little theory exists. Practical examples include *linear probing*, a common approach for interpreting the hidden states (or activations) (Kornblith et al., 2019; He et al., 2020; Kumar et al., 2022), and *projection pursuit*, which explores data via low-dimensional projections, with recent theory developed in high dimensions (Bickel et al., 2018; Montanari & Zhou, 2022).

**Challenge 2: Class imbalance in downstream tasks**  In downstream tasks, the base neural network is typically frozen, and only the last layer is retrained—also known as linear probing. In such classification settings, data imbalance poses key challenges: MLE may be unreliable, label shifts in the minority proportion $\pi_+$ can degrade performance, and misclassifying minority samples is often more costly (He & Garcia, 2009). Common remedies include adjusting decision boundaries, reweighting losses, and sub-/over-sampling (King & Zeng, 2001; Chawla et al., 2002).

Overfitting in high dimensions further exacerbates imbalance, as large deep models often memorize rather than generalize to minority samples (Sagawa et al., 2020). Although various remedies have been proposed (Huang et al., 2016; Cao et al., 2019; Liu et al., 2019; Khan et al., 2019), they remain ad hoc and offer little guidance on hyperparameter choice or feature interpretation.

**Imbalanced classification.**  Classical work on logistic regression shows that large-sample asymptotics fails with small samples, motivating bias corrections (Anderson & Richardson, 1979; McCullagh & Nelder, 1983; Schaefer, 1983). Under label shift, intercept correction and upweighting are proposed (Xie & Manski, 1989; King & Zeng, 2001; Loffredo et al., 2024; Pezzicoli et al., 2025; Sarao Mannelli et al., 2025). For kernel and tree-based methods, resampling, and synthetic data generation are common (Chawla et al., 2002; He et al., 2008; He & Garcia, 2009), yet these are ineffective for separable data (Cao et al., 2019) and do not address high-dimensional overfitting.

**Margin-based methods.**  Margin is central to classification methods such as SVM. For imbalanced settings, promoting unequal margins is proposed for the perceptron algorithm (Li et al., 2002), SVM (Li et al., 2005), and more recently deep networks (Huang et al., 2016; Cao et al., 2019; Khan et al., 2019; Liu et al., 2019). Theoretical work has established margin-based generalization bounds (Bartlett, 1996; Bartlett et al., 1998; Bartlett & Mendelson, 2002; Koltchinskii & Panchenko, 2002; Bartlett et al., 2017), motivating margin-rebalancing losses (Cao et al., 2019). However, the margin-dependent bounds are agnostic to data distributions and may be excessively conservative.

**High-dimensional asymptotics.**  Classical asymptotics in low dimensions is inaccurate (El Karoui et al., 2013; Donoho & Montanari, 2016). A recent line of work develops high-dimensional classification theory (Dobriban & Wager, 2018; Sur & Candès, 2019; Salehi et al., 2019; Sur et al., 2019; Candès & Sur, 2020; Mignacco et al., 2020; Kini et al., 2021; Loureiro et al., 2021; Deng et al., 2022; Montanari et al., 2023; Dandi et al., 2023; Montanari et al., 2024), refining Table 1 mainly via Gordon's theorem (Gordon, 1985; Thrampoulidis et al., 2015). Closest is (Montanari & Zhou, 2022), which analyzes projection pursuit and low-dimensional asymptotics, but none characterize the impact of class imbalance on overfitting or calibration.

Table 1: Qualitative comparison between low/high dimensions for binary classification, where a linear classifier $\widehat{y}(\boldsymbol{x}) = 2\mathbb{1}\{\widehat{f}(\boldsymbol{x}) > 0\} - 1$ with $\widehat{f}(\boldsymbol{x}) = \langle \boldsymbol{x}, \widehat{\boldsymbol{\beta}} \rangle + \widehat{\beta}_0$ is trained on $\{(\boldsymbol{x}_i, y_i)\}_{i=1}^n \overset{\text{i.i.d.}}{\sim} P_{\boldsymbol{x},y}$. Here, the logits $\{\widehat{f}(\boldsymbol{x}_i)\}_{i=1}^n$ are obtained by evaluating $\widehat{f}$ on the training set.

|  | **Low dimensions** | **High dimensions** |
|---|---|---|
| Parameter estimation | $\left\langle \frac{\widehat{\boldsymbol{\beta}}}{\|\widehat{\boldsymbol{\beta}}\|}, \frac{\boldsymbol{\beta}}{\|\boldsymbol{\beta}\|} \right\rangle \approx 1$ | $\left\langle \frac{\widehat{\boldsymbol{\beta}}}{\|\widehat{\boldsymbol{\beta}}\|}, \frac{\boldsymbol{\beta}}{\|\boldsymbol{\beta}\|} \right\rangle < 1$ |
| Generalization | Train error $\approx$ Test error | Train error $<$ Test error |
| Distribution of logits | 1D projection of $P_{\boldsymbol{x}}$ | Skewed/distorted 1D projection of $P_{\boldsymbol{x}}$ |

**Our contribution**  We identify two key gaps in the existing literature. First, the reason why overfitting is more severe in minority classes, though consistently observed, remains unclear. Second, there is no comprehensive analysis of how key factors—such as dimensionality, imbalance, and signal strength—affect performance metrics like test accuracy and uncertainty quantification. The goal of this paper is to develop a statistical theory that addresses these gaps.

- We characterize overfitting via the discrepancy between *logit distributions* on training and test sets, and show that dimensionality induces a *truncation effect* in training logits. (**Section 2**)

- We derive the optimal hyperparameter for *margin rebalancing* to mitigate class imbalance, and show that test error decreases with key model parameters—the *imbalance ratio*, *signal strength* (class center separation), and *aspect ratio*, see Table 2. (**Section 3**)
- We further show that parameter changes that raise test error could simultaneously *worsen model calibration*, revealing an adverse effect of overfitting, see Table 2. (**Section 4**)

Table 2: Monotonicity of test errors and miscalibration metrics on model parameters.

|  | $\text{Err}_+^*, \text{Err}_-^*, \text{Err}_b^*$ | $\text{CalErr}^*$ | $\text{MSE}^*$ | $\text{ConfErr}^*$ |
|---|---|---|---|---|
| imbalance ratio $\pi \uparrow$ | $\downarrow$ (Prop. 3.1) |  | $\downarrow$ (Thm. 4.1) | $\downarrow$ (Claim D.10) |
| signal strength $\|\boldsymbol{\mu}\|_2 \uparrow$ | $\downarrow$ (Prop. 3.1) | $\downarrow$ (Claim D.10) | $\downarrow$ (Thm. 4.1) |  |
| aspect ratio $n/d \to \delta \uparrow$ | $\downarrow$ (Prop. 3.1) | $\downarrow$ (Claim D.10) | $\downarrow$ (Thm. 4.1) | $\downarrow$ (Prop. D.9) |

Building on theoretical tools from high-dimensional statistics, our analysis focuses on a stylized model. Suppose the i.i.d. training data $\{(\boldsymbol{x}_i, y_i)\}_{i=1}^n$ are generated from a two-component Gaussian mixture model (2-GMM):

$$\mathbb{P}(y_i = +1) = \pi, \quad \mathbb{P}(y_i = -1) = 1 - \pi, \quad \boldsymbol{x}_i \,|\, y_i \sim \mathsf{N}(y_i \boldsymbol{\mu}, \mathbf{I}_d), \tag{1}$$

where $\boldsymbol{\mu} \in \mathbb{R}^d$ is the signal vector. Under this model, the Bayes-optimal classifier has the form $y^*(\boldsymbol{x}) = 2\mathbb{1}\{\langle \boldsymbol{x}, \boldsymbol{\beta} \rangle + \beta_0 > 0\} - 1$, where $\boldsymbol{\beta} /\!\!/ \boldsymbol{\mu}$. See Section 5 for extensions of Eq. (1). We study the behavior of two standard approaches for binary classification: (a slightly generalized version of) logistic regression and support vector machines (SVMs). Denoting by $\ell : \mathbb{R} \to \mathbb{R}$ a strictly convex decreasing function, including the logistic function $\log(1 + e^{-x})$ as a special case, we solve

$$\text{(logistic regression)} \quad \underset{\boldsymbol{\beta} \in \mathbb{R}^d, \beta_0 \in \mathbb{R}}{\text{minimize}} \quad \frac{1}{n} \sum_{i=1}^n \ell\big(y_i(\langle \boldsymbol{x}_i, \boldsymbol{\beta} \rangle + \beta_0)\big), \tag{2a}$$

$$\begin{aligned} \text{(SVM)} \quad &\underset{\boldsymbol{\beta} \in \mathbb{R}^d, \, \beta_0, \kappa \in \mathbb{R}}{\text{maximize}} \quad \kappa, \\ &\text{subject to} \quad y_i(\langle \boldsymbol{x}_i, \boldsymbol{\beta} \rangle + \beta_0) \geq \kappa, \quad \forall i \in [n], \\ &\qquad\qquad\quad \|\boldsymbol{\beta}\|_2 \leq 1. \end{aligned} \tag{2b}$$

Both optimization problems are convex and yield solutions $\widehat{\boldsymbol{\beta}}, \widehat{\beta}_0$, which are used to predict class labels for a test data point $\boldsymbol{x}$ based on $\widehat{f}(\boldsymbol{x}) = \langle \boldsymbol{x}, \widehat{\boldsymbol{\beta}} \rangle + \widehat{\beta}_0$. Namely, the predicted binary label of a test data point $\boldsymbol{x}$ is $\widehat{y}(\boldsymbol{x}) = 2\mathbb{1}\{\langle \boldsymbol{x}, \widehat{\boldsymbol{\beta}} \rangle + \widehat{\beta}_0 > 0\} - 1$.

We will analyze both classifiers with a focus on the SVM for the following reason. In modern machine learning, it is common for the labeled data $\{(\boldsymbol{x}_i, y_i)\}_{i=1}^n$ to be linearly separable due to high dimensionality. When data are linearly separable, the hard-margin SVM coincides with the max-margin classifier. It is known that the gradient descent iterates of logistic regression converge in direction to the max-margin solution (Soudry et al., 2018; Ji & Telgarsky, 2019), which is known as a form of inductive bias (Neyshabur et al., 2015). See Appendix C.1 for the background. In this sense, the two classifiers are closely related.

All experiment details and proofs are provided in the appendix. The code for our experiments can be found at **https://github.com/jlyu55/Imbalanced_Classification_iclr**.

## 2 Characterizing overfitting via empirical logit distribution

Explaining why test accuracy drops more for the minority class requires a more refined characterization of overfitting. We therefore study the empirical distribution of logits $\widehat{f}(\boldsymbol{x}_i) = \langle \boldsymbol{x}_i, \widehat{\boldsymbol{\beta}} \rangle + \widehat{\beta}_0$, $i \in [n]$ on the training set. Let $(\boldsymbol{x}_1, y_1), \ldots, (\boldsymbol{x}_n, y_n) \overset{\text{i.i.d.}}{\sim} P_{\boldsymbol{x},y}$ be training data, and $(\boldsymbol{x}_{\text{test}}, y_{\text{test}})$ an independent test point. Consider a binary classifier $\widehat{y} : \mathbb{R}^d \to \{\pm 1\}$ based on $\widehat{f} : \mathbb{R}^d \to \mathbb{R}$ that predicts $\widehat{y} = +1$ if $\widehat{f}(\boldsymbol{x}) > 0$ and $\widehat{y} = -1$ otherwise.

**Definition 2.1** (Logit and margin). *Let $(\boldsymbol{x}, y) \in \mathbb{R}^d \times \{\pm 1\}$ be a data point. For a binary classifier of the form $\widehat{y}(\boldsymbol{x}) = 2\mathbb{1}\{\widehat{f}(\boldsymbol{x}) > 0\} - 1$, we define the* logit *of $\boldsymbol{x}$ as $\widehat{f}(\boldsymbol{x})$, and the* margin *of the classifier $\widehat{f}$ (on the training data) as $\widehat{\kappa}_n = \min_{i \in [n]} y_i \widehat{f}(\boldsymbol{x}_i)$.*

The following definitions highlight the logit distribution on both training and test data.

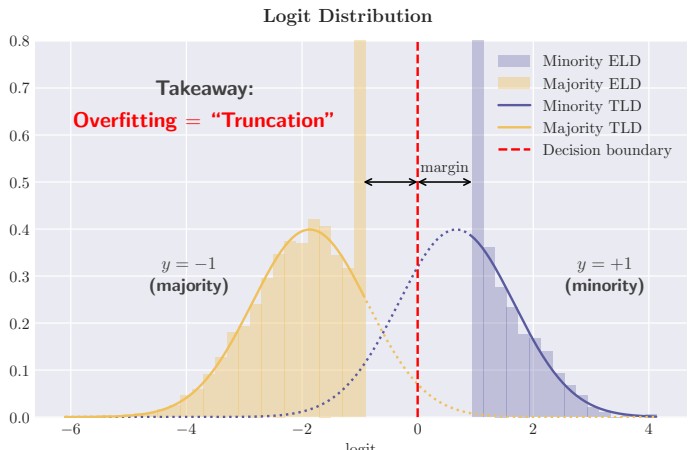

Figure 1: **Empirical logit distribution (ELD) and testing logit distribution (TLD)**. We train a max-margin classifier (namely SVM) $\widehat{f}$ on synthetic data from a 2-component Gaussian mixture model. Colors indicate labels $y_i$ and $x$-axis indicates logits $\widehat{f}(\boldsymbol{x}_i)$. **ELD for both classes:** rectified Gaussian distribution (histogram). **TLD for both classes:** Gaussian distribution (curve). **Overfitting effect:** The density areas below the dotted curves are overlapping in TLD, thus leading to positive test error; however they are "pushed" to respective margin boundaries in ELD, thus leading to linear separability and zero training errors.

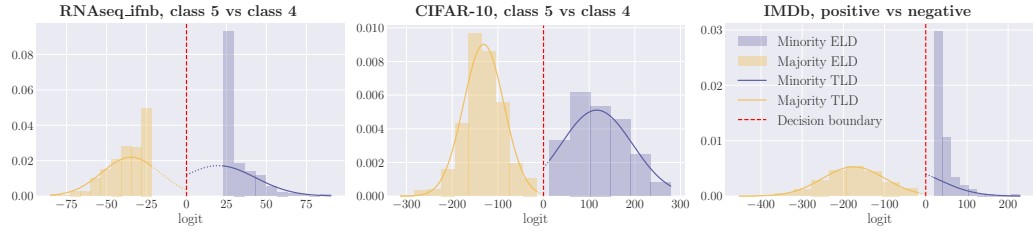

Figure 2: **ELD and TLD of logistic regression classifier (the last fully-connected layer) for real data**. **Left:** IFNB single-cell RNA-seq dataset (tabular data). **Middle:** CIFAR-10 dataset preprocessed by pretrained ResNet-18 model for feature extraction (image data). **Right:** IMDb movie review dataset preprocessed by BERT base model (110M) for feature extraction (text data).

**Definition 2.2** (ELD and TLD). **Empirical logit distribution (ELD)**, *or* training logit distribution, *is defined as the empirical distribution of label-logit pairs based on training data:*

$$\widehat{\nu}_n = \widehat{\nu}_n^{\mathrm{train}} = \frac{1}{n}\sum_{i=1}^n \delta_{(y_i,\widehat{f}(\boldsymbol{x}_i))}^{\,2} \quad \textit{(i.e., the "histogram" of } \{\widehat{f}(\boldsymbol{x}_i)\}_{i=1}^n). \tag{3}$$

Minority ELD *and* majority ELD *are defined respectively as ELD in minority and majority class.*

**Testing logit distribution (TLD)** *is defined as the distribution of the label-logit pair for a test point:*

$$\widehat{\nu}_n^{\mathrm{test}} = \mathtt{Law}\left(y_{\mathrm{test}}, \widehat{f}(\boldsymbol{x}_{\mathrm{test}})\right)^3.$$

Minority TLD *and* majority TLD *are defined respectively as TLD in minority and majority class.*

When $\widehat{\kappa}_n > 0$, the training set is linearly separable, and $\widehat{f}$ attains $100\%$ training accuracy. In high dimensions this is common, yet test accuracy is typically imperfect; this train/test discrepancy is known as *overfitting*. As logit distribution is more informative than train/test accuracies, we analyze overfitting through ELD/TLD.

**Empirical phenomenon**   First, we show a simple yet representative simulated example to illustrate the phenomenon; see Fig. 1. We generate training data according to 2-GMM in Eq. (1) with $n =$

---

[2]$\delta_{\boldsymbol{a}}$ is the delta measure supported at point $\boldsymbol{a}$ (Dirac measure).

[3]$\mathtt{Law}$ means the distribution of random variables/vectors.

10,000, $d = 4{,}000$, $\|\boldsymbol{\mu}\|_2 = 1.75$, and imbalance $\pi = 0.15$, which guarantees linear separability. Training an SVM Eq. (2b) and plotting ELD/TLD by class, we find TLDs are Gaussian, whereas ELDs are the same Gaussians but *truncated* at the margin—such discrepancy characterizes the effect of overfitting in imbalanced classification. The minority ($y_i = +1$) ELD loses over half its mass by truncation, yielding much lower accuracy than the majority ($y_i = -1$). Formally, such distribution of ELD is called *rectified Gaussian* (i.e., $\max\{Z, \kappa\}$ or $\min\{Z, \kappa\}$ where $Z \sim \mathsf{N}(\mu, \sigma^2)$).

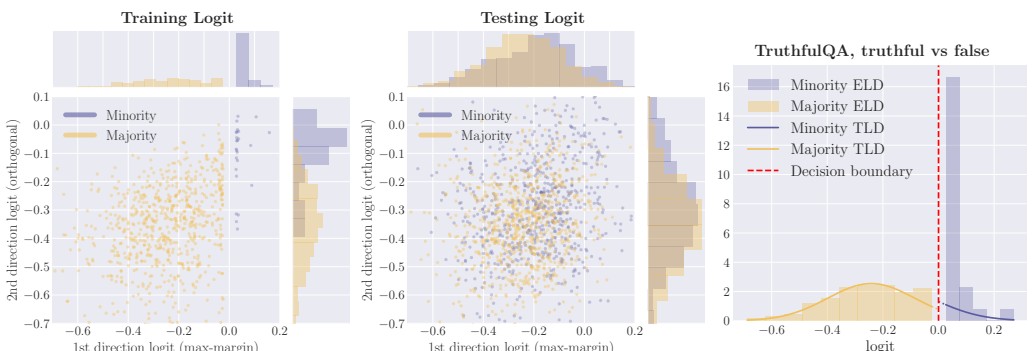

Figure 3: **ELD and TLD of Llama-3-8B-Instruct activation probing on TruthfulQA dataset. Left, Middle:** Scatter plot on training/testing data for activations (31th layer, 26th head) of truthful (minority) and false (majority) QA pairs after projection onto the top-2 directions. Marginal distributions are shown on the upper and right sides. **Right:** Marginal ELD and TLD on 1st direction.

For real-data examples, we fine-tune pretrained DNNs by freezing all parameters except for the last classification layer on imbalanced labeled data, effectively using the pretrained network as a feature extractor—a standard downstream practice (Sharif Razavian et al., 2014; Zhou et al., 2016; Howard & Ruder, 2018; Radford et al., 2021). We evaluate three representative data modalities; see Fig. 2.

1. *Tabular data.* We use a single-cell RNA-seq dataset of peripheral blood mononuclear cells treated with interferon-$\beta$ (IFNB) (Kang et al., 2018), with dimension $d = 2{,}000$. We choose class 5 (CD4 Naive T cells) and class 4 (CD4 Memory T cells), and subsample an imbalanced training set. Class 4 is the minority class, with imbalance $\pi = 0.2$ and sample size $n = 953$.

2. *Image data.* We use CIFAR-10 image dataset (Krizhevsky, 2009). The pretrained ResNet-18 model (He et al., 2016; Dadalto, 2023) is applied to extract the features of dimension $d = 512$. We randomly choose two classes, for example, class 5 (dog) and class 4 (deer), and subsample an imbalanced training set. Class 4 is the minority class, with $\pi = 0.1$ and $n = 555$.

3. *Text data.* We use IMDb movie review dataset (Maas et al., 2011) to perform binary sentiment classification. The BERT base language model (110M) (Devlin, 2018) is applied to extract the features of dimension $d = 768$. An imbalanced training set is sampled, where negative reviews belong to the minority class, with $\pi = 0.02$ and $n = 6{,}377$.

Empirically, we observe a pervasive ELD regularity: for separable data, the ELD of each class can be fitted by rectified Gaussian, and such *distributional truncation solely explains overfitting* in high dimensions. The minority class also suffers more from truncation effect as its test accuracy is worse.

Further, we extend our analysis from last-layer features to intermediate-layer hidden states of large language models (LLMs), by applying Llama-3-8B-Instruct (AI@Meta, 2024) to TruthfulQA (Lin et al., 2021), following the experiment in (Li et al., 2023). For each QA pair we concatenate the question and answer, extract last-token head activations, and build a probing dataset for every head and layer. We fit a linear probe (Eq. (2a), 1st direction) and an orthogonal probe (2nd direction, minimizing the same objective subject to orthogonality). See details in Appendix B.2. Fig. 3 shows the joint and marginal logit distributions along the two directions. Unlike (Li et al., 2023), we use an imbalanced probing set ($\pi = 0.04$, $n = 690$; true answers are minority) and compare ELD (Fig. 3 left) with TLD (middle). We observe truncation in the first direction and distortion in the second, indicating overfitting in LLM probing under class imbalance and offering potential guidance toward understanding unintended memorization in LLMs.

As our theoretical insights below reveal, this truncation phenomenon arises because both classes share a common "overfitting budget", which disproportionately shifts the minority margin boundary.

**Theoretical foundation** We highlight a summary of our result for the separable case here and defers the non-separable case to Appendix D.1. Consider the asymptotic regime $n/d \to \delta$ where $\delta \in (0, \infty)$ is called the limiting aspect ratio. Recall that $(\widehat{\boldsymbol{\beta}}, \widehat{\beta}_0, \widehat{\kappa})$ are the trained parameters in Eq. (2b), where $\widehat{\kappa}$ is the margin of classifier $\widehat{y}(\boldsymbol{x}) = 2\mathbb{1}\{\widehat{f}(\boldsymbol{x}) > 0\} - 1$ with $\widehat{f}(\boldsymbol{x}) = \langle \boldsymbol{x}, \widehat{\boldsymbol{\beta}}\rangle + \widehat{\beta}_0$. Denote $\widehat{\rho} = \langle \frac{\boldsymbol{\beta}}{\|\boldsymbol{\beta}\|_2}, \frac{\boldsymbol{\mu}}{\|\boldsymbol{\mu}\|_2}\rangle$ the cosine similarity between the slope and the optimal direction $\boldsymbol{\mu}$.

On a test point $(\boldsymbol{x}_{\text{test}}, y_{\text{test}}) \sim P_{\boldsymbol{x}, y}$, we consider the minority error and majority error

$$\text{Err}_+ := \mathbb{P}\left(\widehat{f}(\boldsymbol{x}_{\text{test}}) \le 0 \,\big|\, y_{\text{test}} = +1\right), \qquad \text{Err}_- := \mathbb{P}\left(\widehat{f}(\boldsymbol{x}_{\text{test}}) > 0 \,\big|\, y_{\text{test}} = -1\right). \quad (4)$$

Theorem 2.1 below characterizes the precise *asymptotic behavior*, namely the limiting distribution of and logits in the proportional regime as $n/d \to \delta$.

**Theorem 2.1** (Separable data, informal version of Theorem D.1). *Consider 2-GMM with asymptotics $n/d \to \delta \in (0, \infty)$ as $n, d \to \infty$. There is a critical threshold $\delta_c = \delta_c(\pi, \|\boldsymbol{\mu}\|_2)$, such that when $\delta < \delta_c$, the following holds as $n, d \to \infty$:*

(a) ***Parameter convergence.*** *The training set is linearly separable with high probability, and*

$$(\widehat{\rho}, \widehat{\beta}_0, \widehat{\kappa}) \xrightarrow{\text{P}} (\rho^*, \beta_0^*, \kappa^*),$$

*where $(\rho^*, \beta_0^*, \kappa^*)$ is the unique solution to the following variational problem*

$$\underset{\rho \in [-1,1], \beta_0 \in \mathbb{R}, \kappa > 0, \xi \in \mathcal{L}^2}{\text{maximize}} \kappa, \quad s.t. \quad \rho \|\boldsymbol{\mu}\|_2 + G + Y\beta_0 + \sqrt{1 - \rho^2}\xi \ge \kappa, \ \ \mathbb{E}[\xi^2] \le 1/\delta, \quad (5)$$

*where $\mathcal{L}^2$ is the space of square integrable random variables in the probability space $(\Omega, \mathcal{F}, \mathbb{P})$, $(Y, G) \sim P_y \times \mathsf{N}(0, 1)$, and $\xi$ is an unknown random variable (functional) to be optimized.*

*As a consequence, the limits of minority/majority errors are*

$$\text{Err}_+ \to \Phi\left(-\rho^* \|\boldsymbol{\mu}\|_2 - \beta_0^*\right), \qquad \text{Err}_- \to \Phi\left(-\rho^* \|\boldsymbol{\mu}\|_2 + \beta_0^*\right),$$

*where $\Phi$ denotes the cumulative distribution function of standard Gaussian.*

(b) ***ELD convergence.*** *The empirical (training) logit distribution $\widehat{\nu}_n^{\text{train}}$ has limit $\nu_*^{\text{train}}$ in the sense*

$$W_2(\widehat{\nu}_n^{\text{train}}, \nu_*^{\text{train}}) \xrightarrow{\text{P}} 0, \qquad \text{where } \nu_*^{\text{train}} := \text{Law}\left(Y, Y \max\{\kappa^*, \rho^* \|\boldsymbol{\mu}\|_2 + G + Y\beta_0^*\}\right).$$

***TLD convergence.*** *The testing logit distribution $\widehat{\nu}_n^{\text{test}}$ has limit $\nu_*^{\text{test}}$ in the sense that*

$$\widehat{\nu}_n^{\text{test}} \xrightarrow{w} \nu_*^{\text{test}}, \qquad \text{where } \nu_*^{\text{test}} := \text{Law}\left(Y, Y(\rho^* \|\boldsymbol{\mu}\|_2 + G + Y\beta_0^*)\right).$$

**Intuition and proof techniques**

- In Eq. (5), $\rho$ is the cosine similarity between $\boldsymbol{\beta}$ and $\boldsymbol{\mu}$, and the term $\rho \|\boldsymbol{\mu}\|_2 + G + Y\beta_0$ yields the projection of the input distribution $\boldsymbol{x} \sim \pi \cdot \mathsf{N}(\boldsymbol{\mu}, \mathbf{I}_d) + (1 - \pi) \cdot \mathsf{N}(-\boldsymbol{\mu}, \mathbf{I}_d)$ onto $\boldsymbol{\beta}$:

$$y\left(\langle \boldsymbol{x}, \boldsymbol{\beta}\rangle + \beta_0\right) = y\left\langle y\boldsymbol{\mu} + \mathsf{N}(\mathbf{0}, \mathbf{I}_d), \boldsymbol{\beta}\right\rangle + y\beta_0 \qquad (\text{recall } \|\boldsymbol{\beta}\|_2 = 1)$$

$$= \underbrace{\left\langle \frac{\boldsymbol{\mu}}{\|\boldsymbol{\mu}\|_2}, \frac{\boldsymbol{\beta}}{\|\boldsymbol{\beta}\|_2}\right\rangle}_{\rho} \cdot \|\boldsymbol{\mu}\|_2 + \underbrace{\left\langle \mathsf{N}(\mathbf{0}, \mathbf{I}_d), \boldsymbol{\beta}\right\rangle}_{\mathsf{N}(0, 1)} + y\beta_0 \approx \rho \|\boldsymbol{\mu}\|_2 + G + Y\beta_0.$$

  Intuitively, the remaining dimensions are not relevant and instead provide room for overfitting, which is captured by the "free" random variable $\xi$.

- The main idea is to rewrite the linear classifier in Eq. (2b) as a min–max formulation involving

$$\min_{\boldsymbol{\beta}} \max_{\boldsymbol{\lambda}} \left\{\boldsymbol{\lambda}^\mathsf{T} \boldsymbol{G} \boldsymbol{\beta} + \langle \boldsymbol{\lambda}, \mathbf{c}\rangle\right\}, \quad \text{where } \boldsymbol{G} \in \mathbb{R}^{n \times d} \text{ with i.i.d. } \mathsf{N}(0, 1) \text{ entries}.$$

  *Gordon's theorem* (Gordon, 1985; Thrampoulidis et al., 2015) allows us to replace the bilinear term $\boldsymbol{\lambda}^\mathsf{T} \boldsymbol{G} \boldsymbol{\beta}$ with $\|\boldsymbol{\beta}\|_2 \langle \boldsymbol{\lambda}, \mathsf{N}(\mathbf{0}, \mathbf{I}_n)\rangle + \|\boldsymbol{\lambda}\|_2 \langle \boldsymbol{\beta}, \mathsf{N}(\mathbf{0}, \mathbf{I}_d)\rangle$, so the min–max problem involves only random vectors instead of the random matrix $\boldsymbol{G}$. Our proofs then address the technical aspects of this reduction and establish the resulting distributional convergence rigorously.

We now provide several additional remarks on Theorem 2.1.

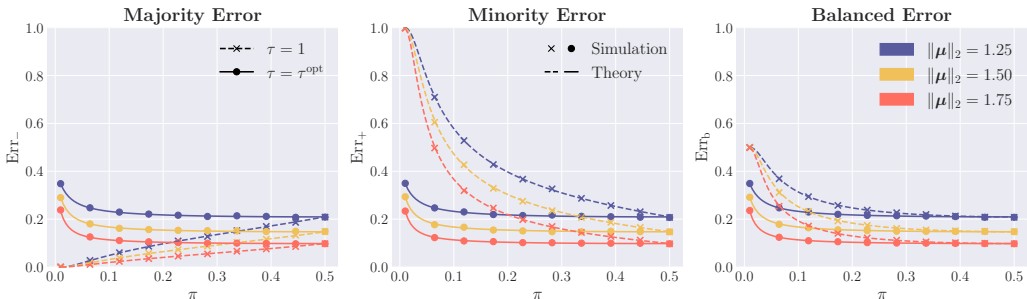

Figure 4: **Impact of imbalance on test errors**. We show test errors from 2-GMM simulations with margin rebalancing (solid curves) and without (dashed curves) at three levels of signal strength $\|\boldsymbol{\mu}\|_2$ under varying imbalanced ratios $\pi$.

- *Compared with the existing literature.* Rather than focusing solely on test errors—as in prior theoretical studies (Kini et al., 2021; Deng et al., 2022)—we analyze overfitting in imbalanced classification by characterizing the ELD/TLD (Theorem 2.1(b)), and further establish monotonicity of test errors in key model parameters (see Section 3 and Table 2).

- *Overfitting effect.* The random variable $\xi$ encodes distortion of ELD due to high dimensions. A smaller aspect ratio $\delta$ loosens the moment constraint $\mathbb{E}[\xi^2] \leq 1/\delta$ when maximizing the margin in Eq. (5), allowing greater distortion of TLD and thus stronger truncation. In fact, in order to maximize $\kappa$, the first inequality constraint must be tight, which yields the explicit formula:

$$\sqrt{1-\rho^2}\,\xi = \max\{\kappa, \rho\,\|\boldsymbol{\mu}\|_2 + G + Y\beta_0\} - (\rho\,\|\boldsymbol{\mu}\|_2 + G + Y\beta_0)$$
$$= (\kappa - \rho\,\|\boldsymbol{\mu}\|_2 - G - Y\beta_0)_+, \qquad \text{where } a_+ := \max\{0, a\}. \tag{6}$$

Thus, $\xi$ maps overlapping TLD masses to ELD margins, explaining ELD/TLD discrepancy.

- *More truncation for minority class.* Due to imbalance, transporting the probability mass in the minority ELD as Eq. (6) incurs less "cost" to the overall "budget" $\mathbb{E}[\xi^2] \leq 1/\delta$. Formally, according to Theorem 2.1, the limiting TLD for each class is

$$\text{minority: } \mathsf{N}\left(\rho^* \|\boldsymbol{\mu}\|_2 + \beta_0^*, 1\right), \qquad \text{majority: } \mathsf{N}\left(-\rho^* \|\boldsymbol{\mu}\|_2 + \beta_0^*, 1\right).$$

It can be shown that $\rho^* > 0$, $\beta_0^* < 0$. So the minority TLD is closer to the decision boundary—the minority class suffers more from the truncation effect (overfitting) than the majority class.

- *Optimal transport perspective.* We show in Appendix D.1.1 that $\mathtt{T}^*(x) = \max\{\kappa^*, x\}$ gives the optimal transport map from $\nu_*$ to $\nu_*^{\text{test}}$ and minimizes the $W_2$ distance between them.

- *Non-separable case.* When $\delta > \delta_c$, the training data is not separable with high probability, so SVM is no longer the limit of logistic regression Eq. (2a). We analyze Eq. (2a) and its ELD: instead of truncation, overfitting appears as nonlinear shrinkage governed by the proximal operator (Moreau-envelope gradient). As $\delta \in (\delta_c, \infty)$ decreases, the shrinkage moves from an identity map (no overfitting) to truncation $\mathtt{T}^*(x) = \max\{\kappa^*, x\}$ (severe overfitting). See Theorem D.3 for the formula and Appendix B.5 for its function plot.

## 3 REBALANCING MARGIN IS CRUCIAL

Rebalancing the margin is a common practice for remedying severe overfitting for the minority class. In binary case, we choose a hyperparameter $\tau > 0$ and consider the margin-rebalanced SVM:

$$\underset{\boldsymbol{\beta}\in\mathbb{R}^d, \beta_0\in\mathbb{R}, \kappa\in\mathbb{R}}{\text{maximize}} \;\; \kappa \qquad \text{subject to} \;\; \widetilde{y}_i(\langle \boldsymbol{x}_i, \boldsymbol{\beta}\rangle + \beta_0) \geq \kappa, \;\; \forall i \in [n], \;\; \|\boldsymbol{\beta}\|_2 \leq 1, \tag{7}$$

where $\widetilde{y}_i = \tau^{-1}$ if $y_i = +1$, otherwise $-1$. For the logistic loss in Eq. (2a), we can similarly incorporate $\tau$ into the objective function. Margin rebalancing is widely used in machine learning (Karakoulas & Shawe-Taylor, 1998; Li et al., 2002; Wu & Chang, 2003; Li et al., 2005; Cao et al., 2019; Kini et al., 2021; Clifford et al., 2024; Hu et al., 2025), but the impact of $\tau$ and other model parameters on test accuracy is not fully explored.

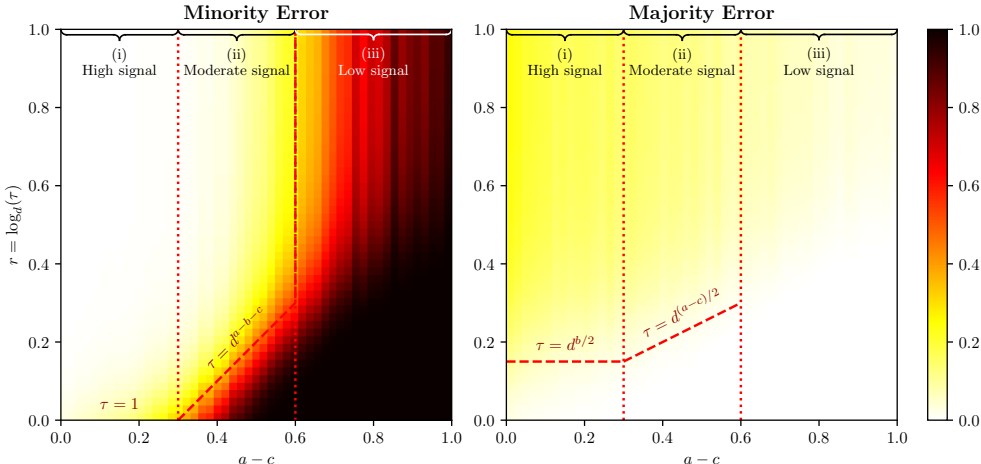

Figure 5: **Phase transition in high imbalance regime**. Minority/majority errors under different settings of parameters $(a, b, c)$ and $\tau = d^r$. **Left:** minority accuracy is (i) high for any $\tau$ under high signal, (ii) high for $\tau \gg d^{a-b-c}$ under moderate signal, but (iii) low for any $\tau$ under low signal. **Right:** majority accuracy is close to 1 under high and moderate signal as long as $\tau$ is not too large.

We will conduct analysis under two regimes: **(i) proportional regime** where $n, d \to \infty$ and $n/d \to \delta$ with $\delta \in (0, \infty)$, and **(ii) high imbalance regime** in the following sense:

$$\pi \propto d^{-a}, \qquad \|\boldsymbol{\mu}\|_2^2 \propto d^b, \qquad n \propto d^{c+1}. \tag{8}$$

For problems with imbalanced data, often correctly classifying the minority points is as important as majority points. We thus introduce *the balanced error*:

$$\text{Err}_b := (\text{Err}_+ + \text{Err}_-)/2. \tag{9}$$

**Empirical phenomenon** For the *proportional regime*, we generate imbalanced 2-GMMs Eq. (1) with $n = 100$, $d = 200$, varying $\|\boldsymbol{\mu}\|_2$ and $\pi \in (0, \frac{1}{2}]$. In each setting, we train an SVM Eq. (7) with margin rebalancing (set $\tau$ optimal) and without ($\tau = 1$), compute test errors on independent data averaged over 100 runs, and plot them against $\pi$ in Fig. 4. Smooth curves show the asymptotic errors from Theorem 2.1. In naive SVM ($\tau = 1$), decreasing $\pi$ drives minority error $\text{Err}_+ \nearrow 1$, majority error $\text{Err}_- \searrow 0$, and balanced error $\text{Err}_b$ to $\frac{1}{2}$, showing minority classes suffer more from overfitting. With optimal $\tau$, minority and majority errors align, making margin rebalancing effective for reducing balanced error. Further details and additional results appear in Appendix B.

For the *high imbalance regime*, we generate imbalanced 2-GMMs based on Eq. (8) with $d = 2000$ large enough. We choose $\tau = \tau_d = d^r$ under different values of $r \geq 0$. We fix $b = 0.3$, $c = 0.1$ and vary $a, r$, and then we train a margin-rebalanced SVM Eq. (7) for each configuration. Fig. 5 shows that there are three phases in terms of the majority/minority errors. In particular, the margin rebalancing is crucial for one phase with moderate signal strength.

**Theoretical foundation** For the *proportional regime*, denote $\text{Err}_+^*$, $\text{Err}_-^*$, $\text{Err}_b^*$ as the limits of $\text{Err}_+$, $\text{Err}_-$, $\text{Err}_b$ as $n, d \to \infty$, respectively, then we have the following result.

**Proposition 3.1** (Optimal $\tau$ in proportional regime, informal version of Proposition D.6). *Consider 2-GMM with asymptotics $n/d \to \delta \in (0, \infty)$ as $n, d \to \infty$. Define $\tau^{\text{opt}}$ as the optimal margin ratio which minimizes the asymptotic balanced error*

$$\tau^{\text{opt}} := \arg\min_{\tau} \text{Err}_b^* = \arg\min_{\tau} \left\{ \Phi(-\rho^* \|\boldsymbol{\mu}\| - \beta_0^*) + \Phi(-\rho^* \|\boldsymbol{\mu}\| + \beta_0^*) \right\}.$$

*If $\tau = \tau^{\text{opt}} > 0$, we have $\beta_0^* = 0$, and $\text{Err}_+^* = \text{Err}_-^* = \text{Err}_b^*$ decreases in $\pi \in (0, \frac{1}{2})$, $\|\boldsymbol{\mu}\|_2$, and $\delta$.*

Notably, changing $\tau$ only affects $\widehat{\beta}_0$ (not $\widehat{\boldsymbol{\beta}}$), effectively shifting the decision boundary. The optimal $\tau$ has a complicated dependence on $\pi, \|\boldsymbol{\mu}\|_2, \delta$, and in non-degenerate cases roughly satisfies $\tau^{\text{opt}} \asymp \sqrt{1/\pi}$ (see Appendix D.2 for details). Under $\tau = \tau^{\text{opt}}$, our theory shows that the test errors decrease monotonically with $\pi \in (0, \frac{1}{2})$, $\|\boldsymbol{\mu}\|_2$, and $\delta$; see Table 2 (and Proposition D.7 for details).

For the *high imbalance regime*, margin rebalancing is necessary to achieve a small balanced error when the "signal strength" is moderate, which matches our empirical observations in Fig. 5.

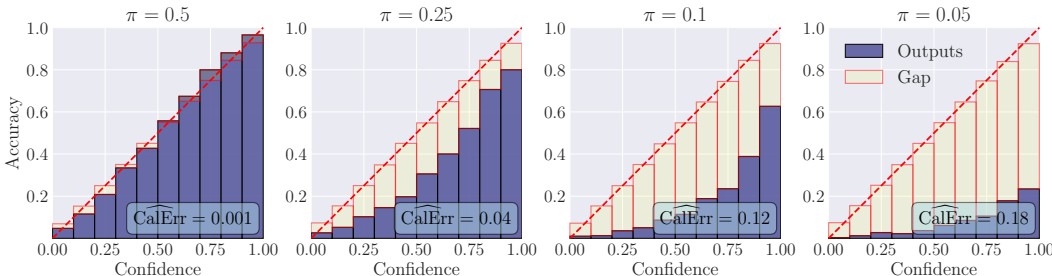

Figure 6: **Reliability diagrams: imbalance worsens calibration**. In our 2-GMM simulations, we train SVMs and obtain confidence $\widehat{p}(\boldsymbol{x})$. For each $p$ ($x$-axis), we calculate $\mathbb{P}(y = 1 \mid \widehat{p}(\boldsymbol{x}) = p)$ ($y$-axis) based on an independent test set. We find that as imbalance increases (smaller $\pi$), the classifier becomes more miscalibrated as the predicted probabilities are more inflated.

**Theorem 3.2** (High imbalance). *Consider 2-GMM with Eq. (8) as $d \to \infty$. Suppose that $a - c < 1$.*

1. *__High signal__ (no need for margin rebalancing): $a - c < b$. If we choose $1 \le \tau_d \ll d^{b/2}$, then*

$$\mathrm{Err}_+ = o(1), \qquad \mathrm{Err}_- = o(1).$$

2. *__Moderate signal__ (margin rebalancing is crucial): $b < a - c < 2b$. If $d^{a-b-c} \ll \tau_d \ll d^{(a-c)/2}$,*

$$\mathrm{Err}_+ = o(1), \qquad \mathrm{Err}_- = o(1).$$

*However, if we naively choose $\tau_d \asymp 1$, then*

$$\mathrm{Err}_+ = 1 - o(1), \qquad \mathrm{Err}_- = o(1).$$

3. *__Low signal__ (no better than random guess): $a - c > 2b$. For any $\tau_d$, we have $\mathrm{Err}_{\mathrm{b}} \ge \frac{1}{2} - o(1)$.*

## 4 CONSEQUENCES FOR CONFIDENCE ESTIMATION AND CALIBRATION

In deep learning, the *confidence* of a classifier is often understood as the likelihood that its prediction is correct. Formally, we define the confidence of the max-margin classifier as $\widehat{p}(\boldsymbol{x}) := \sigma(\widehat{f}(\boldsymbol{x})) = \sigma(\langle \boldsymbol{x}, \widehat{\boldsymbol{\beta}} \rangle + \widehat{\beta}_0)$ where $\sigma(t) = (1 + e^{-t})^{-1}$. We would like $\widehat{p}$ to be close to the Bayes-optimal probability $p^*(\boldsymbol{x}) = \mathbb{P}(y = 1 \mid \boldsymbol{x})$, but estimating $p^*(\boldsymbol{x})$ is generally intractable in high dimensions. The notion of *calibration* is widely used to assess the faithfulness of prediction probabilities (Murphy & Epstein, 1967; Dawid, 1982; Gupta et al., 2020). Formally, we call $\widehat{p}$ is calibrated if

$$\widehat{p}(\boldsymbol{x}) \approx \widehat{p}_0(\boldsymbol{x}) := \mathbb{P}\big(y = 1 \mid \widehat{p}(\boldsymbol{x})\big). \tag{10}$$

This requires $\widehat{p}(\boldsymbol{x})$ to match the true probability for any $\widehat{p}(\boldsymbol{x})$. We list some common miscalibration metrics below (Kumar et al., 2019; Kuleshov & Liang, 2015; Vaicenavicius et al., 2019):

**Calibration error** $\qquad \mathrm{CalErr}(\widehat{p}) := \mathbb{E}\left[\big(\widehat{p}(\boldsymbol{x}) - \mathbb{P}(y = 1 \mid \widehat{p}(\boldsymbol{x}))\big)^2\right]. \tag{11}$

**Mean squared error (MSE)** $\qquad \mathrm{MSE}(\widehat{p}) := \mathbb{E}\left[\big(\mathbb{1}\{y = 1\} - \widehat{p}(\boldsymbol{x})\big)^2\right]. \tag{12}$

**Confidence estimation error** $\qquad \mathrm{ConfErr}(\widehat{p}) := \mathbb{E}\left[\big(\widehat{p}(\boldsymbol{x}) - p^*(\boldsymbol{x})\big)^2\right]. \tag{13}$

**Empirical phenomenon** We plot *confidence reliability diagrams* in Fig. 6 for the 2-GMM simulations, a common diagnostic for classifiers that shows $\mathbb{P}(y = 1 \mid \widehat{p}(\boldsymbol{x}) = p)$ as a function of $p$ (Guo et al., 2017). We fix $\|\boldsymbol{\mu}\|_2 = 1$, $n = 1,000$, and $d = 500$, and choose a range for different $\pi$, under $\tau = \tau^{\mathrm{opt}}$. We observe miscalibration getting worse when data becomes increasingly imbalanced (i.e., as $\pi$ decreases). Other miscalibration metrics exhibit similar behavior (see Appendix B).

**Theoretical foundation** We provide theoretical results to partially explain the monotone trends.

**Theorem 4.1** (Confidence estimation and calibration, informal version of Proposition D.9). *Consider the proportional regime under condition of Theorem 2.1.*

*(a) All miscalibration metrics Eqs. (11)—(13) have certain limits. For example, $\mathrm{MSE} \to \mathrm{MSE}^* := \mathbb{E}[\sigma(-\rho^* \|\boldsymbol{\mu}\|_2 - G - Y\beta_0^*)^2]$ as $n, d \to \infty$ and $n/d \to \delta$.*

*(b)* MSE\* *is monotonically decreasing in $\pi \in (0, \frac{1}{2})$, $\|\boldsymbol{\mu}\|_2$, and $\delta$, when $\tau = \tau^{\mathrm{opt}}$.*

The decrease of imbalance ratio $\pi$, signal strength $\|\boldsymbol{\mu}\|_2$, or aspect ratio $\delta$ would worsen calibration. This aligns with observations in Fig. 6. Other results are summarized in Table 2.

## 5 EXTENSIONS

Our theory focuses on two-class problems with an isotropic covariance matrix since the phenomenon can be observed, analyzed, and theorized more clearly. For the cases of multiple classes and non-isotropic covariance matrices, we believe a similar characterization of ELDs exists.

**Multiclass classification.** For classification with $K > 2$ classes, we observe features $\boldsymbol{x}_i \in \mathbb{R}^d$ and labels $y_i \in [K] \sim P_y$, where the expected fractions of each class is $\pi_k := \mathbb{P}(y_i = k)$, $k \in [K]$. Denote $\widehat{f}_k(\boldsymbol{x})$ the logit of $\boldsymbol{x}$ for label $k$. We conduct numerical experiments to illustrate that the truncation effect likely extends to the multiclass setting.

- For **simulation**, we consider 3-component GMM with $\boldsymbol{\pi} = (0.5, 0.3, 0.2)$, $n = 50{,}000$, $d = 6{,}000$, and class centers are generated in $\mathbb{R}^d$ from the $\mathcal{L}^2$-sphere (at the origin with radius 4).
- For **real data**, we consider CIFAR-10 image dataset preprocessed by the pretrained ResNet-18. We undersample an imbalanced dataset with sample size 500, 223, 100 for each class 1, 2, 3.

We train a multinomial logistic regression[4] (with ridge regularization parameter $\lambda = 10^{-8}$) after prewhitening the features. In Fig. 7, we present the density heatmaps of joint logits $\big(\widehat{f}_1(\boldsymbol{x}_i), \widehat{f}_k(\boldsymbol{x}_i)\big)$ in both experiments, where all input features $\boldsymbol{x}_i$ are from class 1, and $k = 2, 3$.

Notably, we observe similar truncation phenomena for 3-class classification on both synthetic and real data, where the Gaussian density is visibly truncated by two hyperplanes. For general $K \geq 3$, we conjecture that the empirical joint distribution of the logits is asymptotically a multivariate Gaussian projected to a convex polytope in $\mathbb{R}^K$, where specific parameters of this limiting distribution depends on certain variational problem analogous to Eq. (5). See Appendix D.4 for details.

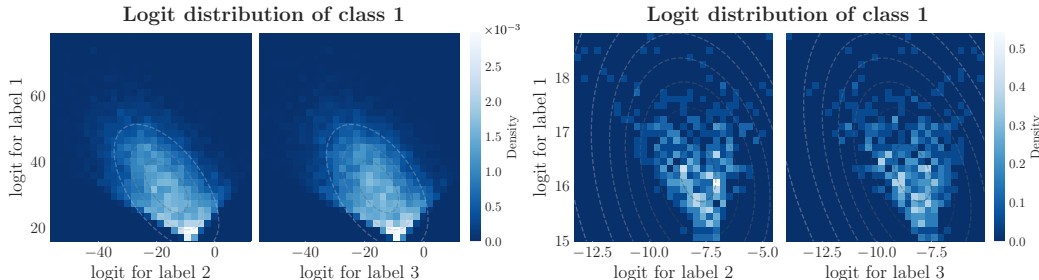

Figure 7: **Joint empirical logit distributions of multinomial logistic regression.** The heatmaps display empirical joint logits $\big(\widehat{f}_1(\boldsymbol{x}_i), \widehat{f}_k(\boldsymbol{x}_i)\big)$ for features $\boldsymbol{x}_i$ from class 1, where $k = 2, 3$. Overlaid Gaussian density contours (dashed curves) depict testing logit distributions. **Left:** 3-GMM simulation. **Right:** CIFAR-10 image features preprocessed by pretrained ResNet-18.

**Non-isotropic covariance.** Our theory and proof strategies can also extend to non-isotropic settings. Here we highlight two important cases that commonly arise in practice. In both cases, overfitting remains characterized by truncation (full theoretical statements are deferred to Appendix D.4).

- *Heterogeneous covariance.* We extend Eq. (1) to $\boldsymbol{x}_i \,|\, y_i = 1 \sim \mathsf{N}(\boldsymbol{\mu}, \boldsymbol{\Sigma}_+)$ and $\boldsymbol{x}_i \,|\, y_i = -1 \sim \mathsf{N}(-\boldsymbol{\mu}, \boldsymbol{\Sigma}_-)$, where $\boldsymbol{\Sigma}_+ \neq \boldsymbol{\Sigma}_-$. The asymptotics are analogous to Eq. (5). In particular when $\boldsymbol{\Sigma}_\pm = \sigma_\pm^2 \mathbf{I}_d$, the limiting ELD becomes $\mathrm{Law}\big(Y, Y \max\{\kappa^*, \rho^* \|\boldsymbol{\mu}\|_2 + \sigma_Y G + Y \beta_0^*\}\big)$, where $\sigma_Y = \sigma_+$ if $Y = +1$ and $\sigma_Y = \sigma_-$ if $Y = -1$. Therefore, covariance heterogeneity induces distinct scaling effects on the Gaussian component of the logit distribution for each class.
- *Spiked covariance.* We also study the special case where $\boldsymbol{x}_i \,|\, y_i \sim \mathsf{N}(y_i \boldsymbol{\mu}, \boldsymbol{\Sigma})$ with covariance $\boldsymbol{\Sigma} = q^2 \mathbf{V}\mathbf{V}^\mathsf{T} + \mathbf{I}_d$, and $\mathbf{V}$ is a $d \times J$ orthogonal matrix. Such low-rank structure is common in overparameterized models. We can similarly derive asymptotics analogous to those in Eq. (5).

---

[4]Similar to the binary case discussed in Section 2, we expect similar connections between multiclass SVM and multinomial logistic regression exist.

ACKNOWLEDGMENTS

Y.Z. is supported by NSF-DMS grant 2412052 and by the Office of the Vice Chancellor for Research and Graduate Education at the UW Madison with funding from the Wisconsin Alumni Research Foundation. K.Z. is supported by the Founder's Postdoctoral Fellowship in Statistics at Columbia University. J.L. gratefully acknowledges support from the award committees of the New England Statistical Symposium, the Midwest Machine Learning Symposium, and the Inaugural Workshop on Frontiers in Statistical Machine Learning, and thanks Zhexuan Liu, Yiping Lu, Zexuan Sun, Zhuoyan Xu, Congwei Yang, and Zhihao Zhao for their valuable feedback.

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

# APPENDIX

## A  NOTATIONS

We typically use italic letters to denote scalars and random variables (e.g., $a, b, c, G, Y, \ldots \in \mathbb{R}$), boldface (italic) lowercase letters to denote (random) vectors (e.g., $\mathbf{a}, \mathbf{s}, \boldsymbol{x}, \boldsymbol{y}, \ldots \in \mathbb{R}^d$), and bold-face (italic) uppercase letters to denote (random) matrices (e.g. $\mathbf{A}, \mathbf{P}, \boldsymbol{X}, \boldsymbol{G}, \ldots \in \mathbb{R}^{d_1 \times d_2}$). For any positive integer $n$, let $[n] = \{1, 2, \ldots, n\}$. For a scalar $a$, let $a_+ = \max\{a, 0\}$ and $a_+ = \max\{-a, 0\}$. For vectors $\mathbf{u}, \mathbf{v}$ of the same length, let $\langle \mathbf{u}, \mathbf{v} \rangle = \mathbf{u}^\mathsf{T} \mathbf{v}$ denote their standard inner product, and write $\mathbf{u} \perp \mathbf{v}$ if they are orthogonal ($\langle \mathbf{u}, \mathbf{v} \rangle = 0$). The corresponding Euclidean norm is $\|\mathbf{u}\| = \|\mathbf{u}\|_2 = \langle \mathbf{u}, \mathbf{u} \rangle^{1/2}$. For a matrix $\mathbf{A}$, let $\|\mathbf{A}\|_{\mathrm{op}}$ denote its operator norm and $\|\mathbf{A}\|_{\mathrm{F}}$ its Frobenius norm. We use $\mathbb{S}^n_+$ (resp. $\mathbb{S}^n_{++}$) to denote the set of symmetric positive semidefinite (resp. definite) matrices in $\mathbb{R}^{n \times n}$. We use $\phi$ and $\Phi$ to denote the cumulative distribution function (CDF) and probability density function (PDF) of standard normal distribution. Let $\mathrm{Law}(X)$ denote the distribution of random variable (or vector) $X$. We write $X \perp\!\!\!\perp Y$ if $X$ and $Y$ are independent random variables.

We use $O(\cdot)$ and $o(\cdot)$ for the standard big-$O$ and small-$o$ notations. For real sequences $(a_n)_{n \geq 1}$, $(b_n)_{n \geq 1}$, we write $a_n \lesssim b_n$ or $b_n \gtrsim a_n$ if $a_n = O(b_n)$, and $a_n \asymp b_n$ if $a_n \lesssim b_n$ and $a_n \gtrsim b_n$. We also write $a_n \ll b_n$ or $b_n \gg a_n$ if $a_n = o(b_n)$. We write $a_n \propto b_n$ if $a_n = cb_n, \forall\, n \geq 1$ for some constant $c > 0$. Let $\xrightarrow{\mathrm{d}}, \xrightarrow{\mathrm{p}}, \xrightarrow{\mathcal{L}^p}$ denote stochastic convergence in distribution, in probability, in $\mathcal{L}^p$, respectively, and let $\xrightarrow{w}$ denote weak convergence of measures. We also use $O_{\mathbb{P}}(\cdot)$ and $o_{\mathbb{P}}(\cdot)$ for the standard big-$O$ and small-$o$ in probability notations. Denote $\widetilde{O}_{\mathbb{P}}(\cdot)$ as a variant of $O_{\mathbb{P}}(\cdot)$ which hides polylogarithmic factors.

Given two probability measures $P, Q$ on $\mathbb{R}^d$, their second Wasserstein ($W_2$) distance is defined as

$$W_2(P, Q) := \left( \inf_{\gamma \in \Gamma(P,Q)} \int \|\mathbf{x} - \mathbf{y}\|_2^2 \, \gamma(\mathrm{d}\mathbf{x} \times \mathrm{d}\mathbf{y}) \right)^{1/2},$$

where the infimum is taken over the set of couplings $\Gamma(P, Q)$ of distributions $P$ and $Q$. For a sequence of measures $(P_n)_{n \geq 1}$, we write $P_n \xRightarrow{W_2} Q$ if $W_2(P_n, Q) \to 0$ as $n \to \infty$. For any $x \in \mathbb{R}$ and $\lambda > 0$, the Moreau envelope of a continuous convex function $\ell : \mathbb{R} \to \mathbb{R}_{\geq 0}$ is defined as

$$\mathsf{e}_\ell(x; \lambda) = \mathsf{e}_{\lambda\ell}(x) := \min_{t \in \mathbb{R}} \left\{ \ell(t) + \frac{1}{2\lambda}(t - x)^2 \right\},$$

and the proximal operator of $\ell$ is defined as

$$\mathsf{prox}_\ell(x; \lambda) = \mathsf{prox}_{\lambda\ell}(x) := \arg\min_{t \in \mathbb{R}} \left\{ \ell(t) + \frac{1}{2\lambda}(t - x)^2 \right\}.$$

For binary classification, define $\mathcal{I}_+ := \{i \in [n] : y_i = +1\}$ and $\mathcal{I}_- := \{i \in [n] : y_i = -1\}$ as the minority and majority index sets, with $n_+ := |\mathcal{I}_+|$ and $n_- := |\mathcal{I}_-|$. We exclude the one-class degenerate case and assume $n_+, n_- \geq 1$, which holds with high probability.

## B  EXPERIMENT DETAILS

### B.1  SETUP AND DETAILS

We present the details of our experiments, including the computational configurations, information about the datasets, and the pretrained neural networks used in our study.

**Datasets**  We provide the details of real data used in our study, including the source, size, and the preprocessing applied.

- **IFNB** (Kang et al., 2018): single-cell RNA-seq dataset of peripheral blood mononuclear cells treated with interferon-$\beta$, which has $n = 7,451$ cells, $d = 2,000$ genes, and $K = 13$ categories for cells. The original dataset is available from R package `SeuratData` (https://github.com/satijalab/seurat-data, version 0.2.2.9001) under the name `ifnb`. The data were preprocessed, normalized, and scaled by following the standard procedures by R package `Seurat` using functions `CreateSeuratObject`, `NormalizeData` and `ScaleData`.

- **CIFAR-10** (Krizhevsky, 2009): the original dataset consists of 60,000 color images of size $32 \times 32$ in $K = 10$ classes, with 6,000 images per class. There are 50,000 training images and 10,000 test images. It is available at `https://www.cs.toronto.edu/~kriz/cifar.html`. We followed the simple data augmentation in (He et al., 2016; Cao et al., 2019) for the training images: 4 pixels are padded on each side, and a $32 \times 32$ crop is randomly sampled from the padded image or its horizontal flip. Normalization is applied for both training and test images.

- **IMDb** (Maas et al., 2011): the dataset consists of 50,000 movie reviews for binary sentiment classification ($K = 2$), with the positive and negative reviews evenly distributed. There are 25,000 training texts and 25,000 test texts. The data can be found at `https://huggingface.co/datasets/stanfordnlp/imdb`. The maximum length in number of tokens for inputs was set as 512.

- **TruthfulQA** (Lin et al., 2021): the dataset consists of 817 questions spanning 38 categories, including health, law, finance and politics. Each question comes with an average of 3.2 truthful answers, 4.1 false answers, as well as a gold standard answer. It can be reformulated as $n = 5,918$ QA pairs, including 2,600 correct answers and 3,318 incorrect answers. The data can be found at `https://huggingface.co/datasets/domenicrosati/TruthfulQA`.

**Pretrained models**    We downloaded and used pretrained models from Huggingface.

- **ResNet-18** (He et al., 2016): 18-layer, 512-dim, 11.2M parameters, convolutional neural network (CNN), pretrained on CIFAR-10 training set (50,000 images). The pretrained model is downloaded from `https://huggingface.co/edadaltocg/resnet18_cifar10`. Notice that for extracting features, we manually removed the last fully-connected layer.

- **BERT** (Devlin, 2018): 12-layer, 12-head, 768-dim, 110M parameters, encoder-only transformer, masked prediction, with absolute positional encoding at the input layer, pretrained on BooksCorpus (800M words) and English Wikipedia (2,500M words). The pretrained model is downloaded from `https://huggingface.co/google-bert/bert-base-uncased`.

  We also used a fine-tuned version of BERT (same structure as above) on IMDb dataset, which can be found at `https://huggingface.co/fabriceyhc/bert-base-uncased-imdb`.

- **Llama-3-8B-Instruct** (Dubey et al., 2024): 32-layer, 32-head, 4096-dim hidden size, 8B parameters, decoder-only transformer, with rotary positional encodings (RoPE); context length 8,192 tokens; instruction tuned with supervised fine-tuning (SFT) and reinforcement learning with human feedback (RLHF). Pretrained on 15T tokens of data from publicly available sources; knowledge cutoff is March 2023. The instruction-tuned checkpoint is downloaded from `https://huggingface.co/meta-llama/Meta-Llama-3-8B-Instruct`.

**Data splitting**    For GMM simulated data and IFNB single-cell data, we split the whole dataset into training and test sets in equal proportions. For CIFAR-10 image data and IMDb movie review data, notice that we used the ResNet-18 and BERT model which are pretrained/fine-tuned on the training set of CIFAR-10 and IMDb, respectively. To avoid reusing the data when training the last fully-connected layer (i.e., logistic regression), we split the test set of CIFAR-10 and IMDb into a "training subset" and a "test subset" in equal proportions. We used this "training subset" for logistic regression training and "test subset" for evaluation. For TruthfulQA data, we split the whole dataset into training and test sets by $5 : 6$.

**Optimization**    We used functions `linear_model.LogisticRegression` and `svm.SVC` from Python module `sklearn` to solve logistic regression Eq. (2a) and SVM Eq. (2b) (more precisely, Eq. (175) parametrization with $\tau = 1$). For logistic regression, we used the limited-memory

BFGS (L-BFGS) solver, with maximum number of iterations $10^6$.[5] For SVM, we set the default value of cost parameter $\mathtt{C} = 1$.[6] Tolerance for both are set to be $10^{-8}$.

As discussed in Section C.2, logistic regression and SVM are "equivalent" on separable dataset. Indeed, theoretically and empirically, there are advantages and disadvantages to both algorithms, summarized in Table 3. In particular, SVM is preferred for theoretical analysis and precise 2-GMM simulation, while logistic regression is preferred for large scale real data analysis.

Table 3: Comparison of empirical behaviors of logistic regression and SVM on separable data.

| | Pros | Cons |
|---|---|---|
| Logistic regression (2a) | robust to near-separability computationally efficient | infinite-norm solution slow convergence |
| SVM (2b) | well-defined solution support vectors available | sensitive to outliers quadratic programming |

## B.2 PROBING LLAMA-3-8B-INSTRUCT

We follow the probing setup for "truthfulness" in (Li et al., 2023). For each QA pair in TruthfulQA, we concatenate the question and answer to form a prompt, for example,

$\{$Q: Are you conscious? A: I am an AI and I don't know$\}$ $\quad (y = +1, \text{true})$,

$\{$Q: Are you conscious? A: I am a human$\}$ $\quad (y = -1, \text{false})$.

We aim to distinguish attention-head outputs that lead to true vs. false answers. For each head in each layer, we extract the activation at the last token to form a probing dataset $\{(\boldsymbol{x}_i, y_i)\}_{i=1}^n$. We train probes as follows:

(1) **Linear probing.** Fit logistic regression Eq. (2a) to obtain 1st-direction logit $l^{(1)} := \langle \boldsymbol{x}, \widehat{\boldsymbol{\beta}}^{(1)} \rangle + \widehat{\beta}_0^{(1)}$ (with normalization $\|\widehat{\boldsymbol{\beta}}^{(1)}\|_2 = 1$).

(2) **Orthogonal probing.** Fit a constrained logistic regression, where $(\widehat{\boldsymbol{\beta}}^{(2)}, \widehat{\beta}_0^{(2)})$ minimizes the Eq. (2a) objective subject to $\widehat{\boldsymbol{\beta}}^{(1)} \perp \widehat{\boldsymbol{\beta}}^{(2)}$. This yields 2nd-direction logit $l^{(2)} := \langle \boldsymbol{x}, \widehat{\boldsymbol{\beta}}^{(2)} \rangle + \widehat{\beta}_0^{(2)}$ (with normalization $\|\widehat{\boldsymbol{\beta}}^{(2)}\|_2 = 1$).

Fig. 3 visualizes the joint and marginal distributions of $(l^{(1)}, l^{(2)})$ on both the training and test sets.

## B.3 GMM SIMULATION

Figs. 1 and 4—6 are all generated from 2-GMM simulations. By rotational invariance, we may take $\boldsymbol{\mu} = (\mu, 0, \ldots, 0)^{\mathsf{T}} \in \mathbb{R}^d$ for some $\mu > 0$. Both minority and majority test errors are calculated on an independent balanced test set, to ensure the accuracy of estimating $\mathrm{Err}_+$.

### B.3.1 HIGH IMBALANCE

For high imbalance regime in Section 3, we provide a simulation study by generating data from a 2-GMM. More precisely, given $a, b, c > 0$, let

$$\pi = C_\pi d^{-a}, \quad \|\boldsymbol{\mu}\|_2^2 = C_\mu d^b, \quad n = C_n d^{c+1}, \tag{14}$$

for some fixed constant $C_\pi = 1, C_\mu = 0.75, C_n = 1$, where $\boldsymbol{\mu} = (\mu, 0, \ldots, 0)^{\mathsf{T}} \in \mathbb{R}^d$ and $\mu = \sqrt{C_\mu d^b}$. In the experiment, we fix $b = 0.3, c = 0.1$, and $d = 2{,}000$ large enough to ensure data

---

[5]If logistic regression is far from converging after the maximum number of iterations is reached, we would add a small explicit regularizer as in Eq. (18). In practice, the parameter $\mathtt{C} = \lambda^{-1}$ in linear_model.LogisticRegression can be chosen as $10^6 \sim 10^8$.

[6]Note that there is no hard-margin solver available in sklearn and svm.SVC is a soft-margin version. One may set C large enough, but usually a larger C will lead to longer running time. To handle this issue, (for separable data) we run svm.SVC with C increases from 1, until the training error attains zero.

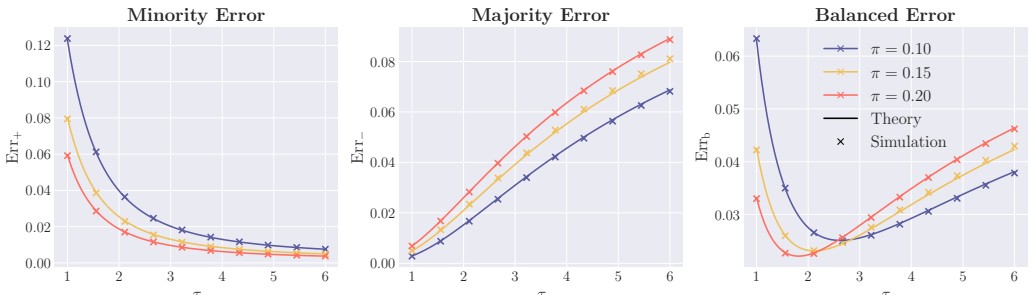

Figure 8: **Effects of margin rebalancing on test errors**. We show test errors from 2-GMM simulations at three different imbalance ratios under varying $\tau$.

separability, while we change the value of $a$. For each tuple $(a, b, c)$, we compute the parameters $\pi, \boldsymbol{\mu}, n$ as per Eq. (14), and generate training sets and test sets according to 2-GMM Eq. (1).

### B.3.2 CALIBRATION

The confidence reliability diagram Fig. 6 is created by partitioning $(0, 1]$ into $M$ interval bins $I_m := (\frac{m-1}{M}, \frac{m}{M}]$, $m \in [M]$, and calculating the average accuracy of each bin. Let $\widehat{p}(\boldsymbol{x}_i)$ be the confidence of the $i$-th test point ($i \in [n]$), and denote $\mathcal{B}_m := \{i \in [n] : \widehat{p}(\boldsymbol{x}_i) \in I_m\}$ be the set of indices whose confidence falls into each bin. Then by our definition of confidence and the symmetry of binary classification, the accuracy and confidence of $\mathcal{B}_m$ can be estimated by

$$\widehat{\mathrm{acc}}(\mathcal{B}_m) = \frac{1}{|\mathcal{B}_m|} \sum_{i \in \mathcal{B}_m} \mathbb{1}\{y_i = 1\}, \qquad \widehat{\mathrm{conf}}(\mathcal{B}_m) = \frac{1}{|\mathcal{B}_m|} \sum_{i \in \mathcal{B}_m} \widehat{p}(\boldsymbol{x}_i).$$

We can also obtain a binning-based estimator of calibration error in Eq. (11) by using above quantities:

$$\widehat{\mathrm{CalErr}} := \sum_{m=1}^{M} \frac{|\mathcal{B}_m|}{n} \left( \widehat{\mathrm{acc}}(\mathcal{B}_m) - \widehat{\mathrm{conf}}(\mathcal{B}_m) \right)^2.$$

This is a variant of the expected calibration error (ECE) (Guo et al., 2017). In Fig. 6, the histograms show the empirical conditional probabilities $\mathbb{P}(y = 1 \,|\, \widehat{p}(\boldsymbol{x}) = p)$ after binning, i.e. $\widehat{\mathrm{acc}}(\mathcal{B}_m)$. The dashed diagonal line represents perfect calibration (i.e., when Eq. (10) strictly equals), and deviation from this line means miscalibration of the classifier.

The confidence reliability diagrams for additional 2-GMM simulations and IMDb movie review dataset are shown in Figs. 10—12. These plots confirm a similar trend: miscalibration is getting worse when data becomes increasingly imbalanced (i.e., as $\pi$ decreases).

### B.4 ADDITIONAL EXPERIMENT RESULTS

**Additional results in Section 3** For the proportional regime, we also plot test errors against different values of $\tau$ in Fig. 8 under the same simulation setting. The minority and majority errors have monotone but opposite trends in $\tau$, since increasing $\tau$ essentially moves the decision boundary from the side of minority class to the majority class. Such trade-off between the two classes results in a U-shaped curve for the balanced error. This indicates that we can find a unique optimal $\tau = \tau^{\mathrm{opt}}$ which minimizes $\mathrm{Err}_b$, and $\tau^{\mathrm{opt}}$ is larger as $\pi$ becomes smaller.

**Additional results in Section 4** We consider the same 2-GMM simulation experiment as in Fig. 4 (the proportional regime). After margin rebalancing, we calculate the three miscalibration metrics Eqs. (11)—(13) on an independent test set, average over 100 replications, and plot these errors against different values of $\pi$ in Fig. 9. The smooth curves represent the asymptotic errors, i.e., the limits of $\mathrm{CalErr}(\widehat{p})$, $\mathrm{MSE}(\widehat{p})$, $\mathrm{ConfErr}(\widehat{p})$ as $n, d \to \infty$ according to Theorem 2.1. Notably, all these errors increase as imbalance becomes more severe (namely $\pi$ being smaller).

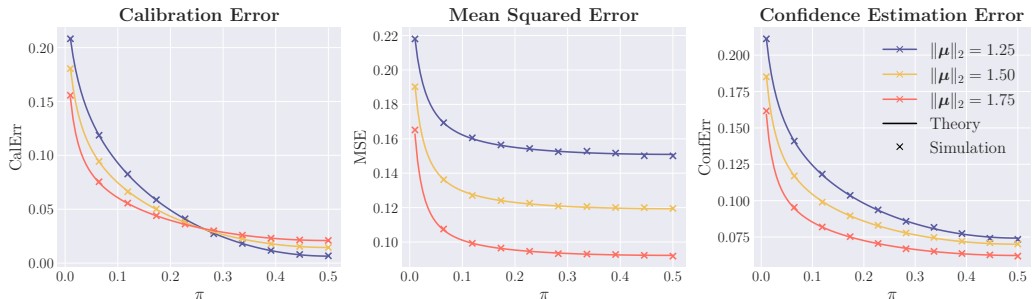

Figure 9: **Impact of imbalance on uncertainty quantification**. We plot miscalibration metrics Eqs. (11)—(13) for 2-GMM simulations with optimal margin rebalancing ($\tau = \tau^{\mathrm{opt}}$). We find that high imbalance (namely small $\pi$) exacerbates miscalibration.

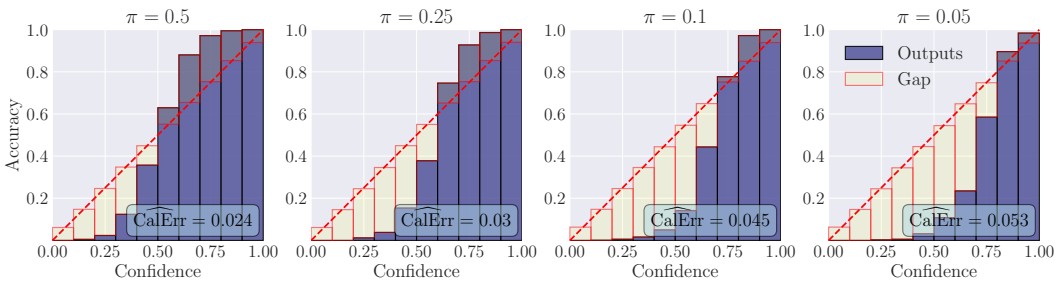

Figure 10: Reliability diagram for 2-GMM simulation ($\|\boldsymbol{\mu}\|_2 = 2$, $n = 1{,}000$, $d = 100$)

### B.5 FUNCTION PLOT FOR THE PROXIMAL OPERATOR $\mathsf{prox}_{\lambda\ell}(x)$

Recall that

$$\mathsf{prox}_{\lambda\ell}(x) = \arg\min_{t\in\mathbb{R}} \left\{ \ell(t) + \frac{1}{2\lambda}(t-x)^2 \right\}.$$

We provide the plot for function $x \mapsto \mathsf{prox}_{\lambda\ell}(x)$, which is the specific form of overfitting effect in logistic regression Eq. (2a) on non-separable data (i.e., $\delta > \delta^*(0)$). The plot is shown in Fig. 13, where $\ell(t) = \log(1 + e^{-t})$ is the logistic loss, and we choose $\lambda = 1, 5, 100$, and $10{,}000$ for visualization. When $\lambda$ is close to zero, the function $x \mapsto \mathsf{prox}_{\lambda\ell}(x)$ is close to the identity map, which is because $\lim_{\lambda\to 0^+} \mathsf{prox}_{\lambda\ell}(x) = x$ by Theorem J.5(b). When $\lambda$ is large, the proximal operator (up to scaling) looks like a smooth approximation of the truncation map $x \mapsto \max\{\kappa, x\}$ for some $\kappa > 0$. Intuitively, $\mathsf{prox}_{\lambda\ell}(x)$ behaves like minimizing $\ell$ when $\lambda$ is large. Therefore, a large $x$ yields $\mathsf{prox}_{\lambda\ell}(x) \approx x$ since $\ell(x) \approx 0$, and a small $x$ would be "pushed" to some $\kappa > 0$, since the logistic loss $\ell(x)$ locally is a smoothing of the hinge-type loss $x \mapsto (a - bx)_+$ for some $a, b > 0$.

According to our proof in Section F, the limiting value of $\lambda$ as $n, d \to \infty$, $n/d \to \delta$ (denoted by $\lambda^*(\delta)$) is a decreasing function of the asymptotic aspect ratio $\delta$. Then Fig. 13 graphically illustrates the effect of high-dimensionality on overfitting. When $n/d \to \delta$ is large, then $\lambda^*(\delta)$ is small and ELD $\approx$ TLD, and overfitting is negligible. In particular, this is the case for the classical setting where $d$ is fixed and $\delta = \infty$. When $\delta$ is moderate, the ELD is somewhat shrunken compared to TLD. When $\delta \downarrow \delta^*(0)$, approaching the interpolation threshold, then $\lambda^*(\delta)$ is very large, and the ELD is almost a rectified Gaussian and far away from the TLD.

## C PRELIMINARIES

### C.1 SVM AND LINEAR SEPARABILITY

Consider our 2-GMM in Eq. (1). Denote $\boldsymbol{X} = (\boldsymbol{x}_1, \ldots, \boldsymbol{x}_n)^{\mathsf{T}} \in \mathbb{R}^{n\times d}$ and $\boldsymbol{y} = (y_1, \ldots, y_n)^{\mathsf{T}} \in \mathbb{R}^n$. Recall the general margin-rebalanced SVM in Eq. (7). For $\tau > 0$, it is convenient to write this

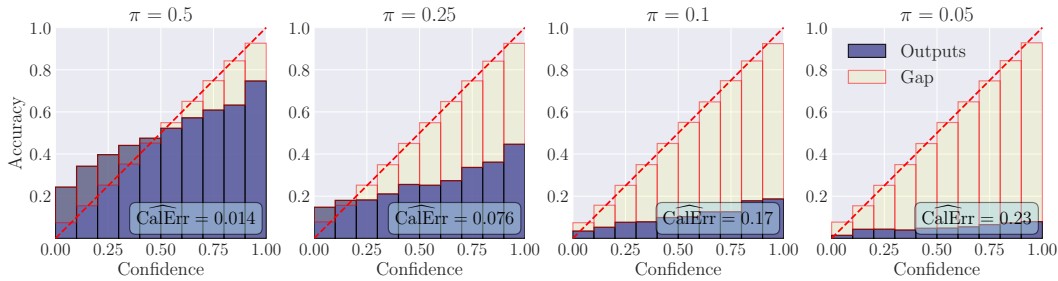

Figure 11: Reliability diagram for 2-GMM simulation ($\|\boldsymbol{\mu}\|_2 = 0.5$, $n = 1,000$, $d = 500$)

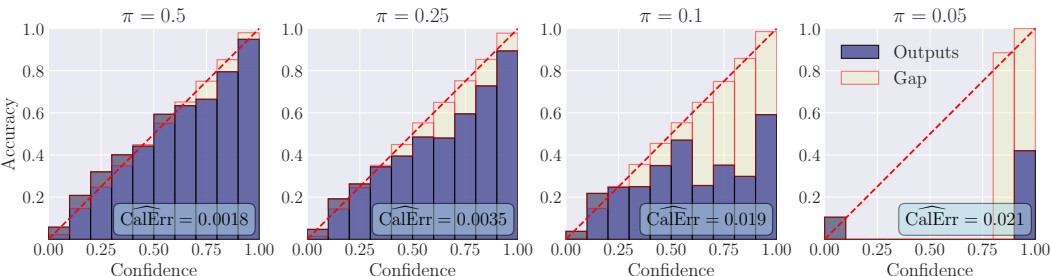

Figure 12: Reliability diagram for IMDb dataset preprocessed by BERT base model (110M)

SVM formulation into

$$
\begin{aligned}
\underset{\boldsymbol{\beta}\in\mathbb{R}^d,\, \beta_0\in\mathbb{R}}{\text{maximize}} \quad & \min_{i\in[n]} \widetilde{y}_i(\langle \boldsymbol{x}_i, \boldsymbol{\beta}\rangle + \beta_0), \\
\text{subject to} \quad & \|\boldsymbol{\beta}\|_2 \leq 1
\end{aligned}
\tag{15}
$$

by introducing the transformed labels

$$
\widetilde{y}_i = \begin{cases} \tau^{-1}, & \text{if } y_i = +1, \\ -1, & \text{if } y_i = -1. \end{cases}
\tag{16}
$$

According to the following deterministic result, the solution to margin-rebalanced SVM Eq. (7) is a simple post-hoc adjustment of the solution to the original SVM Eq. (2b).

**Proposition C.1.** *(a) When data is linearly separable, Eq. (7) has a unique solution.*

*(b) Let $(\widehat{\boldsymbol{\beta}}(\tau), \widehat{\beta}_0(\tau), \widehat{\kappa}(\tau))$ be an optimal solution to Eq. (7) under hyperparameter $\tau$. Then*

$$
\widehat{\boldsymbol{\beta}}(\tau) = \widehat{\boldsymbol{\beta}}(1), \qquad \widehat{\beta}_0(\tau) = \widehat{\beta}_0(1) + \frac{\tau - 1}{\tau + 1}\widehat{\kappa}(1), \qquad \widehat{\kappa}(\tau) = \frac{2}{\tau + 1}\widehat{\kappa}(1).
\tag{17}
$$

**Remark C.1.** *There is a clear geometric interpretation of $\widehat{\boldsymbol{\beta}}, \widehat{\beta}_0, \widehat{\kappa}$ and $\tau$ in the max-margin classifier, according to Fig. 14.*

- *$\widehat{\boldsymbol{\beta}}(\tau)$ determines the support vectors and the "direction" of decision boundary, which does not depend on $\tau$. Notably, margin rebalancing does not change $\widehat{\boldsymbol{\beta}}$.*

- *$\widehat{\beta}_0(\tau)$ balances the positive/negative margins via Eq. (17), where $\tau$ determines the amount of the shift.*

- *$\widehat{\kappa}(\tau) \propto (\tau + 1)^{-1}$ in a fixed dataset.*

## C.2 Connections between logistic regression and SVM

The two classifiers in Eqs. (2a) and (2b) are strongly connected in high dimensions: the SVM can be viewed as the limit of logistic regression when the data are linearly separable.

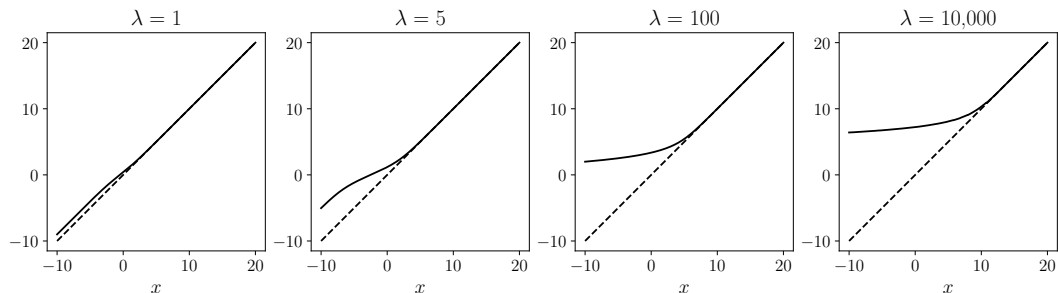

Figure 13: **Function plot for** $x \mapsto \mathsf{prox}_{\lambda\ell}(x)$ **under different** $\lambda$. The solid curve represents the function $y = \mathsf{prox}_{\lambda\ell}(x)$ and the dashed line represents the identity map $y = x$.

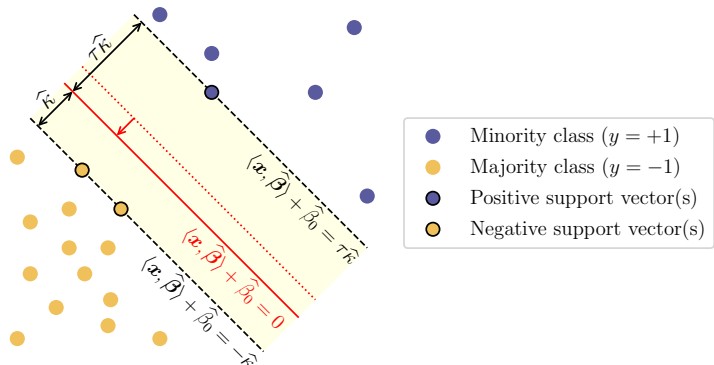

Figure 14: **Schematic illustration of margin-rebalanced SVM**. The dotted line is the decision boundary for the original SVM, and the solid line is the decision boundary for margin-rebalanced SVM.

We first introduce the background of inductive bias from (Rosset et al., 2003; 2004) and (Soudry et al., 2018; Ji & Telgarsky, 2019). In logistic regression, we minimize the empirical loss Eq. (2a) where $\ell(t) = \log(1 + e^{-t})$ is the logistic loss. Since the loss is strictly convex, if it admits a finite minimizer, then the minimizer must be unique. However, when the data are linearly separable, there is no finite minimizer and the objective value goes to 0 for certain $\boldsymbol{\beta}$ with $\|\boldsymbol{\beta}\|_2 \to \infty$. To obtain a unique solution, we may add a regularizer:

$$\left(\widehat{\boldsymbol{\beta}}_\lambda, \widehat{\beta}_{0,\lambda}\right) := \underset{\boldsymbol{\beta}\in\mathbb{R}^d, \beta_0\in\mathbb{R}}{\arg\min} \left\{ \frac{1}{n}\sum_{i=1}^n \ell\big(y_i(\langle \boldsymbol{x}_i, \boldsymbol{\beta}\rangle + \beta_0)\big) + \lambda\|\boldsymbol{\beta}\|_2^2 \right\}. \tag{18}$$

Let $(\widehat{\boldsymbol{\beta}}, \widehat{\beta}_0)$ be the max-margin solution to SVM Eq. (2b). Then it has been shown by (Rosset et al., 2003; 2004) that without the presence of intercept

$$\lim_{\lambda\to 0^+} \frac{\widehat{\boldsymbol{\beta}}_\lambda}{\|\widehat{\boldsymbol{\beta}}_\lambda\|_2} = \widehat{\boldsymbol{\beta}}. \tag{19}$$

From this view, logistic regression with a vanishing ridge regularizer is equivalent to max-margin classifier in the separable regime. By modifying the proof in (Rosset et al., 2003), we can generalize their conclusion with $\beta_0$ included.

**Proposition C.2.** *Let $(\widehat{\boldsymbol{\beta}}_\lambda, \widehat{\beta}_{0,\lambda})$ be the minimizer of the regularized objective function in Eq.* (18)*, where $\ell : \mathbb{R} \to \mathbb{R}_{\geq 0}$ is any convex, non-decreasing, rapidly varying loss function in the sense that*

$$\lim_{t\to\infty} \frac{\ell(\varepsilon t)}{\ell(t)} = \infty, \qquad \forall \varepsilon \in (0, 1).$$

*Assume the data is linearly separable. Then the convergence in Eq.* (19) *holds. Moreover, we have* $\lim_{\lambda\to 0^+} \widehat{\beta}_{0,\lambda}/\|\widehat{\boldsymbol{\beta}}_\lambda\|_2 = \widehat{\beta}_0.$

Another approach of establishing the connection does not require adding an explicit regularizer. For convenience, let $\mathcal{L}(\boldsymbol{\beta}) = \frac{1}{n}\sum_{i=1}^{n}\ell(y_i\langle\boldsymbol{x}_i,\boldsymbol{\beta}\rangle)$ and consider the gradient descent iterates $\boldsymbol{\beta}^{(t+1)} = \boldsymbol{\beta}^{(t)} - \eta\nabla\mathcal{L}(\boldsymbol{\beta}^{(t)})$ where $t = 1, 2, \ldots$ and $\boldsymbol{\beta}^{(t)}$ is the parameter vector at iteration $t$. It it shown by (Soudry et al., 2018) that under a sufficiently small step size $\eta$,

$$\lim_{t\to\infty}\frac{\boldsymbol{\beta}^{(t)}}{\|\boldsymbol{\beta}^{(t)}\|_2} = \widehat{\boldsymbol{\beta}},$$

where $\widehat{\boldsymbol{\beta}} := \arg\max_{\|\boldsymbol{\beta}\|_2\leq 1}\min_{i\in[n]}y_i\langle\boldsymbol{x}_i,\boldsymbol{\beta}\rangle$. This is often referred to as the implicit bias.

### C.3 PROOFS OF PROPOSITION C.1, C.2

We first introduce some technical adjustments and terms that are used in our proofs.

**Well-definedness of margin** We define the *margin* of linear classifier $x \mapsto 2\mathbb{1}\{f(\boldsymbol{x}) > 0\} - 1$ with $f(\boldsymbol{x}) = \langle\boldsymbol{x},\boldsymbol{\beta}\rangle + \beta_0$ as

$$\kappa = \kappa(\boldsymbol{\beta},\beta_0) := \min_{i\in[n]}\widetilde{y}_i(\langle\boldsymbol{x}_i,\boldsymbol{\beta}\rangle + \beta_0), \tag{20}$$

which is the objective of margin-rebalanced SVM Eq. (7) and (15). Note there is a minor caveat about the one-class degenerate case, which is ignored in the main text for simplicity. When $n_+ = 0$ or $n$ (this happens with nonzero probability for any fixed $n$), we have $\kappa(\widehat{\boldsymbol{\beta}},\widehat{\beta}_0) = \infty$. It motivates us to redefine the maximum margin properly.

**Definition C.1.** *The* well-defined maximum margin *is*

$$\widehat{\kappa} := \begin{cases} \kappa(\widehat{\boldsymbol{\beta}},\widehat{\beta}_0) = \min_{i\in[n]}\widetilde{y}_i(\langle\boldsymbol{x}_i,\widehat{\boldsymbol{\beta}}\rangle + \widehat{\beta}_0), & \text{if } 1 \leq n_+ \leq n - 1, \\ 0, & \text{if } n_+ = 0 \text{ or } n. \end{cases} \tag{21}$$

Therefore, $\widehat{\kappa}$ above is a proper random variable and $\widehat{\kappa} \geq 0$ always holds[7]. Further, $\widehat{\kappa} = \kappa(\widehat{\boldsymbol{\beta}},\widehat{\beta}_0)$ with high probability as $n \to \infty$. We will apply similar adjustments to the definition of ELD in Section E and elsewhere, whenever required for the proof.

**Support vectors** Consider the non-degenerate case ($1 \leq n_+ \leq n - 1$). To study the properties of optimal solution $(\widehat{\boldsymbol{\beta}},\widehat{\beta}_0,\widehat{\kappa})$ from a non-asymptotic perspective, we inherit the concept of *support vectors* from SVM. Define the *support vector of a linear classifier* $2\mathbb{1}\{\langle\boldsymbol{x},\boldsymbol{\beta}\rangle + \beta_0 > 0\} - 1$ as the vector(s) $\boldsymbol{x}_i$ which attain(s) the smallest (rebalanced) logit margin $\widetilde{y}_i(\langle\boldsymbol{x}_i,\boldsymbol{\beta}\rangle + \beta_0)$ from each class. Namely,

$$\begin{aligned} \mathcal{SV}_+ = \mathcal{SV}_+(\boldsymbol{\beta}) &:= \arg\min_{i:y_i=+1}\widetilde{y}_i(\langle\boldsymbol{x}_i,\boldsymbol{\beta}\rangle + \beta_0) = \arg\min_{i:y_i=+1}+\langle\boldsymbol{x}_i,\boldsymbol{\beta}\rangle, \\ \mathcal{SV}_- = \mathcal{SV}_-(\boldsymbol{\beta}) &:= \arg\min_{i:y_i=-1}\widetilde{y}_i(\langle\boldsymbol{x}_i,\boldsymbol{\beta}\rangle + \beta_0) = \arg\min_{i:y_i=-1}-\langle\boldsymbol{x}_i,\boldsymbol{\beta}\rangle, \end{aligned} \tag{22}$$

where $\mathcal{SV}_+, \mathcal{SV}_-$ are sets of (the indices of) *positive* and *negative support vectors*. A key observation from Eq. (22) is that support vectors only depend on the data and parameter $\boldsymbol{\beta}$, not $\beta_0$ or $\tau$.[8] Let $\mathsf{sv}_+(\boldsymbol{\beta})$ and $\mathsf{sv}_-(\boldsymbol{\beta})$ be any element in $\mathcal{SV}_+(\boldsymbol{\beta})$ and $\mathcal{SV}_-(\boldsymbol{\beta})$, i.e.,

$$\mathsf{sv}_+(\boldsymbol{\beta}) \in \mathcal{SV}_+(\boldsymbol{\beta}), \qquad \mathsf{sv}_-(\boldsymbol{\beta}) \in \mathcal{SV}_-(\boldsymbol{\beta}),$$

which are (the indices of) arbitrary positive and negative support vectors (only depends on $(\boldsymbol{X},\boldsymbol{y})$ and $\boldsymbol{\beta}$). In particular, $\mathsf{sv}_+(\widehat{\boldsymbol{\beta}}) \in \mathcal{SV}_+(\widehat{\boldsymbol{\beta}})$, $\mathsf{sv}_-(\widehat{\boldsymbol{\beta}}) \in \mathcal{SV}_-(\widehat{\boldsymbol{\beta}})$ are support vectors of the max-margin classifier $2\mathbb{1}\{\langle\boldsymbol{x},\widehat{\boldsymbol{\beta}}\rangle + \widehat{\beta}_0 > 0\} - 1$, which aligns with the definition of support vectors in SVM.

The lemma below summarizes some important properties of the max-margin solution Eq. (7) characterized by support vectors, which is a stronger statement than Theorem C.1.

**Lemma C.3.** *For non-degenerate case, let $(\widehat{\boldsymbol{\beta}},\widehat{\beta}_0,\widehat{\kappa})$ be an optimal solution to Eq. (7). Then*

---

[7]For degenerate case ($n_+ = 0$ or $n$), the dataset is considered as linearly separable.

[8]Hence, we can view $\mathcal{SV}_{\pm}(\boldsymbol{\beta})$ as a mapping from $\mathbb{R}^d$ to the power set of $\{i : y_i = \pm 1\}$.

*(a)* $\widehat{\boldsymbol{\beta}}$ *does NOT depend on* $\tau$*, and*

$$(\tau + 1)\widehat{\kappa} = \max_{\boldsymbol{\beta} \in \S^{d-1}} \langle \boldsymbol{x}_{\mathsf{sv}_+(\boldsymbol{\beta})} - \boldsymbol{x}_{\mathsf{sv}_-(\boldsymbol{\beta})}, \boldsymbol{\beta} \rangle = \langle \boldsymbol{x}_{\mathsf{sv}_+(\widehat{\boldsymbol{\beta}})} - \boldsymbol{x}_{\mathsf{sv}_-(\widehat{\boldsymbol{\beta}})}, \widehat{\boldsymbol{\beta}} \rangle. \tag{23}$$

*(b)* $\widehat{\beta}_0$ *depends on* $\tau$ *by*

$$\widehat{\beta}_0 = -\frac{\tau \langle \boldsymbol{x}_{\mathsf{sv}_-(\widehat{\boldsymbol{\beta}})}, \widehat{\boldsymbol{\beta}} \rangle + \langle \boldsymbol{x}_{\mathsf{sv}_+(\widehat{\boldsymbol{\beta}})}, \widehat{\boldsymbol{\beta}} \rangle}{\tau + 1}. \tag{24}$$

*(c)* *If the data are linearly separable, then* $(\widehat{\boldsymbol{\beta}}, \widehat{\beta}_0)$ *must be unique.*

*Proof.* For any feasible solution $(\boldsymbol{\beta}, \beta_0)$ of Eq. (15), we denote the *positive* and *negative margin* of the classifier $\boldsymbol{x} \mapsto 2\mathbb{1}\{\langle \boldsymbol{x}, \boldsymbol{\beta} \rangle + \beta_0 > 0\} - 1$ as

$$\begin{aligned} \kappa_+(\boldsymbol{\beta}, \beta_0) &:= \min_{i:y_i=+1} \widetilde{y}_i(\langle \boldsymbol{x}_i, \boldsymbol{\beta} \rangle + \beta_0) = \tau^{-1}(\langle \boldsymbol{x}_{\mathsf{sv}_+(\boldsymbol{\beta})}, \boldsymbol{\beta} \rangle + \beta_0), \\ \kappa_-(\boldsymbol{\beta}, \beta_0) &:= \min_{i:y_i=-1} \widetilde{y}_i(\langle \boldsymbol{x}_i, \boldsymbol{\beta} \rangle + \beta_0) = -(\langle \boldsymbol{x}_{\mathsf{sv}_-(\boldsymbol{\beta})}, \boldsymbol{\beta} \rangle + \beta_0). \end{aligned} \tag{25}$$

According to Eq. (15), we have

$$\widehat{\boldsymbol{\beta}} = \arg\max_{\boldsymbol{\beta} \in \S^{d-1}} \kappa(\boldsymbol{\beta}, \check{\beta}_0(\boldsymbol{\beta})) = \arg\max_{\boldsymbol{\beta} \in \S^{d-1}} \min_{i \in [n]} \widetilde{y}_i(\langle \boldsymbol{x}_i, \boldsymbol{\beta} \rangle + \check{\beta}_0(\boldsymbol{\beta})),$$

where

$$\begin{aligned} \check{\beta}_0(\boldsymbol{\beta}) &:= \arg\max_{\beta_0 \in \mathbb{R}} \kappa(\boldsymbol{\beta}, \beta_0) = \arg\max_{\beta_0 \in \mathbb{R}} \min_{i \in [n]} \widetilde{y}_i(\langle \boldsymbol{x}_i, \boldsymbol{\beta} \rangle + \beta_0) \\ &= \arg\max_{\beta_0 \in \mathbb{R}} \left\{ \min_{i:y_i=+1} \widetilde{y}_i(\langle \boldsymbol{x}_i, \boldsymbol{\beta} \rangle + \beta_0), \min_{i:y_i=-1} \widetilde{y}_i(\langle \boldsymbol{x}_i, \boldsymbol{\beta} \rangle + \beta_0) \right\} \\ &= \arg\max_{\beta_0 \in \mathbb{R}} \min \left\{ \kappa_+(\boldsymbol{\beta}, \beta_0), \kappa_-(\boldsymbol{\beta}, \beta_0) \right\}. \end{aligned} \tag{26}$$

Here, $\check{\beta}_0(\boldsymbol{\beta})$ can be viewed as the optimal intercept for a linear classifier with slope given by $\boldsymbol{\beta}$.

**(b):** As defined in Eq. (25), note $\min\{\kappa_+(\boldsymbol{\beta}, \beta_0), \kappa_-(\boldsymbol{\beta}, \beta_0)\}$ is a piecewise linear concave function of $\beta_0$. Therefore, $\check{\beta}_0(\boldsymbol{\beta})$ must satisfy the *margin-balancing* condition[9], i.e.,

$$\kappa_+(\boldsymbol{\beta}, \check{\beta}_0(\boldsymbol{\beta})) = \kappa_-(\boldsymbol{\beta}, \check{\beta}_0(\boldsymbol{\beta})) = \kappa(\boldsymbol{\beta}, \check{\beta}_0(\boldsymbol{\beta})). \tag{27}$$

In particular, recall that $\check{\beta}_0(\widehat{\boldsymbol{\beta}}) = \widehat{\beta}_0$, then $\kappa_+(\widehat{\boldsymbol{\beta}}, \widehat{\beta}_0) = \kappa_-(\widehat{\boldsymbol{\beta}}, \widehat{\beta}_0)$. Substitute this back to Eq. (25) deduce

$$\tau^{-1}(\langle \boldsymbol{x}_{\mathsf{sv}_+(\widehat{\boldsymbol{\beta}})}, \widehat{\boldsymbol{\beta}} \rangle + \widehat{\beta}_0) = -(\langle \boldsymbol{x}_{\mathsf{sv}_-(\widehat{\boldsymbol{\beta}})}, \widehat{\boldsymbol{\beta}} \rangle + \widehat{\beta}_0),$$

which uniquely solves the expression for $\widehat{\beta}_0$ in Eq. (24). This concludes the proof of part (b).

**(a):** Next, we show that $\widehat{\boldsymbol{\beta}}$ does not depend on $\tau$. According to Eq. (27) and Eq. (25),

$$\begin{aligned} \widehat{\boldsymbol{\beta}} &= \arg\max_{\boldsymbol{\beta} \in \S^{d-1}} \kappa(\boldsymbol{\beta}, \check{\beta}_0(\boldsymbol{\beta})) \\ &= \arg\max_{\boldsymbol{\beta} \in \S^{d-1}} \frac{\tau \kappa_+(\boldsymbol{\beta}, \check{\beta}_0(\boldsymbol{\beta})) + \kappa_-(\boldsymbol{\beta}, \check{\beta}_0(\boldsymbol{\beta}))}{\tau + 1} \\ &= \arg\max_{\boldsymbol{\beta} \in \S^{d-1}} \frac{\langle \boldsymbol{x}_{\mathsf{sv}_+(\boldsymbol{\beta})}, \boldsymbol{\beta} \rangle - \langle \boldsymbol{x}_{\mathsf{sv}_-(\boldsymbol{\beta})}, \boldsymbol{\beta} \rangle}{\tau + 1} = \arg\max_{\boldsymbol{\beta} \in \S^{d-1}} \langle \boldsymbol{x}_{\mathsf{sv}_+(\boldsymbol{\beta})} - \boldsymbol{x}_{\mathsf{sv}_+(\boldsymbol{\beta})}, \boldsymbol{\beta} \rangle, \end{aligned}$$

where $\langle \boldsymbol{x}_{\mathsf{sv}_+(\boldsymbol{\beta})} - \boldsymbol{x}_{\mathsf{sv}_+(\boldsymbol{\beta})}, \boldsymbol{\beta} \rangle$ only depends on $\boldsymbol{\beta}$ and $(\boldsymbol{X}, \boldsymbol{y})$ by definition. Hence, it deduces

$$\widehat{\kappa} = \kappa(\widehat{\boldsymbol{\beta}}, \check{\beta}_0(\widehat{\boldsymbol{\beta}})) = \frac{\tau \kappa_+(\widehat{\boldsymbol{\beta}}, \check{\beta}_0(\widehat{\boldsymbol{\beta}})) + \kappa_-(\widehat{\boldsymbol{\beta}}, \check{\beta}_0(\widehat{\boldsymbol{\beta}}))}{\tau + 1} = \frac{\langle \boldsymbol{x}_{\mathsf{sv}_+(\widehat{\boldsymbol{\beta}})}, \widehat{\boldsymbol{\beta}} \rangle - \langle \boldsymbol{x}_{\mathsf{sv}_-(\widehat{\boldsymbol{\beta}})}, \widehat{\boldsymbol{\beta}} \rangle}{\tau + 1}.$$

---

[9]As we have seen, the margin-balancing condition holds regardless of the sign of margin. It holds even if the data is not linearly separable.

This concludes the proof of part (a).

**(c):** Since Eq. (175) is a convex optimization problem with objective function $\|\mathbf{w}\|_2^2$, which is strictly convex in $\mathbf{w}$, by equivalence between Eq. (7), (15) and (175), we know that $\widehat{\mathbf{w}}$ and $\widehat{\boldsymbol{\beta}} = \widehat{\mathbf{w}}/\|\widehat{\mathbf{w}}\|_2$ must be unique. And by (a), $\widehat{\beta}_0$ is also unique. This concludes the proof of part (c). $\qquad\square$

Notice that Theorem C.3 will also be used in the proof of Theorem H.4 for the high imbalance regime. Below, we show that Theorem C.1 is a direct consequence of Theorem C.3.

***Proof of Proposition C.1.*** We only show the relation on $\widehat{\kappa}(\tau)$ and $\widehat{\beta}_0(\tau)$, while the other results are simply restatements of Theorem C.3. According to Eq. (23), for any $\tau$, $(\tau+1)\widehat{\kappa}(\tau)$ equals a quantity which does not depend on $\tau$. Plugging in $\tau = 1$, we get $(\tau + 1)\widehat{\kappa}(\tau) = 2\widehat{\kappa}(1)$.

Combining Eq. (23) and (24), we can solve

$$\langle \boldsymbol{x}_{\mathsf{sv}_+(\widehat{\boldsymbol{\beta}})}, \widehat{\boldsymbol{\beta}} \rangle = \tau\widehat{\kappa}(\tau) - \widehat{\beta}_0(\tau), \qquad \langle \boldsymbol{x}_{\mathsf{sv}_-(\widehat{\boldsymbol{\beta}})}, \widehat{\boldsymbol{\beta}} \rangle = -\widehat{\kappa}(\tau) - \widehat{\beta}_0(\tau).$$

Notice the above holds for any $\tau > 0$. Taking $\tau = 1$ and substituting it into Eq. (24), we get

$$\widehat{\beta}_0(\tau) = -\frac{\tau\big(-\widehat{\kappa}(1) - \widehat{\beta}_0(1)\big) + \big(\widehat{\kappa}(1) - \widehat{\beta}_0(1)\big)}{\tau + 1} = \widehat{\beta}_0(1) + \frac{\tau - 1}{\tau + 1}\widehat{\kappa}(1).$$

This completes the proof. $\qquad\square$

***Proof of Proposition C.2.*** Our argument follows the proof of (Soudry et al., 2018, Theorem 2.1). Assume that $\boldsymbol{\beta}^*$ is a limit point of $\widehat{\boldsymbol{\beta}}_\lambda/\|\widehat{\boldsymbol{\beta}}_\lambda\|_2$ as $\lambda \to 0^+$, with $\|\boldsymbol{\beta}^*\|_2 = 1$. The existence of $\boldsymbol{\beta}^*$ is guaranteed by boundedness. Let $\beta_0^* := \limsup_{\lambda\to0^+} \widehat{\beta}_{0,\lambda}/\|\widehat{\boldsymbol{\beta}}_\lambda\|_2$. Now, suppose the max-margin classifier given by $(\widehat{\boldsymbol{\beta}}, \widehat{\beta}_0)$ (with $\|\widehat{\boldsymbol{\beta}}\|_2 = 1$) has a larger margin than $(\boldsymbol{\beta}^*, \beta_0^*)$, that is,

$$\kappa(\boldsymbol{\beta}^*, \beta_0^*) = \min_{i\in[n]} y_i(\langle \boldsymbol{x}_i, \boldsymbol{\beta}^* \rangle + \beta_0^*) < \kappa(\widehat{\boldsymbol{\beta}}, \widehat{\beta}_0) = \min_{i\in[n]} y_i(\langle \boldsymbol{x}_i, \widehat{\boldsymbol{\beta}} \rangle + \widehat{\beta}_0).$$

By continuity of $\kappa(\boldsymbol{\beta}, \beta_0)$, there exists some open neighborhood of $(\boldsymbol{\beta}^*, \beta_0^*)$:

$$\mathcal{N}_{\boldsymbol{\beta}^*, \beta_0^*} := \big\{ \boldsymbol{\beta} \in \mathbb{R}^d, \beta_0 \in R : \|\boldsymbol{\beta}\|_2 = 1, \|\boldsymbol{\beta} - \boldsymbol{\beta}^*\|_2^2 + |\beta_0 - \beta_0^*|^2 < \delta^2 \big\}$$

and an $\varepsilon > 0$, such that

$$\kappa(\boldsymbol{\beta}, \beta_0) = \min_{i\in[n]} y_i(\langle \boldsymbol{x}_i, \boldsymbol{\beta} \rangle + \beta_0) < \kappa(\widehat{\boldsymbol{\beta}}, \widehat{\beta}_0) - \varepsilon, \qquad \forall\, (\boldsymbol{\beta}, \beta_0) \in \mathcal{N}_{\boldsymbol{\beta}^*, \beta_0^*}.$$

Since $\ell$ is rapidly varying, now by (Soudry et al., 2018, Lemma 2.3) we know that there exists some constant $T > 0$ (depends on $\kappa(\widehat{\boldsymbol{\beta}}, \widehat{\beta}_0)$ and $\varepsilon$), such that

$$\sum_{i=1}^n \ell\big(y_i(\langle \boldsymbol{x}_i, t\widehat{\boldsymbol{\beta}} \rangle + t\widehat{\beta}_0)\big) < \sum_{i=1}^n \ell\big(y_i(\langle \boldsymbol{x}_i, t\boldsymbol{\beta} \rangle + t\beta_0)\big), \qquad \forall\, t > T,\ (\boldsymbol{\beta}, \beta_0) \in \mathcal{N}_{\boldsymbol{\beta}^*, \beta_0^*},$$

which implies $(t\widehat{\boldsymbol{\beta}}, t\widehat{\beta}_0)$ has a smaller loss Eq. (18) than $(t\boldsymbol{\beta}, t\beta_0)$. This indicates that $\boldsymbol{\beta}^*$ cannot be a limit point of $\widehat{\boldsymbol{\beta}}_\lambda/\|\widehat{\boldsymbol{\beta}}_\lambda\|_2$, which is a contradiction. Hence we must have $\kappa(\boldsymbol{\beta}^*, \beta_0^*) = \kappa(\widehat{\boldsymbol{\beta}}, \widehat{\beta}_0)$. Replacing $\limsup$ by $\liminf$ in the definition of $\beta_0^*$ gives the same conclusion. Then we complete the proof by noticing the max-margin solution is unique on separable data by Theorem C.3(c). $\quad\square$

## D  ADDITIONAL DETAILS IN THEORETICAL RESULTS

### D.1  PRECISE ASYMPTOTICS OF EMPIRICAL LOGIT DISTRIBUTION

In this section, we present additional details on the asymptotics of empirical logit distribution introduced in Section 2. Recall that data $\{(\boldsymbol{x}_i, y_i)\}_{i=1}^n$ are i.i.d. generated from a 2-GMM Eq. (1), i.e., $\boldsymbol{x}_i \,|\, y_i \sim \mathsf{N}(y_i\boldsymbol{\mu}, \mathbf{I}_d)$, with label distribution $P_y : \mathbb{P}(y_i = +1) = 1 - \mathbb{P}(y_i = -1) = \pi \in (0, \frac{1}{2}]$. We

consider proportional asymptotics where $n, d \to \infty$ and $n/d \to \delta$ with $\delta \in (0, \infty)$. Based on relations between $\pi, \boldsymbol{\mu}, \delta$, we will consider linearly separable data (fitted by SVM) and non-separable data (fitted by logistic regression) separately.

We define the following functions $\delta^* : \mathbb{R} \to \mathbb{R}_{\geq 0}$ and $H_\kappa : [-1, 1] \times \mathbb{R} \to \mathbb{R}_{\geq 0}$ that are related to the critical threshold of data separability:

$$\delta^*(\kappa) := \max_{\rho \in [-1,1], \beta_0 \in \mathbb{R}} H_\kappa(\rho, \beta_0), \qquad H_\kappa(\rho, \beta_0) := \frac{1 - \rho^2}{\mathbb{E}\left[\left(s(Y)\kappa - \rho \|\boldsymbol{\mu}\|_2 + G - Y\beta_0\right)_+^2\right]}, \tag{28}$$

where $(Y, G) \sim P_y \times \mathsf{N}(0, 1)$ and

$$s(y) := \begin{cases} \tau, & \text{if } y = +1, \\ 1, & \text{if } y = -1. \end{cases} \tag{29}$$

We will show in Theorem D.1 that the relationship between $\delta$ and $\delta^*(0)$ determines separability, where $\delta^*(0)$ does not depend on $\tau$ by definition.

We summarize the asymptotics of logit distribution for both separable and non-separable case in Table 4, which is the main contribution of our theoretical results (Theorem D.1 and D.3).

Table 4: Comparison of logit distributions on separable and non-separable data ($\tau = 1$).

| | **limiting ELD** $(\nu_*)$ | **cause for overfitting** $(\xi^*)$ |
|---|---|---|
| separable data | $\mathtt{Law}\left(Y, Y \max\{\kappa^*, \mathtt{LOGITS}\}\right)$ | $R^* \sqrt{1 - \rho^{*2}}\xi^* = (\kappa^* - \mathtt{LOGITS})_+$ |
| non-separable data | $\mathtt{Law}\left(Y, Y \, \mathtt{prox}_{\lambda^* \ell}(\mathtt{LOGITS})\right)$ | $R^* \sqrt{1 - \rho^{*2}}\xi^* = -\lambda^* \nabla \mathsf{e}_{\lambda^* \ell}(\mathtt{LOGITS})$ |
| **limiting TLD** $(\nu_*^{\text{test}})$ | $\mathtt{Law}\left(Y, Y \cdot \mathtt{LOGITS}\right)$ | |
| | $\mathtt{LOGITS} := \rho^* \|\boldsymbol{\mu}\|_2 R^* + R^* G + Y\beta_0^*$ | $(R^* := 1$ in separable case$)$ |

### D.1.1 SEPARABLE DATA

For linearly separable data, recall the margin-rebalanced SVM in Eq. (7) and (15). The following theorem summarizes the precise asymptotics of SVM under arbitrary $\tau$, including the limits of parameters, margin, and logit distribution. The proofs are deferred to the appendices.

Recall that data $\{(\boldsymbol{x}_i, y_i)\}_{i=1}^n$ are generated from 2-GMM with fixed parameters $\boldsymbol{\mu} \in \mathbb{R}^d$, $\pi \in (0, \frac{1}{2}]$. Let $(\widehat{\boldsymbol{\beta}}_n, \widehat{\beta}_{0,n})$ be an optimal solution to the margin-rebalanced SVM Eq. (15), and $\widehat{\kappa}_n = \min_{i \in [n]} \widetilde{y}_i(\langle \boldsymbol{x}_i, \widehat{\boldsymbol{\beta}} \rangle + \widehat{\beta}_0)$ be the maximum margin. Recall the cosine angle $\widehat{\rho}_n := \widehat{\rho}$ between $\boldsymbol{\mu}$ and $\widehat{\boldsymbol{\beta}}_n$ defined as

$$\widehat{\rho} := \left\langle \frac{\widehat{\boldsymbol{\beta}}}{\|\widehat{\boldsymbol{\beta}}\|}, \frac{\boldsymbol{\mu}}{\|\boldsymbol{\mu}\|} \right\rangle. \tag{30}$$

Let $\delta^*(\kappa)$ be defined as per Eq. (28), and $\rho^*, \beta_0^*, \kappa^*, \xi^*$ be a solution to the variational problem

$$\begin{aligned} \max_{\rho \in [-1,1], \beta_0 \in \mathbb{R}, \kappa \in \mathbb{R}, \xi \in \mathcal{L}^2} \quad & \kappa, \\ \text{subject to} \quad & \rho \|\boldsymbol{\mu}\|_2 + G + Y\beta_0 + \sqrt{1 - \rho^2}\xi \geq s(Y)\kappa, \qquad \mathbb{E}[\xi^2] \leq 1/\delta. \end{aligned} \tag{31}$$

where $\mathcal{L}^2$ is the space of all square integrable random variables in $(\Omega, \mathcal{F}, \mathbb{P})$, and $(Y, G) \sim P_y \times \mathsf{N}(0, 1)$. We define

$$\nu_* := \mathtt{Law}\left(Y, Y \max\{s(Y)\kappa^*, \rho^* \|\boldsymbol{\mu}\|_2 + G + Y\beta_0^*\}\right),$$
$$\nu_*^{\text{test}} := \mathtt{Law}\left(Y, Y(\rho^* \|\boldsymbol{\mu}\|_2 + G + Y\beta_0^*)\right),$$

which we will prove to be the limiting ELD and TLD respectively.

**Theorem D.1** (Separable data). *Assume $n, d \to \infty$ with $n/d \to \delta \in (0, \infty)$. Fix $\tau \in (0, \infty)$.*

(a) (**Phase transition**) *With probability tending to one, the data is linearly separable if $\delta < \delta^*(0)$ and is not linearly separable if $\delta > \delta^*(0)$.*

(b) (**Variational problem**) *In the separable regime $\delta < \delta^*(0)$, $(\rho^*, \beta_0^*, \kappa^*, \xi^*)$ is the unique solution to Eq. (31) with $\rho^* \in (0,1)$ (not depend on $\tau$), $\kappa^* > 0$, and the random variable $\xi^*$ satisfies (a.s.)*

$$\sqrt{1 - \rho^{*2}}\xi^* = \left(s(Y)\kappa^* - \rho^* \|\boldsymbol{\mu}\|_2 - G - Y\beta_0^*)\right)_+. \tag{32}$$

*Moreover, $(\rho^*, \beta_0^*, \kappa^*)$ is also the unique solution to*

$$\begin{array}{cl} \underset{\rho \in [-1,1], \beta_0 \in \mathbb{R}, \kappa \in \mathbb{R}}{\text{maximize}} & \kappa, \\ \text{subject to} & H_\kappa(\rho, \beta_0) \geq \delta \end{array} \tag{33}$$

*and $\kappa^* = \sup\left\{\kappa \in \mathbb{R} : \delta^*(\kappa) \geq \delta\right\}$.*

(c) (**Margin convergence**) *In the separable regime $\delta < \delta^*(0)$,*

$$\widehat{\kappa}_n \xrightarrow{\mathcal{L}^2} \kappa^*.$$

*In the non-separable regime $\delta > \delta^*(0)$ we have negative margin, i.e., with probability tending to one, for some $\overline{\kappa} > 0$,*

$$\max_{\substack{\|\boldsymbol{\beta}\|_2 = 1 \\ \beta_0 \in \mathbb{R}}} \min_{i \in [n]} \widetilde{y}_i(\langle \boldsymbol{x}_i, \boldsymbol{\beta} \rangle + \beta_0) \leq -\overline{\kappa}.$$

(d) (**Parameter convergence**) *In the separable regime $\delta < \delta^*(0)$,*

$$\widehat{\rho}_n \xrightarrow{\mathrm{P}} \rho^*, \qquad \widehat{\beta}_{0,n} \xrightarrow{\mathrm{P}} \beta_0^*.$$

(e) (**Asymptotic errors**) *Recall the minority and majority test prediction errors , $\mathrm{Err}_{+,n}$ and $\mathrm{Err}_{-,n}$ respectively, of the max-margin classifier defined in Eq. (4) (writing subscript $n$ for clarity). Then in the separable regime $\delta < \delta^*(0)$,*

$$\mathrm{Err}_{+,n} \to \Phi\left(-\rho^* \|\boldsymbol{\mu}\|_2 - \beta_0^*\right), \qquad \mathrm{Err}_{-,n} \to \Phi\left(-\rho^* \|\boldsymbol{\mu}\|_2 + \beta_0^*\right).$$

(f) (**ELD/TLD convergence**) *Recall the ELD $\widehat{\nu}_n$ and TLD $\widehat{\nu}_n^{\text{test}}$ defined as per Definition 2.2, where $\widehat{f}(\boldsymbol{x}) = \langle \boldsymbol{x}, \widehat{\boldsymbol{\beta}}_n \rangle + \widehat{\beta}_{0,n}$. Then in the separable regime $\delta < \delta^*(0)$ we have logit convergence for both training and test data, i.e.,*

$$W_2\left(\widehat{\nu}_n, \nu_*\right) \xrightarrow{\mathrm{P}} 0, \qquad \widehat{\nu}_n^{\text{test}} \xrightarrow{w} \nu_*^{\text{test}}.$$

**Remark D.1.** *By taking $\tau = 1$, the ELD convergence $W_2(\widehat{\nu}_n, \nu_*) \xrightarrow{\mathrm{P}} 0$ in Theorem 2.1(b) is a consequence of Theorem D.1(f), and the TLD convergence $\widehat{\nu}_n^{\text{test}} \xrightarrow{w} \nu_*^{\text{test}}$ is a corollary of Theorem D.1(d).*

As discussed in Section 2, random variable $\xi^*$ and the nonlinear transformation $\mathtt{T}^*(x) = \max\{x, \kappa^*\}$ therein characterize the effect of overfitting on logits. The following result provides an optimal transport perspective of this overfitting effect. For ease of description, we reformulate $\nu_*$ and $\nu_*^{\text{test}}$ in terms of the following one-dimensional measures

$$\mathcal{L}_* := \mathtt{Law}\left(\max\{\kappa^*, \rho^* \|\boldsymbol{\mu}\|_2 + G + Y\beta_0^*\}\right), \qquad \mathcal{L}_*^{\text{test}} := \mathtt{Law}\left(\rho^* \|\boldsymbol{\mu}\|_2 + G + Y\beta_0^*\right).$$

**Proposition D.2** (Optimal transport map). *$\mathtt{T}^*(x) = \max\{\kappa^*, x\}$ is the unique optimal transport map from $\mathcal{L}_*^{\text{test}}$ to $\mathcal{L}_*$ under the cost function $c(x, y) = h(x - y)$ for any strictly convex $h : \mathbb{R}^2 \to \mathbb{R}_{\geq 0}$. That is,*

$$\mathtt{T}^* = \underset{\mathtt{T}:\mathbb{R}\to\mathbb{R}}{\arg\min}\left\{\int_{\mathbb{R}} c\left(x, \mathtt{T}(x)\right)\mathrm{d}\mathcal{L}_*^{\text{test}}(x) \,\bigg|\, \mathtt{T}_\sharp \mathcal{L}_*^{\text{test}} = \mathcal{L}_*\right\},$$

*where $\mathtt{T}_\sharp$ is the pushforward operator.*

### D.1.2 NON-SEPARABLE DATA

For non-separable data, SVM yields a trivial solution $\boldsymbol{\beta} = \mathbf{0}, \beta_0 = 0$. A typical approach to fitting a classifier is to solve regression problem Eq. (2a). Similar to the margin-rebalanced SVM Eq. (15), we can also incorporate $\tau$ into the objective function by substituting $y_i$ for $\widetilde{y}_i = y_i/s(y_i)$, that is,

$$\min_{\boldsymbol{\beta} \in \mathbb{R}^d, \beta_0 \in \mathbb{R}} \qquad \frac{1}{n} \sum_{i=1}^{n} \ell\big(\widetilde{y}_i(\langle \boldsymbol{x}_i, \boldsymbol{\beta}\rangle + \beta_0)\big), \tag{34}$$

where $\ell : \mathbb{R} \to \mathbb{R}_{\geq 0}$ is the loss function. We consider a more general form than logistic regression. We say that $\ell$ is *pseudo-Lipschitz* if there exists a constant $L > 0$ such that, for all $x, y \in \mathbb{R}$,

$$|\ell(x) - \ell(y)| \leq L\left(1 + |x| + |y|\right)|x - y|.$$

This condition is satisfied, for instance, by the widely used logistic loss $\ell(t) = \log(1 + e^{-t})$. As the counterpart of Theorem D.1 in the non-separable regime, the following theorem summarizes the precise asymptotics of regression Eq. (34), including the limits of parameters and logit distribution.

We consider the same 2-GMM setting as Section D.1.1. For any non-increasing, strictly convex, pseudo-Lipschitz, twice differentiable function $\ell : \mathbb{R} \to \mathbb{R}_{\geq 0}$, let $(\widehat{\boldsymbol{\beta}}_n, \widehat{\beta}_{0,n})$ be the optimal solution to regression Eq. (34). Recall $\widehat{\rho}_n := \widehat{\rho}$ defined in Eq. (30) and $\delta^*(\kappa)$ defined in Eq. (28). Let $\rho^*, R^*, \beta_0^*, \xi^*$ be a solution to the variational problem

$$\begin{aligned}
\operatorname*{minimize}_{\rho \in [-1,1], R \geq 0, \beta_0 \in \mathbb{R}, \xi \in \mathcal{L}^2} \quad & \mathbb{E}\left[\ell\left(\frac{\rho \|\boldsymbol{\mu}\|_2 R + RG + Y\beta_0 + R\sqrt{1-\rho^2}\xi}{s(Y)}\right)\right], \\
\text{subject to} \quad & \mathbb{E}\left[\xi^2\right] \leq 1/\delta.
\end{aligned} \tag{35}$$

where $(Y, G) \sim P_y \times \mathsf{N}(0, 1)$. We define

$$\nu_* := \mathtt{Law}\left(Y, Ys(Y)\, \mathsf{prox}_{\frac{\lambda^*\ell}{s(Y)}}\left(\frac{\rho^*\|\boldsymbol{\mu}\|_2 R^* + R^*G + Y\beta_0^*}{s(Y)}\right)\right),$$

$$\nu_*^{\text{test}} := \mathtt{Law}\left(Y, Y(R^*\rho^*\|\boldsymbol{\mu}\| + R^*G + Y\beta_0^*)\right),$$

aiming to show they are the limiting ELD and TLD respectively.

**Theorem D.3** (Non-separable data). *Consider the same 2-GMM and proportional settings $n/d \to \delta$ as in Theorem D.1.*

(a) *(Variational problem) In the non-separable regime $\delta > \delta^*(0)$, $(\rho^*, R^*, \beta_0^*, \xi^*)$ is the unique solution to Eq. (35) with $\rho^* \in (0, 1)$, $R^* \in (0, \infty)$, and the random variable $\xi^*$ satisfies (a.s.)*

$$R^*\sqrt{1-\rho^{*2}}\xi^* = -\lambda^*\ell'\left(\mathsf{prox}_{\frac{\lambda^*\ell}{s(Y)}}\left(\frac{\rho^*\|\boldsymbol{\mu}\|_2 R^* + R^*G + Y\beta_0^*}{s(Y)}\right)\right), \tag{36}$$

*where $\lambda^* \in (0, \infty)$ is the unique constant such that $\mathbb{E}[\xi^2] = 1/\delta$. Moreover, $(\rho^*, R^*, \beta_0^*, \lambda^*)$ is also the unique solution to the following system of equations*

$$-\frac{\tau R\rho}{2\pi\lambda\delta\|\boldsymbol{\mu}\|_2} = \mathbb{E}\left[\ell'\left(\mathsf{prox}_{\frac{\lambda\ell}{\tau}}\left(\frac{\rho\|\boldsymbol{\mu}\|_2 R + RG + \beta_0}{\tau}\right)\right)\right],$$

$$-\frac{R\rho}{2(1-\pi)\lambda\delta\|\boldsymbol{\mu}\|_2} = \mathbb{E}\left[\ell'\left(\mathsf{prox}_{\lambda\ell}(\rho\|\boldsymbol{\mu}\|_2 R + RG - \beta_0)\right)\right],$$

$$\frac{1}{\lambda\delta} = \mathbb{E}\left[\frac{1}{s(Y)} \cdot \frac{\ell''\left(\mathsf{prox}_{\frac{\lambda\ell}{s(Y)}}\left(\frac{\rho\|\boldsymbol{\mu}\|_2 R + RG + Y\beta_0}{s(Y)}\right)\right)}{s(Y) + \lambda\ell''\left(\mathsf{prox}_{\frac{\lambda\ell}{s(Y)}}\left(\frac{\rho\|\boldsymbol{\mu}\|_2 R + RG + Y\beta_0}{s(Y)}\right)\right)}\right],$$

$$\frac{R^2(1-\rho^2)}{\lambda^2\delta} = \mathbb{E}\left[\left(\frac{1}{s(Y)} \cdot \ell'\left(\mathsf{prox}_{\frac{\lambda\ell}{s(Y)}}\left(\frac{\rho\|\boldsymbol{\mu}\|_2 R + RG + Y\beta_0}{s(Y)}\right)\right)\right)^2\right].$$

(b) *(Parameter convergence) In the non-separable regime $\delta > \delta^*(0)$, as $n \to \infty$,*

$$\|\widehat{\boldsymbol{\beta}}_n\|_2 \xrightarrow{\mathrm{P}} R^*, \qquad \widehat{\rho}_n \xrightarrow{\mathrm{P}} \rho^*, \qquad \widehat{\beta}_{0,n} \xrightarrow{\mathrm{P}} \beta_0^*.$$

*(c)* ***(Asymptotic errors)*** *Recall the prediction errors defined as per Eq. (4). Then in the non-separable regime $\delta > \delta^*(0)$, as $n \to \infty$,*

$$\mathrm{Err}_{+,n} \to \Phi\left(-\rho^* \|\boldsymbol{\mu}\|_2 - \frac{\beta_0^*}{R^*}\right), \qquad \mathrm{Err}_{-,n} \to \Phi\left(-\rho^* \|\boldsymbol{\mu}\|_2 + \frac{\beta_0^*}{R^*}\right).$$

*(d)* ***(ELD/TLD convergence)*** *Recall the ELD $\widehat{\nu}_n$ and TLD $\widehat{\nu}_n^{\mathrm{test}}$ defined as per Definition 2.2, where $\widehat{f}(\boldsymbol{x}) = \langle \boldsymbol{x}, \widehat{\boldsymbol{\beta}}_n \rangle + \widehat{\beta}_{0,n}$. Then in the non-separable regime $\delta > \delta^*(0)$ we have logit convergence for both training and test data, i.e., as $n \to \infty$,*

$$W_2\big(\widehat{\nu}_n, \nu_*\big) \xrightarrow{\mathrm{p}} 0, \qquad \widehat{\nu}_n^{\mathrm{test}} \xrightarrow{w} \nu_*^{\mathrm{test}}.$$

**Remark D.2.** *Compared to the separable regime, the random variable $\xi$ in the non-separable regime Eq. (35) can also be interpreted as the cause for overfitting, but its distortion effect on ELD is not truncation. When $\tau = 1$, by Eq. (35), (36), the following holds for a "typical" training point:*

$$\begin{aligned}
y_i(\langle \boldsymbol{x}_i, \widehat{\boldsymbol{\beta}}_n \rangle + \widehat{\beta}_{0,n}) &\approx \rho^* \|\boldsymbol{\mu}\|_2 R^* + R^* G + Y\beta_0^* + R^* \sqrt{1 - \rho^{*2}} \xi^* \\
&= \rho^* \|\boldsymbol{\mu}\|_2 R^* + R^* G + Y\beta_0^* - \lambda^* \ell'\big(\mathrm{prox}_{\lambda^* \ell}(\rho^* \|\boldsymbol{\mu}\|_2 R^* + R^* G + Y\beta_0^*)\big) \\
&= \mathrm{prox}_{\lambda^* \ell}(\rho^* \|\boldsymbol{\mu}\|_2 R^* + R^* G + Y\beta_0^*),
\end{aligned}$$

*where the equalities come from Theorem J.5. Hence, the ELD in the non-separable regime is the TLD under nonlinear shrinkage due to the proximal operator of loss function $\ell$.*

## D.2 ANALYSIS OF MARGIN REBALANCING FOR SEPARABLE DATA

In this section, we show how margin rebalancing improves the test accuracies on imbalanced dataset by choosing the hyperparameter $\tau$ in Eq. (7) appropriately.

### D.2.1 PROPORTIONAL REGIME

Consider the same 2-GMM and proportional settings in Section D.1.1 on linearly separable dataset ($\delta < \delta^*(0)$). According to Theorem D.1(e), the asymptotic minority and majority test errors are

$$\mathrm{Err}_+^* := \Phi\left(-\rho^* \|\boldsymbol{\mu}\|_2 - \beta_0^*\right), \qquad \mathrm{Err}_-^* := \Phi\left(-\rho^* \|\boldsymbol{\mu}\|_2 + \beta_0^*\right). \tag{37}$$

For the purpose of imbalanced classification, we define the *asymptotic balanced error* as

$$\mathrm{Err}_{\mathrm{b}}^* := \frac{1}{2}\mathrm{Err}_+^* + \frac{1}{2}\mathrm{Err}_-^*.$$

**Monotonicity analysis.** We first provide some monotone results for test errors, which support our empirical observations in Section 3.

**Proposition D.4.** $\mathrm{Err}_+^*$ *is a decreasing function of $\pi \in (0, \frac{1}{2})$, $\|\boldsymbol{\mu}\|_2$, and $\delta$ when $\tau = 1$.*

However, the majority error $\mathrm{Err}_-^*$ and balanced error $\mathrm{Err}_{\mathrm{b}}^*$ are not necessarily monotone under arbitrary $\tau$. Thus, we will focus on the monotonicity of these test errors when $\tau$ is chosen to be optimal.

According to Fig. 8, by taking $\tau > 1$, we can improve the minority accuracy at the cost of harming majority accuracy. The opposite effects of $\tau$ on $\mathrm{Err}_+^*$ and $\mathrm{Err}_-^*$ are summarized in the following result.

**Proposition D.5.** $\mathrm{Err}_+^*$ *is decreasing in $\tau \in (0, \infty)$, and $\mathrm{Err}_-^*$ is increasing in $\tau \in (0, \infty)$.*

**Choosing the optimal $\tau$.** A natural idea for margin rebalancing is to choose $\tau$ such that the balanced error $\mathrm{Err}_{\mathrm{b}}^*$ is minimized.

**Proposition D.6** (Optimal $\tau$). *Let $\tau^{\mathrm{opt}}$ be the optimal margin ratio $\tau$ defined in Theorem 3.1. Denote $g_1(x) := \mathbb{E}\left[(G + x)_+\right]$ where $G \sim \mathsf{N}(0, 1)$. Then $\tau^{\mathrm{opt}}$ has the explicit expression*

$$\tau^{\mathrm{opt}} = \frac{g_1^{-1}\left(\dfrac{\rho^*}{2\pi \|\boldsymbol{\mu}\|_2 \delta}\right) + \rho^* \|\boldsymbol{\mu}\|_2}{g_1^{-1}\left(\dfrac{\rho^*}{2(1 - \pi) \|\boldsymbol{\mu}\|_2 \delta}\right) + \rho^* \|\boldsymbol{\mu}\|_2}. \tag{38}$$

**Remark D.3.** *The optimal choice of $\tau$ has a complicated dependence on $\pi$. However, we note that the numerator scales as $\tau^{\mathrm{opt}} \asymp \sqrt{1/\pi}$ for small $\pi$ and fixed $\|\boldsymbol{\mu}\|_2$ and $\delta$, which is consistent with the choice of $\tau$ in importance tempering (Lu et al., 2022). In (Cao et al., 2019), $\tau$ is suggested to scale with $\pi^{-1/4}$, however it was proved in (Kini et al., 2021) that their algorithm won't converge to the solution with the desired $\tau$.*

*It is worth noticing that in the near-degenerate cases where $\pi$ is very small or $\|\boldsymbol{\mu}\|_2$, $\delta$ are very large, then $\rho^*$ is close to $0$ and the denominator can be negative, leading to $\tau^{\mathrm{opt}} < 0$. While our theory (Theorem D.7, Theorem D.9) is still valid when we allow potential negative $\tau$, it is rarely used in practice. See Section G.2 for a further discussion. The near-degenerate cases (small $\pi$, large $\|\boldsymbol{\mu}\|_2$ or $\delta$) are better addressed under the high imbalance regime, as we analyze in the next subsection.*

The minority/majority/balanced errors all equal $\Phi(-\rho^* \|\boldsymbol{\mu}\|_2)$ when $\tau = \tau^{\mathrm{opt}}$. We can also obtain the monotonicity of test errors after margin rebalancing.

**Proposition D.7.** *When $\tau = \tau^{\mathrm{opt}} > 0$, all the test errors $\mathrm{Err}^*_+$, $\mathrm{Err}^*_-$, $\mathrm{Err}^*_{\mathrm{b}}$ are decreasing functions of $\pi \in (0, 1/2)$ (imbalance ratio), $\delta$ (aspect ratio), and $\|\boldsymbol{\mu}\|_2$ (signal strength).*

### D.2.2 HIGH IMBALANCE REGIME

Different from the proportional regime considered in Section D.1 and D.2.1, here we focus on a high-imbalanced scenario where $\pi$ is small, $\|\boldsymbol{\mu}\|_2$ is large, and $n$ grows much faster than $d$. In this regime, we can even extend the feature distribution beyond Gaussian, and generalize the 2-GMM settings.

**Definition D.1** (High imbalance). *We say a dataset $\{(\boldsymbol{x}_i, y_i)\}_{i=1}^n$ is i.i.d. generated from a two-component sub-gaussian mixture model (2-subGMM) if for any $i \in [n]$,*

i. *Label distribution:* $\mathbb{P}(y_i = +1) = 1 - \mathbb{P}(y_i = -1) = \pi$,

ii. *Feature distribution:* $\boldsymbol{x}_i = y_i \boldsymbol{\mu} + \boldsymbol{z}_i$, *where $\boldsymbol{z}_i$ has independent coordinates with uniformly bounded sub-gaussian norms. Namely, each coordinate $z_{ij}$ of $\boldsymbol{z}_i$ satisfies $\mathbb{E}[z_{ij}] = 0$, $\mathrm{Var}(z_{ij}) = 1$, and $\|z_{ij}\|_{\psi_2} := \inf\{K > 0 : \mathbb{E}[\exp(X^2/K^2)] \leq 2\} \leq C$ for all $j \in [d]$, where $C$ is an absolute constant.*

*For any constants $a, b, c > 0$, we say a 2-subGMM is $(a, b, c)$-imbalanced if Eq. (8) holds.*

**Remark D.4.** *Parameters $a$, $\frac{b}{2}$, and $c$ each specifies the degenerate rate of imbalance ratio $\pi$, and the growth rate of signal strength $\|\boldsymbol{\mu}\|_2$, aspect ratio $n/d$. We usually require $a < c + 1$ to make sure the minority class sample size $n_+ := \pi n = d^{c-a+1} \to \infty$ does not degenerate.*

Our goal is to study the performance of margin-rebalanced SVM Eq. (7) in this high imbalance regime, asymptotically as $d \to \infty$. Therefore, we allow $\tau = \tau_d$ depends on dimension $d$ and care about what order of $\tau_d$ would make the test errors vanish. We summarize our findings in the following theorem, which is consistent with the empirical observations in Fig. 5 and extends Theorem 3.2 to the case of imbalanced 2-subGMM.

**Theorem D.8** (Phase transition in high imbalance regime). *Consider the high imbalance regime where the training data is i.i.d. generated from an $(a, b, c)$-imbalanced 2-subGMM. Suppose that $a - c < 1$. A margin-rebalanced SVM is trained, with test errors calculated according to Eq. (4). Then as $d \to \infty$, the conclusions of the three phases in Theorem 3.2 still hold.*

### D.3 CONSEQUENCES FOR CONFIDENCE ESTIMATION AND CALIBRATION

Recall the definition of confidence of the max-margin classifier $\widehat{p}(\boldsymbol{x}) := \sigma(\widehat{f}(\boldsymbol{x})) = \sigma(\langle \boldsymbol{x}, \widehat{\boldsymbol{\beta}} \rangle + \widehat{\beta}_0)$ in Section 4. Note that $\widehat{p}(\boldsymbol{x})$ and $1 - \widehat{p}(\boldsymbol{x})$ are the predicted probabilities of $\boldsymbol{x}$ for the minority class ($y = +1$) and the majority class ($y = -1$) respectively.

It is worth noticing that the confidence is sensitive to scales, i.e., $\sigma(t) \neq \sigma(ct)$ if $c \neq 1$, despite the fact that rescaling yields the same label prediction and thus does not affect accuracy. While small models tend to be calibrated, especially when parameter estimation is consistent, larger models such as DNNs are known to suffer from poor calibration (Guo et al., 2017). A simple theoretical explanation is that in a DNN, the last layer (usually a logistic regression) $\boldsymbol{x} \mapsto \sigma(\langle \boldsymbol{x}, \widehat{\boldsymbol{\beta}} \rangle + \widehat{\beta}_0)$

trained by gradient descent on separable features often results in a very large $\|\widehat{\boldsymbol{\beta}}\|_2$ (as mentioned in Section C.2), thereby inflating the predicted probabilities. Here we focus on the common form of SVM (2b) where normalization $\|\widehat{\boldsymbol{\beta}}\|_2 = 1$ is applied.

Some probabilities regarding the confidence are as follows.

1. **Max-margin confidence.** The confidence of the max-margin classifier is

$$\widehat{p}(\boldsymbol{x}) := \sigma\big(\widehat{f}(\boldsymbol{x})\big) = \sigma\big(\langle \boldsymbol{x}, \widehat{\boldsymbol{\beta}} \rangle + \widehat{\beta}_0\big).$$

2. **Bayes optimal probability.** The true conditional probability is

$$p^*(\boldsymbol{x}) := \mathbb{P}(y = 1 \mid \boldsymbol{x}).$$

3. **True posterior probability.** The probability conditioning on max-margin confidence is

$$\widehat{p}_0(\boldsymbol{x}) := \mathbb{P}\big(y = 1 \mid \widehat{p}(\boldsymbol{x})\big).$$

Note that $p^*(\boldsymbol{x})$ is the confidence of the Bayes classifier $y^*(\boldsymbol{x}) := 2\mathbb{1}\{\langle \boldsymbol{x}, 2\boldsymbol{\mu} \rangle + \log \frac{\pi}{1-\pi} > 0\} - 1$.

Recall the definition of calibration Eq. (10) and some miscalibration metrics Eqs. (11)—(13) introduced in Section 4. We offer some further explanations for them.

- **Calibration error:** The $\mathcal{L}^2$ distance between confidence and posteriori, which is the most commonly used metric.

$$\mathrm{CalErr}(\widehat{p}) := \mathbb{E}\left[\big(\widehat{p}(\boldsymbol{x}) - \widehat{p}_0(\boldsymbol{x})\big)^2\right]$$

- **Mean squared error (MSE):** Also known as the Brier score, subject to a calibration budget (Brier, 1950; Gneiting et al., 2007).

$$\mathrm{MSE}(\widehat{p}) := \mathbb{E}\left[\big(\mathbb{1}\{y = 1\} - \widehat{p}(\boldsymbol{x})\big)^2\right]$$

It can be shown that MSE has the following decomposition

$$\mathrm{MSE}(\widehat{p}) \;=\; \underbrace{\mathrm{Var}\big[\mathbb{1}\{y = 1\}\big]}_{\text{irreducible}} + \underbrace{\mathrm{CalErr}(\widehat{p})}_{\text{lack of calibration}} - \underbrace{\mathrm{Var}\big[\widehat{p}_0(\boldsymbol{x})\big]}_{\text{sharpness/resolution}} \;.$$

Calibration error itself does not guarantee a useful predictor. Sharpness, also known as resolution (Murphy, 1973; Kuleshov & Liang, 2015), is another desired property which measures the variance in the response $y$ explained by the probabilistic prediction $\widehat{p}(\boldsymbol{x})$. Hence, a small MSE suggests a classifier to be calibrated with high sharpness.

Note that $\mathrm{Var}[\mathbb{1}\{y = 1\}] = \pi(1 - \pi)$ is an intrinsic quantity unrelated to $\widehat{f}$. When study the effect of $\pi$ on model calibration, we may discard the irreducible variance term and define a modified MSE as

$$\mathrm{mMSE}(\widehat{p}) := \mathrm{CalErr}(\widehat{p}) - \mathrm{Var}\big[\widehat{p}_0(\boldsymbol{x})\big].$$

- **Confidence estimation error:** The $\mathcal{L}^2$ distance between confidence and Bayes optimum.

$$\mathrm{ConfErr}(\widehat{p}) := \mathbb{E}\left[\big(\widehat{p}(\boldsymbol{x}) - p^*(\boldsymbol{x})\big)^2\right].$$

It has the following relation with MSE:

$$\mathrm{MSE}(\widehat{p}) = \mathbb{E}\left[p^*(\boldsymbol{x})\big(1 - p^*(\boldsymbol{x})\big)\right] + \mathrm{ConfErr}(\widehat{p}), \tag{39}$$

where the first term is intrinsic, which only depends on $\pi$ and $\|\boldsymbol{\mu}\|_2$.

The asymptotics of these metrics, and some monotone effect of model parameters $\pi \in (0, \frac{1}{2}]$, $\|\boldsymbol{\mu}\|_2$, $\delta$ on them, are summarized in the following proposition.

**Proposition D.9** (Confidence estimation and calibration). *Consider 2-GMM and the proportional settings in Section D.1.1 on linearly separable dataset ($\delta < \delta^*(0)$).*

(a) *Let* $(\rho^*, \beta_0^*)$ *be defined as per Theorem D.1, and* $(Y, G) \sim P_y \times \mathsf{N}(0, 1)$. *Denote*

$$\mathrm{MSE}^* := \mathbb{E}\left[\sigma\big(-\rho^* \|\boldsymbol{\mu}\|_2 - \beta_0^* Y + G\big)^2\right], \qquad \mathrm{mMSE}^* = \mathrm{MSE}^* - \pi(1 - \pi),$$

$$\mathrm{CalErr}^* := \mathbb{E}\left[\left(\sigma\left(2\rho^* \|\boldsymbol{\mu}\|_2 \left(\rho^* \|\boldsymbol{\mu}\|_2 Y + G\right) + \log\frac{\pi}{1 - \pi}\right) - \sigma\big(\rho^* \|\boldsymbol{\mu}\|_2 Y + G + \beta_0^*\big)\right)^2\right],$$

$$V_{y|\boldsymbol{x}}^* := \mathbb{E}\left[\sigma\left(-2\|\boldsymbol{\mu}\|_2 \left(\|\boldsymbol{\mu}\|_2 + G\right) - \log\frac{\pi}{1 - \pi}Y\right)^2\right], \qquad \mathrm{ConfErr}^* = \mathrm{MSE}^* - V_{y|\boldsymbol{x}}^*.$$

*then*

$$\lim_{n \to \infty} \mathrm{MSE}(\widehat{p}) = \mathrm{MSE}^*, \qquad \lim_{n \to \infty} \mathrm{CalErr}(\widehat{p}) = \mathrm{CalErr}^*,$$
$$\lim_{n \to \infty} \mathrm{mMSE}(\widehat{p}) = \mathrm{mMSE}^*, \qquad \lim_{n \to \infty} \mathrm{ConfErr}(\widehat{p}) = \mathrm{ConfErr}^*.$$

(b) *When* $\tau = \tau^{\mathrm{opt}} > 0$,

- $\mathrm{MSE}^*$ *and* $\mathrm{mMSE}^*$ *are decreasing functions of* $\pi \in (0, \frac{1}{2})$, $\|\boldsymbol{\mu}\|_2$, $\delta$.

- $\mathrm{ConfErr}^*$ *is decreasing in* $\delta$.

In addition, there are some monotone relationships that can be verified numerically. We summarized them in the following claim.

**Claim D.10.** *Consider the same settings as Theorem D.9. When* $\tau = \tau^{\mathrm{opt}} > 0$, *we have*

- $\mathrm{CalErr}^*$ *is decreasing in* $\|\boldsymbol{\mu}\|_2$ *and* $\delta$, *for any* $\pi \leq \overline{\pi}$ *fixed, where* $\overline{\pi} \approx 0.25$ *is some constant.*

- $\mathrm{ConfErr}^*$ *is decreasing in* $\pi \in (0, \frac{1}{2})$.

### D.4 GENERALIZATIONS

Below we present two possible extensions of our main results stated in previous sections.

**Multiclass classification.** In the $K$-class setting, we observe features $\boldsymbol{x}_i \in \mathbb{R}^d$ and labels $y_i \in [K] \sim P_y$, sampled from a $K$-component Gaussian mixture model ($K$-GMM):

$$\pi_k := \mathbb{P}(y_i = k), \ \ k \in [K], \qquad \boldsymbol{x}_i \mid y_i = k \sim \mathsf{N}(\boldsymbol{\mu}_k, \mathbf{I}_d), \tag{40}$$

where $\{\boldsymbol{\mu}_k\}_{k \in [K]}$ are the class means. Let $\widehat{\boldsymbol{f}}(\boldsymbol{x}) = \widehat{\mathbf{W}}\boldsymbol{x} + \widehat{\mathbf{w}}_0$ be the logits of multinomial logistic regression for $\{(\boldsymbol{x}_i, y_i)\}_{i=1}^n$, where $\widehat{\mathbf{W}} \in \mathbb{R}^{K \times d}$, $\widehat{\mathbf{w}}_0 \in \mathbb{R}^K$ are the solution to $K$-class SVM:

$$\begin{aligned} \underset{\mathbf{W} \in \mathbb{R}^{K \times d}, \mathbf{w}_0 \in \mathbb{R}^K}{\mathrm{minimize}} \quad & \|\mathbf{W}\|_{\mathrm{F}}^2, \\ \mathrm{subject\ to} \quad & \langle \boldsymbol{x}_i, \mathbf{w}_{y_i} \rangle + w_{0, y_i} \geq \langle \boldsymbol{x}_i, \mathbf{w}_k \rangle + w_{0, k} + 1, \quad \forall i \in [n], \ \forall k \neq y_i, \end{aligned} \tag{41}$$

where $\mathbf{w}_k$ is the $k$-th row of $\mathbf{W}$, and $w_{0, k}$ is the $k$-th element of $\mathbf{w}_0$. The prediction is given by $\widehat{y}(\boldsymbol{x}) := \arg\max_{k \in [K]} \widehat{f}_k(\boldsymbol{x})$, where $\widehat{f}_k(\boldsymbol{x})$ is the logit of $\boldsymbol{x}$ for label $k$, i.e., the $k$-th component of $\boldsymbol{f}(\boldsymbol{x})$. Using the non-rigorous replica method from statistical physics, we conjecture the limiting empirical logits distribution as follows.

**Conjecture D.11.** *Assume that* $n/d \to \delta$ *as* $n, d \to \infty$. *Denote by* $\boldsymbol{\mu} = [\boldsymbol{\mu}_1, \cdots, \boldsymbol{\mu}_K] \in \mathbb{R}^{d \times K}$ *the matrix of class means, and assume that* $\boldsymbol{\mu}^\mathsf{T} \boldsymbol{\mu}$ *converges to a deterministic matrix* $\mathbf{Q}_\mu \in \mathbb{S}_+^K$ *as* $d \to \infty$. *Then as* $n, d \to \infty$,

$$W_2\left(\frac{1}{n}\sum_{i=1}^n \delta_{(y_i, \widehat{\mathbf{W}}\boldsymbol{x}_i + \widehat{\mathbf{w}}_0)}, \ \mathtt{Law}\left(Y, (\mathbf{R}_Y^*)^\mathsf{T} + \boldsymbol{U}^* + \mathbf{w}_0^*\right)\right) \xrightarrow{\mathrm{P}} 0,$$

where $\mathbf{R}_Y^*$ is the $Y$-th row of $\mathbf{R}^* \in \mathbb{R}^{K \times K}$, and $(\mathbf{R}^*, \boldsymbol{U}^*, \mathbf{w}_0^*)$ is the optimal solution to the following variational problem

$$
\begin{aligned}
\text{minimize} \quad & \mathrm{Tr}(\mathbf{Q}), \\
\text{w.r.t.} \quad & \mathbf{w}_0 \in \mathbb{R}^K, \quad \mathbf{Q}, \mathbf{Q}_R \in \mathbb{S}_+^K, \quad \mathbf{O}, \mathbf{R} \in \mathbb{R}^{K \times K}, \quad \boldsymbol{H}, \boldsymbol{U} \in \mathcal{L}^2(\Omega, \mathbb{R}^K), \\
\text{subject to} \quad & \mathbf{Q}_R \preceq \mathbf{I}_K, \quad \mathbf{O}^\mathsf{T}\mathbf{O} \preceq \mathbf{I}_K, \quad \mathbb{E}[\boldsymbol{H}\boldsymbol{H}^\mathsf{T}] \preceq \frac{1}{\delta}\mathbf{I}_K, \\
& \mathbf{R} = \mathbf{Q}_\mu^{1/2}\mathbf{O}\mathbf{Q}_R^{1/2}\mathbf{Q}^{1/2}, \quad \boldsymbol{U} = \mathbf{Q}^{1/2}\boldsymbol{G} + \mathbf{Q}^{1/2}(\mathbf{I}_K - \mathbf{Q}_R)^{1/2}\boldsymbol{H}, \\
& \textit{On the event } \{Y = k\}, \; R_{kk} - R_{kl} + U_k - U_l + w_{0,k} - w_{0,l} \geq 1 \\
& \qquad\qquad \textit{almost surely}, \quad \forall k \in [K], \; \forall l \neq k,
\end{aligned}
\tag{42}
$$

where $\mathcal{L}^2(\Omega, \mathbb{R}^K)$ is the space of square integrable $K$-dimensional random vectors in the probability space $(\Omega, \mathcal{F}, \mathbb{P})$, $(Y, \boldsymbol{G}) \sim P_y \times \mathsf{N}(\mathbf{0}, \mathbf{I}_K)$, and $\boldsymbol{H}, \boldsymbol{U}$ are random vectors to be optimized.

The last almost-sure linear constraint in Eq. (42) induces a truncation effect, captured by the random vector $\boldsymbol{H}$ (analogous to $\xi$ in Eq. (5)), which in turn characterizes the resulting overfitting behavior.

**Heterogeneous non-isotropic covariance.** We can extend our theory to the setting where the two classes have general non-isotropic covariances, i.e.,

$$
\boldsymbol{x}_i \,|\, y_i = +1 \sim \mathsf{N}(\boldsymbol{\mu}, \boldsymbol{\Sigma}_+), \quad \boldsymbol{x}_i \,|\, y_i = -1 \sim \mathsf{N}(-\boldsymbol{\mu}, \boldsymbol{\Sigma}_-).
\tag{43}
$$

Under the proportional asymptotics $n, d \to \infty$ with $n/d \to \delta$, we make the following assumptions.

(A1) $\boldsymbol{\Sigma}_+$ and $\boldsymbol{\Sigma}_-$ have eigenvalue decompositions $\boldsymbol{\Sigma}_+ = \sum_{i=1}^d \lambda_{+,i} \mathbf{v}_i \mathbf{v}_i^\mathsf{T}$, $\boldsymbol{\Sigma}_- = \sum_{i=1}^d \lambda_{-,i} \mathbf{v}_i \mathbf{v}_i^\mathsf{T}$, i.e., $\boldsymbol{\Sigma}_+$ and $\boldsymbol{\Sigma}_-$ commute.

(A2) There exists a random vector $(S_+, S_-, M) \in \mathbb{R}_{>0} \times \mathbb{R}_{>0} \times \mathbb{R}$ in probability space $(\Omega, \mathcal{F}, \mathbb{P})$ such that the empirical distribution of spectrum $\{(\lambda_{+,i}, \lambda_{-,i}, \sqrt{d}\langle \mathbf{v}_i, \boldsymbol{\mu}\rangle)\}_{i=1}^d$ converges to the distribution of $(S_+, S_-, M)$ in Wasserstein-2 sense:

$$
\frac{1}{d}\sum_{i=1}^d \delta_{(\lambda_{+,i}, \lambda_{-,i}, \sqrt{d}\langle \mathbf{v}_i, \boldsymbol{\mu}\rangle)} \xRightarrow{W_2} \mathrm{Law}\,(S_+, S_-, M).
$$

In particular, $\|\boldsymbol{\mu}\|_2 \equiv (\mathbb{E}[M^2])^{1/2}$ is fixed and $\mathbb{E}[S_\pm^2] < \infty$.

Recall that $\widehat{\rho} = \langle \frac{\widehat{\boldsymbol{\beta}}}{\|\widehat{\boldsymbol{\beta}}\|_2}, \frac{\boldsymbol{\mu}}{\|\boldsymbol{\mu}\|_2}\rangle$ and $(\widehat{\boldsymbol{\beta}}, \widehat{\beta}_0)$ is the max-margin solution to Eq. (2b). The result below generalizes Theorem 2.1 under Eq. (43).

**Conjecture D.12.** *Consider model Eq. (43) with asymptotics $n/d \to \delta \in (0, \infty)$ as $n, d \to \infty$. Under assumptions (A1)—(A2), there is a critical threshold $\delta_c = \delta_c(\pi, S_+, S_-, M)$, such that when $\delta < \delta_c$, the following holds as $n, d \to \infty$:*

(a) ***Parameter convergence.*** *The training set is linearly separable with high probability, and*

$$
(\widehat{\rho}, \widehat{\beta}_0, \widehat{\kappa}) \xrightarrow{\mathrm{P}} (\rho^*, \beta_0^*, \kappa^*),
$$

*where $\rho^* = \mathbb{E}[MB_*]/(\mathbb{E}[M^2])^{1/2}$ and $(B_*, \beta_0^*, \kappa^*)$ is the solution to the variational problem*

$$
\begin{aligned}
\max_{B \in \mathcal{L}^2, \beta_0 \in \mathbb{R}, \kappa > 0} \quad & \kappa, \\
\text{subject to} \quad & \mathbb{E}\left[B^2\right] \leq 1, \\
& \mathbb{E}\left[\left(\kappa - \mathbb{E}[MB] + (\mathbb{E}[S_+ B^2])^{1/2} G - \beta_0\right)_+^2\right] \leq \frac{(\mathbb{E}\left[S_+^{1/2} G_+ B\right])^2}{\pi\delta}, \\
& \mathbb{E}\left[\left(\kappa - \mathbb{E}[MB] + (\mathbb{E}[S_- B^2])^{1/2} G + \beta_0\right)_+^2\right] \leq \frac{(\mathbb{E}\left[S_-^{1/2} G_- B\right])^2}{(1-\pi)\delta},
\end{aligned}
$$

*where $\mathcal{L}^2$ is the space of square integrable random variables in the probability space $(\Omega, \mathcal{F}, \mathbb{P})$, $(G, G_+, G_-) \sim_{\mathrm{i.i.d.}} \mathsf{N}(0, 1)$ is independent of $(S_+, S_-, M)$, and $B$ is a random variable to be optimized.*

*As a consequence, the limits of minority/majority errors are*

$$\text{Err}_+ \to \Phi\left(\frac{-\rho^* \|\boldsymbol{\mu}\|_2 - \beta_0^*}{\sigma^*(+1)}\right), \qquad \text{Err}_- \to \Phi\left(\frac{-\rho^* \|\boldsymbol{\mu}\|_2 + \beta_0^*}{\sigma^*(-1)}\right),$$

*where $\sigma^*(+1) = (\mathbb{E}[S_+ B_*^2])^{1/2}$ and $\sigma^*(-1) = (\mathbb{E}[S_- B_*^2])^{1/2}$.*

*(b) **ELD convergence.** The empirical (training) logit distribution $\widehat{\nu}_n$ has limit $\nu_*$ in the sense that*

$$W_2(\widehat{\nu}_n, \nu_*) \xrightarrow{\text{P}} 0, \qquad \text{where } \nu_* := \text{Law}\left(Y, Y \max\{\kappa^*, \rho^* \|\boldsymbol{\mu}\|_2 + \sigma^*(Y)G + Y\beta_0^*\}\right).$$

*   **TLD convergence.** The testing logit distribution $\widehat{\nu}_n^{\text{test}}$ has limit $\nu_*^{\text{test}}$ in the sense that*

$$\widehat{\nu}_n^{\text{test}} \xrightarrow{w} \nu_*^{\text{test}}, \qquad \text{where } \nu_*^{\text{test}} := \text{Law}\left(Y, Y(\rho^* \|\boldsymbol{\mu}\|_2 + \sigma^*(Y)G + Y\beta_0^*)\right).$$

Compared to Theorem 2.1, we find that covariance heterogeneity, captured by $\sigma^*(Y)$, induces distinct scaling effects on the testing errors $(\text{Err}_+, \text{Err}_-)$. It also rescales the Gaussian component of the logit distribution $(G)$ for each class.

**Spiked non-isotropic covariance.** We also provide a characterization of the empirical logits distribution under spiked covariance. In particular, we assume that

$$\boldsymbol{x}_i \mid y_i \sim \text{N}(y_i \boldsymbol{\mu}, \boldsymbol{\Sigma}), \qquad \boldsymbol{\Sigma} = q^2 \mathbf{V}\mathbf{V}^\mathsf{T} + \mathbf{I}_d, \tag{44}$$

where the spike $\mathbf{V}$ is a $d \times J$ orthogonal matrix. For this model, we have the following analogous result on the empirical logits distribution of SVM:

**Conjecture D.13.** *Assume that $n/d \to \delta$, $J/d \to \psi_1$ as $n, d, J \to \infty$. Denote $\psi_2 = 1 - \psi_1$, and further assume that*

$$\lim_{n\to\infty} \left\|\mathbf{V}^\mathsf{T}\boldsymbol{\mu}\right\|_2 = c_1, \qquad \lim_{n\to\infty} \sqrt{\|\boldsymbol{\mu}\|_2^2 - \|\mathbf{V}^\mathsf{T}\boldsymbol{\mu}\|_2^2} = c_2.$$

*Let $(\widehat{\boldsymbol{\beta}}, \widehat{\beta}_0)$ be the max-margin solution to Eq. (2b). Then, as $n \to \infty$,*

$$W_2\left(\frac{1}{n}\sum_{i=1}^n \delta_{(y_i, \langle \boldsymbol{x}_i, \widehat{\boldsymbol{\beta}}\rangle + \widehat{\beta}_0)}, \text{Law}\left(Y, Y\rho_1^* c_1 + Y\rho_2^* c_2 + \sqrt{1+q^2} r_1^* G_1 + r_2^* G_2 + \beta_0^*\right)\right) \xrightarrow{\text{P}} 0.$$

*In the above display, $Y \sim P_y$ is independent of $(G_1, G_2) \sim \text{N}(0,1)^{\otimes 2}$, and $(\rho_1^*, \rho_2^*, r_1^*, r_2^*, \beta_0^*)$ solves the following convex optimization problem:*

$$\begin{aligned}
\underset{\rho_1, \rho_2, r_1, r_2, \beta_0, \kappa}{\text{maximize}} \quad & \kappa, \\
\text{subject to} \quad & \mathbb{E}\left[\left(\kappa - \rho_1 c_1 - \rho_2 c_2 - \sqrt{1+q^2} r_1 G_1 - r_2 G_2 - \beta_0 Y\right)_+^2\right] \\
& \leq \frac{1}{\delta}\left(\sqrt{1+q^2}\sqrt{r_1^2 - \rho_1^2}\sqrt{\psi_1} + \sqrt{r_2^2 - \rho_2^2}\sqrt{\psi_2}\right)^2, \\
& r_1^2 + r_2^2 = 1, \quad \rho_1^2 \leq r_1^2, \quad \rho_2^2 \leq r_2^2.
\end{aligned} \tag{45}$$

**Future work.** In deep learning, the features are learned by optimizing the loss over all weights in a neural network, and data imbalance impacts on feature learning in a complex way as observed in (Cao et al., 2019). Also, models tend to erroneously find spurious features if data imbalance is severe (Sagawa et al., 2020). It would be interesting to analyze overfitting and propose remedies for these scenarios.

# E    LOGIT DISTRIBUTION FOR SEPARABLE DATA: PROOFS FOR SECTION D.1.1

## E.1    PROOF OF THEOREM D.1

Recall that the margin-rebalanced SVM can be rewritten as

$$\begin{aligned}
\underset{\boldsymbol{\beta} \in \mathbb{R}^d, \beta_0, \kappa \in \mathbb{R}}{\text{maximize}} \quad & \kappa, \\
\text{subject to} \quad & \widetilde{y}_i(\langle \boldsymbol{x}_i, \boldsymbol{\beta}\rangle + \beta_0) \geq \kappa, \quad \forall i \in [n], \\
& \|\boldsymbol{\beta}\|_2 \leq 1.
\end{aligned} \tag{46}$$

Let $(\widehat{\boldsymbol{\beta}}_n, \widehat{\beta}_{0,n})$ be an optimal solution and $\widehat{\kappa}_n = \mathbb{1}_{1 \leq n_+ \leq n-1} \kappa(\widehat{\boldsymbol{\beta}}_n, \widehat{\beta}_{0,n})$ be the well-defined maximum margin as per Definition C.1. Our goal is to derive exact asymptotics for $(\widehat{\boldsymbol{\beta}}_n, \widehat{\beta}_{0,n}, \widehat{\kappa}_n)$. Similar to the development in (Montanari et al., 2023), for any positive margin $\kappa > 0$, we define the event

$$
\begin{aligned}
\mathcal{E}_{n,\kappa} &= \left\{ \kappa(\widehat{\boldsymbol{\beta}}_n, \widehat{\beta}_{0,n}) \geq \kappa \right\} \\
&= \left\{ \exists \boldsymbol{\beta} \in \mathbb{R}^d, \|\boldsymbol{\beta}\|_2 \leq 1, \beta_0 \in \mathbb{R}, \text{ such that } \widetilde{y}_i \big( \langle \boldsymbol{x}_i, \boldsymbol{\beta} \rangle + \beta_0 \big) \geq \kappa \text{ for all } i \in [n] \right\} \\
&= \left\{ \exists \boldsymbol{\beta} \in \mathbb{R}^d, \|\boldsymbol{\beta}\|_2 \leq 1, \beta_0 \in \mathbb{R}, \text{ such that } \left\| \left( \kappa \mathbf{s}_{\boldsymbol{y}} - \boldsymbol{y} \odot \boldsymbol{X}\boldsymbol{\beta} - \beta_0 \boldsymbol{y} \right)_+ \right\|_2 = 0 \right\},
\end{aligned}
$$

where $\mathbf{s}_{\boldsymbol{y}} = (s(y_1), \ldots, s(y_n))^\mathsf{T}$ and $s$ is the function defined in Eq. (29). Therefore, the data $(\boldsymbol{X}, \boldsymbol{y})$ is linearly separable if and only if $\mathcal{E}_{n,\kappa}$ holds for some $\kappa > 0$. We would like to determine for which sets of parameters $(\pi, \boldsymbol{\mu}, \delta, \tau)$ we have $\mathbb{P}(\mathcal{E}_{n,\kappa}) \to 1$ and for which instead $\mathbb{P}(\mathcal{E}_{n,\kappa}) \to 0$ as $n, d \to \infty$. To this end, we also define

$$
\begin{aligned}
\xi_{n,\kappa} &:= \min_{\substack{\|\boldsymbol{\beta}\|_2 \leq 1 \\ \beta_0 \in \mathbb{R}}} \frac{1}{\sqrt{d}} \left\| \left( \kappa \mathbf{s}_{\boldsymbol{y}} - \boldsymbol{y} \odot \boldsymbol{X}\boldsymbol{\beta} - \beta_0 \boldsymbol{y} \right)_+ \right\|_2 \\
&\overset{(i)}{=} \min_{\substack{\|\boldsymbol{\beta}\|_2 \leq 1 \\ \beta_0 \in \mathbb{R}}} \max_{\substack{\|\boldsymbol{\lambda}\|_2 \leq 1 \\ \boldsymbol{\lambda} \odot \boldsymbol{y} \geq \mathbf{0}}} \frac{1}{\sqrt{d}} \boldsymbol{\lambda}^\mathsf{T} \left( \kappa \mathbf{s}_{\boldsymbol{y}} \odot \boldsymbol{y} - \boldsymbol{X}\boldsymbol{\beta} - \beta_0 \mathbf{1} \right),
\end{aligned}
\tag{47}
$$

where $(i)$ is a consequence of Lagrange duality (dual norm) $\|(\mathbf{a})_+\|_2 = \max_{\|\boldsymbol{\lambda}\|_2 \leq 1, \boldsymbol{\lambda} \geq \mathbf{0}} \boldsymbol{\lambda}^\mathsf{T} \mathbf{a}$. Then we established the following equivalence

$$
\{\xi_{n,\kappa} = 0\} \Longleftrightarrow \mathcal{E}_{n,\kappa} \qquad \{\xi_{n,\kappa} > 0\} \Longleftrightarrow \mathcal{E}_{n,\kappa}^c.
$$

Keep in mind that we are only concerned with the sign (positivity) of $\xi_{n,\kappa}$, not its magnitude. As a consequence, we have

$$
\widehat{\kappa}_n = \mathbb{1}_{1 \leq n_+ \leq n-1} \cdot \sup\{\kappa \in \mathbb{R} : \xi_{n,\kappa} = 0\}.
$$

Let $\mathcal{D}_n := \{n_+ = 0 \text{ or } n\}$ be the event of degeneration for any datasets of size $n$. Clearly $\mathbb{P}(\mathcal{D}_n) = \pi^n + (1-\pi)^n \to 0$ as $n \to \infty$. Technically, the empirical logit distribution (ELD) in Eq. (3) is not well-defined on $\mathcal{D}_n$. Similar as Definition C.1, we can also redefine it as follows:

$$
\widehat{\nu}_n := \frac{1}{n} \sum_{i=1}^n \delta_{(y_i, \langle \boldsymbol{x}_i, \widehat{\boldsymbol{\beta}} \rangle + \widehat{\beta}_0) \cdot \mathbb{1}\{1 \leq n_+ \leq n-1\}}.
\tag{48}
$$

We provide an outline for the main parts of the proofs of Theorem D.1(a)—(c), which involves several steps of transforming and simplifying the random variable $\xi_{n,\kappa}$.

$$
\left.
\begin{aligned}
\xi_{n,\kappa} \xrightarrow[\text{Theorem E.1}]{\textbf{Step 1}} \xi'_{n,\kappa,B} \xrightarrow[\text{Theorem E.2}]{\textbf{Step 2}} \xi'^{(1)}_{n,\kappa,B} \xrightarrow[\text{Theorem E.3}]{\textbf{Step 3}} \overline{\xi}'^{(2)}_{\kappa,B} \Rightarrow \overline{\xi}^{(2)}_\kappa \\
\xrightarrow[\text{Theorem E.4}]{\textbf{Step 4}} \overline{\xi}^{(3)}_\kappa \Rightarrow \widetilde{\xi}^{(3)}_\kappa, F_\kappa(\rho, \beta_0) \xrightarrow[\text{Theorem E.5}]{\textbf{Step 5}} \delta^*(\kappa), H_\kappa(\rho, \beta_0).
\end{aligned}
\right\} \text{Theorem E.6}
$$

**Step 1: Boundedness of the intercept (from $\xi_{n,\kappa}$ to $\xi'_{n,\kappa,B}$)** According to the definition of $\xi_{n,\kappa}$, parameters $\boldsymbol{\beta}$ and $\boldsymbol{\lambda}$ are optimized in compact sets, but $\beta_0$ is not. Such non-compactness might cause technical difficulties in the following steps, for example, when applying Gordon's Gaussian comparison inequality and establishing uniform convergence. However, it turns out that $\beta_0$ is asymptotically bounded on the event $\mathcal{E}_{n,\kappa}$. More precisely, we define

$$
\xi'_{n,\kappa,B} := \min_{\substack{\|\boldsymbol{\beta}\|_2 \leq 1 \\ |\beta_0| \leq B}} \max_{\substack{\|\boldsymbol{\lambda}\|_2 \leq 1 \\ \boldsymbol{\lambda} \odot \boldsymbol{y} \geq \mathbf{0}}} \frac{1}{\sqrt{d}} \boldsymbol{\lambda}^\mathsf{T} \left( \kappa \mathbf{s}_{\boldsymbol{y}} \odot \boldsymbol{y} - \boldsymbol{X}\boldsymbol{\beta} - \beta_0 \mathbf{1} \right),
\tag{49}
$$

where $B = B(\tau, \kappa, \pi, \|\boldsymbol{\mu}\|_2, \delta)$ is a sufficiently large constant. Then we can show that $\xi_{n,\kappa}$ and $\xi'_{n,\kappa,B}$ have the same sign with high probability, which enables us to work with $\xi'_{n,\kappa,B}$ instead of $\xi_{n,\kappa}$.

**Lemma E.1** (Boundedness of $\beta_0$). *There exists some constant $B \in (0, \infty)$ (depends on $\tau, \kappa, \pi, \|\boldsymbol{\mu}\|_2, \delta$) such that*

$$
\lim_{n \to \infty} \left| \mathbb{P}\big(\xi_{n,\kappa} = 0\big) - \mathbb{P}\big(\xi'_{n,\kappa,B} = 0\big) \right| = 0.
$$

See Section E.1.1 for the proof.

**Step 2: Reduction via Gaussian comparison (from $\xi'_{n,\kappa,B}$ to $\xi'^{(1)}_{n,\kappa,B}$)** According to the expression of $\xi'_{n,\kappa,B}$, it is not hard to see the objective function (of $(\boldsymbol{\beta}, \boldsymbol{\lambda})$) is a bilinear form of the Gaussian random matrix $\boldsymbol{X}$. To simplify the bilinear term and make the calculation easier, we will use the convex Gaussian minimax theorem (CGMT, see Theorem J.1), i.e., Gordon's comparison inequality (Gordon, 1985; Thrampoulidis et al., 2015). To do so, we introduce another quantity:

$$\xi'^{(1)}_{n,\kappa,B} := \min_{\substack{\rho^2 + \|\boldsymbol{\theta}\|_2^2 \leq 1 \\ |\beta_0| \leq B}} \max_{\substack{\|\boldsymbol{\lambda}\|_2 \leq 1 \\ \boldsymbol{\lambda} \odot \boldsymbol{y} \geq 0}} \frac{1}{\sqrt{d}} \left( \|\boldsymbol{\lambda}\|_2 \, \boldsymbol{g}^\top \boldsymbol{\theta} + \|\boldsymbol{\theta}\|_2 \, \boldsymbol{h}^\top \boldsymbol{\lambda} + \boldsymbol{\lambda}^\top \big( \kappa \mathbf{s}_{\boldsymbol{y}} \odot \boldsymbol{y} - \rho \|\boldsymbol{\mu}\|_2 \, \boldsymbol{y} + \rho \boldsymbol{u} - \beta_0 \mathbf{1} \big) \right),$$
(50)

where $\rho \in \mathbb{R}$, $\boldsymbol{\theta} \in \mathbb{R}^{d-1}$ are parameters, $\boldsymbol{g} \sim \mathsf{N}(\mathbf{0}, \mathbf{I}_{d-1})$, $\boldsymbol{h} \sim \mathsf{N}(\mathbf{0}, \mathbf{I}_n)$, $\boldsymbol{u} \sim \mathsf{N}(\mathbf{0}, \mathbf{I}_n)$ are independent Gaussian vectors. The following lemma connects $\xi'_{n,\kappa,B}$ with $\xi'^{(1)}_{n,\kappa,B}$.

**Lemma E.2** (Reduction via CGMT). *For any $v \in \mathbb{R}$ and $t \geq 0$,*

$$\mathbb{P}\Big( \big| \xi'_{n,\kappa,B} - v \big| \geq t \Big) \leq 2\mathbb{P}\Big( \big| \xi'^{(1)}_{n,\kappa,B} - v \big| \geq t \Big).$$

See Section E.1.2 for the proof.

**Step 3: Dimension reduction (from $\xi'^{(1)}_{n,\kappa,B}$ to $\overline{\xi}'^{(2)}_{\kappa,B}$)** It turns out that $\xi'^{(1)}_{n,\kappa,B}$ can be further simplified for analytical purposes. We define a new (deterministic) quantity

$$\overline{\xi}'^{(2)}_{\kappa,B} := \min_{\substack{\rho^2 + r^2 \leq 1, r \geq 0 \\ |\beta_0| \leq B}} -r + \sqrt{\delta} \left( \mathbb{E}\left[ \big( s(Y)\kappa - \rho \|\boldsymbol{\mu}\|_2 + \rho G_1 + r G_2 - \beta_0 Y \big)_+^2 \right] \right)^{1/2},$$

which is a constrained minimization over only three variables $\rho$, $r$, and $\beta_0$, with random variables $(Y, G_1, G_2) \sim P_y \times \mathsf{N}(0,1) \times \mathsf{N}(0,1)$. The two quantities of interest can be related via the uniform law of large numbers (ULLN) as shown in the following lemma.

**Lemma E.3** (ULLN). *As $n, d \to \infty$, we have*

$$\xi'^{(1)}_{n,\kappa,B} \xrightarrow{\mathrm{P}} \left( \overline{\xi}'^{(2)}_{\kappa,B} \right)_+.$$

See Section E.1.3 for the proof.

**Step 4: Investigation of the positivity (from $\overline{\xi}'^{(2)}_{\kappa,B}$ to $\overline{\xi}^{(3)}_{\kappa}$)** To further simplify the problem, we define the following quantities that are closely related to $\overline{\xi}'^{(2)}_{\kappa,B}$:

$$\overline{\xi}^{(2)}_{\kappa} := \min_{\substack{\rho^2 + r^2 \leq 1, r \geq 0 \\ \beta_0 \in \mathbb{R}}} -r + \sqrt{\delta} \left( \mathbb{E}\left[ \big( s(Y)\kappa - \rho \|\boldsymbol{\mu}\|_2 + \rho G_1 + r G_2 - \beta_0 Y \big)_+^2 \right] \right)^{1/2}$$

$$\overline{\xi}^{(3)}_{\kappa} := \min_{\substack{\rho \in [-1,1] \\ \beta_0 \in \mathbb{R}}} -\sqrt{1 - \rho^2} + \sqrt{\delta} \left( \mathbb{E}\left[ \big( s(Y)\kappa - \rho \|\boldsymbol{\mu}\|_2 + G - \beta_0 Y \big)_+^2 \right] \right)^{1/2}. \tag{51}$$

Firstly, we argue that $\overline{\xi}'^{(2)}_{\kappa,B} = \overline{\xi}^{(2)}_{\kappa}$ for constant $B$ large enough, by noticing the optimal (unique) $\beta_0$ in $\overline{\xi}^{(2)}_{\kappa}$ is always bounded by some constant (depends on $\tau, \kappa, \pi, \|\boldsymbol{\mu}\|_2, \delta$). Secondly, notice $\overline{\xi}^{(3)}_{\kappa}$ can be viewed as fixing $r = \sqrt{1 - \rho^2}$ in the optimization of $\overline{\xi}^{(2)}_{\kappa}$, and $G := \rho G_1 + \sqrt{1 - \rho^2} G_2 \sim \mathsf{N}(0,1)$. The following lemma shows that the sign won't change from $\overline{\xi}^{(2)}_{\kappa}$ to $\overline{\xi}^{(3)}_{\kappa}$.

**Lemma E.4** (Sign invariance). *For any $\kappa > 0$, the following result holds:*

*(a)* $\operatorname{sign}(\overline{\xi}^{(2)}_{\kappa}) = \operatorname{sign}(\overline{\xi}^{(3)}_{\kappa})$.

*(b) If $\overline{\xi}^{(2)}_{\kappa} \leq 0$, then $\overline{\xi}^{(2)}_{\kappa} = \overline{\xi}^{(3)}_{\kappa}$.*

See Section E.1.4 for the proof.

**Step 5: Phase transition and margin convergence**  Note the function $\delta^* : \mathbb{R} \to \mathbb{R}_{\geq 0}$ defined in Eq. (28) is closely related to $\overline{\xi}_\kappa^{(3)}$. Let $\kappa^* := \sup\{\kappa \in \mathbb{R} : \delta^*(\kappa) \geq \delta\}$. By combining the results from previous steps, we have the following relation.

**Lemma E.5** (Phase transition). *For any $\kappa > 0$, we have*

$$\lim_{n \to \infty} \mathbb{P}\left(\xi_{n,\kappa} = 0\right) = 1, \qquad \text{if } \delta \leq \delta^*(\kappa) \text{ (i.e., } \kappa \leq \kappa^*),$$
$$\lim_{n \to \infty} \mathbb{P}\left(\xi_{n,\kappa} > 0\right) = 1, \qquad \text{if } \delta > \delta^*(\kappa) \text{ (i.e., } \kappa > \kappa^*).$$

*In particular,*

$$\lim_{n \to \infty} \mathbb{P}\left\{(\boldsymbol{X}, \boldsymbol{y}) \text{ is linearly separable}\right\} = 1, \qquad \text{if } \delta < \delta^*(0),$$
$$\lim_{n \to \infty} \mathbb{P}\left\{(\boldsymbol{X}, \boldsymbol{y}) \text{ is not linearly separable}\right\} = 0, \qquad \text{if } \delta > \delta^*(0).$$

As a consequence, we can also derive the convergence of margin in probability. Notice that the following result is weaker than $\mathcal{L}^2$ convergence Theorem D.1(c). However, we need this preliminary result for the subsequent proof of ELD convergence in Theorem E.7.

**Lemma E.6** (Margin convergence, in probability). *If $\delta < \delta^*(0)$, we have $\widehat{\kappa}_n \xrightarrow{\text{P}} \kappa^*$.*

See Section E.1.5 for the proof.

### E.1.1 STEP 1 — BOUNDEDNESS OF THE INTERCEPT: PROOF OF LEMMA E.1

***Proof of Lemma E.1.*** Recall that

$$\xi_{n,\kappa} = \min_{\substack{\|\boldsymbol{\beta}\|_2 \leq 1 \\ \beta_0 \in \mathbb{R}}} \frac{1}{\sqrt{d}} \left\| \left(\kappa \mathbf{s}_{\boldsymbol{y}} - \boldsymbol{y} \odot \boldsymbol{X}\boldsymbol{\beta} - \beta_0 \boldsymbol{y}\right)_+ \right\|_2.$$

Let $(\widetilde{\boldsymbol{\beta}}_n, \widetilde{\beta}_{0,n})$ be a minimizer of the function above[10]. On the event $\mathcal{D}_n^c \cap \mathcal{E}_{n,\kappa}$ $(\xi_{n,\kappa} = 0)$, we have

$$\left\| \left(\kappa \mathbf{s}_{\boldsymbol{y}} - \boldsymbol{y} \odot \boldsymbol{X}\widetilde{\boldsymbol{\beta}}_n - \widetilde{\beta}_{0,n}\boldsymbol{y}\right)_+ \right\|_2 = 0, \quad \implies \quad \begin{cases} \tau\kappa - \langle \boldsymbol{x}_i, \widetilde{\boldsymbol{\beta}}_n \rangle - \widetilde{\beta}_{0,n} \leq 0, & \text{if } y_i = +1, \\ \kappa + \langle \boldsymbol{x}_i, \widetilde{\boldsymbol{\beta}}_n \rangle + \widetilde{\beta}_{0,n} \leq 0, & \text{if } y_i = -1. \end{cases}$$

Write $\boldsymbol{x}_i = y_i\boldsymbol{\mu} + \boldsymbol{z}_i$, where $\boldsymbol{z}_i \overset{\text{i.i.d.}}{\sim} \mathsf{N}(\mathbf{0}, \mathbf{I}_d)$ and $y_i \perp\!\!\!\perp \boldsymbol{z}_i$. Then we obtain

$$\begin{cases} \widetilde{\beta}_{0,n} \geq \tau\kappa - \langle \boldsymbol{\mu}, \widetilde{\boldsymbol{\beta}}_n \rangle - \langle \boldsymbol{z}_i, \widetilde{\boldsymbol{\beta}}_n \rangle, & \text{if } y_i = +1, \\ \widetilde{\beta}_{0,n} \leq -\kappa + \langle \boldsymbol{\mu}, \widetilde{\boldsymbol{\beta}}_n \rangle - \langle \boldsymbol{z}_i, \widetilde{\boldsymbol{\beta}}_n \rangle, & \text{if } y_i = -1, \end{cases}$$

which implies for all $i, j$ such that $y_i = +1, y_j = -1$,

$$|\widetilde{\beta}_{0,n}| \leq \left| \tau\kappa - \langle \boldsymbol{\mu}, \widetilde{\boldsymbol{\beta}}_n \rangle - \langle \boldsymbol{z}_i, \widetilde{\boldsymbol{\beta}}_n \rangle \right| + \left| \kappa - \langle \boldsymbol{\mu}, \widetilde{\boldsymbol{\beta}}_n \rangle + \langle \boldsymbol{z}_j, \widetilde{\boldsymbol{\beta}}_n \rangle \right|$$
$$\leq (\tau + 1)\kappa + 2\left|\langle \boldsymbol{\mu}, \widetilde{\boldsymbol{\beta}}_n \rangle\right| + \left|\langle \boldsymbol{z}_i, \widetilde{\boldsymbol{\beta}}_n \rangle\right| + \left|\langle \boldsymbol{z}_j, \widetilde{\boldsymbol{\beta}}_n \rangle\right|.$$

Using the inequality $(a + b + c)^2 \leq 3(a^2 + b^2 + c^2)$, we have

$$|\widetilde{\beta}_{0,n}|^2 \leq 3 \left\{ \left((\tau + 1)\kappa + 2\left|\langle \boldsymbol{\mu}, \widetilde{\boldsymbol{\beta}}_n \rangle\right|\right)^2 + \min_{i:y_i=+1} \left|\langle \boldsymbol{z}_i, \widetilde{\boldsymbol{\beta}}_n \rangle\right|^2 + \min_{j:y_j=-1} \left|\langle \boldsymbol{z}_j, \widetilde{\boldsymbol{\beta}}_n \rangle\right|^2 \right\}$$

$$\leq 3 \left\{ \left((\tau + 1)\kappa + 2\left|\langle \boldsymbol{\mu}, \widetilde{\boldsymbol{\beta}}_n \rangle\right|\right)^2 + \frac{1}{n_+} \sum_{i:y_i=+1} \left|\langle \boldsymbol{z}_i, \widetilde{\boldsymbol{\beta}}_n \rangle\right|^2 + \frac{1}{n_-} \sum_{j:y_j=-1} \left|\langle \boldsymbol{z}_j, \widetilde{\boldsymbol{\beta}}_n \rangle\right|^2 \right\}$$

$$\overset{\text{(i)}}{=} 3 \left\{ \left((\tau + 1)\kappa + 2\left|\langle \boldsymbol{\mu}, \widetilde{\boldsymbol{\beta}}_n \rangle\right|\right)^2 + \frac{1}{n_+} \left\|\boldsymbol{Z}_+\widetilde{\boldsymbol{\beta}}_n\right\|_2^2 + \frac{1}{n_-} \left\|\boldsymbol{Z}_-\widetilde{\boldsymbol{\beta}}_n\right\|_2^2 \right\}$$

$$\overset{\text{(ii)}}{\leq} 3 \left\{ \left((\tau + 1)\kappa + 2\left\|\boldsymbol{\mu}\right\|_2\right)^2 + \frac{1}{n_+} \left\|\boldsymbol{Z}_+\right\|_{\text{op}}^2 + \frac{1}{n_-} \left\|\boldsymbol{Z}_-\right\|_{\text{op}}^2 \right\} =: \widetilde{B}_{0,n},$$

---

[10]In general $(\widetilde{\boldsymbol{\beta}}_n, \widetilde{\beta}_{0,n})$ may not be unique and may not be equal to $(\widehat{\boldsymbol{\beta}}_n, \widehat{\beta}_{0,n})$.

where in (i) we denote $\boldsymbol{Z}_+ \in \mathbb{R}^{n_+ \times d}$ as a Gaussian random matrix with rows $\boldsymbol{z}_i$ such that $y_i = +1$, $\boldsymbol{Z}_- \in \mathbb{R}^{n_- \times d}$ with rows $\boldsymbol{z}_j$ such that $y_j = +1$, while in (ii) we use Cauchy–Schwarz inequality, the definition of operator norm, and $\|\widetilde{\boldsymbol{\beta}}_n\|_2 \le 1$.

Next, we show that $\widetilde{B}_{0,n}$ is asymptotically bounded. Notice $\boldsymbol{Z}_+, \boldsymbol{Z}_-$ have i.i.d. standard Gaussian entries. According to the tail bound of Gaussian matrices (Vershynin, 2018, Corollary 7.3.3), for any $t_n \ge 0$ such that $t_n = o(\sqrt{n})$ and some absolute constants $c, C \in (0, \infty)$, we have

$$\widetilde{B}_{0,n} \overset{(i)}{\le} 3\left\{ \left((\tau+1)\kappa + 2\|\boldsymbol{\mu}\|_2\right)^2 + \frac{1}{n_+}\left(\sqrt{n_+} + \sqrt{d} + t_n\right)^2 + \frac{1}{n_-}\left(\sqrt{n_-} + \sqrt{d} + t_n\right)^2 \right\}$$

$$\overset{(ii)}{\le} 3\left\{ \left((\tau+1)\kappa + 2\|\boldsymbol{\mu}\|_2\right)^2 + \left(C + \frac{1}{\sqrt{\pi\delta}}\right)^2 + \left(C + \frac{1}{\sqrt{(1-\pi)\delta}}\right)^2 \right\} =: B_0,$$

where (i) holds with probability as least $1 - 4\exp(-ct_n^2)$, and (ii) holds with probability one based on the fact that $n_+/n \to \pi$, $n_-/n \to 1 - \pi$ a.s. (by strong law of large numbers), and $n/d \to \delta$ as $n \to \infty$. Notice the upper bound $B_0$ is a constant which depends on $(\tau, \kappa, \pi, \|\boldsymbol{\mu}\|_2, \delta)$. Let $t_n \to \infty$, then we conclude $\widetilde{B}_{0,n} \le B_0$ with high probability.

Combining these results, for any $B > \sqrt{B_0}$,

$$\left(\{\xi_{n,\kappa} = 0\} \cap \mathcal{D}_n^c \cap \{\widetilde{B}_{0,n} \le B_0\}\right) \subseteq \left(\{\xi_{n,\kappa} = 0\} \cap \mathcal{D}_n^c \cap \{|\widetilde{\beta}_{0,n}| \le B\}\right) \subseteq \{\xi'_{n,\kappa,B} = 0\}.$$

Therefore, by union bound we have

$$\mathbb{P}\left(\xi_{n,\kappa} = 0\right) = \mathbb{P}\left(\{\xi_{n,\kappa} = 0\} \cap \left(\mathcal{D}_n^c \cap \{\widetilde{B}_{0,n} \le B_0\}\right)\right) + \mathbb{P}\left(\{\xi_{n,\kappa} = 0\} \cap \left(\mathcal{D}_n \cup \{\widetilde{B}_{0,n} > B_0\}\right)\right)$$

$$\le \mathbb{P}\left(\xi'_{n,\kappa,B} = 0\right) + \mathbb{P}(\mathcal{D}_n) + \mathbb{P}\left(\widetilde{B}_{0,n} > B_0\right).$$

Finally, by noticing $\xi_{n,\kappa} \le \xi'_{n,\kappa,B}$, we conclude

$$0 \le \mathbb{P}\left(\xi_{n,\kappa} = 0\right) - \mathbb{P}\left(\xi'_{n,\kappa,B} = 0\right) \le \mathbb{P}(\mathcal{D}_n) + \mathbb{P}\left(\widetilde{B}_{0,n} > B_0\right) \to 0, \qquad \text{as } n \to \infty.$$

This completes the proof. $\qquad\square$

### E.1.2 STEP 2 — REDUCTION VIA GAUSSIAN COMPARISON: PROOF OF LEMMA E.2

***Proof of Lemma E.2.*** Rewrite $\boldsymbol{x}_i = y_i\boldsymbol{\mu} + \boldsymbol{z}_i$, where $\boldsymbol{z}_i \overset{\text{i.i.d.}}{\sim} \mathsf{N}(\boldsymbol{0}, \mathbf{I}_d)$. Note that $y_i \perp\!\!\!\perp \boldsymbol{z}_i$. Denote the projection matrices

$$\mathbf{P}_{\boldsymbol{\mu}} := \frac{1}{\|\boldsymbol{\mu}\|_2^2}\boldsymbol{\mu}\boldsymbol{\mu}^\mathsf{T}, \qquad \mathbf{P}_{\boldsymbol{\mu}}^\perp := \mathbf{I}_d - \frac{1}{\|\boldsymbol{\mu}\|_2^2}\boldsymbol{\mu}\boldsymbol{\mu}^\mathsf{T},$$

where $\mathbf{P}_{\boldsymbol{\mu}}$ is the orthogonal projection onto $\mathrm{span}\{\boldsymbol{\mu}\}$ and $\mathbf{P}_{\boldsymbol{\mu}}^\perp$ is the orthogonal projection onto the orthogonal complement of $\mathrm{span}\{\boldsymbol{\mu}\}$. Then we have the following decomposition:

$$\langle \boldsymbol{x}_i, \boldsymbol{\beta} \rangle = y_i\langle \boldsymbol{\mu}, \boldsymbol{\beta} \rangle + \langle \boldsymbol{z}_i, \boldsymbol{\beta} \rangle = y_i\langle \boldsymbol{\mu}, \boldsymbol{\beta} \rangle + \langle \boldsymbol{z}_i, \mathbf{P}_{\boldsymbol{\mu}}\boldsymbol{\beta} \rangle + \langle \boldsymbol{z}_i, \mathbf{P}_{\boldsymbol{\mu}}^\perp\boldsymbol{\beta} \rangle$$

$$= y_i\left\langle \boldsymbol{\beta}, \frac{\boldsymbol{\mu}}{\|\boldsymbol{\mu}\|_2} \right\rangle \|\boldsymbol{\mu}\|_2 + \left\langle \boldsymbol{\beta}, \frac{\boldsymbol{\mu}}{\|\boldsymbol{\mu}\|_2} \right\rangle \left\langle \boldsymbol{z}_i, \frac{\boldsymbol{\mu}}{\|\boldsymbol{\mu}\|_2} \right\rangle + \langle \boldsymbol{z}_i, \mathbf{P}_{\boldsymbol{\mu}}^\perp\boldsymbol{\beta} \rangle$$

$$= y_i\rho \|\boldsymbol{\mu}\|_2 + \rho u_i + \langle \boldsymbol{z}_i, \mathbf{P}_{\boldsymbol{\mu}}^\perp\boldsymbol{\beta} \rangle,$$

where

$$\rho := \left\langle \boldsymbol{\beta}, \frac{\boldsymbol{\mu}}{\|\boldsymbol{\mu}\|_2} \right\rangle, \qquad u_i := \left\langle \boldsymbol{z}_i, \frac{\boldsymbol{\mu}}{\|\boldsymbol{\mu}\|_2} \right\rangle \sim \mathsf{N}(0, 1).$$

Let $\mathbf{Q} \in \mathbb{R}^{n \times (n-1)}$ be an orthonormal basis for the subspace $\mathrm{span}\{\boldsymbol{\mu}\}^\perp$ ($\mathbf{Q}^\mathsf{T}\mathbf{Q} = \mathbf{I}_{n-1}$). Note that

$$\langle \boldsymbol{z}_i, \mathbf{P}_{\boldsymbol{\mu}}^\perp\boldsymbol{\beta} \rangle = \langle \boldsymbol{z}_i, \mathbf{Q}\mathbf{Q}^\mathsf{T}\boldsymbol{\beta} \rangle = \langle \mathbf{Q}^\mathsf{T}\boldsymbol{z}_i, \mathbf{Q}^\mathsf{T}\boldsymbol{\beta} \rangle = \langle \boldsymbol{g}_i, \boldsymbol{\theta} \rangle,$$

where

$$\boldsymbol{g}_i := \mathbf{Q}^\mathsf{T}\boldsymbol{z}_i \sim \mathsf{N}(\boldsymbol{0}, \mathbf{I}_{d-1}), \qquad \boldsymbol{g}_i \perp\!\!\!\perp u_i,$$

$$\boldsymbol{\theta} := \mathbf{Q}^\mathsf{T}\boldsymbol{\beta} \in \mathbb{R}^{n-1}, \qquad \|\boldsymbol{\theta}\|_2 = \sqrt{\|\boldsymbol{\beta}\|_2^2 - \|\mathbf{P}_{\boldsymbol{\mu}}\boldsymbol{\beta}\|_2^2} \le \sqrt{1 - \rho^2}.$$

We obtain a one-to-one map $\boldsymbol{\beta} \leftrightarrow (\rho, \boldsymbol{\theta})$ in the unit ball. Therefore, we can reparametrize

$$\langle \boldsymbol{x}_i, \boldsymbol{\beta} \rangle + \beta_0 \overset{\mathrm{d}}{=} y_i \rho \| \boldsymbol{\mu} \|_2 - \rho u_i - \langle \boldsymbol{g}_i, \boldsymbol{\theta} \rangle + \beta_0,$$

where $\rho^2 + \| \boldsymbol{\theta} \|_2^2 \leq 1$, and $\{(y_i, u_i, \boldsymbol{g}_i)\}_{i=1}^n$ are i.i.d., each has joint distribution:

$$y_i \perp\!\!\!\perp u_i \perp\!\!\!\perp \boldsymbol{g}_i, \qquad \mathbb{P}(y_i = +1) = 1 - \mathbb{P}(y_i = -1) = \pi, \quad u_i \sim \mathsf{N}(0,1), \quad \boldsymbol{g}_i \sim \mathsf{N}(\boldsymbol{0}, \mathbf{I}_{d-1}).$$

Now denote

$$\boldsymbol{u} = (u_1, \ldots, u_n)^\mathsf{T} \in \mathbb{R}^n, \qquad \boldsymbol{G} = (\boldsymbol{g}_1, \ldots, \boldsymbol{g}_n)^\mathsf{T} \in \mathbb{R}^{n \times (d-1)}.$$

Therefore, $\xi_{n,\kappa,B}^{\prime(0)} := \xi_{n,\kappa,B}^{\prime}$ defined in Eq. (49) can be written as

$$\xi_{n,\kappa,B}^{\prime(0)} = \min_{\substack{\|\boldsymbol{\beta}\|_2 \leq 1 \\ |\beta_0| \leq B}} \max_{\substack{\|\boldsymbol{\lambda}\|_2 \leq 1 \\ \boldsymbol{\lambda} \odot \boldsymbol{y} \geq \boldsymbol{0}}} \frac{1}{\sqrt{d}} \boldsymbol{\lambda}^\mathsf{T} \left( \kappa \mathbf{s}_{\boldsymbol{y}} \odot \boldsymbol{y} - \boldsymbol{X} \boldsymbol{\beta} - \beta_0 \mathbf{1} \right)$$

$$\overset{\mathrm{d}}{=} \min_{\substack{\rho^2 + \|\boldsymbol{\theta}\|_2^2 \leq 1 \\ |\beta_0| \leq B}} \max_{\substack{\|\boldsymbol{\lambda}\|_2 \leq 1 \\ \boldsymbol{\lambda} \odot \boldsymbol{y} \geq \boldsymbol{0}}} \frac{1}{\sqrt{d}} \boldsymbol{\lambda}^\mathsf{T} \left( \kappa \mathbf{s}_{\boldsymbol{y}} \odot \boldsymbol{y} - \rho \| \boldsymbol{\mu} \|_2 \boldsymbol{y} + \rho \boldsymbol{u} + \boldsymbol{G} \boldsymbol{\theta} - \beta_0 \mathbf{1} \right)$$

$$= \min_{\substack{\rho^2 + \|\boldsymbol{\theta}\|_2^2 \leq 1 \\ |\beta_0| \leq B}} \max_{\substack{\|\boldsymbol{\lambda}\|_2 \leq 1 \\ \boldsymbol{\lambda} \odot \boldsymbol{y} \geq \boldsymbol{0}}} \frac{1}{\sqrt{d}} \left( \boldsymbol{\lambda}^\mathsf{T} \boldsymbol{G} \boldsymbol{\theta} + \boldsymbol{\lambda}^\mathsf{T} (\kappa \mathbf{s}_{\boldsymbol{y}} \odot \boldsymbol{y} - \rho \| \boldsymbol{\mu} \|_2 \boldsymbol{y} + \rho \boldsymbol{u} - \beta_0 \mathbf{1}) \right).$$

On the other hand, recall $\xi_{n,\kappa}^{(1)}$ defined in Eq. (50):

$$\xi_{n,\kappa,B}^{\prime(1)} = \min_{\substack{\rho^2 + \|\boldsymbol{\theta}\|_2^2 \leq 1 \\ |\beta_0| \leq B}} \max_{\substack{\|\boldsymbol{\lambda}\|_2 \leq 1 \\ \boldsymbol{\lambda} \odot \boldsymbol{y} \geq \boldsymbol{0}}} \frac{1}{\sqrt{d}} \left( \| \boldsymbol{\lambda} \|_2 \boldsymbol{g}^\mathsf{T} \boldsymbol{\theta} + \| \boldsymbol{\theta} \|_2 \boldsymbol{h}^\mathsf{T} \boldsymbol{\lambda} + \boldsymbol{\lambda}^\mathsf{T} (\kappa \mathbf{s}_{\boldsymbol{y}} \odot \boldsymbol{y} - \rho \| \boldsymbol{\mu} \|_2 \boldsymbol{y} + \rho \boldsymbol{u} - \beta_0 \mathbf{1}) \right).$$

Note that both minimization and maximization above are defined over compact and convex constraint sets, and the objective function in $\xi_{n,\kappa,B}^{\prime(0)}$ is a bilinear in $(\boldsymbol{\theta}, \boldsymbol{\lambda})$ (not $\beta_0$). In addition, $(\boldsymbol{y}, \boldsymbol{u})$ is independent of $\boldsymbol{G}, (\boldsymbol{g}, \boldsymbol{h})$, so we can apply a variant of CGMT (Theorem J.1) by conditioning on $(\boldsymbol{y}, \boldsymbol{u})$, which yields for any $v \in \mathbb{R}$ and $t \geq 0$:

$$\mathbb{P} \left( \xi_{n,\kappa,B}^{\prime(0)} \leq v + t \mid \boldsymbol{y}, \boldsymbol{u} \right) \leq 2 \mathbb{P} \left( \xi_{n,\kappa,B}^{\prime(1)} \leq v + t \mid \boldsymbol{y}, \boldsymbol{u} \right),$$

$$\mathbb{P} \left( \xi_{n,\kappa,B}^{\prime(0)} \geq v - t \mid \boldsymbol{y}, \boldsymbol{u} \right) \leq 2 \mathbb{P} \left( \xi_{n,\kappa,B}^{\prime(1)} \geq v - t \mid \boldsymbol{y}, \boldsymbol{u} \right).$$

Taking expectation over $(\boldsymbol{y}, \boldsymbol{u})$ on both sides of the equation gives for any $v \in \mathbb{R}$ and $t \geq 0$:

$$\mathbb{P} \left( \xi_{n,\kappa,B}^{\prime(0)} \leq v + t \right) \leq 2 \mathbb{P} \left( \xi_{n,\kappa,B}^{\prime(1)} \leq v + t \right), \quad \mathbb{P} \left( \xi_{n,\kappa,B}^{\prime(0)} \geq v - t \right) \leq 2 \mathbb{P} \left( \xi_{n,\kappa,B}^{\prime(1)} \geq v - t \right),$$

which proves Theorem E.2. $\qquad\square$

### E.1.3   STEP 3 — DIMENSION REDUCTION: PROOF OF LEMMA E.3

***Proof of Lemma E.3.*** The expression of $\xi_{n,\kappa,B}^{\prime(1)}$ can be further simplified to

$$\xi_{n,\kappa,B}^{\prime(1)} = \min_{\substack{\rho^2 + \|\boldsymbol{\theta}\|_2^2 \leq 1 \\ |\beta_0| \leq B}} \max_{\substack{\|\boldsymbol{\lambda}\|_2 \leq 1 \\ \boldsymbol{\lambda} \odot \boldsymbol{y} \geq \boldsymbol{0}}} \frac{1}{\sqrt{d}} \left( \| \boldsymbol{\lambda} \|_2 \boldsymbol{g}^\mathsf{T} \boldsymbol{\theta} + \boldsymbol{\lambda}^\mathsf{T} (\kappa \mathbf{s}_{\boldsymbol{y}} \odot \boldsymbol{y} - \rho \| \boldsymbol{\mu} \|_2 \boldsymbol{y} + \rho \boldsymbol{u} + \| \boldsymbol{\theta} \|_2 \boldsymbol{h} - \beta_0 \mathbf{1}) \right)$$

$$\overset{\mathrm{(i)}}{=} \min_{\substack{\rho^2 + \|\boldsymbol{\theta}\|_2^2 \leq 1 \\ |\beta_0| \leq B}} \frac{1}{\sqrt{d}} \left( \boldsymbol{g}^\mathsf{T} \boldsymbol{\theta} + \left\| (\kappa \mathbf{s}_{\boldsymbol{y}} - \rho \| \boldsymbol{\mu} \|_2 + \rho \boldsymbol{u} \odot \boldsymbol{y} + \| \boldsymbol{\theta} \|_2 \boldsymbol{h} \odot \boldsymbol{y} - \beta_0 \boldsymbol{y})_+ \right\|_2 \right)_+$$

$$\overset{\mathrm{(ii)}}{=} \min_{\substack{\rho^2 + r^2 \leq 1 \\ r \geq 0, |\beta_0| \leq B}} \frac{1}{\sqrt{d}} \left( -r \| \boldsymbol{g} \|_2 + \left\| (\kappa \mathbf{s}_{\boldsymbol{y}} - \rho \| \boldsymbol{\mu} \|_2 + \rho \boldsymbol{u} \odot \boldsymbol{y} + r \boldsymbol{h} \odot \boldsymbol{y} - \beta_0 \boldsymbol{y})_+ \right\|_2 \right)_+,$$

where in (i) we use the fact

$$\max_{\|\boldsymbol{\lambda}\|_2 \leq 1, \boldsymbol{\lambda} \geq \boldsymbol{0}} \left( a \| \boldsymbol{\lambda} \|_2 + \boldsymbol{\lambda}^\mathsf{T} \mathbf{b} \right) = \max_{r \in [0,1]} \max_{\|\mathbf{v}\|_2 = 1, \mathbf{v} \geq \boldsymbol{0}} r \left( a + \mathbf{v}^\mathsf{T} \mathbf{b} \right) = \left( \max_{\|\mathbf{v}\|_2 = 1, \mathbf{v} \geq \boldsymbol{0}} \left( a + \mathbf{v}^\mathsf{T} \mathbf{b} \right) \right)_+$$

$$= \left( a + \| (\mathbf{b})_+ \|_2 \right)_+,$$

in (ii) we use Cauchy–Schwarz inequality $\boldsymbol{g}^\mathsf{T}\boldsymbol{\theta} \geq -\|\boldsymbol{\theta}\|_2 \|\boldsymbol{g}\|_2$ and denote $r = \|\boldsymbol{\theta}\|_2$. For convenience, we write the parameter space as $\overline{\Theta}_B := \{(\rho, r, \beta_0) : \rho^2 + r^2 \leq 1, r \geq 0, |\beta_0| \leq B\}$. Now, define

$$\overline{\xi}'^{(1)}_{n,\kappa,B} := \min_{(\rho,r,\beta_0)\in\overline{\Theta}_B} \frac{1}{\sqrt{d}} \left( -r \|\boldsymbol{g}\|_2 + \left\| (\kappa \mathbf{s}_{\boldsymbol{y}} - \rho \|\boldsymbol{\mu}\|_2 + \rho \boldsymbol{u} \odot \boldsymbol{y} + r\boldsymbol{h} \odot \boldsymbol{y} - \beta_0 \boldsymbol{y})_+ \right\|_2 \right)$$

$$= \min_{(\rho,r,\beta_0)\in\overline{\Theta}_B} \left\{ -r \frac{\|\boldsymbol{g}\|_2}{\sqrt{d}} + \sqrt{\frac{n}{d}} \sqrt{\frac{1}{n} \sum_{i=1}^n \left( s(y_i)\kappa - \rho \|\boldsymbol{\mu}\|_2 + \rho u_i y_i + r h_i y_i - \beta_0 y_i \right)_+^2} \right\}$$

$$=: \min_{(\rho,r,\beta_0)\in\overline{\Theta}_B} f^{(1)}_{n,\kappa}(\rho, r, \beta_0),$$

then $\xi'^{(1)}_{n,\kappa,B} = \left( \overline{\xi}'^{(1)}_{n,\kappa,B} \right)_+$. Recall that

$$\overline{\xi}'^{(2)}_{\kappa,B} = \min_{(\rho,r,\beta_0)\in\overline{\Theta}_B} -r + \sqrt{\delta} \left( \mathbb{E}\left[ \left( s(Y)\kappa - \rho \|\boldsymbol{\mu}\|_2 + \rho Y G_1 + r Y G_2 - \beta_0 Y \right)_+^2 \right] \right)^{1/2}$$

$$= \min_{(\rho,r,\beta_0)\in\overline{\Theta}_B} -r + \sqrt{\delta} \left( \mathbb{E}\left[ \left( s(Y)\kappa - \rho \|\boldsymbol{\mu}\|_2 + \rho G_1 + r G_2 - \beta_0 Y \right)_+^2 \right] \right)^{1/2}$$

$$=: \min_{(\rho,r,\beta_0)\in\overline{\Theta}_B} f^{(2)}_{\kappa}(\rho, r, \beta_0),$$

where $Y \perp\!\!\!\perp G_1 \perp\!\!\!\perp G_2$, $\mathbb{P}(Y = +1) = 1 - \mathbb{P}(Y = -1) = \pi$, and $G_1, G_2 \sim \mathsf{N}(0,1)$. We also define

$$\phi^{(1)}_{n,\kappa}(\rho, r, \beta_0) := \frac{1}{n} \sum_{i=1}^n \left( s(y_i)\kappa - \rho \|\boldsymbol{\mu}\|_2 + \rho u_i y_i + r h_i y_i - \beta_0 y_i \right)_+^2 =: \mathbb{E}_n\left[ f(Y, G_1, G_2; \rho, r, \beta_0) \right],$$

$$\phi^{(2)}_{\kappa}(\rho, r, \beta_0) := \mathbb{E}\left[ \left( s(Y)\kappa - \rho \|\boldsymbol{\mu}\|_2 + \rho G_1 Y + r G_2 Y - \beta_0 Y \right)_+^2 \right] =: \mathbb{E}\left[ f(Y, G_1, G_2; \rho, r, \beta_0) \right],$$

where $\mathbb{E}_n[\cdot]$ denotes the expectation over the empirical distribution of $\{(y_i, u_i, h_i)\}_{i=1}^n$. In order to apply the uniform law of large numbers (ULLN), note that

- $\overline{\Theta}_B$ is compact. $(\rho, r, \beta_0) \mapsto f$ is continuous in $\overline{\Theta}_B$ for each $(Y, G_1, G_2)$, and $(Y, G_1, G_2) \mapsto f$ is measurable for each $(\rho, r, \beta_0)$

- $|f(Y, G_1, G_2; \rho, r, \beta_0)| \leq 3 \left( (\kappa\tau + \|\boldsymbol{\mu}\|_2 + B)^2 + G_1^2 + G_2^2 \right)$ for all $(\rho, r, \beta_0) \in \overline{\Theta}_B$ and $\mathbb{E}[G_1^2] = \mathbb{E}[G_2^2] = 1 < \infty$.

Therefore, by ULLN (Newey & McFadden, 1994, Lemma 2.4), we have

$$\sup_{(\rho,r,\beta_0)\in\overline{\Theta}_B} \left| \left( \phi^{(1)}_{n,\kappa}(\rho, r, \beta_0) \right)^{1/2} - \left( \phi^{(2)}_{\kappa}(\rho, r, \beta_0) \right)^{1/2} \right|$$

$$\leq \sup_{(\rho,r,\beta_0)\in\overline{\Theta}_B} \left| \phi^{(1)}_{n,\kappa}(\rho, r, \beta_0) - \phi^{(2)}_{\kappa}(\rho, r, \beta_0) \right|^{1/2} = o_{\mathbb{P}}(1),$$

where the inequality comes from the fact that $x \mapsto \sqrt{x}$ is $1/2$-Hölder continuous on $[0, \infty)$. Then

$$\sup_{(\rho,r,\beta_0)\in\overline{\Theta}_B} \left| f^{(1)}_{n,\kappa}(\rho, r, \beta_0) - f^{(2)}_{\kappa}(\rho, r, \beta_0) \right|$$

$$\leq \sup_{r\in[-1,1]} \left| r - r \frac{\|\boldsymbol{g}\|_2}{\sqrt{d}} \right| + \sup_{(\rho,r,\beta_0)\in\overline{\Theta}_B} \left| \sqrt{\frac{n}{d}} \left( \phi^{(1)}_{n,\kappa}(\rho, r, \beta_0) \right)^{1/2} - \sqrt{\delta} \left( \phi^{(2)}_{\kappa}(\rho, r, \beta_0) \right)^{1/2} \right|$$

$$\leq \left| 1 - \frac{\|\boldsymbol{g}\|_2}{\sqrt{d}} \right| + \sqrt{\frac{n}{d}} \sup_{(\rho,r,\beta_0)\in\overline{\Theta}_B} \left| \left( \phi^{(1)}_{n,\kappa}(\rho, r, \beta_0) \right)^{1/2} - \left( \phi^{(2)}_{\kappa}(\rho, r, \beta_0) \right)^{1/2} \right|$$

$$+ \left| \sqrt{\frac{n}{d}} - \sqrt{\delta} \right| \sup_{(\rho,r,\beta_0)\in\overline{\Theta}_B} \left( \phi^{(2)}_{\kappa}(\rho, r, \beta_0) \right)^{1/2}$$

$$= o_{\mathbb{P}}(1),$$

by using $n/d \to \delta$ and law of large numbers $\|g\|_2^2/(d-1) \xrightarrow{\mathrm{P}} 1$. Finally, since the function $x \mapsto (x)_+$ is 1-Lipschitz, we conclude

$$\left| \xi_{n,\kappa,B}^{\prime(1)} - \left( \overline{\xi}_{\kappa,B}^{\prime(2)} \right)_+ \right| \leq \left| \overline{\xi}_{n,\kappa,B}^{\prime(1)} - \overline{\xi}_{\kappa,B}^{\prime(2)} \right| \leq \sup_{(\rho,r,\beta_0) \in \overline{\Theta}_B} \left| f_{n,\kappa}^{(1)}(\rho,r,\beta_0) - f_{\kappa}^{(2)}(\rho,r,\beta_0) \right| = o_{\mathbb{P}}(1).$$

This completes the proof. $\qquad\square$

### E.1.4  STEP 4 — INVESTIGATION OF THE POSITIVITY: PROOF OF LEMMA E.4

**Proof of Lemma E.4.** We claim $\overline{\xi}_{\kappa,B}^{\prime(2)} = \overline{\xi}_{\kappa}^{(2)}$ when $B$ is large enough. Recall that

$$\overline{\xi}_{\kappa}^{(2)} = \min_{\substack{\rho^2+r^2 \leq 1, r \geq 0 \\ \beta_0 \in \mathbb{R}}} -r + \sqrt{\delta} \left( \mathbb{E}\left[ \left( s(Y)\kappa - \rho\|\boldsymbol{\mu}\|_2 + \rho G_1 + r G_2 - \beta_0 Y \right)_+^2 \right] \right)^{1/2}.$$

Let $(\widetilde{\rho}, \widetilde{r}, \widetilde{\beta}_0)$ be a minimizer above and notice $\widetilde{\rho} G_1 + \widetilde{r} G_2 \overset{\mathrm{d}}{=} \widetilde{R} G$, where $\widetilde{R} = \sqrt{\widetilde{\rho}^2 + \widetilde{r}^2}$ and $G \sim \mathsf{N}(0,1)$. Then

$$\begin{aligned}
\widetilde{\beta}_0 \in \ & \operatorname*{arg\,min}_{\beta_0 \in \mathbb{R}} \mathbb{E}\left[ \left( s(Y)\kappa - \widetilde{\rho}\|\boldsymbol{\mu}\|_2 + \widetilde{R} G - \beta_0 Y \right)_+^2 \right] \\
= \ & \operatorname*{arg\,min}_{\beta_0 \in \mathbb{R}} \left\{ \pi \mathbb{E}\left[ \left( \tau\kappa - \widetilde{\rho}\|\boldsymbol{\mu}\|_2 + \widetilde{R} G - \beta_0 \right)_+^2 \right] + (1-\pi)\mathbb{E}\left[ \left( \kappa - \widetilde{\rho}\|\boldsymbol{\mu}\|_2 + \widetilde{R} G + \beta_0 \right)_+^2 \right] \right\} \\
=: \ & \operatorname*{arg\,min}_{\beta_0 \in \mathbb{R}} g_{\widetilde{\rho},\widetilde{r}}(\beta_0).
\end{aligned}$$

Notice that $g_{\widetilde{\rho},\widetilde{r}}(\beta_0)$ is convex and continuously differentiable, since

$$g_{\widetilde{\rho},\widetilde{r}}'(\beta_0) = -2\pi \mathbb{E}\left[ \left( \tau\kappa - \widetilde{\rho}\|\boldsymbol{\mu}\|_2 + \widetilde{R} G - \beta_0 \right)_+ \right] + 2(1-\pi)\mathbb{E}\left[ \left( \kappa - \widetilde{\rho}\|\boldsymbol{\mu}\|_2 + \widetilde{R} G + \beta_0 \right)_+ \right]$$

is non-decreasing, which is based on the fact that $x \mapsto \mathbb{E}[(G+x)_+]$ is increasing. Then $\widetilde{\beta}_0$ must satisfy $g_{\widetilde{\rho},\widetilde{r}}'(\widetilde{\beta}_0) = 0$. Since $g_{\widetilde{\rho},\widetilde{r}}'(+\infty) = +\infty$, $g_{\widetilde{\rho},\widetilde{r}}'(-\infty) = -\infty$, by our construction in the proof of Theorem E.1, we can choose $B$ large enough such that $\overline{\xi}_{\kappa,B}^{\prime(2)} = \overline{\xi}_{\kappa}^{(2)}$.

We can rewrite $\overline{\xi}_{\kappa}^{(2)}$ as follows by introducing an auxiliary parameter $c$:

$$\overline{\xi}_{\kappa}^{(2)} = \min_{\substack{\rho^2+r^2 \leq 1, r \geq 0, \beta_0 \in \mathbb{R}, \\ \rho^2+r^2+\beta_0^2=c^2, c \geq 0}} -r + \sqrt{\delta} \left( \mathbb{E}\left[ \left( s(Y)\kappa - \rho\|\boldsymbol{\mu}\|_2 + \rho G_1 + r G_2 - \beta_0 Y \right)_+^2 \right] \right)^{1/2},$$

and we also define the following quantity

$$\begin{aligned}
\widetilde{\xi}_{\kappa}^{(2)} &:= \min_{\substack{\rho^2+r^2 \leq 1, r \geq 0, \beta_0 \in \mathbb{R}, \\ \rho^2+r^2+\beta_0^2=c^2, c \geq 0}} \frac{1}{c}\left\{ -r + \sqrt{\delta} \left( \mathbb{E}\left[ \left( s(Y)\kappa - \rho\|\boldsymbol{\mu}\|_2 + \rho G_1 + r G_2 - \beta_0 Y \right)_+^2 \right] \right)^{1/2} \right\} \\
&= \min_{\substack{\rho^2+r^2 \leq 1, r \geq 0, \beta_0 \in \mathbb{R}, \\ \rho^2+r^2+\beta_0^2=c^2, c \geq 0}} -\frac{r}{c} + \sqrt{\delta} \left( \mathbb{E}\left[ \left( s(Y)\frac{\kappa}{c} - \frac{\rho}{c}\|\boldsymbol{\mu}\|_2 + \frac{\rho}{c} G_1 + \frac{r}{c} G_2 - \frac{\beta_0}{c} Y \right)_+^2 \right] \right)^{1/2}.
\end{aligned}$$

Then for any $\kappa > 0$, we have the following observations:

- $\operatorname{sign}(\overline{\xi}_{\kappa}^{(2)}) = \operatorname{sign}(\widetilde{\xi}_{\kappa}^{(2)})$. (Their objective functions differ only by a multiplier $c \geq 0$.[11])

- The minimizer in $\widetilde{\xi}_{\kappa}^{(2)}$ must satisfy $\rho^2 + r^2 = 1$.

  Suppose $(\widetilde{\rho}, \widetilde{r}, \widetilde{\beta}_0, \widetilde{c})$ is a minimizer in $\widetilde{\xi}_{\kappa}^{(2)}$ such that $\widetilde{\rho}^2 + \widetilde{r}^2 < 1$. We can increase $(\widetilde{\rho}, \widetilde{r}, \widetilde{\beta}_0, \widetilde{c})$ proportionally, which results in a better solution. That is, define

  $$\check{\rho} := \frac{1}{\sqrt{\widetilde{\rho}^2+\widetilde{r}^2}}\widetilde{\rho}, \qquad \check{r} := \frac{1}{\sqrt{\widetilde{\rho}^2+\widetilde{r}^2}}\widetilde{r}, \qquad \check{\beta}_0 := \frac{1}{\sqrt{\widetilde{\rho}^2+\widetilde{r}^2}}\widetilde{\beta}_0, \qquad \check{c} := \frac{1}{\sqrt{\widetilde{\rho}^2+\widetilde{r}^2}}\widetilde{c},$$

---

[11] We allow $c = 0$. If $c = 0$, then $\rho = r = \beta_0 = 0$ and the objective value in $\overline{\xi}_{\kappa}^{(2)}$ is $\sqrt{\delta(\pi\tau^2\kappa^2 + (1-\pi)\kappa^2)} > 0$, and the objective value in $\widetilde{\xi}_{\kappa}^{(2)}$ is defined as $+\infty$. Both of them are positive.

then $(\widetilde{\rho}, \widetilde{r}, \widetilde{\beta}_0, \widetilde{c})$ has a smaller objective value (because $r/c$, $\rho/c$, $\beta_0/c$ all remain unchanged, but $\kappa/c$ decreases since $\check{c} > \widetilde{c}$), which contradicts the optimality of $(\widetilde{\rho}, \widetilde{r}, \widetilde{\beta}_0, \widetilde{c})$.

As a consequence, we can simplify

$$\widetilde{\xi}_\kappa^{(2)} = \min_{\substack{\rho \in [-1,1], \beta_0 \in \mathbb{R}, \\ \beta_0^2 = c^2 - 1, c \geq 1}} \frac{1}{c} \left\{ -\sqrt{1-\rho^2} + \sqrt{\delta} \left( \mathbb{E}\left[ \left( s(Y)\kappa - \rho\|\boldsymbol{\mu}\|_2 + \rho G_1 + \sqrt{1-\rho^2}G_2 - \beta_0 Y \right)_+^2 \right] \right)^{1/2} \right\}$$

$$= \min_{\substack{\rho \in [-1,1], \beta_0 \in \mathbb{R}, \\ \beta_0^2 = c^2 - 1, c \geq 1}} \frac{1}{c} \left\{ -\sqrt{1-\rho^2} + \sqrt{\delta} \left( \mathbb{E}\left[ \left( s(Y)\kappa - \rho\|\boldsymbol{\mu}\|_2 + G - \beta_0 Y \right)_+^2 \right] \right)^{1/2} \right\},$$

where $G := \rho G_1 + \sqrt{1-\rho^2}G_2 \sim \mathsf{N}(0,1)$. By the same argument, $\mathrm{sign}(\overline{\xi}_\kappa^{(3)}) = \mathrm{sign}(\widetilde{\xi}_\kappa^{(2)})$, where

$$\overline{\xi}_\kappa^{(3)} = \min_{\substack{\rho \in [-1,1] \\ \beta_0 \in \mathbb{R}}} -\sqrt{1-\rho^2} + \sqrt{\delta} \left( \mathbb{E}\left[ \left( s(Y)\kappa - \rho\|\boldsymbol{\mu}\|_2 + G - \beta_0 Y \right)_+^2 \right] \right)^{1/2}.$$

Therefore, $\mathrm{sign}(\overline{\xi}_\kappa^{(2)}) = \mathrm{sign}(\overline{\xi}_\kappa^{(3)})$.

In order to show $\overline{\xi}_\kappa^{(2)} = \overline{\xi}_\kappa^{(3)}$ when $\overline{\xi}_\kappa^{(2)} < 0$, we define the objective function of $\overline{\xi}_\kappa^{(2)}$ as

$$T_\kappa(\rho, r, \beta_0) := -r + \sqrt{\delta} \left( \mathbb{E}\left[ \left( s(Y)\kappa - \rho\|\boldsymbol{\mu}\|_2 + \rho G_1 + r G_2 - \beta_0 Y \right)_+^2 \right] \right)^{1/2}.$$

Then it suffices to show the minimizer of $T_\kappa$ must satisfy $\rho^2 + r^2 = 1$. Again, suppose $(\widetilde{\rho}, \widetilde{r}, \widetilde{\beta}_0)$ is a minimizer of $T_\kappa$ such that $\widetilde{\rho}^2 + \widetilde{r}^2 < 1$. We can increase $(\widetilde{\rho}, \widetilde{r}, \widetilde{\beta}_0)$ proportionally by defining

$$\check{\rho} := \frac{1}{\sqrt{\widetilde{\rho}^2 + \widetilde{r}^2}}\widetilde{\rho}, \qquad \check{r} := \frac{1}{\sqrt{\widetilde{\rho}^2 + \widetilde{r}^2}}\widetilde{r}, \qquad \check{\beta}_0 := \frac{1}{\sqrt{\widetilde{\rho}^2 + \widetilde{r}^2}}\widetilde{\beta}_0, \qquad \kappa' := \frac{1}{\sqrt{\widetilde{\rho}^2 + \widetilde{r}^2}}\kappa,$$

then

$$0 > \overline{\xi}_\kappa^{(2)} = T_\kappa(\widetilde{\rho}, \widetilde{r}, \widetilde{\beta}_0) > \frac{T_\kappa(\widetilde{\rho}, \widetilde{r}, \widetilde{\beta}_0)}{\sqrt{\widetilde{\rho}^2 + \widetilde{r}^2}} = T_{\kappa'}(\check{\rho}, \check{r}, \check{\beta}_0) > T_\kappa(\check{\rho}, \check{r}, \check{\beta}_0),$$

where the last inequality is because $x \mapsto \mathbb{E}[(G + c_1 x + c_2)_+^2]$ strictly increasing for any $c_1 > 0$ and $c_2 \in \mathbb{R}$, and the fact that $\kappa' > \kappa$. Therefore, a contradiction occurs and we complete the proof. $\square$

### E.1.5 STEP 5 — PHASE TRANSITION, MARGIN CONVERGENCE: PROOFS OF LEMMAS E.5, E.6

***Proof of Lemma E.5.*** We define the following two functions:

$$T_\kappa(\rho, \beta_0) := -\sqrt{1-\rho^2} + \sqrt{\delta} \left( \mathbb{E}\left[ \left( s(Y)\kappa - \rho\|\boldsymbol{\mu}\|_2 + G - \beta_0 Y \right)_+^2 \right] \right)^{1/2},$$

$$F_\kappa(\rho, \beta_0) := -(1 - \rho^2) + \delta\mathbb{E}\left[ \left( s(Y)\kappa - \rho\|\boldsymbol{\mu}\|_2 + G - \beta_0 Y \right)_+^2 \right]$$

$$= \pi\delta\mathbb{E}\left[ \left( G - \rho\|\boldsymbol{\mu}\|_2 - \beta_0 + \kappa\tau \right)_+^2 \right] + (1-\pi)\delta\mathbb{E}\left[ \left( G - \rho\|\boldsymbol{\mu}\|_2 + \beta_0 + \kappa \right)_+^2 \right] + \rho^2 - 1,$$
$$(52)$$

and then

$$\overline{\xi}_\kappa^{(3)} = \min_{\rho \in [-1,1], \beta_0 \in \mathbb{R}} T_\kappa(\rho, \beta_0), \qquad \widetilde{\xi}_\kappa^{(3)} := \min_{\rho \in [-1,1], \beta_0 \in \mathbb{R}} F_\kappa(\rho, \beta_0).$$

Clearly, $\mathrm{sign}(T_\kappa(\rho, \beta_0)) = \mathrm{sign}(F_\kappa(\rho, \beta_0))$ for any $\rho, \beta_0$ and $\mathrm{sign}(\overline{\xi}_\kappa^{(3)}) = \mathrm{sign}(\widetilde{\xi}_\kappa^{(3)})$. Also recall that

$$\delta^*(\kappa) = \max_{\substack{\rho \in [-1,1] \\ \beta_0 \in \mathbb{R}}} H_\kappa(\rho, \beta_0), \qquad H_\kappa(\rho, \beta_0) = \frac{1 - \rho^2}{\mathbb{E}\left[ \left( s(Y)\kappa - \rho\|\boldsymbol{\mu}\|_2 + G - \beta_0 Y \right)_+^2 \right]}.$$

We can see that $H_\kappa(\rho, \beta_0)$ is well-defined since $\mathbb{E}[(s(Y)\kappa - \rho\|\boldsymbol{\mu}\|_2 + G - \beta_0 Y)_+^2]$ is bounded away from zero for any $\rho \in [-1, 1]$ and $\beta_0 \in \mathbb{R} \cup \{\pm\infty\}$.

Since $x \mapsto \mathbb{E}[(G + c_1 x + c_2)_+^2]$ is continuous and strictly increasing for any $c_1 > 0, c_2 \in \mathbb{R}$, it can be shown that both $\kappa \mapsto \overline{\xi}_\kappa^{(3)}$, $\kappa \mapsto \widetilde{\xi}_\kappa^{(3)}$ are continuous strictly increasing, and $\delta^*(\kappa)$ is continuous strictly decreasing (by restricting $\beta_0 : |\beta_0| \leq B$ for some constant $B$ large enough, similar as Step 4, and then use compactness). Therefore, we have the following equivalent definitions of $\kappa^*$:

$$
\begin{aligned}
\kappa^* &:= \sup \left\{ \kappa \in \mathbb{R} : \delta^*(\kappa) \geq \delta \right\} \\
&= \left\{ \kappa \in \mathbb{R} : \delta^*(\kappa) = \delta \right\} = \left\{ \kappa \in \mathbb{R} : \overline{\xi}_\kappa^{(3)} = 0 \right\} = \left\{ \kappa \in \mathbb{R} : \widetilde{\xi}_\kappa^{(3)} = 0 \right\}.
\end{aligned}
\tag{53}
$$

Now we can consider the following two regimes, each with a chain of equivalence:

$$
\begin{aligned}
\delta \leq \delta^*(\kappa) \quad &\overset{\text{(i)}}{\Longleftrightarrow} \quad \kappa \leq \kappa^* \quad \overset{\text{(i)}}{\Longleftrightarrow} \quad \overline{\xi}_\kappa^{(3)}, \widetilde{\xi}_\kappa^{(3)} \leq 0 \quad \overset{\text{(ii)}}{\Longleftrightarrow} \quad \overline{\xi}_\kappa^{(2)} \leq 0 \\
&\overset{\text{(iii)}}{\Longleftrightarrow} \quad \xi'_{n,\kappa,B} \overset{\mathrm{P}}{\to} \left( \overline{\xi}_\kappa^{(2)} \right)_+ = 0 \quad \overset{\text{(iv)}}{\Longleftrightarrow} \quad \mathbb{P}(\xi_{n,\kappa} = 0) \to 1, \\
\delta > \delta^*(\kappa) \quad &\overset{\text{(i)}}{\Longleftrightarrow} \quad \kappa > \kappa^* \quad \overset{\text{(i)}}{\Longleftrightarrow} \quad \overline{\xi}_\kappa^{(3)}, \widetilde{\xi}_\kappa^{(3)} > 0 \quad \overset{\text{(ii)}}{\Longleftrightarrow} \quad \overline{\xi}_\kappa^{(2)} > 0 \\
&\overset{\text{(iii)}}{\Longleftrightarrow} \quad \xi'_{n,\kappa,B} \overset{\mathrm{P}}{\to} \left( \overline{\xi}_\kappa^{(2)} \right)_+ > 0 \quad \overset{\text{(iv)}}{\Longleftrightarrow} \quad \mathbb{P}(\xi_{n,\kappa} > 0) \to 1,
\end{aligned}
\tag{54}
$$

where (i) is from Eq. (53), (ii) is from Theorem E.4, (iii) is from Theorem E.2, E.3, and (iv) is from Theorem E.1. Linear separability considers the special case $\kappa = 0$. From definition Eq. (47), for any $\kappa \leq 0$ we have $\xi_{n,\kappa} = 0$ (by taking $\boldsymbol{\beta} = \mathbf{0}$, $\beta_0 = 0$). Therefore,

- If $\delta < \delta^*(0)$, by Eq. (54) $\kappa^* > 0$ and $\mathbb{P}(\mathcal{E}_{n,\kappa^*}) = \mathbb{P}(\xi_{n,\kappa^*} = 0) \to 1$, which deduces the data is linearly separable with high probability.

- If $\delta > \delta^*(0)$, by Eq. (54) $\kappa^* < 0$ and $\mathbb{P}(\mathcal{E}_{n,\kappa}) = \mathbb{P}(\xi_{n,\kappa} = 0) \to 0$ for any $\kappa > 0$ (as $\kappa \mapsto \xi_{n,\kappa}$ is non-decreasing), which implies the data is not linearly separable with high probability.

$\square$

**Proof of Lemma E.6.** If $\delta < \delta^*(0)$, then $\kappa^* > 0$ and $\overline{\xi}_{\kappa^*}^{(3)} = 0$. According to Eq. (54), for any $\varepsilon > 0$ small enough, we have

$$
\begin{aligned}
\overline{\xi}_{\kappa^* - \varepsilon}^{(3)} < 0 \quad &\Longrightarrow \quad \mathbb{P}(\mathcal{E}_{n,\kappa^* - \varepsilon}) = \mathbb{P}(\xi_{n,\kappa^* - \varepsilon} = 0) \to 1, \\
\overline{\xi}_{\kappa^* + \varepsilon}^{(3)} > 0 \quad &\Longrightarrow \quad \mathbb{P}(\mathcal{E}_{n,\kappa^* + \varepsilon}) = \mathbb{P}(\xi_{n,\kappa^* + \varepsilon} = 0) \to 0.
\end{aligned}
$$

Recall that $\widehat{\kappa}_n = \mathbb{1}_{1 \leq n_+ \leq n-1} \sup\{\kappa \in \mathbb{R} : \xi_{n,\kappa} = 0\}$. By combining these arguments, we can see that $\kappa^* - \varepsilon \leq \widehat{\kappa}_n \leq \kappa^* + \varepsilon$ holds on the event $\mathcal{D}_n^c$, with high probability. This proves $\widehat{\kappa}_n \overset{\mathrm{P}}{\to} \kappa^*$. $\square$

### E.1.6 Convergence of ELD and parameters for $\tau = 1$: Proofs of Lemmas E.7, E.8

In this section, we provide a proof of parameter convergence in Theorem D.1(d) and ELD convergence in (f) for the special case $\tau = 1$. For convenience of notation, we drop the subscripts and simply write $\widehat{\rho} := \widehat{\rho}_n$, $\widehat{\beta}_0 := \widehat{\beta}_{0,n}$. Recall the ELD (well-defined version, i.e., Eq. (48)) and its asymptotics are respectively defined as

$$
\widehat{\nu}_n = \frac{1}{n} \sum_{i=1}^n \delta_{(y_i, \langle \boldsymbol{x}_i, \widehat{\boldsymbol{\beta}} \rangle + \widehat{\beta}_0) \cdot \mathbb{1}\{\mathcal{D}_n^c\}}, \qquad \nu_* = \mathtt{Law}\left( Y, Y \max\{\kappa^*, \rho^* \|\boldsymbol{\mu}\|_2 + G + \beta_0^* Y\} \right).
$$

Here $(\rho^*, \beta_0^*, \kappa^*)$ is defined as the maximizer of Eq. (33), and obviously $\kappa^*$ also satisfies Eq. (53). The uniqueness of $(\rho^*, \beta_0^*)$ will be given by Theorem E.8. Analogous to the proof of (Montanari & Zhou, 2022, Theorem 4.6), by using the theory of projection pursuit therein, we have the following results.

**Lemma E.7** (ELD and parameter convergence). *Consider $\tau = 1$. As $n, d \to \infty$, we have*

$$
W_2\left( \widehat{\nu}_n, \nu_* \right) \overset{\mathrm{P}}{\to} 0.
$$

*The convergence of $\widehat{\rho} \overset{\mathrm{P}}{\to} \rho^*$ and $\widehat{\beta}_0 \overset{\mathrm{P}}{\to} \beta_0^*$ are followed by continuity and convexity of $H_\kappa$ in Eq. (28).*

*Proof.* Our proof primarily follows the setup in (Montanari & Zhou, 2022, Section 4.1) and techniques in (Montanari & Zhou, 2022, Section 4.3). Recall that we can rewrite $\boldsymbol{x}_i = y_i \boldsymbol{\mu} + \boldsymbol{z}_i$, where $\boldsymbol{z}_i \overset{\text{i.i.d.}}{\sim} \mathsf{N}(\boldsymbol{0}, \mathbf{I}_d)$ and $y_i \perp\!\!\!\perp \boldsymbol{z}_i$. Using notation from (Montanari & Zhou, 2022), $\mathbb{P}(y_i = 1 \mid \boldsymbol{z}_i) = \varphi(\boldsymbol{\mu}_0^\mathsf{T} \boldsymbol{z}_i)$, where $\boldsymbol{\mu}_0 = \boldsymbol{\mu} / \|\boldsymbol{\mu}\|_2$ and $\varphi(x) \equiv \pi$ is a constant function. Recall that we reparametrize $\widehat{\rho} = \boldsymbol{\mu}_0^\mathsf{T} \widehat{\boldsymbol{\beta}}$. Now, define random variables with joint distribution

$$Y \perp\!\!\!\perp G \perp\!\!\!\perp Z, \quad \mathbb{P}(Y = +1 \mid G) = 1 - \mathbb{P}(Y = -1 \mid G) = \varphi(G) \equiv \pi, \quad G, Z \sim \mathsf{N}(0, 1).$$

Let $(Y, G, Z) \perp\!\!\!\perp \widehat{\boldsymbol{\beta}}$. According to the definition in (Montanari & Zhou, 2022, Lemma 4.2), we have

$$\mathtt{Law}\left(Y, \boldsymbol{\mu}_0^\mathsf{T} \widehat{\boldsymbol{\beta}} \cdot G + \sqrt{1 - (\boldsymbol{\mu}_0^\mathsf{T} \widehat{\boldsymbol{\beta}})^2} \cdot Z\right) = \mathtt{Law}\left(Y, \widehat{\rho} G + \sqrt{1 - \widehat{\rho}^2} Z\right) = \mathtt{Law}(Y, Z).$$

Therefore, by using (Montanari & Zhou, 2022, Theorem 4.3), for any $\varepsilon, \eta > 0$, with high probability we have

$$W_2^{(\eta)}\left(\frac{1}{n} \sum_{i=1}^{n} \delta_{(y_i, \langle \boldsymbol{z}_i, \widehat{\boldsymbol{\beta}} \rangle)}, \mathtt{Law}(Y, Z)\right) \leq \frac{\sqrt{1 - \widehat{\rho}^2}}{\sqrt{\delta}} + \varepsilon,$$

where $W_2^{(\eta)}$ is the $\eta$-constrained $W_2$ distance (Montanari & Zhou, 2022, Definition 4.1). Formally, for any $\eta > 0$, the $\eta$-constrained $W_2$ distance between any two probability measures $P$ and $Q$ in $\mathbb{R}^d$ is defined by

$$W_2^{(\eta)}(P, Q) := \left(\inf_{\gamma \in \Gamma^{(\eta)}(P, Q)} \int_{\mathbb{R}^d \times \mathbb{R}^d} \|\boldsymbol{x} - \boldsymbol{y}\|_2^2 \, \gamma(\mathrm{d}\boldsymbol{x} \times \mathrm{d}\boldsymbol{y})\right)^{1/2},$$

where $\Gamma^{(\eta)}(P, Q)$ denotes the set of all couplings $\gamma$ of $P$ and $Q$ which satisfy

$$\left(\int_{\mathbb{R}^d \times \mathbb{R}^d} |\langle \mathbf{e}_1, \boldsymbol{x} - \boldsymbol{y} \rangle|^2 \gamma(\mathrm{d}\boldsymbol{x} \times \mathrm{d}\boldsymbol{y})\right)^{1/2} \leq \eta, \tag{55}$$

where $\mathbf{e}_1 = (1, 0, \ldots, 0)^\mathsf{T}$.

The following proof is analogous to the proof of (Montanari & Zhou, 2022, Theorem 4.6). We show the convergence of logit margins $W_2(\widehat{\mathcal{L}}_n, \mathcal{L}_*) \overset{\mathrm{P}}{\to} 0$ first, where

$$\widehat{\mathcal{L}}_n := \frac{1}{n} \sum_{i=1}^{n} \delta_{y_i(\langle \boldsymbol{x}_i, \widehat{\boldsymbol{\beta}} \rangle + \widehat{\beta}_0)}, \qquad \mathcal{L}_* := \mathtt{Law}\left(\max\{\kappa^*, \rho^* \|\boldsymbol{\mu}\|_2 + G + \beta_0^* Y\}\right). \tag{56}$$

Throughout this subsection, all the expectations (including the one in $H_\kappa$) are conditional on $\{(y_i, \boldsymbol{z}_i)\}_{i=1}^{n}$, which will be denoted as $\mathbb{E}_{\cdot|n}[\cdot]$. Now, let

$$\frac{1}{n} \sum_{i=1}^{n} \delta_{(y_i, \langle \boldsymbol{z}_i, \widehat{\boldsymbol{\beta}} \rangle)} =: \mathtt{Law}(Y', Z'),$$

then by definition in Eq. (55) and the same arguments in the proof of (Montanari & Zhou, 2022, Theorem 4.6), there exists a coupling $(Y, Z, Y', Z')$ and a sufficiently small $\eta$ ($\eta < \varepsilon^2 / 4$), such that

$$\left(\mathbb{E}_{\cdot|n}\left[(Y - Y')^2\right]\right)^{1/2} \leq \eta, \qquad \left(\mathbb{E}_{\cdot|n}\left[(YZ - Y'Z')^2\right]\right)^{1/2} \leq \frac{\sqrt{1 - \widehat{\rho}^2}}{\sqrt{\delta}} + 2\varepsilon \tag{57}$$

holds with high probability. We can express the empirical distribution of logit margins Eq. (56) as

$$\widehat{\mathcal{L}}_n = \frac{1}{n} \sum_{i=1}^{n} \delta_{y_i \langle \boldsymbol{z}_i, \widehat{\boldsymbol{\beta}} \rangle + \widehat{\rho} \|\boldsymbol{\mu}\|_2 + y_i \widehat{\beta}_0} = \mathtt{Law}\Big(\underbrace{Y'Z' + \widehat{\rho} \|\boldsymbol{\mu}\|_2 + \widehat{\beta}_0 Y'}_{=:V}\Big). \tag{58}$$

For convenience, denote $\widehat{V} := YZ + \widehat{\rho} \|\boldsymbol{\mu}\|_2 + \widehat{\beta}_0 Y$, then with high probability we have

$$\begin{aligned}
\left(\mathbb{E}_{\cdot|n}\left[(V - \widehat{V})^2\right]\right)^{1/2} &\overset{\text{(i)}}{\leq} \left(\mathbb{E}_{\cdot|n}\left[(YZ - Y'Z')^2\right]\right)^{1/2} + \left(\mathbb{E}_{\cdot|n}\left[(Y - Y')^2\right]\right)^{1/2} |\widehat{\beta}_0| \\
&\overset{\text{(ii)}}{\leq} \frac{\sqrt{1 - \widehat{\rho}^2}}{\sqrt{\delta}} + 2\varepsilon + \eta B \\
&\overset{\text{(iii)}}{\leq} \frac{\sqrt{1 - \widehat{\rho}^2}}{\sqrt{\delta}} + 3\varepsilon,
\end{aligned} \tag{59}$$

where (i) follows from Minkowski inequality, (ii) uses Eq. (57) and $|\widehat{\beta}_0| \leq B$ from Theorem E.1, by recalling that $\delta < \delta^*(0)$ and the data is linearly separable with high probability, while in (iii) we choose $\eta < \min\{\varepsilon^2/4, \varepsilon/B\}$. According to $\widehat{\kappa}_n \xrightarrow{\mathrm{P}} \kappa^*$ from Theorem E.6, we know that

$$\lim_{n\to\infty} \mathbb{P}\left(y_i(\langle\widehat{\boldsymbol{\beta}}, \boldsymbol{x}_i\rangle + \widehat{\beta}_0) \geq \kappa^* - \varepsilon, \forall\, i \in [n]\right) = 1.$$

Then by definition of $V$ in Eq. (58), with high probability we have

$$V \geq \kappa^* - \varepsilon, \qquad \text{almost surely.} \tag{60}$$

Now, recall $\delta = \delta^*(\kappa^*) = H_{\kappa^*}(\rho^*, \beta_0^*)$ by Eq. (53), where $(\rho^*, \beta_0^*) = \arg\min_{\rho\in[-1,1], \beta_0\in\mathbb{R}} H_{\kappa^*}(\rho, \beta_0)$. Therefore,

$$\left(\mathbb{E}_{\cdot|n}\big[(V-\widehat{V})^2\big]\right)^{1/2} \leq \frac{\sqrt{1-\widehat{\rho}^2}}{\sqrt{\delta}} + 3\varepsilon = \frac{\sqrt{1-\widehat{\rho}^2}}{\sqrt{H_{\kappa^*}(\rho^*, \beta_0^*)}} + 3\varepsilon \tag{61}$$

holds with high probability. For $\rho \in [-1, 1], \beta_0 \in \mathbb{R}$, let us define

$$h_{\kappa^*}^*(\rho, \beta_0) := \frac{1}{\sqrt{H_{\kappa^*}(\rho, \beta_0)}} - \frac{1}{\sqrt{H_{\kappa^*}(\rho^*, \beta_0^*)}}.$$

Note that $h_{\kappa^*}^*(\rho, \beta_0) \geq 0$. Hence, Eq. (61) implies that (reminding $\tau = 1$) with high probability

$$\left(\mathbb{E}_{\cdot|n}\big[(V-\widehat{V})^2\big]\right)^{1/2} \leq \sqrt{1-\widehat{\rho}^2}\left(\sqrt{\frac{1}{H_{\kappa^*}(\widehat{\rho}, \widehat{\beta}_0)}} - h_{\kappa^*}^*(\widehat{\rho}, \widehat{\beta}_0)\right) + 3\varepsilon$$

$$= \left(\mathbb{E}_{\cdot|n}\Big[\big(\kappa^* - \widehat{\rho}\|\boldsymbol{\mu}\|_2 + G - \widehat{\beta}_0 Y\big)_+^2\Big]\right)^{1/2} - \sqrt{1-\widehat{\rho}^2}\cdot h_{\kappa^*}^*(\widehat{\rho}, \widehat{\beta}_0) + 3\varepsilon$$

$$\overset{\text{(i)}}{=} \left(\mathbb{E}_{\cdot|n}\big[(\kappa^* - \widehat{V})_+^2\big]\right)^{1/2} - \sqrt{1-\widehat{\rho}^2}\cdot h_{\kappa^*}^*(\widehat{\rho}, \widehat{\beta}_0) + 3\varepsilon,$$

which can be further written as (with high probability)

$$\left(\mathbb{E}_{\cdot|n}\big[(V-\widehat{V})^2\big]\right)^{1/2} + \sqrt{1-\widehat{\rho}^2}\cdot h_{\kappa^*}^*(\widehat{\rho}, \widehat{\beta}_0) \leq \left(\mathbb{E}_{\cdot|n}\big[(\kappa^* - \widehat{V})_+^2\big]\right)^{1/2} + 3\varepsilon$$

$$\overset{\text{(ii)}}{\leq} \left(\mathbb{E}_{\cdot|n}\big[(\kappa^* - \varepsilon - \widehat{V})_+^2\big]\right)^{1/2} + 4\varepsilon. \tag{62}$$

In the derivation above, equation (i) follows from $\widehat{V} = YZ + \widehat{\rho}\|\boldsymbol{\mu}\|_2 + \widehat{\beta}_0 Y \overset{\text{d}}{=} -G + \widehat{\rho}\|\boldsymbol{\mu}\|_2 + \widehat{\beta}_0 Y$ when conditioning on $\{(y_i, \boldsymbol{z}_i)\}_{i=1}^n$, and (ii) follows from the fact that

$$\frac{\mathrm{d}}{\mathrm{d}\kappa}\left(\mathbb{E}_{\cdot|n}\big[(\kappa - \widehat{V})_+^2\big]\right)^{1/2} = \frac{\mathbb{E}_{\cdot|n}\big[(\kappa - \widehat{V})_+^2\big]}{\left(\mathbb{E}_{\cdot|n}\big[(\kappa - \widehat{V})_+^2\big]\right)^{1/2}} \leq 1.$$

Besides, by using Eq. (60) and exactly the same arguments in the proof of (Montanari & Zhou, 2022, Theorem 4.6), we can show that with high probability,

$$\mathbb{E}_{\cdot|n}\left[\big(V - \max\{\kappa^* - \varepsilon, \widehat{V}\}\big)^2\right] \leq \mathbb{E}_{\cdot|n}\big[(V-\widehat{V})^2\big] - \mathbb{E}_{\cdot|n}\big[(\kappa^* - \varepsilon - \widehat{V})_+^2\big]. \tag{63}$$

Combining Eq. (63) with (62) gives the following implications:

- Eq. (63) implies
$$\mathbb{E}_{\cdot|n}\big[(\kappa^* - \varepsilon - \widehat{V})_+^2\big] \leq \mathbb{E}_{\cdot|n}\big[(V-\widehat{V})^2\big].$$
Plugging this into Eq. (62) yields that with high probability,
$$\sqrt{1-\widehat{\rho}^2}\cdot h_{\kappa^*}^*(\widehat{\rho}, \widehat{\beta}_0) \leq 4\varepsilon,$$
i.e., $\sqrt{1-\widehat{\rho}^2}\cdot h_{\kappa^*}^*(\widehat{\rho}, \widehat{\beta}_0) \xrightarrow{\mathrm{P}} 0$. Note that if $|\rho| \to 1$ (i.e., $\sqrt{1-\rho^2} = o_\varepsilon(1)$), the quantity
$$\sqrt{1-\rho^2}\cdot h_{\kappa^*}^*(\rho, \beta_0) = \left(\mathbb{E}\left[\big(\kappa^* - \rho\|\boldsymbol{\mu}\|_2 + G - \beta_0 Y\big)_+^2\right]\right)^{1/2} - \frac{\sqrt{1-\rho^2}}{\sqrt{H_{\kappa^*}(\rho^*, \beta_0^*)}}$$
is bounded away from 0, for any $\beta_0 \in \mathbb{R}\cup\{\pm\infty\}$. Therefore, we must have $h_{\kappa^*}^*(\widehat{\rho}, \widehat{\beta}_0) \xrightarrow{\mathrm{P}} 0$. By Theorem E.8 (proof is deferred to the end of this subsection), we know $h_{\kappa^*}^*(\rho, \beta_0) \geq 0$ for all $\rho \in [-1, 1], \beta_0 \in \mathbb{R}$, and $(\rho, \beta_0) \to (\rho^*, \beta_0^*)$ if and only if $h_{\kappa^*}^*(\rho, \beta_0) \to 0$. Hence, we conclude
$$(\widehat{\rho}, \widehat{\beta}_0) \xrightarrow{\mathrm{P}} (\rho^*, \beta_0^*),$$
which gives parameter convergence.

- Let

$$I := \big(\mathbb{E}_{\cdot|n}\big[(V - \widehat{V})^2\big]\big)^{1/2}, \qquad II := \big(\mathbb{E}_{\cdot|n}\big[(\kappa^* - \varepsilon - \widehat{V})_+^2\big]\big)^{1/2}.$$

Then Eq. (62) implies $I - II \le 4\varepsilon$, and we also have (for $\varepsilon > 0$ small enough)

$$\begin{aligned}
II &\le |\kappa^* - \varepsilon| + \big(\mathbb{E}_{\cdot|n}\big[\widehat{V}^2\big]\big)^{1/2} \le \kappa^* + \big(\mathbb{E}_{\cdot|n}\big[(G + \widehat{\rho}\,\|\boldsymbol{\mu}\|_2 + \widehat{\beta}_0 Y)^2\big]\big)^{1/2} \\
&\le \kappa^* + \big(\mathbb{E}[G^2]\big)^{1/2} + |\widehat{\rho}|\,\|\boldsymbol{\mu}\|_2 + |\widehat{\beta}_0| \\
&\le \kappa^* + 1 + \|\boldsymbol{\mu}\|_2 + B,
\end{aligned}$$

by using Minkowski inequality and $|\widehat{\beta}_0| \le B$ (with high probability) from Theorem E.1. Based on these results and Eq. (63), with high probability, we have

$$\begin{aligned}
\mathbb{E}_{\cdot|n}\Big[\big(V - \max\{\kappa^* - \varepsilon, \widehat{V}\}\big)^2\Big] &\le I^2 - II^2 = (I - II)(I - II + 2II) \\
&\le 4\varepsilon\big(4\varepsilon + 2(\kappa^* + 1 + \|\boldsymbol{\mu}\|_2 + B)\big) \\
&\le C\varepsilon,
\end{aligned}$$

where $C \in (0, \infty)$ is some constant depending on $(\pi, \|\boldsymbol{\mu}\|_2, \delta)$ (through $\kappa^*, B$). Therefore, by recalling $\widehat{V} \stackrel{\mathrm{d}}{=} G + \widehat{\rho}\,\|\boldsymbol{\mu}\|_2 + \widehat{\beta}_0 Y$, we obtain that with high probability,

$$W_2\Big(\widehat{\mathcal{L}}_n, \texttt{Law}\big(\max\{\kappa^* - \varepsilon, G + \widehat{\rho}\,\|\boldsymbol{\mu}\|_2 + \widehat{\beta}_0 Y\}\big)\Big) \le \sqrt{C\varepsilon}. \tag{64}$$

As a consequence, the following holds with high probability:

$$\begin{aligned}
&W_2\big(\widehat{\mathcal{L}}_n, \mathcal{L}_*\big) \\
&= W_2\Big(\widehat{\mathcal{L}}_n, \texttt{Law}\big(\max\{\kappa^*, G + \rho^*\,\|\boldsymbol{\mu}\|_2 + \beta_0^* Y\}\big)\Big) \\
&\le W_2\Big(\widehat{\mathcal{L}}_n, \texttt{Law}\big(\max\{\kappa^* - \varepsilon, G + \widehat{\rho}\,\|\boldsymbol{\mu}\|_2 + \widehat{\beta}_0 Y\}\big)\Big) \\
&\quad + W_2\Big(\texttt{Law}\big(\max\{\kappa^* - \varepsilon, G + \widehat{\rho}\,\|\boldsymbol{\mu}\|_2 + \widehat{\beta}_0 Y\}\big), \texttt{Law}\big(\max\{\kappa^*, G + \widehat{\rho}\,\|\boldsymbol{\mu}\|_2 + \widehat{\beta}_0 Y\}\big)\Big) \\
&\quad + W_2\Big(\texttt{Law}\big(\max\{\kappa^*, G + \widehat{\rho}\,\|\boldsymbol{\mu}\|_2 + \widehat{\beta}_0 Y\}\big), \texttt{Law}\big(\max\{\kappa^*, G + \rho^*\,\|\boldsymbol{\mu}\|_2 + \beta_0^* Y\}\big)\Big) \\
&\le \sqrt{C\varepsilon} + \varepsilon + o_\varepsilon(1) = o_\varepsilon(1),
\end{aligned}$$

where in the last inequality, we use that (i) the result from Eq. (64), (ii) the fact that the mapping $\kappa \mapsto \max\{\kappa, G + \widehat{\rho}\,\|\boldsymbol{\mu}\|_2 + \widehat{\beta}_0 Y\}$ is 1-Lipschitz, and (iii) the consequence of $(\widehat{\rho}, \widehat{\beta}_0) \xrightarrow{\mathrm{P}} (\rho^*, \beta_0^*)$ and $|\widehat{\rho}| \le 1$, $|\widehat{\beta}_0| \le B$ (with high probability).

Now we prove the convergence of ELD. Denote $\widehat{\mathcal{L}}_n =: \texttt{Law}(L')$, $\mathcal{L}_* =: \texttt{Law}(L)$, where $(L, L')$ is a coupling such that

$$\big(\mathbb{E}_{\cdot|n}\big[(L - L')^2\big]\big)^{1/2} = o_\varepsilon(1). \tag{65}$$

Therefore, for some constants $C_1, C_2 > 0$, with high probability, we have

$$\begin{aligned}
W_2\big(\widehat{\nu}_n, \nu_*\big) &\le \big(\mathbb{E}_{\cdot|n}\big[(Y - Y')^2\big]\big)^{1/2} + \big(\mathbb{E}_{\cdot|n}\big[(YL - Y'L')^2\big]\big)^{1/2} \\
&\stackrel{\text{(i)}}{\le} \eta + \big(\mathbb{E}_{\cdot|n}\big[(YL - Y'L)^2\big]\big)^{1/2} + \big(\mathbb{E}_{\cdot|n}\big[(Y'L - Y'L')^2\big]\big)^{1/2} \\
&\stackrel{\text{(ii)}}{\le} \eta + C_1\big(\mathbb{E}_{\cdot|n}\big[(Y - Y')^2\big]\big)^{1/4}\big(\mathbb{E}[L^4]\big)^{1/4} + \big(\mathbb{E}_{\cdot|n}\big[(L - L')^2\big]\big)^{1/2} \\
&\stackrel{\text{(iii)}}{\le} \eta + C_2\sqrt{\eta} + o_\varepsilon(1) \stackrel{\text{(iv)}}{\le} o_\varepsilon(1),
\end{aligned}$$

where in (i) we use Eq. (57) and Minkowski inequality, in (ii) use Cauchy–Schwarz inequality and $Y, Y' \in \{\pm 1\}$, in (iii) use Eq. (57) and (65), while in (iv) recall that $\eta < \min\{\varepsilon^2/4, \varepsilon/B\} = o_\varepsilon(1)$. By taking $\varepsilon \to 0$, we can show that $W_2\big(\widehat{\nu}_n, \nu_*\big) \xrightarrow{\mathrm{P}} 0$ for $\tau = 1$. This completes the proof. $\qquad\square$

Finally, we prove the following technical lemma.

**Lemma E.8.** *For any fixed $\kappa \in \mathbb{R}$ and $\tau > 0$, the function $H_\kappa(\rho, \beta_0)$ in Eq. (28) admits a unique maximizer $(\rho^*(\kappa), \beta_0^*(\kappa)) \in [0, 1) \times \mathbb{R}$.*

*Proof.* For simplicity, write $\rho^* := \rho^*(\kappa)$, $\beta_0^* := \beta_0^*(\kappa)$. First, note that

$$H_\kappa(\rho, \beta_0) = \frac{1 - \rho^2}{\mathbb{E}\left[\left(s(Y)\kappa - \rho \|\boldsymbol{\mu}\|_2 + G - \beta_0 Y\right)_+^2\right]} \leq \frac{1}{\mathbb{E}\left[\left(s(Y)\kappa - \|\boldsymbol{\mu}\|_2 + G - \beta_0 Y\right)_+^2\right]},$$

which converges to 0 as $\beta_0 \to \pm\infty$. Moreover, $H_\kappa(-\rho, \beta_0) < H_\kappa(\rho, \beta_0)$ for any $\rho \in (0, 1]$. Therefore, $H_\kappa(\rho, \beta_0)$ must have a maximizer $(\rho^*, \beta_0^*) \in [0, 1] \times \mathbb{R}$. Further, $\rho^* \in [0, 1)$ since $H_\kappa(1, \beta_0) \equiv 0$. We prove the uniqueness of $(\rho^*, \beta_0^*)$ by contradiction. For future convenience, we denote $H_{\max} := H_\kappa(\rho^*, \beta_0^*)$. Assume that there exist $(\rho_1, \beta_{0,1})$ and $(\rho_2, \beta_{0,2})$ such that $(\rho_1, \beta_{0,1}) \neq (\rho_2, \beta_{0,2})$, and

$$H_\kappa(\rho_1, \beta_{0,1}) = H_\kappa(\rho_2, \beta_{0,2}) = H_{\max},$$

which implies

$$G_\kappa(\rho_1, \beta_{0,1}) = \frac{\sqrt{1 - \rho_1^2}}{\sqrt{H_{\max}}}, \qquad G_\kappa(\rho_2, \beta_{0,2}) = \frac{\sqrt{1 - \rho_2^2}}{\sqrt{H_{\max}}},$$

where we define

$$G_\kappa(\rho, \beta_0) := \left(\mathbb{E}\left[\left(s(Y)\kappa - \rho \|\boldsymbol{\mu}\|_2 + G - \beta_0 Y\right)_+^2\right]\right)^{1/2}.$$

Similar to (Montanari et al., 2023, Lemma 6.3), we can show that $G_\kappa$ is strictly convex. Hence,

$$G_\kappa\left(\frac{\rho_1 + \rho_2}{2}, \frac{\beta_{0,1} + \beta_{0,2}}{2}\right) < \frac{1}{2}\left(G_\kappa(\rho_1, \beta_{0,1}) + G_\kappa(\rho_2, \beta_{0,2})\right)$$

$$= \frac{1}{\sqrt{H_{\max}}} \frac{1}{2}\left(\sqrt{1 - \rho_1^2} + \sqrt{1 - \rho_2^2}\right)$$

$$\leq \frac{1}{\sqrt{H_{\max}}} \sqrt{1 - \left(\frac{\rho_1 + \rho_2}{2}\right)^2},$$

where in the last line we use the concavity of the mapping $x \mapsto \sqrt{1 - x^2}$. It finally follows that

$$H_\kappa\left(\frac{\rho_1 + \rho_2}{2}, \frac{\beta_{0,1} + \beta_{0,2}}{2}\right) > H_{\max},$$

a contradiction. This concludes the proof. $\square$

### E.1.7 COMPLETING THE PROOF OF THEOREM D.1

***Proof of Theorem D.1.*** **(a)** is established by Theorem E.5.

**(b):** Notice the definition of $(\rho^*, \beta_0^*, \kappa^*)$ we used in our proof (Section E.1.5, E.1.6) is based on Eq. (33). It suffices to show the equivalence of two optimization problems Eq. (31) and (33). Now we fix $\rho, \beta_0$ in Eq. (31) and $X := \rho\|\boldsymbol{\mu}\|_2 + G + Y\beta_0$. Then Eq. (31) can be written as

$$\begin{aligned} \underset{\kappa > 0, \, \xi \in \mathcal{L}^2}{\text{maximize}} \quad & \kappa, \\ \text{subject to} \quad & X + \sqrt{1 - \rho^2}\xi \geq s(Y)\kappa, \qquad \mathbb{E}[\xi^2] \leq 1/\delta. \end{aligned} \tag{66}$$

Note that it can be written as a convex optimization problem, and it is infeasible if $\rho = \pm 1$ (since $X$ has support $\mathbb{R}$). Take $\rho \in (-1, 1)$. According to the Karush–Kuhn–Tucker (KKT) and Slater's conditions for variational problems (Zalinescu, 2002, Theorem 2.9.2), $(\kappa, \xi)$ is the solution to Eq. (66) if and only if it satisfies the following for some $\Lambda \in \mathcal{L}^1, \Lambda \geq 0$ (a.s.) and $\nu \geq 0$:

$$-1 + \mathbb{E}[s(Y)\Lambda] = 0, \qquad -\sqrt{1 - \rho^2}\Lambda + 2\nu\xi = 0 \text{ (a.s.)},$$

$$\nu\left(\mathbb{E}[\xi^2] - \delta^{-1}\right) = 0, \qquad \Lambda\left(s(Y)\kappa - X - \sqrt{1 - \rho^2}\xi\right) = 0 \text{ (a.s.)}.$$

Clearly $\nu > 0$ (otherwise, $\Lambda = 0$ a.s., a contradiction). Consider the following two cases:

- On the event $\{s(Y(\omega))\kappa - X(\omega) < 0\}$, we obtain $s(Y(\omega))\kappa - X(\omega) - \sqrt{1-\rho^2}\xi(\omega) < 0$, which implies $\Lambda(\omega) = 0$. Therefore, $\xi(\omega) = 0$.

- On the event $\{s(Y(\omega))\kappa - X(\omega) > 0\}$, we obtain $\sqrt{1-\rho^2}\xi(\omega) \geq s(Y(\omega))\kappa - X(\omega) > 0$, which implies $\xi(\omega) > 0$. Therefore, $\Lambda(\omega) > 0$, and thus $s(Y(\omega))\kappa - X(\omega) - \sqrt{1-\rho^2}\xi(\omega) = 0$.

(Note $\mathbb{P}(s(Y)\kappa - X = 0) = 0$.) By combining these, we get $\sqrt{1-\rho^2}\xi = (s(Y)\kappa - X)_+$. This proves Eq. (32). Plug in it into Eq. (31) gives Eq. (33). The proof of $\rho^* \in (0,1)$ and its independence of $\tau$ is given by Theorem E.9 in Section E.2.

This concludes the proof of part (b). $\qquad\qquad\qquad\qquad\qquad\qquad\qquad\qquad\qquad\qquad\qquad\square$

**(c), $\delta < \delta^*(0)$:** We show that $\widehat{\kappa}_n \xrightarrow{\mathrm{P}} \kappa^*$ in Theorem E.6 can be strengthened to $\widehat{\kappa}_n \xrightarrow{\mathcal{L}^2} \kappa^*$. To this end, we show that $\widehat{\kappa}_n^2$ is uniformly integrable (u.i.). Recall that $\kappa(\widehat{\boldsymbol{\beta}}_n, \widehat{\beta}_{0,n}) \geq 0$ and

$$\kappa(\widehat{\boldsymbol{\beta}}_n, \widehat{\beta}_{0,n}) = \min_{i \in [n]} \widetilde{y}_i\big(\langle \boldsymbol{x}_i, \widehat{\boldsymbol{\beta}}_n \rangle + \widehat{\beta}_{0,n}\big) = \min_{i \in [n]} \widetilde{y}_i\big(y_i\langle \boldsymbol{\mu}, \widehat{\boldsymbol{\beta}}_n \rangle + \langle \boldsymbol{z}_i, \widehat{\boldsymbol{\beta}}_n \rangle + \widehat{\beta}_{0,n}\big)$$

$$= \min\left\{ \min_{i:y_i=+1} \tau^{-1}\big(\langle \boldsymbol{\mu}, \widehat{\boldsymbol{\beta}}_n \rangle + \langle \boldsymbol{z}_i, \widehat{\boldsymbol{\beta}}_n \rangle + \widehat{\beta}_{0,n}\big), \min_{i:y_i=-1}\big(\langle \boldsymbol{\mu}, \widehat{\boldsymbol{\beta}}_n \rangle - \langle \boldsymbol{z}_i, \widehat{\boldsymbol{\beta}}_n \rangle - \widehat{\beta}_{0,n}\big)\right\}.$$

Hence, on the event $\mathcal{D}_n^c$ (non-degenerate case), we have $\widehat{\kappa}_n = \kappa(\widehat{\boldsymbol{\beta}}_n, \widehat{\beta}_{0,n})$ and it can be bounded by the average from each class:

$$\kappa(\widehat{\boldsymbol{\beta}}_n, \widehat{\beta}_{0,n}) \leq \tau^{-1}\big(\langle \boldsymbol{\mu}, \widehat{\boldsymbol{\beta}}_n \rangle + \langle \overline{\boldsymbol{z}}_n^+, \widehat{\boldsymbol{\beta}}_n \rangle + \widehat{\beta}_{0,n}\big) := \overline{\kappa}_n^+,$$

$$\kappa(\widehat{\boldsymbol{\beta}}_n, \widehat{\beta}_{0,n}) \leq \langle \boldsymbol{\mu}, \widehat{\boldsymbol{\beta}}_n \rangle - \langle \overline{\boldsymbol{z}}_n^-, \widehat{\boldsymbol{\beta}}_n \rangle - \widehat{\beta}_{0,n} := \overline{\kappa}_n^-,$$

where

$$\overline{\boldsymbol{z}}_n^+ := \frac{1}{n_+} \sum_{i:y_i=+1} \boldsymbol{z}_i, \qquad \overline{\boldsymbol{z}}_n^- := \frac{1}{n_-} \sum_{i:y_i=-1} \boldsymbol{z}_i.$$

Combine these two bounds and apply Cauchy–Schwarz inequality, we obtain

$$\kappa(\widehat{\boldsymbol{\beta}}_n, \widehat{\beta}_{0,n}) \leq \frac{\tau\overline{\kappa}_n^+ + \overline{\kappa}_n^-}{\tau + 1} = \frac{2}{\tau + 1}\left(\langle \boldsymbol{\mu}, \widehat{\boldsymbol{\beta}}_n \rangle + \left\langle \frac{\overline{\boldsymbol{z}}_n^+ - \overline{\boldsymbol{z}}_n^-}{2}, \widehat{\boldsymbol{\beta}}_n \right\rangle\right) \leq \frac{2}{\tau + 1}\big(\|\boldsymbol{\mu}\|_2 + \|\widetilde{\boldsymbol{z}}_n\|_2\big),$$

where

$$\widetilde{\boldsymbol{z}}_n := \frac{\overline{\boldsymbol{z}}_n^+ - \overline{\boldsymbol{z}}_n^-}{2}, \quad \widetilde{\boldsymbol{z}}_n \mid \boldsymbol{y} \sim \mathsf{N}\left(\boldsymbol{0}, \frac{1}{4}\Big(\frac{1}{n_+} + \frac{1}{n_-}\Big)\mathbf{I}_d\right).$$

Therefore,

$$0 \leq \widehat{\kappa}_n = \kappa(\widehat{\boldsymbol{\beta}}_n, \widehat{\beta}_{0,n})\mathbb{1}_{1\leq n_+ \leq n-1} \leq \frac{2}{\tau + 1}\big(\|\boldsymbol{\mu}\|_2 + \|\widetilde{\boldsymbol{z}}_n\|_2\,\mathbb{1}_{1\leq n_+ \leq n-1}\big).$$

In order to prove $\widehat{\kappa}_n^2$ is u.i., it suffices to show that $\|\widetilde{\boldsymbol{z}}_n\|_2^2\,\mathbb{1}_{1\leq n_+ \leq n-1}$ is u.i.. Next, we prove this by establishing that $\|\widetilde{\boldsymbol{z}}_n\|_2^2\,\mathbb{1}_{1\leq n_+ \leq n-1}$ converges in $\mathcal{L}^1$. It requires two steps (by Scheffé's Lemma):

$$\mathbb{E}\left[\|\widetilde{\boldsymbol{z}}_n\|_2^2\,\mathbb{1}_{1\leq n_+ \leq n-1}\right] \to \frac{1}{4\delta}\left(\frac{1}{\pi} + \frac{1}{1-\pi}\right), \tag{67a}$$

$$\|\widetilde{\boldsymbol{z}}_n\|_2^2\,\mathbb{1}_{1\leq n_+ \leq n-1} \xrightarrow{\mathrm{P}} \frac{1}{4\delta}\left(\frac{1}{\pi} + \frac{1}{1-\pi}\right). \tag{67b}$$

For Eq. (67a), Observe

$$\mathbb{E}\left[\|\widetilde{\boldsymbol{z}}_n\|_2^2\,\mathbb{1}_{1\leq n_+ \leq n-1}\right] = \mathbb{E}\left[\mathbb{E}\big[\|\widetilde{\boldsymbol{z}}_n\|_2^2\,\mathbb{1}_{1\leq n_+ \leq n-1} \mid \boldsymbol{y}\big]\right]$$

$$= \mathbb{E}\left[\frac{d}{4}\Big(\frac{1}{n_+} + \frac{1}{n_-}\Big)\mathbb{1}_{1\leq n_+ \leq n-1}\right] = \frac{d}{4n}\mathbb{E}\left[\Big(\frac{n}{n_+} + \frac{n}{n_-}\Big)\mathbb{1}_{1\leq n_+ \leq n-1}\right].$$

To evaluate the expected value, note that (by law of large numbers)

$$\frac{n}{n_+}\mathbb{1}_{1\leq n_+ \leq n-1} \leq \frac{2n}{n_+ + 1}, \qquad \frac{n}{n_+}\mathbb{1}_{1\leq n_+ \leq n-1} \xrightarrow{\mathrm{P}} \frac{1}{\pi}, \qquad \frac{2n}{n_+ + 1} \xrightarrow{\mathrm{P}} \frac{2}{\pi}.$$

A classical result (Chao & Strawderman, 1972) gives

$$\lim_{n\to\infty} \mathbb{E}\left[\frac{2n}{n_+ + 1}\right] = \lim_{n\to\infty} \frac{2n\left(1 - (1-\pi)^{n+1}\right)}{(n+1)\pi} = \frac{2}{\pi}.$$

So $\frac{2n}{n_+ + 1} \xrightarrow{\mathcal{L}^1} \frac{2}{\pi}$, which implies $\frac{2n}{n_+ + 1}$ is u.i., and so is $\frac{n}{n_+}\mathbb{1}_{1\le n_+\le n-1}$. Therefore, by Vitali convergence theorem, we have $\frac{n}{n_+}\mathbb{1}_{1\le n_+\le n-1} \xrightarrow{\mathcal{L}^1} \frac{1}{\pi}$. Similar arguments give $\frac{n}{n_-}\mathbb{1}_{1\le n_+\le n-1} \xrightarrow{\mathcal{L}^1} \frac{1}{1-\pi}$. Hence

$$\lim_{n\to\infty}\mathbb{E}\left[\|\widetilde{\boldsymbol{z}}_n\|_2^2\,\mathbb{1}_{1\le n_+\le n-1}\right] = \lim_{n\to\infty}\frac{d}{4n}\cdot\lim_{n\to\infty}\mathbb{E}\left[\left(\frac{n}{n_+} + \frac{n}{n_-}\right)\mathbb{1}_{1\le n_+\le n-1}\right] = \frac{1}{4\delta}\left(\frac{1}{\pi} + \frac{1}{1-\pi}\right).$$

For Eq. (67b), notice that $\|\widetilde{\boldsymbol{z}}_n\|_2^2 \mid \boldsymbol{y} \sim a_n\chi_d^2$, where $a_n = \frac{1}{4}(\frac{1}{n_+} + \frac{1}{n_-})$. By concentration inequality (e.g., Theorem J.3(a)), we have

$$\mathbb{P}\left(\left|\|\widetilde{\boldsymbol{z}}_n\|_2^2 - da_n\right| \ge \varepsilon \mid \boldsymbol{y}\right) \le 2\exp\left(-c\min\left\{\frac{\varepsilon^2}{da_n^2}, \frac{\varepsilon}{a_n}\right\}\right) = o_{\mathbb{P}}(1),$$

where $c > 0$ is a constant, $a_n = o_{\mathbb{P}}(1)$, $da_n \xrightarrow{\mathrm{P}} \frac{1}{4\delta}(\frac{1}{\pi} + \frac{1}{1-\pi})$. By taking expectation on both sides and using bounded convergence theorem, we have $\|\widetilde{\boldsymbol{z}}_n\|_2^2 - da_n = o_{\mathbb{P}}(1)$. Then we get Eq. (67b).

Finally, Eq. (67a) and (67b) imply that $\|\widetilde{\boldsymbol{z}}_n\|_2^2\,\mathbb{1}_{1\le n_+\le n-1}$ converges in $\mathcal{L}^1$, and thus is u.i.. So $\widehat{\kappa}_n^2$ is also u.i.. By Vitali convergence theorem, convergence in probability of $\widehat{\kappa}_n$ can be strengthen to $\mathcal{L}^2$ convergence.

This concludes the proof of part (c) for $\delta < \delta^*(0)$. $\qquad\qquad\square$

**(c), $\delta > \delta^*(0)$:** For non-separable regime, we cannot work with $\xi_{n,\kappa}$ in Eq. (47) to show a negative margin, since $\widehat{\kappa}_n \ge 0$ always holds (by taking $\boldsymbol{\beta} = 0$, $\beta_0 = 0$). To this end, we define

$$\Xi_{n,\kappa} := \min_{\substack{\|\boldsymbol{\beta}\|_2=1 \\ \beta_0\in\mathbb{R}}} \frac{1}{\sqrt{d}}\left\|\left(\kappa\mathbf{s}_{\boldsymbol{y}} - \boldsymbol{y}\odot\boldsymbol{X}\boldsymbol{\beta} - \beta_0\boldsymbol{y}\right)_+\right\|_2,$$

which replace the constraint $\|\boldsymbol{\beta}\|_2 \le 1$ in $\xi_{n,\kappa}$ by $\|\boldsymbol{\beta}\|_2 = 1$. Here we define the margin as

$$\widetilde{\kappa}_n := \sup\{\kappa \in \mathbb{R} : \Xi_{n,\kappa} = 0\}. \tag{68}$$

Note that $\widetilde{\kappa}_n = \widehat{\kappa}_n$ on separable data, but $\widetilde{\kappa}_n$ is allowed to be negative. Then our goal is to show

$$\widetilde{\kappa}_n \le -\overline{\kappa} \tag{69}$$

holds for some $\overline{\kappa} > 0$ with high probability. Then followed by the proof outline at the beginning of Section E.1, we can also define a series of random variables in a similar way:

$$\Xi'_{n,\kappa,B} := \min_{\substack{\|\boldsymbol{\beta}\|_2=1 \\ |\beta_0|\le B}} \max_{\substack{\|\boldsymbol{\lambda}\|_2\le 1 \\ \boldsymbol{\lambda}\odot\boldsymbol{y}\ge 0}} \frac{1}{\sqrt{d}}\boldsymbol{\lambda}^{\mathsf{T}}\left(\kappa\mathbf{s}_{\boldsymbol{y}}\odot\boldsymbol{y} - \boldsymbol{X}\boldsymbol{\beta} - \beta_0\mathbf{1}\right),$$

$$\Xi'^{(1)}_{n,\kappa,B} := \min_{\substack{\rho^2+\|\boldsymbol{\theta}\|_2^2=1 \\ |\beta_0|\le B}} \max_{\substack{\|\boldsymbol{\lambda}\|_2\le 1 \\ \boldsymbol{\lambda}\odot\boldsymbol{y}\ge 0}} \frac{1}{\sqrt{d}}\left(\|\boldsymbol{\lambda}\|_2\,\boldsymbol{g}^{\mathsf{T}}\boldsymbol{\theta} + \|\boldsymbol{\theta}\|_2\,\boldsymbol{h}^{\mathsf{T}}\boldsymbol{\lambda} + \boldsymbol{\lambda}^{\mathsf{T}}\left(\kappa\mathbf{s}_{\boldsymbol{y}}\odot\boldsymbol{y} - \rho\|\boldsymbol{\mu}\|_2\,\boldsymbol{y} + \rho\boldsymbol{u} - \beta_0\mathbf{1}\right)\right),$$

$$\overline{\Xi}'^{(2)}_{\kappa,B} := \min_{\substack{\rho^2+r^2=1,\,r\ge 0 \\ |\beta_0|\le B}} -r + \sqrt{\delta}\left(\mathbb{E}\left[\left(s(Y)\kappa - \rho\|\boldsymbol{\mu}\|_2 + \rho G_1 + rG_2 - \beta_0 Y\right)_+^2\right]\right)^{1/2},$$

where the constraints $\|\boldsymbol{\beta}\|_2 \le 1$, $\rho^2 + \|\boldsymbol{\theta}\|_2^2 \le 1$, and $\rho^2 + r^2 \le 1$ in $\xi'_{n,\kappa,B}$, $\xi'^{(1)}_{n,\kappa,B}$, and $\overline{\xi}'^{(2)}_{\kappa,B}$ all become equality constraints. Then we follow the same arguments in Step 1—5 (Theorem E.1—E.6).

- Analogous to the proof of Theorem E.1, we have $\left|\mathbb{P}\big(\Xi_{n,\kappa} = 0\big) - \mathbb{P}\big(\Xi'_{n,\kappa,B} = 0\big)\right| \to 0$.

- Analogous to the proof of Theorem E.2, we can apply CGMT Theorem J.1 to connect $\Xi'_{n,\kappa,B}$ with $\Xi'^{(1)}_{n,\kappa,B}$.

$$\mathbb{P}\left(\Xi'_{n,\kappa,B} \le t\right) \le 2\,\mathbb{P}\left(\Xi'^{(1)}_{n,\kappa,B} \le t\right).$$

Here we only get a one-sided inequality since $\{(\rho,\boldsymbol{\theta}) : \rho^2 + \|\boldsymbol{\theta}\|_2^2 = 1\}$ is non-convex.

- Analogous to the proof of Theorem E.3, we have $\Xi_{n,\kappa,B}^{\prime(1)} \xrightarrow{\text{P}} (\overline{\Xi}_{\kappa,B}^{\prime(2)})_+$.

- Notice that the optimal $r$ in $\overline{\Xi}_{\kappa,B}^{\prime(2)}$ must be nonnegative. Hence, by substituting $r = \sqrt{1-\rho^2}$, we have $\overline{\Xi}_{\kappa,B}^{\prime(2)} = \overline{\xi}_\kappa^{(3)}$ (Eq. (51)) for some $B > 0$ large enough.

Recall that in the proof of Theorem E.5 and E.6, if $\delta > \delta^*(0)$, then there exists a $\kappa_0 < 0$, such that $\overline{\xi}_{\kappa_0}^{(3)} = 0$. According to Eq. (54), for any $\varepsilon > 0$ small enough, by using above relations, we have

$$\overline{\xi}_{\kappa_0+\varepsilon}^{(3)} = \overline{\Xi}_{\kappa_0+\varepsilon,B}^{\prime(2)} > 0 \quad \Longrightarrow \quad \Xi_{n,\kappa_0+\varepsilon,B}^{\prime(1)} \xrightarrow{\text{P}} (\overline{\xi}_{\kappa_0+\varepsilon}^{(3)})_+ > 0$$

$$\Longrightarrow \quad \Xi_{n,\kappa_0+\varepsilon,B}' > 0 \text{ w.h.p.} \quad \Longrightarrow \quad \Xi_{n,\kappa_0+\varepsilon} > 0 \text{ w.h.p.}.$$

By Eq. (68), $\widetilde{\kappa}_n < \kappa_0 + \varepsilon < 0$ holds with high probability (by taking $\varepsilon$ to be sufficiently small), which proves Eq. (69).

This concludes the proof of part (c) for $\delta > \delta^*(0)$. $\qquad\qquad\qquad\qquad\qquad\qquad\square$

**(d), (f):** We have shown parameter and ELD convergence for the case $\tau = 1$ in Theorem E.7. Now for any $\tau \geq 1$, denote $\widehat{\boldsymbol{\beta}}_n(\tau), \widehat{\beta}_{0,n}(\tau), \widehat{\kappa}_n(\tau)$ as the max-margin solution to Eq. (46), and define

$$\widehat{\rho}_n(\tau) := \left\langle \widehat{\boldsymbol{\beta}}_n(\tau), \frac{\boldsymbol{\mu}}{\|\boldsymbol{\mu}\|_2} \right\rangle.$$

Similarly, denote $\rho^*(\tau), \beta_0^*(\tau), \kappa^*(\tau)$ as the optimal solution to Eq. (33). By Theorem C.1,

- We have

$$\widehat{\rho}_n(\tau) = \widehat{\rho}_n(1), \qquad \widehat{\beta}_{0,n}(\tau) = \widehat{\beta}_{0,n}(1) + \frac{\tau-1}{\tau+1}\widehat{\kappa}_n(1). \tag{70}$$

- We can write

$$\widehat{\nu}_n = \frac{1}{n}\sum_{i=1}^n \delta_{\left(y_i, \langle \boldsymbol{x}_i, \widehat{\boldsymbol{\beta}}_n\rangle + \widehat{\beta}_{0,n}(\tau)\right)\mathbb{1}\{\mathcal{D}_n^c\}} =: \text{Law}\left(Y'\mathbb{1}_{\mathcal{D}_n^c}, \left(\langle \boldsymbol{x}', \widehat{\boldsymbol{\beta}}_n\rangle + \widehat{\beta}_{0,n}(\tau)\right)\mathbb{1}_{\mathcal{D}_n^c}\right)$$

$$= \text{Law}\left(Y'\mathbb{1}_{\mathcal{D}_n^c}, \left(\langle \boldsymbol{x}', \widehat{\boldsymbol{\beta}}_n\rangle + \widehat{\beta}_{0,n}(1)\right)\mathbb{1}_{\mathcal{D}_n^c} + \frac{\tau-1}{\tau+1}\widehat{\kappa}_n(1)\right). \tag{71}$$

Besides, according to Theorem E.10,

- We have

$$\rho^*(\tau) = \rho^*(1), \qquad \beta_0^*(\tau) = \beta_0^*(1) + \frac{\tau-1}{\tau+1}\kappa^*(1). \tag{72}$$

- We can also write

$$\nu_* = \text{Law}\left(Y, Y\max\{s(Y)\kappa^*(\tau), G + \rho^*\|\boldsymbol{\mu}\|_2 + \beta_0^*(\tau)Y\}\right)$$

$$= \text{Law}\left(Y, Y\max\{\kappa^*(1), G + \rho^*\|\boldsymbol{\mu}\|_2 + \beta_0^*(1)Y\} + \frac{\tau-1}{\tau+1}\kappa^*(1)\right). \tag{73}$$

We have shown $\widehat{\kappa}_n(1) \xrightarrow{\mathcal{L}^2} \kappa^*(1)$ and $\widehat{\beta}_{0,n}(1) \xrightarrow{\text{P}} \beta_0^*(1)$ in Theorem E.7. Then by continuous mapping theorem, comparing Eq. (70) and (72), it follows that $\widehat{\beta}_{0,n}(\tau) \xrightarrow{\text{P}} \beta_0^*(\tau)$ for any $\tau > 0$.

In Theorem E.7, we have shown that $W_2\big(\widehat{\nu}_n, \nu_*\big) = o_\varepsilon(1)$ for $\tau = 1$ with high probability, i.e.,

$$W_2\left(\text{Law}\left(Y'\mathbb{1}_{\mathcal{D}_n^c}, \underbrace{\left(\langle \boldsymbol{x}', \widehat{\boldsymbol{\beta}}_n\rangle + \widehat{\beta}_{0,n}(1)\right)\mathbb{1}_{\mathcal{D}_n^c}}_{=:U_n}\right), \text{Law}\left(Y, \underbrace{Y\max\{\kappa^*(1), G + \rho^*\|\boldsymbol{\mu}\|_2 + \beta_0^*(1)Y\}}_{=:U^*}\right)\right) = o_\varepsilon(1).$$

$$\tag{74}$$

Then there exists a coupling $(Y', Y, U_n, U^*)$ such that, with high probability,

$$W_2(\widehat{\nu}_n, \nu_*) \le \left(\mathbb{E}_{\cdot|n}\left[(Y'\mathbb{1}_{\mathcal{D}_n^c} - Y)^2\right]\right)^{\frac{1}{2}} + \left(\mathbb{E}_{\cdot|n}\left[\left(U_n - U^* + \frac{\tau-1}{\tau+1}\widehat{\kappa}_n(1) - \frac{\tau-1}{\tau+1}\kappa^*(1)\right)^2\right]\right)^{\frac{1}{2}}$$

$$\overset{\text{(i)}}{\le} \left(\mathbb{E}_{\cdot|n}\left[(Y'\mathbb{1}_{\mathcal{D}_n^c} - Y)^2\right]\right)^{\frac{1}{2}} + \left(\mathbb{E}_{\cdot|n}\left[(U_n - U^*)^2\right]\right)^{\frac{1}{2}} + \frac{\tau-1}{\tau+1}\left(\mathbb{E}_{\cdot|n}\left[(\widehat{\kappa}_n(1) - \kappa^*(1))^2\right]\right)^{\frac{1}{2}}$$

$$\overset{\text{(ii)}}{=} o_\varepsilon(1),$$

where in (i) we use Minkowski inequality, while in (ii) we use $\widehat{\kappa}_n(1) \xrightarrow{\mathcal{L}^2} \kappa^*(1)$ in (c) and Eq. (74). By taking $\varepsilon \to 0$, we can show that $W_2(\widehat{\nu}_n, \nu_*) \xrightarrow{\text{p}} 0$ holds for any $\tau > 0$.

For TLD convergence, we give a proof of $\widehat{\nu}_n^{\text{test}} \xrightarrow{w} \nu_*^{\text{test}}$. Write $x_{\text{new}} = y_{\text{new}}\mu + z_{\text{new}}$, $z_{\text{new}} \sim \mathsf{N}(\mathbf{0}, \mathbf{I}_d)$, and recall $\widehat{\nu}_n^{\text{test}} = \mathtt{Law}(y^{\text{new}}, \langle x^{\text{new}}, \widehat{\beta}_n \rangle + \widehat{\beta}_{0,n})$. Let $G \sim \mathsf{N}(0,1)$ and $G \perp\!\!\!\perp y^{\text{new}}$, then

$$\langle x^{\text{new}}, \widehat{\beta}_n \rangle + \widehat{\beta}_{0,n} = \langle y^{\text{new}}\mu + z^{\text{new}}, \widehat{\beta}_n \rangle + \widehat{\beta}_{0,n}$$

$$= y^{\text{new}}\widehat{\rho}_n \|\mu\|_2 + \langle z^{\text{new}}, \widehat{\beta}_n \rangle + \widehat{\beta}_{0,n}$$

$$\xrightarrow{\text{d}} y^{\text{new}}(\rho^* \|\mu\|_2 + G + y^{\text{new}}\beta_0^*),$$

where in the last line we use Slutsky's theorem and $y^{\text{new}} \perp\!\!\!\perp (y^{\text{new}} z^{\text{new}}, \widehat{\beta}_n, \widehat{\beta}_{0,n})$.

This concludes the proof of part (d) and (f). □

**(e):** In (f) above, we showed that

$$\widehat{f}(x_{\text{new}}) = \langle x_{\text{new}}, \widehat{\beta}_n \rangle + \widehat{\beta}_{0,n} \xrightarrow{\text{d}} y_{\text{new}}\rho^* \|\mu\|_2 + G + \beta_0^*.$$

Therefore, by bounded convergence theorem, the errors have their limits

$$\lim_{n \to \infty} \text{Err}_{+,n} = \mathbb{P}\left(+\rho^* \|\mu\|_2 + G + \beta_0^* \le 0\right) = \Phi\left(-\rho^* \|\mu\|_2 - \beta_0^*\right),$$

$$\lim_{n \to \infty} \text{Err}_{-,n} = \mathbb{P}\left(-\rho^* \|\mu\|_2 + G + \beta_0^* > 0\right) = \Phi\left(-\rho^* \|\mu\|_2 + \beta_0^*\right).$$

This concludes the proof of part (e). □

Finally, we complete the proof of Theorem D.1. □

### E.2 ANALYSIS OF THE ASYMPTOTIC OPTIMIZATION PROBLEM: PROOF OF LEMMA E.9

We provide an analysis of the low dimensional asymptotic optimization problem Eq. (33) in this subsection. The conclusion below has been used in the proofs of Theorem D.1(b), (d) and (f). It will be also used in Section G to obtain monotonicity results.

For $G \sim \mathsf{N}(0,1)$ and $t \in \mathbb{R}$, we define two auxiliary functions

$$g_1(t) := \mathbb{E}\left[(G+t)_+\right], \qquad g_2(t) := \mathbb{E}\left[(G+t)_+^2\right]. \tag{75}$$

Clearly both $g_1$ and $g_2$ are strictly increasing mappings from $\mathbb{R}$ to $\mathbb{R}_{>0}$. Then $g := g_2 \circ g_1^{-1}$ is also strictly increasing. The following lemma shows that the limiting parameters $(\rho^*, \beta_0^*, \kappa^*)$ defined in Theorem D.1 can be characterized by the following system of equations, involving $g$ and $g_1^{-1}$.

**Lemma E.9** (Analysis of the asymptotic problem). *In the separable regime $\delta < \delta^*(0)$, $(\rho^*, \beta_0^*, \kappa^*)$ is the unique solution to the system of equations*

$$\pi\delta \cdot g\left(\frac{\rho}{2\pi \|\mu\|_2 \delta}\right) + (1-\pi)\delta \cdot g\left(\frac{\rho}{2(1-\pi) \|\mu\|_2 \delta}\right) = 1 - \rho^2, \tag{76a}$$

$$-\beta_0 + \kappa\tau = \rho \|\mu\|_2 + g_1^{-1}\left(\frac{\rho}{2\pi \|\mu\|_2 \delta}\right), \tag{76b}$$

$$\beta_0 + \kappa = \rho \|\mu\|_2 + g_1^{-1}\left(\frac{\rho}{2(1-\pi) \|\mu\|_2 \delta}\right), \tag{76c}$$

*where $\rho^* \in (0,1)$ does not depend on $\tau$ and $\kappa^* > 0$.*

*Proof.* Recall that in the proof of Theorem E.5 and E.7, we established that $(\rho^*, \beta_0^*, \kappa^*)$ is the unqiue solution to

$$\underset{\rho \in [0,1], \beta_0 \in \mathbb{R}, \kappa \in \mathbb{R}}{\text{maximize}} \quad \kappa,$$
$$\text{subject to} \quad H_\kappa(\rho, \beta_0) \geq \delta.$$

Let $F(\rho, \beta_0, \kappa) := F_\kappa(\rho, \beta_0)$, where $F_\kappa$ is defined in Eq. (52). Then the above optimization problem is equivalent to

$$\underset{\rho \in [0,1], \beta_0 \in \mathbb{R}, \kappa \in \mathbb{R}}{\text{maximize}} \quad \kappa,$$
$$\text{subject to} \quad F(\rho, \beta_0, \kappa) \leq 0,$$

Note $F$ is convex (since $x \mapsto (x)_+^2$ is a convex map, and expectation preserves convexity). Setting $\partial_\rho F = 0$ and $\partial_{\beta_0} F = 0$, we obtain the first-order conditions satisfied by $(\rho^*, \beta_0^*)$:

$$\mathbb{E}\left[\left(G - \rho \|\boldsymbol{\mu}\|_2 - \beta_0 + \kappa\tau\right)_+\right] = \frac{\rho}{2\pi \|\boldsymbol{\mu}\|_2 \delta},$$
$$\mathbb{E}\left[\left(G - \rho \|\boldsymbol{\mu}\|_2 + \beta_0 + \kappa\right)_+\right] = \frac{\rho}{2(1-\pi) \|\boldsymbol{\mu}\|_2 \delta}. \tag{77}$$

Moreover, we have $\delta^*(\kappa^*) = \delta$ and thus $F(\rho, \kappa, \beta_0) = 0$ at $(\rho^*, \kappa^*, \beta_0^*)$, which leads to

$$\pi\delta\mathbb{E}\left[\left(G - \rho \|\boldsymbol{\mu}\|_2 - \beta_0 + \kappa\tau\right)_+^2\right] + (1-\pi)\delta\mathbb{E}\left[\left(G - \rho \|\boldsymbol{\mu}\|_2 + \beta_0 + \kappa\right)_+^2\right] = 1 - \rho^2. \tag{78}$$

Using $g_1, g_2$ defined in Eq. (75), the first-order conditions Eq. (77) can be rewritten as

$$g_1\left(-\rho \|\boldsymbol{\mu}\|_2 - \beta_0 + \kappa\tau\right) = \frac{\rho}{2\pi \|\boldsymbol{\mu}\|_2 \delta},$$
$$g_1\left(-\rho \|\boldsymbol{\mu}\|_2 + \beta_0 + \kappa\right) = \frac{\rho}{2(1-\pi) \|\boldsymbol{\mu}\|_2 \delta}. \tag{79}$$

Similarly, we recast Eq. (78) into

$$\pi\delta g_2\left(-\rho \|\boldsymbol{\mu}\|_2 - \beta_0 + \kappa\tau\right) + (1-\pi)\delta g_2\left(-\rho \|\boldsymbol{\mu}\|_2 + \beta_0 + \kappa\right) = 1 - \rho^2. \tag{80}$$

By combining Eq. (79) and (80), we get Eq. (76a). Eq. (76b) and (76c) directly come from Eq. (79).

Note that function $g : \mathbb{R}_{>0} \to \mathbb{R}_{>0}$ satisfies $g(0^+) = 0$. As $\rho$ varies from 0 to 1, the L.H.S. of Eq. (76a) increases from 0 to a positive number while the R.H.S. decays to 0, which guarantees the existence and uniqueness of $\rho^* > 0$. Since Eq. (76a) does not depend on $\tau$ and $\kappa^*$, we know that $\rho^*$ does not depend on $\tau$ and $\kappa^*$. This concludes the proof. □

In parallel to Theorem C.1 for the original non-asymptotic problem, we provide the following similar result on the asymptotic problem Eq. (33).

**Corollary E.10.** *In the separable regime $\delta < \delta^*(0)$, let $(\rho^*(\tau), \beta_0^*(\tau), \kappa^*(\tau))$ be the optimal solution to Eq.* (33) *under hyperparameter $\tau$. Then*

$$\rho^*(\tau) = \rho^*(1), \qquad \beta_0^*(\tau) = \beta_0^*(1) + \frac{\tau - 1}{\tau + 1}\kappa^*(1), \qquad \kappa^*(\tau) = \frac{2}{\tau + 1}\kappa^*(1). \tag{81}$$

*Proof.* Conclusion for $\rho^*$ is already shown in Theorem E.9. For $\beta_0^*$ and $\kappa^*$, note that the R.H.S. of Eq. (76b) and (76c) are constants under $\rho = \rho^*$ (depending on $\pi$, $\|\boldsymbol{\mu}\|_2$ and $\delta$). Then we have

$$-\beta_0^*(\tau) + \kappa^*(\tau)\tau = -\beta_0^*(1) + \kappa^*(1),$$
$$\beta_0^*(\tau) + \kappa^*(\tau) = \beta_0^*(1) + \kappa^*(1).$$

Combining these two equations gives the expression of $\beta_0^*(\tau), \kappa^*(\tau)$ in terms of $\beta_0^*(1), \kappa^*(1)$ as in Eq. (81), completing the proof. □

### E.3 Proof of Proposition D.2

***Proof of Proposition D.2.*** We can prove a more general result by replacing $\mathcal{L}_*^{\text{test}}$ with $\mu$ and $\mathcal{L}_*$ with $\nu := \text{Law}(\max\{\kappa^*, X\})$, where $X \sim \mu$ and $\mu$ is any probability measure with atomless (continuous) CDF $F_\mu$. As a special case, in Theorem D.2 we consider $\mu$ as a mixture of two Gaussian distributions, and the cost function $c(x, y) = (x - y)^2$.

We now prove the general statement. Note the CDF of $\nu$ has the form

$$F_\nu(t) := \begin{cases} F_\mu(t), & \text{if } t < \kappa^*, \\ 1, & \text{if } t \geq \kappa^*. \end{cases}$$

According to the optimal transport theory (Santambrogio, 2015, Theorem 2.5), the unique (also monotone) optimal transport map from $\mu$ to $\nu$ is given by $\text{T}^* := F_\nu^- \circ F_\mu$, where $F_\nu^-$ is the quantile function of $\nu$:

$$F_\nu^-(x) = \inf\{t \in \mathbb{R} : F_\nu(t) \geq x\} = \begin{cases} F_\mu^{-1}(x), & \text{if } x < F_\mu(\kappa^*), \\ \kappa^*, & \text{if } x \geq F_\mu(\kappa^*). \end{cases}$$

Then we have $\text{T}^*(x) := F_\nu^-\big(F_\mu(x)\big) = \max\{\kappa^*, x\}$, which concludes the proof. $\qquad\square$

## F LOGIT DISTRIBUTION FOR NON-SEPARABLE DATA: PROOFS FOR SECTION D.1.2

### F.1 PROOF OF THEOREM D.3

Throughout this section, we assume the loss function $\ell : \mathbb{R} \to \mathbb{R}_{\geq 0}$ is non-increasing, strictly convex, and twice differentiable. Based on these assumptions, we establish the following properties of $\ell$.

**Lemma F.1.** *Let $\ell \in C^1(\mathbb{R})$ be a nonnegative, non-increasing, and strictly convex function. Then*

*(a) $\ell$ is strictly decreasing.*

*(b) $\ell(-\infty) = +\infty$ and $\ell(+\infty) = \underline{\ell}$ for some $\underline{\ell} \in [0, +\infty)$.*

*Proof.* Notice that $\ell'(u) \leq 0$ (by non-increasing) and $\ell'(u)$ is strictly increasing (by strict convexity), which implies that $\ell'(u) < 0$ for all $u \in \mathbb{R}$ and hence deduces part (a). For part (b), the limits $\lim_{u \to \pm\infty} \ell(u)$ are well-defined, and $\ell(+\infty) = \underline{\ell}$ for some $\underline{\ell} \in [0, +\infty)$ since $\ell$ is monotone and bounded from below. It remains to show $\ell(-\infty) = +\infty$.

Assume $\ell(-\infty) = \overline{\ell} < \infty$ by contradiction. By convexity, we have $\ell(u) \leq \frac{1}{2}(\ell(2u) + \ell(0))$ for any $u \in \mathbb{R}$. Taking $u \to -\infty$ on both sides yields $\overline{\ell} \leq \frac{1}{2}(\overline{\ell} + \ell(0))$, hence $\overline{\ell} \leq \ell(0)$, which contradicts the fact that $\ell$ is strictly decreasing. Therefore, we must have $\ell(-\infty) = +\infty$. $\qquad\square$

Without loss of generality, assume $\underline{\ell} := \ell(+\infty) = 0$. Otherwise, we can just consider $\ell - \underline{\ell}$ instead of $\ell$. In addition, we also assume $\ell$ is pseudo-Lipschitz, i.e., there exists a constant $L > 0$ such that, for all $x, y \in \mathbb{R}$,

$$|\ell(x) - \ell(y)| \leq L(1 + |x| + |y|)|x - y|.$$

For ease of exposition, we assume $\tau = 1$, as it is not fundamentally different from the case of arbitrary $\tau > 0$. In Section F.1.7, we will discuss how to extend our proof to general $\tau > 0$.

Recall the original unconstrained empirical risk minimization (ERM) problem Eq. (2a):

$$M_n := \min_{\boldsymbol{\beta} \in \mathbb{R}^d, \beta_0 \in \mathbb{R}} \widehat{R}_n(\boldsymbol{\beta}, \beta_0) := \min_{\boldsymbol{\beta} \in \mathbb{R}^d, \beta_0 \in \mathbb{R}} \frac{1}{n} \sum_{i=1}^n \ell\big(y_i(\langle \boldsymbol{x}_i, \boldsymbol{\beta}\rangle + \beta_0)\big). \tag{82}$$

We first provide an outline for the proof of Theorem D.3, which involves several intermediate steps of simplifying the random optimization problem $M_n$.

$$M_n \xrightarrow[\text{Theorem F.2}]{\textbf{Step 1}} M_n(\boldsymbol{\Theta}_{\boldsymbol{\beta}}, \boldsymbol{\Xi}_{\mathbf{u}}) \xrightarrow[\text{Theorem F.3}]{\textbf{Step 2}} M_n^{(1)}(\boldsymbol{\Theta}_{\boldsymbol{\beta}}, \boldsymbol{\Xi}_{\mathbf{u}}) \Rightarrow M_n^{(2)}(\boldsymbol{\Theta}_c, \boldsymbol{\Xi}_{\mathbf{u}}) \left.\vphantom{\begin{matrix}a\\b\end{matrix}}\right\}$$
$$\xrightarrow[\text{Theorem F.4}]{\textbf{Step 3}} M_n^{(3)}(\boldsymbol{\Theta}_c, \boldsymbol{\Xi}_{\mathbf{u}}) \Rightarrow M_n^{(3)}(\boldsymbol{\Theta}_c) \xrightarrow[\text{Theorem F.5}]{\textbf{Step 4}} M^*(\boldsymbol{\Theta}_c) \Rightarrow M^*. \left.\vphantom{\begin{matrix}a\\b\end{matrix}}\right\} \text{Theorem F.6}$$

**Step 1: Boundedness of $\boldsymbol{\beta}$ and $\beta_0$ (from $M_n$ to $M_n(\boldsymbol{\Theta_\beta}, \boldsymbol{\Xi_u})$)**   Notice that by introducing the auxiliary variable $\mathbf{u} = (u_1, \ldots, u_n)^\mathsf{T} \in \mathbb{R}^n$ and Lagrangian multiplier $\mathbf{v} = (v_1, \ldots, v_n)^\mathsf{T} \in \mathbb{R}^n$, we can rewrite Eq. (82) as a minimax problem

$$
M_n = \min_{\substack{\boldsymbol{\beta} \in \mathbb{R}^d, \beta_0 \in \mathbb{R} \\ \mathbf{u} \in \mathbb{R}^n}} \max_{\mathbf{v} \in \mathbb{R}^n} \left\{ \frac{1}{n} \sum_{i=1}^n \ell(u_i) + \frac{1}{n} \sum_{i=1}^n v_i \big( y_i(\langle \boldsymbol{x}_i, \boldsymbol{\beta} \rangle + \beta_0) - u_i \big) \right\}
$$

$$
= \min_{\substack{\boldsymbol{\beta} \in \mathbb{R}^d, \beta_0 \in \mathbb{R} \\ \mathbf{u} \in \mathbb{R}^n}} \max_{\mathbf{v} \in \mathbb{R}^n} \left\{ \frac{1}{n} \sum_{i=1}^n \ell(u_i) + \frac{1}{n} \sum_{i=1}^n v_i (\langle \boldsymbol{\mu}, \boldsymbol{\beta} \rangle + \langle \boldsymbol{z}_i, \boldsymbol{\beta} \rangle + y_i \beta_0 - u_i) \right\},
$$

where in the second line, we reformulate $\boldsymbol{x}_i = y_i(\boldsymbol{\mu} + \boldsymbol{z}_i)$, $\boldsymbol{z}_i \sim \mathsf{N}(\mathbf{0}, \mathbf{I}_d)$, $y_i \perp\!\!\!\perp \boldsymbol{z}_i$. For any closed subsets $\boldsymbol{\Theta_\beta} \subset \mathbb{R}^d \times \mathbb{R}$, $\boldsymbol{\Xi_u} \subset \mathbb{R}^n$, we also define the quantity $M_n(\boldsymbol{\Theta_\beta}, \boldsymbol{\Xi_u})$, which can be viewed as the constrained version of ERM problem $M_n$.

$$
M_n(\boldsymbol{\Theta_\beta}, \boldsymbol{\Xi_u}) := \min_{\substack{(\boldsymbol{\beta}, \beta_0) \in \boldsymbol{\Theta_\beta} \\ \mathbf{u} \in \boldsymbol{\Xi_u}}} \max_{\mathbf{v} \in \mathbb{R}^n} \left\{ \frac{1}{n} \sum_{i=1}^n \ell(u_i) + \frac{1}{n} \sum_{i=1}^n v_i (\langle \boldsymbol{\mu}, \boldsymbol{\beta} \rangle + \langle \boldsymbol{z}_i, \boldsymbol{\beta} \rangle + y_i \beta_0 - u_i) \right\}
$$

$$
= \min_{\substack{(\boldsymbol{\beta}, \beta_0) \in \boldsymbol{\Theta_\beta} \\ \mathbf{u} \in \boldsymbol{\Xi_u}}} \max_{\mathbf{v} \in \mathbb{R}^n} \left\{ \frac{1}{n} \sum_{i=1}^n \ell(u_i) + \frac{1}{n} \mathbf{v}^\mathsf{T} \mathbf{1} \langle \boldsymbol{\mu}, \boldsymbol{\beta} \rangle + \frac{1}{n} \mathbf{v}^\mathsf{T} \boldsymbol{Z} \boldsymbol{\beta} + \frac{1}{n} \beta_0 \mathbf{v}^\mathsf{T} \boldsymbol{y} - \frac{1}{n} \mathbf{v}^\mathsf{T} \mathbf{u} \right\},
$$
(83)

where $\boldsymbol{Z} = (\boldsymbol{z}_1, \ldots, \boldsymbol{z}_n)^\mathsf{T} \in \mathbb{R}^{n \times d}$. Let $(\widehat{\boldsymbol{\beta}}_n, \widehat{\beta}_{0,n})$ be the unique minimizer of Eq. (82). The following lemma implies that $\widehat{\boldsymbol{\beta}}_n$ and $\widehat{\beta}_{0,n}$ are bounded with high probability, which enables us to work with $M_n(\boldsymbol{\Theta_\beta}, \boldsymbol{\Xi_u})$ instead of $M_n$ for some compact sets $\boldsymbol{\Theta_\beta}$ and $\boldsymbol{\Xi_u}$.

**Lemma F.2** (Boundedness of $\boldsymbol{\beta}$ and $\beta_0$). *In the non-separable regime $\delta > \delta^*(0)$, there exists some constants $C_{\boldsymbol{\beta}}, C_{\beta_0}, C_{\mathbf{u}} \in (0, \infty)$, such that $M_n = M_n(\boldsymbol{\Theta_\beta}, \boldsymbol{\Xi_u})$ with high probability, where*

$$
\boldsymbol{\Theta_\beta} = \{ (\boldsymbol{\beta}, \beta_0) \in \mathbb{R}^d \times \mathbb{R} : \|\boldsymbol{\beta}\|_2 \leq C_{\boldsymbol{\beta}}, |\beta_0| \leq C_{\beta_0} \}, \qquad \boldsymbol{\Xi_u} = \{ \mathbf{u} \in \mathbb{R}^n : \|\mathbf{u}\|_2 \leq C_{\mathbf{u}} \sqrt{n} \}.
$$

See Section F.1.1 for the proof.

**Step 2: Reduction via Gaussian comparison (from $M_n(\boldsymbol{\Theta_\beta}, \boldsymbol{\Xi_u})$ to $M_n^{(1)}(\boldsymbol{\Theta_\beta}, \boldsymbol{\Xi_u})$)**   The objective function of $M_n(\boldsymbol{\Theta_\beta}, \boldsymbol{\Xi_u})$ in Eq. (83) is a bilinear form of the Gaussian random matrix $\boldsymbol{Z}$. To simplify the bilinear term, we will use the convex Gaussian minimax theorem (CGMT), i.e., Gordon's comparison inequality (Gordon, 1985; Thrampoulidis et al., 2015). To do so, we introduce another quantity:

$$
M_n^{(1)}(\boldsymbol{\Theta_\beta}, \boldsymbol{\Xi_u}) := \min_{\substack{(\boldsymbol{\beta}, \beta_0) \in \boldsymbol{\Theta_\beta} \\ \mathbf{u} \in \boldsymbol{\Xi_u}}} \max_{\mathbf{v} \in \mathbb{R}^n} \left\{ \frac{1}{n} \sum_{i=1}^n \ell(u_i) + \frac{1}{n} \mathbf{v}^\mathsf{T} \mathbf{1} \langle \boldsymbol{\mu}, \boldsymbol{\beta} \rangle + \frac{1}{n} \|\mathbf{v}\|_2 \, \boldsymbol{h}^\mathsf{T} \boldsymbol{\beta} + \frac{1}{n} \|\boldsymbol{\beta}\|_2 \, \boldsymbol{g}^\mathsf{T} \mathbf{v} \right.
$$

$$
\left. + \frac{1}{n} \beta_0 \mathbf{v}^\mathsf{T} \boldsymbol{y} - \frac{1}{n} \mathbf{v}^\mathsf{T} \mathbf{u} \right\},
$$

where $\boldsymbol{h} \sim \mathsf{N}(\mathbf{0}, \mathbf{I}_d)$, $\boldsymbol{g} \sim \mathsf{N}(\mathbf{0}, \mathbf{I}_n)$ are independent Gaussian vectors. However, the classical CGMT cannot be directly applied to $M_n^{(1)}(\boldsymbol{\Theta_\beta}, \boldsymbol{\Xi_u})$ since $\mathbf{v}$ is maximized over an unbounded set. To this end, we proved the following version of CGMT, which connects $M_n^{(1)}(\boldsymbol{\Theta_\beta}, \boldsymbol{\Xi_u})$ with $M_n(\boldsymbol{\Theta_\beta}, \boldsymbol{\Xi_u})$.

**Lemma F.3** (CGMT, unbounded for maximum). *For any compact sets $\boldsymbol{\Theta_\beta}$ and $\boldsymbol{\Xi_u}$ (not necessarily convex) and $t \in \mathbb{R}$, we have*

$$
\mathbb{P}\left( M_n(\boldsymbol{\Theta_\beta}, \boldsymbol{\Xi_u}) \leq t \right) \leq 2\, \mathbb{P}\left( M_n^{(1)}(\boldsymbol{\Theta_\beta}, \boldsymbol{\Xi_u}) \leq t \right).
$$
(84)

*Additionally, if $\boldsymbol{\Theta_\beta}$ and $\boldsymbol{\Xi_u}$ are convex, then*

$$
\mathbb{P}\left( M_n(\boldsymbol{\Theta_\beta}, \boldsymbol{\Xi_u}) \geq t \right) \leq 2\, \mathbb{P}\left( M_n^{(1)}(\boldsymbol{\Theta_\beta}, \boldsymbol{\Xi_u}) \geq t \right).
$$
(85)

See Section F.1.2 for the proof.

**Reparametrization in low dimensions (from $M_n^{(1)}(\boldsymbol{\Theta}_{\boldsymbol{\beta}}, \boldsymbol{\Xi}_{\mathbf{u}})$ to $M_n^{(2)}(\boldsymbol{\Theta}_c, \boldsymbol{\Xi}_{\mathbf{u}})$)** To simplify $M_n^{(1)}(\boldsymbol{\Theta}_{\boldsymbol{\beta}}, \boldsymbol{\Xi}_{\mathbf{u}})$, we consider the following change of variables

$$\rho := \cos(\boldsymbol{\mu}, \boldsymbol{\beta}) := \begin{cases} \left\langle \dfrac{\boldsymbol{\mu}}{\|\boldsymbol{\mu}\|_2}, \dfrac{\boldsymbol{\beta}}{\|\boldsymbol{\beta}\|_2} \right\rangle, & \text{if } \boldsymbol{\beta} \neq \mathbf{0}, \\ 0, & \text{if } \boldsymbol{\beta} = \mathbf{0}, \end{cases} \qquad R := \|\boldsymbol{\beta}\|_2 . \tag{86}$$

Now, for any closed subset $\boldsymbol{\Theta}_c \subset [-1, 1] \times \mathbb{R}_{\geq 0} \times \mathbb{R}$, we define the quantity $M_n^{(2)}(\boldsymbol{\Theta}_c, \boldsymbol{\Xi}_{\mathbf{u}})$ by

$$M_n^{(2)}(\boldsymbol{\Theta}_{\boldsymbol{\beta}}, \boldsymbol{\Xi}_{\mathbf{u}}) := \min_{\substack{(\boldsymbol{\beta}, \beta_0) \in \mathbb{R}^d \times \mathbb{R}: \\ (\cos(\boldsymbol{\mu}, \boldsymbol{\beta}), \|\boldsymbol{\beta}\|_2, \beta_0) \in \boldsymbol{\Theta}_c \\ \mathbf{u} \in \boldsymbol{\Xi}_{\mathbf{u}}}} \max_{\mathbf{v} \in \mathbb{R}^n} \left\{ \frac{1}{n} \sum_{i=1}^n \ell(u_i) + \frac{1}{n} \mathbf{v}^\mathsf{T} \mathbf{1} \langle \boldsymbol{\mu}, \boldsymbol{\beta} \rangle + \frac{1}{n} \|\mathbf{v}\|_2 \, \boldsymbol{h}^\mathsf{T} \boldsymbol{\beta} \right.$$
$$\left. + \frac{1}{n} \|\boldsymbol{\beta}\|_2 \, \boldsymbol{g}^\mathsf{T} \mathbf{v} + \frac{1}{n} \beta_0 \mathbf{v}^\mathsf{T} \boldsymbol{y} - \frac{1}{n} \mathbf{v}^\mathsf{T} \mathbf{u} \right\}.$$

Therefore, $M_n^{(2)}(\boldsymbol{\Theta}_c, \boldsymbol{\Xi}_{\mathbf{u}})$ can be viewed as reparametrization of $M_n^{(1)}(\boldsymbol{\Theta}_{\boldsymbol{\beta}}, \boldsymbol{\Xi}_{\mathbf{u}})$ when $\boldsymbol{\Theta}_{\boldsymbol{\beta}} \subset \mathbb{R}^d \times \mathbb{R}$ takes the form

$$\boldsymbol{\Theta}_{\boldsymbol{\beta}} = \left\{ (\boldsymbol{\beta}, \beta_0) \in \mathbb{R}^d \times \mathbb{R} : \big( \cos(\boldsymbol{\mu}, \boldsymbol{\beta}), \|\boldsymbol{\beta}\|_2, \beta_0 \big) \in \boldsymbol{\Theta}_c \right\} .$$

Then we can simplify $M_n^{(2)}(\boldsymbol{\Theta}_{\boldsymbol{\beta}}, \boldsymbol{\Xi}_{\mathbf{u}})$ as follows:

$$M_n^{(2)}(\boldsymbol{\Theta}_c, \boldsymbol{\Xi}_{\mathbf{u}})$$

$$\overset{\text{(i)}}{=} \min_{\substack{(\rho, R, \beta_0) \in \boldsymbol{\Theta}_c \\ \mathbf{u} \in \boldsymbol{\Xi}_{\mathbf{u}}}} \min_{\substack{\|\boldsymbol{\beta}\|_2 = R \\ \cos(\boldsymbol{\mu}, \boldsymbol{\beta}) = \rho}} \max_{\gamma \geq 0} \max_{\|\mathbf{v}_0\|_2 = 1} \left\{ \frac{1}{n} \sum_{i=1}^n \ell(u_i) + \frac{\gamma}{n} \mathbf{v}_0^\mathsf{T} (\rho \|\boldsymbol{\mu}\|_2 R \mathbf{1} + R\boldsymbol{g} + \beta_0 \boldsymbol{y} - \mathbf{u}) + \frac{\gamma}{n} \boldsymbol{h}^\mathsf{T} \boldsymbol{\beta} \right\}$$

$$\overset{\text{(ii)}}{=} \min_{\substack{(\rho, R, \beta_0) \in \boldsymbol{\Theta}_c \\ \mathbf{u} \in \boldsymbol{\Xi}_{\mathbf{u}}}} \min_{\substack{\|\boldsymbol{\beta}\|_2 = R \\ \cos(\boldsymbol{\mu}, \boldsymbol{\beta}) = \rho}} \max_{\gamma \geq 0} \left\{ \frac{1}{n} \sum_{i=1}^n \ell(u_i) + \frac{\gamma}{n} \big\| \rho \|\boldsymbol{\mu}\|_2 R \mathbf{1} + R\boldsymbol{g} + \beta_0 \boldsymbol{y} - \mathbf{u} \big\|_2 + \frac{\gamma}{n} \boldsymbol{h}^\mathsf{T} \boldsymbol{\beta} \right\}$$

$$\overset{\text{(iii)}}{=} \min_{\substack{(\rho, R, \beta_0) \in \boldsymbol{\Theta}_c \\ \mathbf{u} \in \boldsymbol{\Xi}_{\mathbf{u}}}} \max_{\gamma \geq 0} \left\{ \frac{1}{n} \sum_{i=1}^n \ell(u_i) + \frac{\gamma}{n} \big\| \rho \|\boldsymbol{\mu}\|_2 R \mathbf{1} + R\boldsymbol{g} + \beta_0 \boldsymbol{y} - \mathbf{u} \big\|_2 + \frac{\gamma}{n} \min_{\substack{\|\boldsymbol{\beta}\|_2 = R \\ \cos(\boldsymbol{\mu}, \boldsymbol{\beta}) = \rho}} \boldsymbol{h}^\mathsf{T} \boldsymbol{\beta} \right\}$$

$$\overset{\text{(iv)}}{=} \min_{\substack{(\rho, R, \beta_0) \in \boldsymbol{\Theta}_c \\ \mathbf{u} \in \boldsymbol{\Xi}_{\mathbf{u}}}} \max_{\gamma \geq 0} \left\{ \frac{1}{n} \sum_{i=1}^n \ell(u_i) + \frac{\gamma}{n} \big\| \rho \|\boldsymbol{\mu}\|_2 R \mathbf{1} + R\boldsymbol{g} + \beta_0 \boldsymbol{y} - \mathbf{u} \big\|_2 \right.$$

$$\left. + \frac{\gamma}{n} R \left( \rho \frac{\boldsymbol{h}^\mathsf{T} \boldsymbol{\mu}}{\|\boldsymbol{\mu}\|_2} - \sqrt{1 - \rho^2} \|\mathbf{P}_{\boldsymbol{\mu}}^\perp \boldsymbol{h}\|_2 \right) \right\}, \tag{87}$$

where in (i) we apply the change of variables Eq. (86) and optimize $\mathbf{v}$ by its length $\gamma$ and direction $\mathbf{v}_0$ separately, (ii) follows from Cauchy–Schwarz inequality, (iii) is from the linearity of objective function in $\gamma$, and (iv) is based on direct calculation by decomposing $\boldsymbol{\beta}$:

$$\min_{\substack{\boldsymbol{\beta} \in \mathbb{R}^d: \|\boldsymbol{\beta}\|_2 = 1 \\ \cos(\boldsymbol{\mu}, \boldsymbol{\beta}) = \rho}} \boldsymbol{h}^\mathsf{T} \boldsymbol{\beta} = \min_{\substack{\boldsymbol{\theta} \in \mathbb{R}^d: \|\boldsymbol{\theta}\|_2 = 1 \\ \langle \boldsymbol{\mu}, \boldsymbol{\theta} \rangle = 0}} \boldsymbol{h}^\mathsf{T} \left( \rho \frac{\boldsymbol{\mu}}{\|\boldsymbol{\mu}\|_2} + \sqrt{1 - \rho^2} \boldsymbol{\theta} \right) = \rho \frac{\boldsymbol{h}^\mathsf{T} \boldsymbol{\mu}}{\|\boldsymbol{\mu}\|_2} - \sqrt{1 - \rho^2} \|\mathbf{P}_{\boldsymbol{\mu}}^\perp \boldsymbol{h}\|_2,$$

where $\mathbf{P}_{\boldsymbol{\mu}}^\perp := \mathbf{I}_d - \boldsymbol{\mu}\boldsymbol{\mu}^\mathsf{T} / \|\boldsymbol{\mu}\|_2^2$.

**Step 3: Convergence in variational forms (from $M_n^{(2)}(\boldsymbol{\Theta}_c, \boldsymbol{\Xi}_{\mathbf{u}})$ to $M_n^{(3)}(\boldsymbol{\Theta}_c, \boldsymbol{\Xi}_{\mathbf{u}})$)** To proceed from Eq. (87), we adopt the following trick from (Montanari et al., 2023), where $\mathbf{u}$ could be viewed as a functional of the empirical measure given by $\boldsymbol{g} = (g_1, \ldots, g_n)^\mathsf{T}$ and $\boldsymbol{y} = (y_1, \ldots, y_n)^\mathsf{T}$. Formally, let $\mathbb{Q}_n$ be the empirical distribution of the coordinates of $(\boldsymbol{g}, \boldsymbol{y})$, i.e., the probability measure on $\mathbb{R}^2$ defined by

$$\mathbb{Q}_n := \frac{1}{n} \sum_{i=1}^n \delta_{(g_i, y_i)}.$$

Let $\mathcal{L}^2(\mathbb{Q}_n) := \mathcal{L}^2(\mathbb{Q}_n, \mathbb{R}^2)$ be the space of functions $U : \mathbb{R}^2 \to \mathbb{R}$, $(g, y) \mapsto U(g, y)$ that are square integrable with respect to $\mathbb{Q}_n$. Notice that the $n$ points that form $\mathbb{Q}_n$ are almost surely distinct, and therefore we can identify this space with the space of vectors $\mathbf{u} \in \mathbb{R}^n$. We also define the two random variables in the same space by $G(g, y) = g$, $Y(g, y) = y$. Denote $\mathbb{E}_{\mathbb{Q}_n}$, $\| \cdot \|_{\mathbb{Q}_n}$ the integral and norm with respect to $\mathbb{Q}_n$ in $\mathcal{L}^2(\mathbb{Q}_n)$, i.e.,

$$\mathbb{E}_{\mathbb{Q}_n}[U] := \int_{\mathbb{R}^2} U(g, y)\, \mathrm{d}\mathbb{Q}_n(g, y) = \frac{1}{n} \sum_{i=1}^{n} U(g_i, y_i), \qquad \|U\|_{\mathbb{Q}_n} := (\mathbb{E}_{\mathbb{Q}_n}[U^2])^{1/2}.$$

Let $\Xi_u \subseteq \mathcal{L}^2(\mathbb{Q}_n)$ be the corresponding subset identified by $\mathbf{\Xi_u} \subseteq \mathbb{R}^n$, that is,

$$\Xi_u := \left\{ U \in \mathcal{L}^2(\mathbb{Q}_n) : \mathbf{u} := (U(g_1, y_1), \ldots, U(g_n, y_n))^\mathsf{T} \in \mathbf{\Xi_u} \right\}.$$

Then with these definitions, we can rewrite the expression of $M_n^{(2)}(\mathbf{\Theta}_c, \mathbf{\Xi_u})$ as

$$M_n^{(2)}(\mathbf{\Theta}_c, \mathbf{\Xi_u}) = \min_{\substack{(\rho, R, \beta_0) \in \mathbf{\Theta}_c \\ U \in \Xi_u}} \max_{\gamma \geq 0} \Bigg\{ \mathbb{E}_{\mathbb{Q}_n}[\ell(U)] + \frac{\gamma}{\sqrt{n}} \big\| \rho \|\boldsymbol{\mu}\|_2 R + RG + \beta_0 Y - U \big\|_{\mathbb{Q}_n}$$

$$+ \frac{\gamma}{n} R \bigg( \rho \frac{\boldsymbol{h}^\mathsf{T} \boldsymbol{\mu}}{\|\boldsymbol{\mu}\|_2} - \sqrt{1 - \rho^2} \|\mathbf{P}_{\boldsymbol{\mu}}^\perp \boldsymbol{h}\|_2 \bigg) \Bigg\}$$

$$= \min_{\substack{(\rho, R, \beta_0) \in \mathbf{\Theta}_c}} \min_{U \in \Xi_u \cap \mathcal{N}_n} \mathbb{E}_{\mathbb{Q}_n}[\ell(U)],$$

where we define the (stochastic) subset $\mathcal{N}_n = \mathcal{N}_n(\rho, R, \beta_0)$ by

$$\mathcal{N}_n := \left\{ U \in \mathcal{L}^2(\mathbb{Q}_n) : \big\| \rho \|\boldsymbol{\mu}\|_2 R + RG + \beta_0 Y - U \big\|_{\mathbb{Q}_n} \leq \frac{R}{\sqrt{n}} \bigg( \sqrt{1 - \rho^2} \|\mathbf{P}_{\boldsymbol{\mu}}^\perp \boldsymbol{h}\|_2 - \rho \frac{\boldsymbol{h}^\mathsf{T} \boldsymbol{\mu}}{\|\boldsymbol{\mu}\|_2} \bigg) \right\}. \tag{88}$$

It can be shown that as $n, d \to \infty$,

$$\frac{R}{\sqrt{n}} \bigg( \sqrt{1 - \rho^2} \|\mathbf{P}_{\boldsymbol{\mu}}^\perp \boldsymbol{h}\|_2 - \rho \frac{\boldsymbol{h}^\mathsf{T} \boldsymbol{\mu}}{\|\boldsymbol{\mu}\|_2} \bigg) \xrightarrow{\mathrm{P}} \frac{R\sqrt{1 - \rho^2}}{\sqrt{\delta}}.$$

This convergence then motivates us to define another quantity

$$M_n^{(3)}(\mathbf{\Theta}_c, \mathbf{\Xi_u}) := \min_{(\rho, R, \beta_0) \in \mathbf{\Theta}_c} \min_{U \in \Xi_u \cap \mathcal{N}_n^\delta} \mathbb{E}_{\mathbb{Q}_n}[\ell(U)], \tag{89}$$

where the subset $\mathcal{N}_n^\delta = \mathcal{N}_n^\delta(\rho, R, \beta_0)$ is given by

$$\mathcal{N}_n^\delta := \left\{ U \in \mathcal{L}^2(\mathbb{Q}_n) : \big\| \rho \|\boldsymbol{\mu}\|_2 R + RG + \beta_0 Y - U \big\|_{\mathbb{Q}_n} \leq \frac{R\sqrt{1 - \rho^2}}{\sqrt{\delta}} \right\}. \tag{90}$$

The following lemma shows that $M_n^{(2)}$ and $M_n^{(3)}$ are close to each other:

**Lemma F.4.** *For any compact sets $\mathbf{\Theta}_c \subset [-1, 1] \times \mathbb{R}_{\geq 0} \times \mathbb{R}$ and $\mathbf{\Xi_u} \subset \mathbb{R}^n$ (not necessarily convex), as $n \to \infty$, we have*

$$\left| M_n^{(2)}(\mathbf{\Theta}_c, \mathbf{\Xi_u}) - M_n^{(3)}(\mathbf{\Theta}_c, \mathbf{\Xi_u}) \right| \xrightarrow{\mathrm{P}} 0.$$

See Section F.1.3 for the proof.

**Step 4: Asymptotic characterization (from $M_n^{(3)}(\mathbf{\Theta}_c, \mathbf{\Xi_u})$, $M_n^{(3)}(\mathbf{\Theta}_c)$ to $M^*(\mathbf{\Theta}_c)$, $M^*$)**  For any closed subsets $\mathbf{\Theta}_c \subset [-1, 1] \times \mathbb{R}_{\geq 0} \times \mathbb{R}$, we define the quantity $M_n^{(3)}(\mathbf{\Theta}_c)$ by

$$M_n^{(3)}(\mathbf{\Theta}_c) := \min_{(\rho, R, \beta_0) \in \mathbf{\Theta}_c} \min_{U \in \mathcal{N}_n^\delta} \mathbb{E}_{\mathbb{Q}_n}[\ell(U)].$$

Compared with Eq. (89), clearly $M_n^{(3)}(\boldsymbol{\Theta}_c, \boldsymbol{\Xi}_{\mathbf{u}}) = M_n^{(3)}(\boldsymbol{\Theta}_c)$ when $\boldsymbol{\Xi}_{\mathbf{u}}$ is large enough. To analyze $M_n^{(3)}(\boldsymbol{\Theta}_c)$, we consider the change of variable[12]

$$\xi := -\frac{\rho \|\boldsymbol{\mu}\|_2 R + RG + \beta_0 Y - U}{R\sqrt{1 - \rho^2}},$$

Then we have

$$M_n^{(3)}(\boldsymbol{\Theta}_c) = \min_{\substack{(\rho, R, \beta_0) \in \boldsymbol{\Theta}_c \\ \xi \in \mathcal{L}^2(\mathbb{Q}_n), \|\xi\|_{\mathbb{Q}_n} \leq 1/\sqrt{\delta}}} \mathbb{E}_{\mathbb{Q}_n} \left[ \ell\big(\rho \|\boldsymbol{\mu}\|_2 R + RG + \beta_0 Y + R\sqrt{1 - \rho^2}\xi\big) \right].$$

Denote $\mathbb{Q}_\infty := \mathbb{P}$ the population measure of $(G, Y)$ (so that $(G, Y) \sim \mathsf{N}(0, 1) \times P_y$ under $\mathbb{Q} = \mathbb{Q}_\infty$, and we have $\mathbb{E}_{\mathbb{Q}_\infty} := \mathbb{E}$, $\|U\|_{\mathbb{Q}_\infty} := (\mathbb{E}[U^2])^{1/2}$). Then we also define the asymptotic counterpart of $M_n^{(3)}(\boldsymbol{\Theta}_c)$ by replacing $\mathbb{Q}_n$ with $\mathbb{Q}_\infty$:

$$M^*(\boldsymbol{\Theta}_c) := \min_{\substack{(\rho, R, \beta_0) \in \boldsymbol{\Theta}_c \\ \xi \in \mathcal{L}^2(\mathbb{Q}_\infty), \|\xi\|_{\mathbb{Q}_\infty} \leq 1/\sqrt{\delta}}} \mathbb{E} \left[ \ell\big(\rho \|\boldsymbol{\mu}\|_2 R + RG + \beta_0 Y + R\sqrt{1 - \rho^2}\xi\big) \right].$$

The following lemma shows that $M_n^{(3)}(\boldsymbol{\Theta}_c)$ converges to the deterministic quantity $M^*(\boldsymbol{\Theta}_c)$:

**Lemma F.5.** *For any compact subset* $\boldsymbol{\Theta}_c \subset [-1, 1] \times \mathbb{R}_{\geq 0} \times \mathbb{R}$, *as* $n \to \infty$, *we have*

$$M_n^{(3)}(\boldsymbol{\Theta}_c) \xrightarrow{\mathrm{P}} M^*(\boldsymbol{\Theta}_c).$$

See Section F.1.4 for the proof.

Finally, combining Theorem F.3—F.5, we obtain the following theorem.

**Theorem F.6.** *Consider any compact sets* $\boldsymbol{\Theta}_{\boldsymbol{\beta}}$ *and* $\boldsymbol{\Xi}_{\mathbf{u}}$ *such that* $\boldsymbol{\Theta}_{\boldsymbol{\beta}}$ *has the form of*

$$\boldsymbol{\Theta}_{\boldsymbol{\beta}} = \big\{ (\boldsymbol{\beta}, \beta_0) \in \mathbb{R}^d \times \mathbb{R} : \big(\cos(\boldsymbol{\mu}, \boldsymbol{\beta}), \|\boldsymbol{\beta}\|_2, \beta_0\big) \in \boldsymbol{\Theta}_c \big\} \tag{91}$$

*for some compact domain* $\boldsymbol{\Theta}_c \subset [-1, 1] \times \mathbb{R}_{\geq 0} \times \mathbb{R}$ *of* $(\rho, R, \beta_0)$. *Assume* $\boldsymbol{\Xi}_{\mathbf{u}}$ *is large enough. Then, for any* $\varepsilon > 0$, *as* $n \to \infty$, *we have*

$$\mathbb{P}\big(M_n(\boldsymbol{\Theta}_{\boldsymbol{\beta}}, \boldsymbol{\Xi}_{\mathbf{u}}) \leq M^*(\boldsymbol{\Theta}_c) - \varepsilon\big) \to 0.$$

*Further, if both* $\boldsymbol{\Theta}_{\boldsymbol{\beta}}$ *and* $\boldsymbol{\Xi}_{\mathbf{u}}$ *are convex, then*

$$M_n(\boldsymbol{\Theta}_{\boldsymbol{\beta}}, \boldsymbol{\Xi}_{\mathbf{u}}) \xrightarrow{\mathrm{P}} M^*(\boldsymbol{\Theta}_c).$$

*Proof.* According to Theorem F.4 and F.5, we have $M_n^{(2)}(\boldsymbol{\Theta}_c, \boldsymbol{\Xi}_{\mathbf{u}}) \xrightarrow{\mathrm{P}} M^*(\boldsymbol{\Theta}_c)$ for any compact sets $\boldsymbol{\Theta}_c \subset [-1, 1] \times \mathbb{R}_{\geq 0} \times \mathbb{R}$ and $\boldsymbol{\Xi}_{\mathbf{u}} \subset \mathbb{R}^n$ large enough such that $\Xi_u \subset \mathcal{N}_n^\delta$. When $\boldsymbol{\Theta}_{\boldsymbol{\beta}}$ takes the form Eq. (91), by CGMT Theorem F.3, for any $\varepsilon > 0$ we have

$$\mathbb{P}\left(M_n(\boldsymbol{\Theta}_{\boldsymbol{\beta}}, \boldsymbol{\Xi}_{\mathbf{u}}) \leq M^*(\boldsymbol{\Theta}_c) - \varepsilon\right) \leq 2\,\mathbb{P}\left(M_n^{(1)}(\boldsymbol{\Theta}_{\boldsymbol{\beta}}, \boldsymbol{\Xi}_{\mathbf{u}}) \leq M^*(\boldsymbol{\Theta}_c) - \varepsilon\right)$$
$$= 2\,\mathbb{P}\left(M_n^{(2)}(\boldsymbol{\Theta}_c, \boldsymbol{\Xi}_{\mathbf{u}}) \leq M^*(\boldsymbol{\Theta}_c) - \varepsilon\right) \xrightarrow{n\to\infty} 0.$$

If both $\boldsymbol{\Theta}_{\boldsymbol{\beta}}$ and $\boldsymbol{\Xi}_{\mathbf{u}}$ are also convex, then we can similarly show that

$$\mathbb{P}\left(M_n(\boldsymbol{\Theta}_{\boldsymbol{\beta}}, \boldsymbol{\Xi}_{\mathbf{u}}) \geq M^*(\boldsymbol{\Theta}_c) + \varepsilon\right) \leq 2\,\mathbb{P}\left(M_n^{(2)}(\boldsymbol{\Theta}_c, \boldsymbol{\Xi}_{\mathbf{u}}) \geq M^*(\boldsymbol{\Theta}_c) + \varepsilon\right) \xrightarrow{n\to\infty} 0.$$

Combining these implies $M_n(\boldsymbol{\Theta}_{\boldsymbol{\beta}}, \boldsymbol{\Xi}_{\mathbf{u}}) \xrightarrow{\mathrm{P}} M^*(\boldsymbol{\Theta}_c)$, which concludes the proof. $\qquad\square$

---

[12]We will show in Theorem F.11 later that the minimizer of $M_n^{(3)}(\boldsymbol{\Theta}_c)$ must satisfy $R\sqrt{1 - \rho^2} > 0$, hence the change of variable $\xi$ can be well-defined.

**Parameter convergence**  Next, we define $M^* := M^*([-1, 1] \times \mathbb{R}_{\geq 0} \times \mathbb{R})$ to be the unconstrained optimization problem Eq. (35), i.e.,

$$M^* = \min_{\substack{\rho \in [-1,1], R \geq 0, \beta_0 \in \mathbb{R} \\ \xi \in \mathcal{L}^2(\mathbb{Q}_\infty), \|\xi\|_{\mathbb{Q}_\infty} \leq 1/\sqrt{\delta}}} \mathbb{E}\left[\ell\left(\rho \|\boldsymbol{\mu}\|_2 R + RG + \beta_0 Y + R\sqrt{1 - \rho^2}\xi\right)\right].$$

An analysis of the Karush–Kuhn–Tucker (KKT) conditions shows that $M^*$ has the unique solution $(\rho^*, R^*, \beta_0^*, \xi^*)$, with $\rho^* \in (0, 1)$, $R^* \in (0, \infty)$, and $\beta_0^* \in (-\infty, \infty)$. Combined with Theorem F.6, it implies $M_n \xrightarrow{\text{P}} M^*$, which leads to the convergence of parameters:

**Lemma F.7** (Parameter convergence). *As $n, d \to \infty$, we have $M_n \xrightarrow{\text{P}} M^*$, which implies*

$$\|\widehat{\boldsymbol{\beta}}_n\|_2 \xrightarrow{\text{P}} R^*, \qquad \widehat{\rho}_n = \left\langle \frac{\widehat{\boldsymbol{\beta}}_n}{\|\widehat{\boldsymbol{\beta}}_n\|_2}, \frac{\boldsymbol{\mu}}{\|\boldsymbol{\mu}\|_2} \right\rangle \xrightarrow{\text{P}} \rho^*, \qquad \widehat{\beta}_{0,n} \xrightarrow{\text{P}} \beta_0^*.$$

See Section F.1.5 for the proof.

**ELD convergence**  Finally, to establish the ELD convergence, we use a proof strategy similar to that in Theorem E.7 by first defining the following measures

$$\widehat{\mathcal{L}}_n := \frac{1}{n} \sum_{i=1}^n \delta_{y_i(\langle \boldsymbol{x}_i, \widehat{\boldsymbol{\beta}}\rangle + \widehat{\beta}_0)}, \quad \mathcal{L}_* := \text{Law}(U^*) = \text{Law}\left(\rho^* \|\boldsymbol{\mu}\|_2 R^* + R^*G + \beta_0^*Y + R^*\sqrt{1 - \rho^{*2}}\xi^*\right).$$

Let $\mathsf{B}_{W_2}(\varepsilon)$ ($\varepsilon > 0$) be the $\varepsilon$-$W_2$ ball at $\mathcal{L}_*$, i.e.,

$$\mathsf{B}_{W_2}(\varepsilon) := \left\{\mathbf{u} \in \mathbb{R}^n : W_2\left(\frac{1}{n}\sum_{i=1}^n \delta_{u_i}, \mathcal{L}_*\right) < \varepsilon\right\}.$$

Then by showing that

$$\lim_{n \to \infty} \mathbb{P}\left(M_n(\mathbb{R}^{d+1}, \mathsf{B}_{W_2}^c(\varepsilon)) > M_n\right) = 1,$$

we can prove the convergence of logit margins $W_2(\widehat{\mathcal{L}}_n, \mathcal{L}_*) \xrightarrow{\text{P}} 0$, and hence the ELD convergence. The result in summarized in the following lemma.

**Lemma F.8** (ELD convergence). *As $n, d \to \infty$, we have $W_2(\widehat{\mathcal{L}}_n, \mathcal{L}_*) \xrightarrow{\text{P}} 0$ and $W_2(\widehat{\nu}_n, \nu_*) \xrightarrow{\text{P}} 0$.*

See Section F.1.6 for the proof.

### F.1.1  STEP 1 — BOUNDEDNESS OF $\boldsymbol{\beta}$ AND $\beta_0$: PROOF OF LEMMA F.2

***Proof of Lemma F.2.*** We first assume $\widehat{\boldsymbol{\beta}} \neq \mathbf{0}$. By Theorem D.1(c), if $\delta > \delta^*(0)$, there exists $k \in [n]$ and constant $\overline{\kappa} > 0$, such that

$$y_k\left(\left\langle \boldsymbol{x}_k, \frac{\widehat{\boldsymbol{\beta}}}{\|\widehat{\boldsymbol{\beta}}\|_2} \right\rangle + \frac{\widehat{\beta}_0}{\|\widehat{\boldsymbol{\beta}}\|_2}\right) \leq -\overline{\kappa} \tag{92}$$

holds with high probability. Therefore, we have

$$\ell(0) \overset{(i)}{\geq} \frac{1}{n}\sum_{i=1}^n \ell\left(y_i(\langle \boldsymbol{x}_i, \widehat{\boldsymbol{\beta}}\rangle + \widehat{\beta}_0)\right) \overset{(ii)}{\geq} \frac{1}{n}\ell\left(y_k(\langle \boldsymbol{x}_k, \widehat{\boldsymbol{\beta}}\rangle + \widehat{\beta}_0)\right) \overset{(iii)}{\geq} \frac{1}{n}\ell(-\overline{\kappa}\|\widehat{\boldsymbol{\beta}}\|_2),$$

where in (i) we note that $\widehat{R}_n(\mathbf{0}, 0) \geq \widehat{R}_n(\widehat{\boldsymbol{\beta}}, \widehat{\beta}_0) = M_n$, in (ii) we use $\ell \geq 0$, and in (iii) we use (92). Clearly the above inequalities also hold for $\widehat{\boldsymbol{\beta}} = \mathbf{0}$. Notice that $\frac{1}{n}\ell(-\overline{\kappa}\|\widehat{\boldsymbol{\beta}}\|_2) \to +\infty$ as $\|\widehat{\boldsymbol{\beta}}\|_2 \to \infty$, which contradicts $\ell(0) < +\infty$. Hence, it implies $\|\widehat{\boldsymbol{\beta}}\|_2$ is bounded with high probability.

Meanwhile, let $j, k \in [n]$ be any two indices $y_j = +1$, $y_k = -1$. Then as $\widehat{\beta}_0 \to \pm\infty$, we have

$$\ell(0) \geq \frac{1}{n}\sum_{i=1}^n \ell\left(y_i(\langle \boldsymbol{x}_i, \widehat{\boldsymbol{\beta}}\rangle + \widehat{\beta}_0)\right) \geq \frac{1}{n}\ell\left(\langle \boldsymbol{x}_j, \widehat{\boldsymbol{\beta}}\rangle + \widehat{\beta}_0\right) + \frac{1}{n}\ell\left(-\langle \boldsymbol{x}_k, \widehat{\boldsymbol{\beta}}\rangle - \widehat{\beta}_0\right) \to +\infty,$$

which leads to a contradiction. So $|\widehat{\beta}_0|$ is also bounded with high probability.

Finally, in the minimax representation of $M_n$, the optimal $\mathbf{u}$ must satisfy $u_i = y_i(\langle \boldsymbol{x}_i, \widehat{\boldsymbol{\beta}}\rangle + \widehat{\beta}_0)$ for all $i \in [n]$. Therefore, according to the tail bound of Gaussian matrices (Vershynin, 2018, Corollary 7.3.3),

$$
\begin{aligned}
\|\mathbf{u}\|_2 = \|\boldsymbol{y} \odot (\boldsymbol{X}\widehat{\boldsymbol{\beta}} + \widehat{\beta}_0 \mathbf{1}_n)\|_2 &= \|\langle \boldsymbol{\mu}, \widehat{\boldsymbol{\beta}}\rangle \mathbf{1}_n + \boldsymbol{Z}\widehat{\boldsymbol{\beta}} + \widehat{\beta}_0 \boldsymbol{y}\|_2 \\
&\leq \sqrt{n}\|\boldsymbol{\mu}\|_2\|\widehat{\boldsymbol{\beta}}\|_2 + \|\boldsymbol{Z}\|_{\mathrm{op}}\|\widehat{\boldsymbol{\beta}}\|_2 + \sqrt{n}|\widehat{\beta}_0| \\
&\leq \sqrt{n}\|\boldsymbol{\mu}\|_2 C_{\boldsymbol{\beta}} + \left(\sqrt{n}(1 + o(1)) + \sqrt{d}\right)C_{\boldsymbol{\beta}} + \sqrt{n}C_{\beta_0} \\
&\leq \sqrt{n}C_{\mathbf{u}}
\end{aligned}
$$

with high probability, where $C_{\mathbf{u}} > 0$ is some constant. This completes the proof. $\qquad\square$

### F.1.2 STEP 2 — REDUCTION VIA GAUSSIAN COMPARISON: PROOF OF LEMMA F.3

***Proof of Lemma F.3.*** For $m \in \mathbb{N}_+$, denote $K_m = \{\mathbf{v} \in \mathbb{R}^n : \|\mathbf{v}\|_2 \leq m\}$, and define

$$
M_n(\boldsymbol{\Theta}_{\boldsymbol{\beta}}, \boldsymbol{\Xi}_{\mathbf{u}}; K_m) := \min_{\substack{(\boldsymbol{\beta}, \beta_0) \in \boldsymbol{\Theta}_{\boldsymbol{\beta}} \\ \mathbf{u} \in \boldsymbol{\Xi}_{\mathbf{u}}}} \max_{\mathbf{v} \in K_m} \left\{ \frac{1}{n}\sum_{i=1}^{n}\ell(u_i) + \frac{1}{n}\mathbf{v}^{\mathsf{T}}\mathbf{1}\langle\boldsymbol{\mu}, \boldsymbol{\beta}\rangle + \frac{1}{n}\mathbf{v}^{\mathsf{T}}\boldsymbol{Z}\boldsymbol{\beta} + \frac{1}{n}\beta_0 \mathbf{v}^{\mathsf{T}}\boldsymbol{y} - \frac{1}{n}\mathbf{v}^{\mathsf{T}}\mathbf{u} \right\},
$$

$$
\begin{aligned}
M_n^{(1)}(\boldsymbol{\Theta}_{\boldsymbol{\beta}}, \boldsymbol{\Xi}_{\mathbf{u}}; K_m) := \min_{\substack{(\boldsymbol{\beta}, \beta_0) \in \boldsymbol{\Theta}_{\boldsymbol{\beta}} \\ \mathbf{u} \in \boldsymbol{\Xi}_{\mathbf{u}}}} \max_{\mathbf{v} \in K_m} \Bigg\{ &\frac{1}{n}\sum_{i=1}^{n}\ell(u_i) + \frac{1}{n}\mathbf{v}^{\mathsf{T}}\mathbf{1}\langle\boldsymbol{\mu}, \boldsymbol{\beta}\rangle + \frac{1}{n}\|\mathbf{v}\|_2\,\boldsymbol{h}^{\mathsf{T}}\boldsymbol{\beta} + \frac{1}{n}\|\boldsymbol{\beta}\|_2\,\boldsymbol{g}^{\mathsf{T}}\mathbf{v} \\
&+ \frac{1}{n}\beta_0 \mathbf{v}^{\mathsf{T}}\boldsymbol{y} - \frac{1}{n}\mathbf{v}^{\mathsf{T}}\mathbf{u} \Bigg\}.
\end{aligned}
$$

We first show that

$$
\lim_{m \to \infty} M_n(\boldsymbol{\Theta}_{\boldsymbol{\beta}}, \boldsymbol{\Xi}_{\mathbf{u}}; K_m) = M_n(\boldsymbol{\Theta}_{\boldsymbol{\beta}}, \boldsymbol{\Xi}_{\mathbf{u}}).
$$

To this end, note that for any fixed $(\boldsymbol{\beta}, \beta_0, \mathbf{u})$, by Cauchy–Schwarz inequality we have

$$
\begin{aligned}
\max_{\mathbf{v} \in K_m} &\left\{ \frac{1}{n}\sum_{i=1}^{n}\ell(u_i) + \frac{1}{n}\mathbf{v}^{\mathsf{T}}\mathbf{1}\langle\boldsymbol{\mu}, \boldsymbol{\beta}\rangle + \frac{1}{n}\mathbf{v}^{\mathsf{T}}\boldsymbol{Z}\boldsymbol{\beta} + \frac{1}{n}\beta_0 \mathbf{v}^{\mathsf{T}}\boldsymbol{y} - \frac{1}{n}\mathbf{v}^{\mathsf{T}}\mathbf{u} \right\} \\
&= \frac{1}{n}\sum_{i=1}^{n}\ell(u_i) + \frac{m}{n}\big\|\mathbf{u} - \langle\boldsymbol{\mu}, \boldsymbol{\beta}\rangle\mathbf{1} - \boldsymbol{Z}\boldsymbol{\beta} - \beta_0\boldsymbol{y}\big\|_2.
\end{aligned} \tag{93}
$$

Let $(\boldsymbol{\beta}_*^{(m)}, \beta_{0,*}^{(m)}, \mathbf{u}_*^{(m)})$ be the minimizer of $M_n(\boldsymbol{\Theta}_{\boldsymbol{\beta}}, \boldsymbol{\Xi}_{\mathbf{u}}; K_m)$. Since $\ell \geq 0$, we know that

$$
\begin{aligned}
\frac{m}{n}\left\|\mathbf{u}_*^{(m)} - \langle\boldsymbol{\mu}, \boldsymbol{\beta}_*^{(m)}\rangle\mathbf{1} - \boldsymbol{Z}\boldsymbol{\beta}_*^{(m)} - \beta_{0,*}^{(m)}\boldsymbol{y}\right\|_2 &\leq M_n(\boldsymbol{\Theta}_{\boldsymbol{\beta}}, \boldsymbol{\Xi}_{\mathbf{u}}; K_m) \leq M_n(\boldsymbol{\Theta}_{\boldsymbol{\beta}}, \boldsymbol{\Xi}_{\mathbf{u}}) \\
\implies \frac{1}{n}\left\|\mathbf{u}_*^{(m)} - \langle\boldsymbol{\mu}, \boldsymbol{\beta}_*^{(m)}\rangle\mathbf{1} - \boldsymbol{Z}\boldsymbol{\beta}_*^{(m)} - \beta_{0,*}^{(m)}\boldsymbol{y}\right\|_2 &\leq \frac{1}{m}M_n(\boldsymbol{\Theta}_{\boldsymbol{\beta}}, \boldsymbol{\Xi}_{\mathbf{u}}).
\end{aligned}
$$

Let $\mathbf{u}' := \langle\boldsymbol{\mu}, \boldsymbol{\beta}_*^{(m)}\rangle\mathbf{1} + \boldsymbol{Z}\boldsymbol{\beta}_*^{(m)} + \beta_{0,*}^{(m)}\boldsymbol{y}$, then we have

$$
\frac{1}{n}\left\|\mathbf{u}_*^{(m)} - \mathbf{u}'\right\|_2 \leq \frac{1}{m}M_n(\boldsymbol{\Theta}_{\boldsymbol{\beta}}, \boldsymbol{\Xi}_{\mathbf{u}}), \tag{94}
$$

which implies that $(\mathbf{u}_*^{(m)} = (u_{*,1}^{(m)}, \ldots, u_{*,n}^{(m)})^\mathsf{T}, \mathbf{u}' = (u_1', \ldots, u_n')^\mathsf{T})$

$$M_n(\Theta_{\boldsymbol\beta}, \Xi_{\mathbf{u}}) = \min_{\substack{(\boldsymbol\beta, \beta_0) \in \Theta_{\boldsymbol\beta} \\ \mathbf{u} \in \Xi_{\mathbf{u}}}} \left\{ \frac{1}{n} \sum_{i=1}^n \ell(u_i) \,\middle|\, \langle \boldsymbol\mu, \boldsymbol\beta \rangle + \langle \mathbf{z}_i, \boldsymbol\beta \rangle + y_i \beta_0 - u_i = 0, \forall\, i \in [n] \right\}$$

$$\leq \frac{1}{n} \sum_{i=1}^n \ell(u_i') \leq \frac{1}{n} \sum_{i=1}^n \ell(u_{*,i}^{(m)}) + \frac{1}{n} \sum_{i=1}^n \left| \ell(u_{*,i}^{(m)}) - \ell(u_i') \right|$$

$$\overset{(i)}{\leq} \frac{1}{n} \sum_{i=1}^n \ell(u_{*,i}^{(m)}) + \frac{C_L}{n} \left\| \mathbf{u}_*^{(m)} - \mathbf{u}' \right\|_1$$

$$\overset{(ii)}{\leq} \frac{1}{n} \sum_{i=1}^n \ell(u_{*,i}^{(m)}) + O_m \left( \frac{1}{m} \right) \overset{(iii)}{\leq} M_n(\Theta_{\boldsymbol\beta}, \Xi_{\mathbf{u}}; K_m) + O_m \left( \frac{1}{m} \right),$$

where (i) follows from the pseudo-Lipschitzness of $\ell$, the compactness of $\Xi_{\mathbf{u}}$, and $C_L > 0$ is some constant, (ii) follows from Eq. (94), while (iii) follows from Eq. (93). This proves that

$$\lim_{m \to \infty} M_n(\Theta_{\boldsymbol\beta}, \Xi_{\mathbf{u}}; K_m) = M_n(\Theta_{\boldsymbol\beta}, \Xi_{\mathbf{u}}).$$

Similarly, one can show that

$$\lim_{m \to \infty} M_n^{(1)}(\Theta_{\boldsymbol\beta}, \Xi_{\mathbf{u}}; K_m) = M_n^{(1)}(\Theta_{\boldsymbol\beta}, \Xi_{\mathbf{u}}).$$

Now for any fixed $m$, applying Theorem J.1(a) yields that $\forall\, t \in \mathbb{R}$:

$$\mathbb{P}\left( M_n(\Theta_{\boldsymbol\beta}, \Xi_{\mathbf{u}}; K_m) \leq t \right) \leq 2\, \mathbb{P}\left( M_n^{(1)}(\Theta_{\boldsymbol\beta}, \Xi_{\mathbf{u}}; K_m) \leq t \right),$$

thus leading to Eq. (84) (by continuity and using the two limits above)

$$\mathbb{P}\left( M_n(\Theta_{\boldsymbol\beta}, \Xi_{\mathbf{u}}) \leq t \right) = \lim_{m \to \infty} \mathbb{P}\left( M_n(\Theta_{\boldsymbol\beta}, \Xi_{\mathbf{u}}; K_m) \leq t \right)$$

$$\leq 2 \lim_{m \to \infty} \mathbb{P}\left( M_n^{(1)}(\Theta_{\boldsymbol\beta}, \Xi_{\mathbf{u}}; K_m) \leq t \right) = 2\, \mathbb{P}\left( M_n^{(1)}(\Theta_{\boldsymbol\beta}, \Xi_{\mathbf{u}}) \leq t \right).$$

Further, if $\Theta_{\boldsymbol\beta}$ and $\Xi_{\mathbf{u}}$ are convex, Theorem J.1(b) implies that

$$\mathbb{P}\left( M_n(\Theta_{\boldsymbol\beta}, \Xi_{\mathbf{u}}; K_m) \geq t \right) \leq 2\, \mathbb{P}\left( M_n^{(1)}(\Theta_{\boldsymbol\beta}, \Xi_{\mathbf{u}}; K_m) \geq t \right).$$

Sending $m \to \infty$ similarly proves the other inequality Eq. (85). $\qquad\square$

### F.1.3   STEP 3 — CONVERGENCE IN VARIATIONAL FORMS: PROOF OF LEMMA F.4

**Proof of Lemma F.4.** First, by definition of $M_n^{(2)}$ and $M_n^{(3)}$:

$$\left| M_n^{(2)}(\Theta_c, \Xi_{\mathbf{u}}) - M_n^{(3)}(\Theta_c, \Xi_{\mathbf{u}}) \right| \leq \sup_{(\rho, R, \beta_0) \in \Theta_c} \left| \min_{U \in \Xi_u \cap \mathcal{N}_n} \mathbb{E}_{\mathbb{Q}_n}[\ell(U)] - \min_{U \in \Xi_u \cap \mathcal{N}_n^\delta} \mathbb{E}_{\mathbb{Q}_n}[\ell(U)] \right|.$$

For any fixed $(\rho, R, \beta_0) \in \Theta_c$, by definition of $\mathcal{N}_n$ in Eq. (88) and $\mathcal{N}_n^\delta$ in Eq. (90), we have

$$\left| \min_{U \in \Xi_u \cap \mathcal{N}_n} \mathbb{E}_{\mathbb{Q}_n}[\ell(U)] - \min_{U \in \Xi_u \cap \mathcal{N}_n^\delta} \mathbb{E}_{\mathbb{Q}_n}[\ell(U)] \right| \leq \max_{\substack{U, U' \in \Xi_u \cap \mathcal{N}_n \cap \mathcal{N}_n^\delta \\ \|U - U'\|_{\mathbb{Q}_n} \leq \varepsilon_n(\rho, R, \beta_0)}} \left| \mathbb{E}_{\mathbb{Q}_n}[\ell(U)] - \mathbb{E}_{\mathbb{Q}_n}[\ell(U')] \right|,$$

where

$$\varepsilon_n(\rho, R, \beta_0) := \left| \frac{R}{\sqrt{n}} \left( \sqrt{1 - \rho^2} \|\mathbf{P}_{\boldsymbol\mu}^\perp \boldsymbol{h}\|_2 - \rho \frac{\boldsymbol{h}^\mathsf{T} \boldsymbol\mu}{\|\boldsymbol\mu\|_2} \right) - R \frac{\sqrt{1 - \rho^2}}{\sqrt{\delta}} \right|.$$

By our assumption that $\ell$ is pseudo-Lipschitz, the following estimate holds:

$$\left| \mathbb{E}_{\mathbb{Q}_n}[\ell(U)] - \mathbb{E}_{\mathbb{Q}_n}[\ell(U')] \right| \leq \frac{1}{n} \sum_{i=1}^n |\ell(u_i) - \ell(u_i')| \leq \frac{L}{n} \sum_{i=1}^n (1 + |u_i| + |u_i'|) |u_i - u_i'|$$

$$\overset{(i)}{\leq} L \left( 1 + \|U\|_{\mathbb{Q}_n} + \|U'\|_{\mathbb{Q}_n} \right) \|U - U'\|_{\mathbb{Q}_n} \overset{(ii)}{\leq} C(1 + o_{\mathbb{P}}(1)) \varepsilon_n(\rho, R, \beta_0),$$

where (i) follows from Cauchy–Schwarz inequality, (ii) follows from the compactness of $\mathcal{N}_n^\delta$ and $\boldsymbol{\Theta}_c$, and the upper bound below:

$$
\begin{aligned}
\|U\|_{\mathbb{Q}_n} &\leq \sup_{(\rho, R, \beta_0) \in \boldsymbol{\Theta}_c} \left\| \rho \, \|\boldsymbol{\mu}\|_2 \, R + RG + \beta_0 Y \right\|_{\mathbb{Q}_n} + \frac{R\sqrt{1-\rho^2}}{\sqrt{\delta}} \\
&\leq \rho \, \|\boldsymbol{\mu}\|_2 \, R_{\max} + R_{\max} \|G\|_{\mathbb{Q}_n} + B_{0,\max} \|Y\|_{\mathbb{Q}_n} + \frac{R_{\max}}{\sqrt{\delta}} \\
&\overset{(*)}{=} \rho \, \|\boldsymbol{\mu}\|_2 \, R_{\max} + R_{\max}\big(1 + o_{\mathbb{P}}(1)\big) + B_{0,\max} + \frac{R_{\max}}{\sqrt{\delta}},
\end{aligned}
$$

by denoting $R_{\max} := \max_{(\rho, R, \beta_0) \in \boldsymbol{\Theta}_c} R$, $B_{0,\max} := \max_{(\rho, R, \beta_0) \in \boldsymbol{\Theta}_c} |\beta_0|$, and $C > 0$ is some constant. Here, $(*)$ is from the law of large numbers: $\|G\|_{\mathbb{Q}_n} \overset{\mathrm{P}}{\to} \|G\|_{\mathbb{Q}_\infty} = (\mathbb{E}[G^2])^{1/2} = 1$. Combining these estimates, we finally deduce that

$$
\begin{aligned}
\left| M_n^{(2)}(\boldsymbol{\Theta}_c, \boldsymbol{\Xi}_{\mathbf{u}}) - M_n^{(3)}(\boldsymbol{\Theta}_c, \boldsymbol{\Xi}_{\mathbf{u}}) \right| &\leq C(1 + o_{\mathbb{P}}(1)) \max_{(\rho, R, \beta_0) \in \boldsymbol{\Theta}_c} \varepsilon_n(\rho, R, \beta_0) \\
&= C(1 + o_{\mathbb{P}}(1)) \max_{(\rho, R, \beta_0) \in \boldsymbol{\Theta}_c} \left| \frac{R}{\sqrt{n}} \left( \sqrt{1-\rho^2} \|\mathbf{P}_{\boldsymbol{\mu}}^\perp \boldsymbol{h}\|_2 - \rho \frac{\boldsymbol{h}^\mathsf{T} \boldsymbol{\mu}}{\|\boldsymbol{\mu}\|_2} \right) - R \frac{\sqrt{1-\rho^2}}{\sqrt{\delta}} \right| \\
&\leq C(1 + o_{\mathbb{P}}(1)) \cdot R_{\max} \left( \left| \frac{1}{\sqrt{n}} \|\mathbf{P}_{\boldsymbol{\mu}}^\perp \boldsymbol{h}\|_2 - \frac{1}{\sqrt{\delta}} \right| + \frac{1}{\sqrt{n}} \frac{|\boldsymbol{h}^\mathsf{T} \boldsymbol{\mu}|}{\|\boldsymbol{\mu}\|_2} \right) \overset{\mathrm{P}}{\to} 0.
\end{aligned}
$$

The convergence in the last line follows from

$$
\frac{\|\mathbf{P}_{\boldsymbol{\mu}}^\perp \boldsymbol{h}\|_2}{\sqrt{n}} = \frac{\|\mathbf{P}_{\boldsymbol{\mu}}^\perp \boldsymbol{h}\|_2}{\|\mathbf{P}_{\boldsymbol{\mu}}^\perp\|_{\mathrm{F}}} \cdot \frac{\sqrt{d-1}}{\sqrt{n}} \overset{\mathrm{P}}{\to} \frac{1}{\sqrt{\delta}}, \qquad \frac{\boldsymbol{h}^\mathsf{T} \boldsymbol{\mu}}{\sqrt{n} \|\boldsymbol{\mu}\|_2} \overset{\mathrm{P}}{\to} 0,
$$

according to Theorem J.3(b)(c) and $\|\mathbf{P}_{\boldsymbol{\mu}}^\perp\|_{\mathrm{op}} = 1$, $\|\mathbf{P}_{\boldsymbol{\mu}}^\perp\|_{\mathrm{F}} = \sqrt{d-1}$. This completes the proof. $\qquad\square$

### F.1.4 Step 4 — Asymptotic characterization: Proofs of Lemmas F.5, F.9

We need the following auxiliary result, which studies a general variational problem for both $\mathbb{Q} = \mathbb{Q}_n$ and $\mathbb{Q} = \mathbb{Q}_\infty$ with parameters $(\rho, R, \beta_0)$ fixed. In particular, we are able to express the random variable $\xi$ by $(\rho, R, \beta_0)$, $(G, Y)$, and an additional scalar (Lagrange multiplier). Then, we can rewrite $M_n^{(3)}(\boldsymbol{\Theta}_c)$, $M^*(\boldsymbol{\Theta}_c)$ as low-dimensional convex-concave minimax problems.

**Lemma F.9.** *For any fixed parameters $\rho \in (-1, 1)$, $R > 0$, $\beta_0 \in \mathbb{R}$, and the probability measure $\mathbb{Q} = \mathbb{Q}_n$ or $\mathbb{Q} = \mathbb{Q}_\infty$, consider the following variational problem*

$$
\zeta_{\rho, R, \beta_0}(\mathbb{Q}) := \min_{\xi \in \mathcal{L}^2(\mathbb{Q}), \|\xi\|_{\mathbb{Q}}^2 \leq 1/\delta} \mathscr{R}_{\mathbb{Q}}(\xi), \quad \mathscr{R}_{\mathbb{Q}}(\xi) := \mathbb{E}_{\mathbb{Q}}\left[ \ell\big(\rho \, \|\boldsymbol{\mu}\|_2 \, R + RG + \beta_0 Y + R\sqrt{1-\rho^2} \xi\big) \right].
$$
(95)

(a) *$\mathscr{R}_{\mathbb{Q}}(\xi)$ has a unique minimizer $\xi^* := \xi_{\mathbb{Q}}^*(\rho, R, \beta_0)$, which must satisfy*

$$
\xi_{\mathbb{Q}}^*(\rho, R, \beta_0) = -\frac{\lambda^*}{R\sqrt{1-\rho^2}} \ell'\big(\mathsf{prox}_{\lambda^* \ell}(\rho \, \|\boldsymbol{\mu}\|_2 \, R + RG + \beta_0 Y)\big),
$$
(96)

*where $\lambda^*$ is the unique solution such that $\|\xi^*\|_{\mathbb{Q}}^2 = 1/\delta$. As a consequence, we have*

$$
\zeta_{\rho, R, \beta_0}(\mathbb{Q}) = \mathbb{E}_{\mathbb{Q}}\left[ \ell\big(\mathsf{prox}_{\lambda^* \ell}(\rho \, \|\boldsymbol{\mu}\|_2 \, R + RG + \beta_0 Y)\big) \right],
$$

*where $\mathsf{prox}_{\lambda^* \ell}$ and $\mathsf{e}_{\lambda^* \ell}$ are the proximal operator and Moreau envelope of $\ell$ defined in Section J.3. Moreover, $\lambda^*$ is a decreasing function of $\delta$.*

(b) *With change of variables $A := R\rho$, $B := R\sqrt{1-\rho^2}$, the variational problem Eq. (95) can be recast as $\zeta_{\rho, R, \beta_0}(\mathbb{Q}) = \sup_{\nu > 0} \mathscr{R}_{\nu, \mathbb{Q}}(A, B, \beta_0)$, where*

$$
\mathscr{R}_{\nu, \mathbb{Q}}(A, B, \beta_0) := -\frac{B\nu}{2\delta} + \mathbb{E}_{\mathbb{Q}}\left[ \mathsf{e}_\ell\Big(A \, \|\boldsymbol{\mu}\|_2 + AG_1 + BG_2 + \beta_0 Y; \frac{B}{\nu}\Big) \right],
$$

*and* $(Y, G_1, G_2) \sim P_y \times \mathsf{N}(0,1) \times \mathsf{N}(0,1)$ *under* $\mathbb{Q} = \mathbb{Q}_\infty$.[13] *Moreover,* $\mathscr{R}_{\nu,\mathbb{Q}}(A, B, \beta_0)$ *is convex in* $(A, B, \beta_0)$ *over* $\mathbb{R}_{>0} \times \mathbb{R}_{>0} \times \mathbb{R}$ *and concave in* $\nu$.

*Proof.* For **(a)**, we first show the existence of a minimizer. The proof is a standard application of direct method in calculus of variations. Since $\ell$ is lower bounded, we know that

$$\inf_{\xi \in \mathcal{L}^2(\mathbb{Q}), \|\xi\|_\mathbb{Q}^2 \leq 1/\delta} \mathscr{R}_\mathbb{Q}(\xi) > -\infty.$$

Let $\{\xi_m\}_{m \in \mathbb{N}} \in \mathcal{L}^2(\mathbb{Q})$ be a minimizing sequence such that $\|\xi_m\|_\mathbb{Q}^2 \leq 1/\delta$, and

$$\lim_{m \to \infty} \mathscr{R}_\mathbb{Q}(\xi_m) = \inf_{\xi \in \mathcal{L}^2(\mathbb{Q}), \|\xi\|_\mathbb{Q}^2 \leq 1/\delta} \mathscr{R}_\mathbb{Q}(\xi).$$

Since $\mathcal{L}^2(\mathbb{Q})$ is a Hilbert space (and hence self-reflexive), Banach-Alaoglu theorem implies that $\{\xi_m\}$ has a weak-* convergent (and hence weak convergent) subsequence, which we still denote as $\{\xi_m\}$. Let $\xi^*$ denote the weak limit of $\{\xi_m\}$. By using Mazur's lemma, we know that there exists another sequence $\{\xi'_m\}_{m \in \mathbb{N}}$, such that each $\xi'_m$ is a finite convex combination of $\{\xi_k\}_{m \leq k \leq m+N(m)}$ ($N(m) \geq 0$ depends on $m$), and that $\xi'_m$ strongly converges to $\xi^*$. Now since $\mathscr{R}_\mathbb{Q}$ is convex (this follows from convexity of $\ell$ and the fact that integration $\mathbb{E}_\mathbb{Q}$ preserves convexity), we have

$$\liminf_{m \to \infty} \mathscr{R}_\mathbb{Q}(\xi'_m) \leq \liminf_{m \to \infty} \mathscr{R}_\mathbb{Q}(\xi_m) = \inf_{\xi \in \mathcal{L}^2(\mathbb{Q}), \|\xi\|_\mathbb{Q}^2 \leq 1/\delta} \mathscr{R}_\mathbb{Q}(\xi).$$

On the other hand, Fatou's lemma implies that

$$\mathscr{R}_\mathbb{Q}(\xi^*) \leq \liminf_{m \to \infty} \mathscr{R}_\mathbb{Q}(\xi'_m).$$

This immediately leads to

$$\mathscr{R}_\mathbb{Q}(\xi^*) = \inf_{\xi \in \mathcal{L}^2(\mathbb{Q}), \|\xi\|_\mathbb{Q}^2 \leq 1/\delta} \mathscr{R}_\mathbb{Q}(\xi),$$

i.e., $\xi^*$ is a minimizer of $\mathscr{R}_\mathbb{Q}$. In order to prove uniqueness of the minimizer, we will show that $\mathscr{R}_\mathbb{Q} : \mathcal{L}^2(\mathbb{Q}) \to \mathbb{R}_{>0}$ is strictly convex. For any $\alpha \in (0,1)$ and $\xi_1, \xi_2 \in \mathcal{L}^2(\mathbb{Q})$, with a shorthand $V := \rho \|\boldsymbol{\mu}\|_2 R + RG + \beta_0 Y$, we notice that

$$\mathscr{R}_\mathbb{Q}(\alpha\xi_1 + (1-\alpha)\xi_2)$$
$$= \mathbb{E}_\mathbb{Q}\left[\ell\left(\alpha\left(V + R\sqrt{1-\rho^2}\xi_1\right) + (1-\alpha)\left(V + R\sqrt{1-\rho^2}\xi_2\right)\right)\right]$$
$$\leq \mathbb{E}_\mathbb{Q}\left[\alpha\ell\left(V + R\sqrt{1-\rho^2}\xi_1\right) + (1-\alpha)\ell\left(V + R\sqrt{1-\rho^2}\xi_2\right)\right] = \alpha\mathscr{R}_\mathbb{Q}(\xi_1) + (1-\alpha)\mathscr{R}_\mathbb{Q}(\xi_2),$$

where the inequality follows from strong convexity of $\ell$, and it becomes equality if and only if $\mathbb{Q}(\xi_1 \neq \xi_2) = 0$. Hence we conclude $\mathscr{R}_\mathbb{Q}$ is strictly convex. Since $\{\xi : \|\xi\|_\mathbb{Q}^2 \leq 1/\delta\}$ is a convex set, it implies the uniqueness ($\mathbb{Q}$-a.s.) of the minimizer $\xi^*$.

As a consequence, the unique minimizer is determined by the Karush–Kuhn–Tucker (KKT) and Slater's conditions for variational problems (Zalinescu, 2002, Theorem 2.9.2). $\xi$ is the minimizer if and only if, for some scalar $\nu$ (dual variable), the followings hold:

$$U = \rho \|\boldsymbol{\mu}\|_2 R + RG + \beta_0 Y + R\sqrt{1-\rho^2}\xi, \qquad \ell'(U) + \nu\xi = 0,$$
$$\|\xi\|_\mathbb{Q}^2 - \delta^{-1} \leq 0, \qquad \nu \geq 0, \qquad \nu(\|\xi\|_\mathbb{Q}^2 - \delta^{-1}) = 0. \tag{97}$$

We claim that the KKT conditions imply that any minimizer $\xi$ and its associated dual variable $\nu$ must satisfy

$$0 < \nu < \infty, \qquad \xi > 0 \ (\mathbb{Q}\text{-a.s.}), \qquad \|\xi\|_\mathbb{Q}^2 = \delta^{-1}.$$

To show this, we notice that $R\sqrt{1-\rho^2} > 0$ and $\ell$ is decreasing. Therefore, for any $\xi \in \mathcal{L}^2(\mathbb{Q})$, $\mathscr{R}_\mathbb{Q}(\xi) \geq \mathscr{R}_\mathbb{Q}(|\xi|)$. It implies that $\xi \geq 0$ if $\xi$ is the minimizer. Hence, by stationarity in Eq. (97):

---

[13]According to the change of variables, we have relation $AG_1 + BG_2 \overset{\mathrm{d}}{=} RG$ under $\mathbb{Q} = \mathbb{Q}_\infty$. We can also construct the realizations $\{G_1(g_i, y_i), G_2(g_i, y_i)\}_{i=1}^n$ such that $AG_1 + BG_2 = RG$, $\mathbb{Q}_n$-a.s., that is, $AG_1(g_i, y_i) + BG_2(g_i, y_i) = RG(g_i, y_i)$, for all $i \in [n]$.

$\nu\xi = -\ell'(U) > 0$, which implies the positivity of $\nu$, $\xi$. Then $\|\xi\|_{\mathbb{Q}}^2 = \delta^{-1}$ comes from complementary slackness in Eq. (97). To show $\nu$ must be finite, notice that $\nu \to +\infty$ implies $\ell'(U) \to -\infty$. Then $U \to -\infty$ since $\ell'$ is strictly increasing, while it contradicts $\xi > 0$ and $\|\xi\|_{\mathbb{Q}}^2 = \delta^{-1}$.

By change of variable $\lambda := R\sqrt{1-\rho^2}/\nu$, now we can rewrite KKT conditions Eq. (97) as

$$U + \lambda\ell'(U) = \rho \|\boldsymbol{\mu}\|_2 R + RG + \beta_0 Y, \qquad 0 < \lambda < \infty, \qquad \|\xi\|_{\mathbb{Q}}^2 = \delta^{-1}, \tag{98}$$

where $\xi$ and $U$ are related by

$$\xi = -\frac{\lambda}{R\sqrt{1-\rho^2}}\ell'(U). \tag{99}$$

Notice that Eq. (98) has a unique solution for $U$, since $x \mapsto x + \lambda\ell'(x)$ is a strictly increasing continuous function from $\mathbb{R}$ to $\mathbb{R}$, for any $\lambda \in (0, \infty)$. Then, according to Theorem J.5, $U$ can be expressed by the proximal operator of $\ell$,

$$U = \mathsf{prox}_{\lambda\ell}(\rho \|\boldsymbol{\mu}\|_2 R + RG + \beta_0 Y). \tag{100}$$

Combine it with Eq. (99) gives the expression of $\xi^*$ in Eq. (96). To establish the uniqueness of $\lambda$, we show that $\nu$ satisfying Eq. (97) must be unique. Note that $\xi = \xi(\nu)$ is determined by

$$\nu\xi(\nu) + \ell'\left(\rho \|\boldsymbol{\mu}\|_2 R + RG + \beta_0 Y + R\sqrt{1-\rho^2}\xi(\nu)\right) = 0.$$

Since $\nu, \xi(\nu) > 0$ and $\ell'$ is strictly increasing (by strong convexity), we know that $\xi(\nu)$ is strictly decreasing in $\nu$. The uniqueness of $\nu$ immediately follows from the condition $\|\xi(\nu)\|_{\mathbb{Q}}^2 = \delta^{-1}$. This also implies that $\xi(\nu) > 0$ is decreasing in $\delta$. Then we conclude $\nu$ is increasing in $\delta$, or equivalently $\lambda$ is decreasing in $\delta$. This completes the proof of part (a).

For **(b)**, as a consequence we have

$$\begin{aligned}
\zeta_{\rho,R,\beta_0}(\mathbb{Q}) &= \min_{\xi \in \mathcal{L}^2(\mathbb{Q}), \|\xi\|_{\mathbb{Q}}^2 \leq 1/\delta} \mathscr{R}_{\mathbb{Q}}(\xi) \\
&= \min_{\xi \in \mathcal{L}^2(\mathbb{Q}), \|\xi\|_{\mathbb{Q}}^2 \leq 1/\delta} \mathbb{E}_{\mathbb{Q}}\left[\ell\left(\rho \|\boldsymbol{\mu}\|_2 R + RG + \beta_0 Y + R\sqrt{1-\rho^2}\xi\right)\right] \\
&= \min_{\xi \in \mathcal{L}^2(\mathbb{Q})} \sup_{\nu \geq 0} \mathbb{E}_{\mathbb{Q}}\left[\ell\left(\rho \|\boldsymbol{\mu}\|_2 R + RG + \beta_0 Y + R\sqrt{1-\rho^2}\xi\right) + R\sqrt{1-\rho^2} \cdot \frac{\nu}{2}\left(\xi^2 - \frac{1}{\delta}\right)\right] \\
&\overset{(i)}{=} \sup_{\nu \geq 0} \min_{\xi \in \mathcal{L}^2(\mathbb{Q})} \mathbb{E}_{\mathbb{Q}}\left[\ell\left(\rho \|\boldsymbol{\mu}\|_2 R + RG + \beta_0 Y + R\sqrt{1-\rho^2}\xi\right) + R\sqrt{1-\rho^2} \cdot \frac{\nu}{2}\left(\xi^2 - \frac{1}{\delta}\right)\right] \\
&\overset{(ii)}{=} \sup_{\lambda > 0} \min_{U \in \mathcal{L}^2(\mathbb{Q})} \mathbb{E}_{\mathbb{Q}}\left[\ell(U) + \frac{1}{2\lambda}\left(U - \rho \|\boldsymbol{\mu}\|_2 R - RG - \beta_0 Y\right)^2 - \frac{R^2(1-\rho^2)}{2\lambda\delta}\right] \\
&\overset{(iii)}{=} \sup_{\lambda > 0} \left\{\mathbb{E}_{\mathbb{Q}}\left[\mathsf{e}_\ell\left(\rho \|\boldsymbol{\mu}\|_2 R + RG + \beta_0 Y; \lambda\right)\right] - \frac{R^2(1-\rho^2)}{2\lambda\delta}\right\},
\end{aligned}$$

where (i) comes from strong duality in part (a), (ii) is by change of variable $U := \rho \|\boldsymbol{\mu}\|_2 R + RG + \beta_0 Y + R\sqrt{1-\rho^2}\xi$ and $\lambda = R\sqrt{1-\rho^2}/\nu$, (iii) is from the definition of Moreau envelope Eq. (174). Now, consider change of variable

$$A = R\rho, \qquad B = R\sqrt{1-\rho^2}, \qquad \nu = R\sqrt{1-\rho^2}/\lambda.$$

Note that $0 < \nu < \infty$ by part (a), then $\zeta_{\rho,R,\beta_0}(\mathbb{Q})$ can be expressed as

$$\zeta_{\rho,R,\beta_0}(\mathbb{Q}) = \min_{\xi \in \mathcal{L}^2(\mathbb{Q}), \|\xi\|_{\mathbb{Q}}^2 \leq 1/\delta} \mathscr{R}_{\mathbb{Q}}(\xi) = \sup_{\nu > 0} \mathscr{R}_{\nu,\mathbb{Q}}(A, B, \beta_0),$$

where

$$\mathscr{R}_{\nu,\mathbb{Q}}(A, B, \beta_0) = -\frac{B\nu}{2\delta} + \mathbb{E}_{\mathbb{Q}}\left[\mathsf{e}_\ell\left(A \|\boldsymbol{\mu}\|_2 + AG_1 + BG_2 + \beta_0 Y; \frac{B}{\nu}\right)\right].$$

Finally, we complete the proof by the following arguments:

- $\mathscr{R}_{\nu,\mathbb{Q}}(A, B, \beta_0)$ is convex in $(A, B, \beta_0)$. It comes from Theorem J.5(a) that $(x, \lambda) \mapsto \mathsf{e}_\ell(x; \lambda)$ is convex, and the fact that integration $\mathbb{E}_\mathbb{Q}$ preserves convexity.

- $\mathscr{R}_{\nu,\mathbb{Q}}(A, B, \beta_0)$ is concave in $\nu$. This comes from Eq. (174) that

$$\mathsf{e}_\ell\left(A\left\|\boldsymbol{\mu}\right\|_2 + AG_1 + BG_2 + \beta_0 Y; \frac{B}{\nu}\right) = \min_{t \in \mathbb{R}}\left\{\ell(t) + \frac{\nu}{2B}(A\left\|\boldsymbol{\mu}\right\|_2 + AG_1 + BG_2 - t)^2\right\},$$

with the fact that pointwise minimum and integration $\mathbb{E}_\mathbb{Q}$ preserves concavity.

This concludes the proof of part (b). $\qquad \square$

Then we can use Theorem F.9 to show convergence $M_n^{(3)}(\boldsymbol{\Theta}_c) \xrightarrow{\mathrm{P}} M^*(\boldsymbol{\Theta}_c)$ in Theorem F.5.

***Proof of Lemma F.5.*** Recall the change of variables $A = R\rho$ and $B = R\sqrt{1 - \rho^2}$ defined in Theorem F.9(b). Note that $f : (\rho, R, \beta_0) \mapsto (R\rho, R\sqrt{1 - \rho^2}, \beta_0)$ is a continuous map. Then $f(\boldsymbol{\Theta}_c) \subset \mathbb{R}_{\geq 0} \times \mathbb{R}_{\geq 0} \times \mathbb{R}$ is still compact. Hence, by Theorem F.9 we have

$$M_n^{(3)}(\boldsymbol{\Theta}_c) = \min_{(A,B,\beta_0) \in f(\boldsymbol{\Theta}_c)} \sup_{\nu > 0} \mathscr{R}_{\nu,\mathbb{Q}_n}(A, B, \beta_0), \quad M^*(\boldsymbol{\Theta}_c) = \min_{(A,B,\beta_0) \in f(\boldsymbol{\Theta}_c)} \sup_{\nu > 0} \mathscr{R}_{\nu,\mathbb{Q}_\infty}(A, B, \beta_0).$$

For any fixed $A, B \geq 0$, $\beta_0 \in \mathbb{R}$, $\nu > 0$, by law of large numbers,

$$\mathscr{R}_{\nu,\mathbb{Q}_n}(A, B, \beta_0) = -\frac{B\nu}{2\delta} + \mathbb{E}_{\mathbb{Q}_n}\left[\mathsf{e}_\ell\left(A\left\|\boldsymbol{\mu}\right\|_2 + AG_1 + BG_2 + \beta_0 Y; \frac{B}{\nu}\right)\right]$$

$$\xrightarrow{\mathrm{P}} \mathscr{R}_{\nu,\mathbb{Q}_\infty}(A, B, \beta_0) = -\frac{B\nu}{2\delta} + \mathbb{E}\left[\mathsf{e}_\ell\left(A\left\|\boldsymbol{\mu}\right\|_2 + AG_1 + BG_2 + \beta_0 Y; \frac{B}{\nu}\right)\right].$$

Recall $\mathscr{R}_{\nu,\mathbb{Q}_n}(A, B, \beta_0)$ is concave in $\nu$. Also, note that $\mathscr{R}_{\nu,\mathbb{Q}_\infty}(A, B, \beta_0) \to -\infty$ as $\nu \to \infty$, since by Theorem J.5(a), we have

$$\lim_{\nu \to \infty} \mathbb{E}\left[\mathsf{e}_\ell\left(A\left\|\boldsymbol{\mu}\right\|_2 + AG_1 + BG_2 + \beta_0 Y; \frac{B}{\nu}\right)\right] = \mathbb{E}\left[\ell(A\left\|\boldsymbol{\mu}\right\|_2 + AG_1 + BG_2 + \beta_0 Y)\right] < \infty.$$

This implies there exits $\overline{\nu} \in \mathbb{R}_{>0}$, such that $\sup_{\nu \geq \overline{\nu}} \mathscr{R}_{\nu,\mathbb{Q}_\infty}(A, B, \beta_0) < \sup_{\nu > 0} \mathscr{R}_{\nu,\mathbb{Q}_\infty}(A, B, \beta_0)$. So, we can apply (Thrampoulidis et al., 2018, Lemma 10) and conclude the uniform convergence

$$\sup_{\nu > 0} \mathscr{R}_{\nu,\mathbb{Q}_n}(A, B, \beta_0) \xrightarrow{\mathrm{P}} \sup_{\nu > 0} \mathscr{R}_{\nu,\mathbb{Q}_\infty}(A, B, \beta_0).$$

Recall that both $\sup_{\nu > 0} \mathscr{R}_{\nu,\mathbb{Q}_n}(A, B, \beta_0)$ and $\sup_{\nu > 0} \mathscr{R}_{\nu,\mathbb{Q}_\infty}(A, B, \beta_0)$ are convex in $(A, B, \beta_0)$ (since pointwise supremum preserves convexity). Then we could obtain uniform convergence on compact set $f(\boldsymbol{\Theta}_c)$ by convexity (Liese & Miescke, 2008, Lemma 7.75):

$$\left|M_n^{(3)}(\boldsymbol{\Theta}_c) - M^*(\boldsymbol{\Theta}_c)\right| \leq \sup_{(A,B,\beta_0) \in f(\boldsymbol{\Theta}_c)}\left|\sup_{\nu > 0} \mathscr{R}_{\nu,\mathbb{Q}_n}(A, B, \beta_0) - \sup_{\nu > 0} \mathscr{R}_{\nu,\mathbb{Q}_\infty}(A, B, \beta_0)\right| \xrightarrow{\mathrm{P}} 0.$$

This completes the proof. $\qquad \square$

### F.1.5 Parameter convergence, optimality analysis: Proofs of Lemmas F.7, F.10, F.11

Recall that

$$M^* = \min_{\substack{\rho \in [-1,1], R \geq 0, \beta_0 \in \mathbb{R} \\ \xi \in \mathcal{L}^2(\mathbb{Q}_\infty), \|\xi\|_{\mathbb{Q}_\infty} \leq 1/\sqrt{\delta}}} \mathbb{E}\left[\ell\left(\rho\left\|\boldsymbol{\mu}\right\|_2 R + RG + \beta_0 Y + R\sqrt{1 - \rho^2}\xi\right)\right], \quad (101)$$

where $R, \beta_0$ are optimized over unbounded sets. The following lemma shows that any minimizer $R^*, \beta_0^*$ of Eq. (101) must be bounded.

**Lemma F.10** (Boundedness of $R^*$ and $\beta_0^*$). *Let $(\rho^*, R^*, \beta_0^*, \xi^*)$ be any minimizer of Eq. (101). Then in the non-separable regime ($\delta > \delta^*(0)$), we have $R^* < \infty$ and $|\beta_0^*| < \infty$.*

*Proof.* We first prove the following claim: There exists an $\varepsilon > 0$, such that for any $(a, b) \in \mathbb{R}_{>0} \times \mathbb{R}$ satisfying $a^2 + b^2 = 1$, any $\rho \in [-1, 1]$, and any $\xi \in \mathcal{L}^2(\mathbb{Q}_\infty)$, $\|\xi\|_{\mathbb{Q}_\infty} \leq 1/\sqrt{\delta}$:

$$\mathbb{P}\left(a\rho \|\boldsymbol{\mu}\|_2 + aG + bY + a\sqrt{1 - \rho^2}\xi \leq -\varepsilon\right) \geq \varepsilon. \tag{102}$$

We prove this claim by contradiction. Assume it is not true, then for any $m \in \mathbb{N}$, there exists the corresponding $(a_m, b_m, \rho_m, \xi_m)$ such that $(a_m, b_m) \in \mathbb{R}_{>0} \times \mathbb{R}$, with $a_m^2 + b_m^2 = 1$, $\rho_m \in [-1, 1]$, and $\xi_m \in \mathcal{L}^2(\mathbb{Q}_\infty)$, $\|\xi_m\|_{\mathbb{Q}_\infty} \leq 1/\sqrt{\delta}$, which satisfy

$$\mathbb{P}\left(a_m \rho_m \|\boldsymbol{\mu}\|_2 + a_m G + b_m Y + a_m \sqrt{1 - \rho_m^2}\xi_m \leq -\frac{1}{m}\right) < \frac{1}{m}. \tag{103}$$

We can always assume that $(a_m, b_m, \rho_m) \to (a, b, \rho)$ and $\xi_m \to \xi$ weakly in $\mathcal{L}^2(\mathbb{Q}_\infty)$ when $m \to \infty$. Otherwise, such a convergent subsequence always exists according to Heine–Borel Theorem and Banach–Alaoglu Theorem. Therefore, $a_m \rho_m \|\boldsymbol{\mu}\|_2 + a_m G + b_m Y + a_m \sqrt{1 - \rho_m^2}\xi_m$ weakly converges to $a\rho \|\boldsymbol{\mu}\|_2 + aG + bY + a\sqrt{1 - \rho^2}\xi$ in $\mathcal{L}^2(\mathbb{Q}_\infty)$. For any nonnegative $Z \in \mathcal{L}^2(\mathbb{Q}_\infty)$, one has

$$\mathbb{E}\left[\left(a\rho \|\boldsymbol{\mu}\|_2 + aG + bY + a\sqrt{1 - \rho^2}\xi\right)Z\right]$$
$$= \lim_{m \to \infty} \mathbb{E}\left[\left(a_m \rho_m \|\boldsymbol{\mu}\|_2 + a_m G + b_m Y + a_m \sqrt{1 - \rho_m^2}\xi_m\right)Z\right].$$

Denote $U_m := a_m \rho_m \|\boldsymbol{\mu}\|_2 + a_m G + b_m Y + a_m \sqrt{1 - \rho_m^2}\xi_m$, then we obtain the following estimate:

$$\mathbb{E}[U_m Z] = \mathbb{E}\left[U_m \mathbb{1}_{U_m > -1/m} Z\right] + \mathbb{E}\left[U_m \mathbb{1}_{U_m \leq -1/m} Z\right]$$
$$\geq -\frac{1}{m}\mathbb{E}[Z] - \left(\mathbb{E}[U_m^2]\right)^{1/2}\left(\mathbb{E}[Z^2 \mathbb{1}_{U_m \leq -1/m}]\right)^{1/2},$$

where the last line follows from Cauchy–Schwarz inequality. By definition of $U_m$, we know that $\mathbb{E}[U_m^2]$ is uniformly bounded for any $m \in \mathbb{N}$. Further, since $Z \in \mathcal{L}^2(\mathbb{Q}_\infty)$ and $\mathbb{P}(U_m \leq -1/m) \leq 1/m \to 0$ as $m \to \infty$ by Eq. (103), we know that $\mathbb{E}[Z^2 \mathbb{1}_{U_m \leq -1/m}] \to 0$. It finally follows that

$$\mathbb{E}\left[\left(a\rho \|\boldsymbol{\mu}\|_2 + aG + bY + a\sqrt{1 - \rho^2}\xi\right)Z\right] = \lim_{m \to \infty} \mathbb{E}[U_m Z] \geq 0.$$

Since this is true for any nonnegative $Z \in \mathcal{L}^2(\mathbb{Q}_\infty)$, we know that

$$a\rho \|\boldsymbol{\mu}\|_2 + aG + bY + a\sqrt{1 - \rho^2}\xi \geq 0, \quad \text{almost surely,}$$

or equivalently, there exists $(\rho, R, \beta_0) \in [-1, 1] \times \mathbb{R}_{>0} \times \mathbb{R}$ and $\xi \in \mathcal{L}^2(\mathbb{Q}_\infty)$, $\mathbb{E}[\xi^2] \leq 1/\delta$ satisfying

$$R\rho \|\boldsymbol{\mu}\|_2 + RG + \beta_0 Y + R\sqrt{1 - \rho^2}\xi \geq 0, \quad \text{almost surely.}$$

It implies the constraint of the variational problem for the separable regime (SVM) Eq. (31), i.e., $\rho \|\boldsymbol{\mu}\|_2 + G + \beta_0' Y + \sqrt{1 - \rho^2}\xi \geq \kappa$ holds for some $\kappa \geq 0$ (with change of variable $\beta_0' := \beta_0/R$). According to Theorem D.1(b), we obtain $\kappa^* \geq 0$, or equivalently $\delta \leq \delta^*(0)$, which contradicts the non-separable regime $\delta > \delta^*(0)$. Our claim Eq. (102) is thus proved.

Now for any $(\rho, R, \beta_0, \xi)$ such that $R > 0$, denote

$$V(\rho, R, \beta_0, \xi) := \frac{1}{\sqrt{R^2 + \beta_0^2}}\left(\rho \|\boldsymbol{\mu}\|_2 R + RG + \beta_0 Y + R\sqrt{1 - \rho^2}\xi\right).$$

We know that $\mathbb{P}(V(\rho, R, \beta_0, \xi) \leq -\varepsilon) \geq \varepsilon$ by Eq. (102). Therefore,

$$\mathbb{E}\left[\ell\left(\rho \|\boldsymbol{\mu}\|_2 R + RG + \beta_0 Y + R\sqrt{1 - \rho^2}\xi\right)\right]$$
$$= \mathbb{E}\left[\ell\left(\sqrt{R^2 + \beta_0^2}\, V(\rho, R, \beta_0, \xi)\right)\right]$$
$$\geq \mathbb{E}\left[\ell\left(\sqrt{R^2 + \beta_0^2}\, V(\rho, R, \beta_0, \xi)\right)\mathbb{1}_{V(\rho, R, \beta_0, \xi) \leq -\varepsilon}\right]$$
$$\geq \varepsilon\ell\left(-\varepsilon\sqrt{R^2 + \beta_0^2}\right),$$

which diverges to infinity as $R^2 + \beta_0^2 \to \infty$. This completes the proof. $\qquad \square$

A direct consequence of Theorem F.10 is that $M^* = M^*(\boldsymbol{\Theta}_c)$ for $\boldsymbol{\Theta}_c$ large enough. The following result shows that $M^*$ in Eq. (101) has a unique minimizer.

**Lemma F.11.** *Consider the variational problem $M^*$ defined in Eq. (101).*

*(a) $M^*$ has a unique minimizer $(\rho^*, R^*, \beta_0^*, \xi^*)$, which must satisfy*

$$\xi^* = -\frac{\lambda^*}{R^*\sqrt{1-\rho^{*2}}}\ell'\big(\mathsf{prox}_{\lambda^*\ell}(\rho^*\,\|\boldsymbol{\mu}\|_2\,R^* + R^*G + \beta_0^*Y)\big),$$

*where $\lambda^*$ is the unique solution such that $\mathbb{E}[\xi^{*2}] = 1/\delta$. As a consequence, we have*

$$M^* = \mathbb{E}\left[\ell\big(\mathsf{prox}_{\lambda^*\ell}(\rho^*\,\|\boldsymbol{\mu}\|_2\,R^* + R^*G + \beta_0^*Y)\big)\right].$$

*(b) $(\rho^*, R^*, \beta_0^*, \lambda^*)$ is also the unique solution to the system of equations*

$$
\begin{aligned}
-\frac{R\rho}{\lambda\delta\,\|\boldsymbol{\mu}\|_2} &= \mathbb{E}\left[\ell'\big(\mathsf{prox}_{\lambda\ell}(\rho\,\|\boldsymbol{\mu}\|_2\,R + RG + \beta_0Y)\big)\right], \\
\frac{R}{\lambda\delta} &= \mathbb{E}\left[\ell'\big(\mathsf{prox}_{\lambda\ell}(\rho\,\|\boldsymbol{\mu}\|_2\,R + RG + \beta_0Y)\big)G\right], \\
0 &= \mathbb{E}\left[\ell'\big(\mathsf{prox}_{\lambda\ell}(\rho\,\|\boldsymbol{\mu}\|_2\,R + RG + \beta_0Y)\big)Y\right], \\
\frac{R^2(1-\rho^2)}{\lambda^2\delta} &= \mathbb{E}\left[\big(\ell'\big(\mathsf{prox}_{\lambda\ell}(\rho\,\|\boldsymbol{\mu}\|_2\,R + RG + \beta_0Y)\big)\big)^2\right],
\end{aligned}
\tag{104}
$$

*where $(\rho^*, R^*, \beta_0^*, \lambda^*) \in (0,1) \times \mathbb{R}_{>0} \times \mathbb{R} \times \mathbb{R}_{>0}$.*

*(c) With change of variables $A := R\rho$, $B := R\sqrt{1-\rho^2}$, the original variational problem Eq. (101) can be reduced to the following minimax problem*

$$M^* = \min_{\substack{A\geq 0, B\geq 0 \\ \beta_0\in\mathbb{R}}} \sup_{\nu>0} \left\{-\frac{B\nu}{2\delta} + \mathbb{E}\left[\mathsf{e}_\ell\Big(A\,\|\boldsymbol{\mu}\|_2 + AG_1 + BG_2 + \beta_0Y; \frac{B}{\nu}\Big)\right]\right\},$$

*where $(Y, G_1, G_2) \sim P_y \times \mathsf{N}(0,1) \times \mathsf{N}(0,1)$, and the objective function is convex-concave.*

*Proof.* We first show the optimization problem Eq. (101) has a unique minimizer. Since its original formulation is non-convex, we make the following change of variables:

$$A := R\rho, \qquad B := R\sqrt{1-\rho^2}, \qquad \xi_B := B\xi. \tag{105}$$

Then, the optimization problem is recast as

$$\min_{\substack{A,B\geq 0, \beta_0\in\mathbb{R} \\ \xi_B\in\mathcal{L}^2(\mathbb{Q}_\infty)}} \mathbb{E}\left[\ell\big(A\,\|\boldsymbol{\mu}\|_2 + AG_1 + BG_2 + \beta_0Y + \xi_B\big)\right], \quad \text{subject to } \|\xi_B\|_{\mathbb{Q}} \leq \frac{B}{\sqrt{\delta}}, \tag{106}$$

which is convex, where $(Y, G_1, G_2) \sim P_y \times \mathsf{N}(0,1) \times \mathsf{N}(0,1)$ (recall that $AG_1 + BG_2 \stackrel{\mathrm{d}}{=} RG$). Now we show that the above optimization problem has a unique minimizer. Note that Theorem F.10 also implies that any minimizer of this optimization problem is finite. Therefore, a similar argument as in the proof of Theorem F.9(a) shows that Eq. (106) has a unique minimizer. Since the mapping $(\rho, R, \xi) \mapsto (A, B, \xi_B)$ is one-to-one, this also proves the original optimization problem Eq. (101) has a unique minimizer.

As a consequence, the unique minimizer is determined by the KKT and Slater's conditions for variational problems (Zalinescu, 2002, Theorem 2.9.2). $(A, B, \beta_0, \xi_B)$ is the minimizer of Eq. (106) if and only if, for some scalar $\nu_B$ (Lagrange multiplier), the followings hold:

$$
\begin{aligned}
A\,\|\boldsymbol{\mu}\|_2 + AG_1 + BG_2 + \beta_0Y + \xi_B &= U, \\
\mathbb{E}\left[\ell'(U)(\|\boldsymbol{\mu}\|_2 + G_1)\right] &= 0, \\
\mathbb{E}\left[\ell'(U)G_2\right] - \nu_B\frac{B}{\delta} &= 0, \\
\mathbb{E}\left[\ell'(U)Y\right] &= 0, \\
\ell'(U) + \nu_B\xi_B &= 0, \\
\delta\,\mathbb{E}[\xi_B^2] \leq B^2, \quad \nu_B \geq 0, \quad \nu_B\big(\delta\,\mathbb{E}[\xi_B^2] - B^2\big) &= 0.
\end{aligned}
\tag{107}
$$

Using a similar argument as in the proof of Theorem F.9(a), we can also show that

$$0 < \nu_B < \infty, \qquad \xi_B > 0 \text{ (a.s.)}, \qquad \mathbb{E}[\xi_B^2] = B^2/\delta,$$

which implies $B > 0$. Plugging this into Eq. (107) solves two conditions

$$\mathbb{E}\left[\left(\ell'(U)\right)^2\right] = \nu_B^2 \frac{B^2}{\delta}, \qquad \mathbb{E}\left[\ell'(U)Y\right] = 0. \tag{108}$$

By Stein's identity, we also have relation

$$\mathbb{E}\left[\ell'(U)G_1\right] = A\,\mathbb{E}\left[\ell''(U)\right], \qquad \mathbb{E}\left[\ell'(U)G_2\right] = B\,\mathbb{E}\left[\ell''(U)\right].$$

Combine the above with Eq. (107), we obtain

$$\mathbb{E}\left[\ell'(U)\right] = -\nu_B \frac{A}{\delta \left\|\boldsymbol{\mu}\right\|_2}, \qquad \mathbb{E}\left[\ell'(U)G_1\right] = \nu_B \frac{A}{\delta}, \qquad \mathbb{E}\left[\ell'(U)G_2\right] = \nu_B \frac{B}{\delta},$$

which is equivalent to (recall that $AG_1 + BG_2 \stackrel{\mathrm{d}}{=} RG$)

$$\mathbb{E}\left[\ell'(U)\right] = -\nu_B \frac{A}{\delta \left\|\boldsymbol{\mu}\right\|_2}, \qquad \mathbb{E}\left[\ell'(U)G\right] = \nu_B \frac{R}{\delta}. \tag{109}$$

The above implies $A > 0$ since $\ell' < 0$ by Theorem F.1. Since both $A, B > 0$, by Eq. (105) we have $\rho \in (-1, 1) \setminus \{0\}$ and $R > 0$. Moreover, notice that for any $\rho > 0$,

$$\mathbb{E}\left[\ell\big(-\rho \left\|\boldsymbol{\mu}\right\|_2 R + RG + \beta_0 Y + R\sqrt{1-\rho^2}\xi\big)\right] > \mathbb{E}\left[\ell\big(\rho \left\|\boldsymbol{\mu}\right\|_2 R + RG + \beta_0 Y + R\sqrt{1-\rho^2}\xi\big)\right].$$

Therefore, we must have $\rho \in (0, 1)$. Then we prove $(\rho^*, R^*, \beta_0^*, \lambda^*) \in (0, 1) \times \mathbb{R}_{>0} \times \mathbb{R} \times \mathbb{R}_{>0}$. Lastly, by combining Eq. (108) and (109) with change of variable $\lambda := 1/\nu_B$, and recalling Eq. (100) in the proof of Theorem F.9, we obtain the KKT conditions Eq. (104) expressed in $(\rho, R, \beta_0, \lambda)$. Then we complete the proof of part (b). Finally, part (c) directly follows from Theorem F.9. $\qquad\square$

We are now in position to establish the convergence of parameters.

***Proof of Lemma F.7.*** Consider any $\varepsilon \geq 0$ and $C_R, C_{\beta_0} \in (0, \infty)$, let

$$\boldsymbol{\Theta}_c^*(\varepsilon) := \{(\rho, R, \beta_0) \in [-1, 1] \times [0, C_R] \times [-C_{\beta_0}, C_{\beta_0}] : \left\|(\rho, R, \beta_0) - (\rho^*, R^*, \beta_0^*)\right\|_2 \geq \varepsilon\}$$

and let $\boldsymbol{\Theta}_{\boldsymbol{\beta}}^*(\varepsilon)$ defined as Eq. (91). By Theorem F.2 and F.11, we can choose some $C_R, C_{\beta_0} > 0$ and compact convex set $\boldsymbol{\Xi_u} \subset \mathbb{R}^n$ large enough, such that as $n, d \to \infty$,

$$M_n = M_n(\boldsymbol{\Theta}_{\boldsymbol{\beta}}^*(0), \boldsymbol{\Xi_u}) \text{ (w.h.p.)}, \qquad M^* = M^*(\boldsymbol{\Theta}_c^*(0)).$$

Then according to Theorem F.6, we have global convergence

$$M_n \xrightarrow{\mathrm{P}} M^*.$$

However, for any $\varepsilon > 0$ and $\zeta > 0$, by Theorem F.6 we have

$$M_n(\boldsymbol{\Theta}_{\boldsymbol{\beta}}^*(\varepsilon), \mathbb{R}^n) = M_n(\boldsymbol{\Theta}_{\boldsymbol{\beta}}^*(\varepsilon), \boldsymbol{\Xi_u}) \text{ (w.h.p.)}, \quad \mathbb{P}\left(M_n(\boldsymbol{\Theta}_{\boldsymbol{\beta}}^*(\varepsilon), \boldsymbol{\Xi_u}) \leq M^*(\boldsymbol{\Theta}_c^*(\varepsilon)) - \zeta\right) \to 0.$$

This implies

$$\text{p-}\liminf_{n \to \infty} M_n(\boldsymbol{\Theta}_{\boldsymbol{\beta}}^*(\varepsilon), \mathbb{R}^n) \geq M^*(\boldsymbol{\Theta}_c^*(\varepsilon)) > M^*,$$

where the strict inequality comes from the uniqueness of minimizer $(\rho^*, R^*, \beta_0^*, \xi^*)$, established in Theorem F.11(a). Since $\varepsilon > 0$ can be arbitrarily small, this proves $(\widehat{\rho}_n, \|\widehat{\boldsymbol{\beta}}_n\|_2, \widehat{\beta}_{0,n}) \xrightarrow{\mathrm{P}} (\rho^*, R^*, \beta_0^*)$. Moreover, we know that $R^* > 0$ by Theorem F.11(b). So $\widehat{\boldsymbol{\beta}}_n \neq \mathbf{0}$ and therefore $\widehat{\rho}_n$ is well-defined with high probability. This concludes the proof of Theorem F.7. $\qquad\square$

### F.1.6 ELD CONVERGENCE: PROOF OF LEMMA F.8

***Proof of Lemma F.8***. We first establish the convergence of logit margins. Recall that

$$
\widehat{\mathcal{L}}_n = \frac{1}{n} \sum_{i=1}^n \delta_{y_i(\langle \boldsymbol{x}_i, \widehat{\boldsymbol{\beta}} \rangle + \widehat{\beta}_0)},
$$

$$
\mathcal{L}_* = \mathrm{Law}\,(U^*) := \mathrm{Law}\left(\rho^* \|\boldsymbol{\mu}\|_2 R^* + R^* G + \beta_0^* Y + R^* \sqrt{1 - \rho^{*2}} \xi^*\right)
$$

$$
= \mathrm{Law}\left(\mathrm{prox}_{\lambda^* \ell}(\rho^* \|\boldsymbol{\mu}\|_2 R^* + R^* G + \beta_0^* Y)\right).
$$

For any $\varepsilon > 0$ small enough, we have defined the $\varepsilon$-$W_2$ open ball by

$$
\mathsf{B}_{W_2}(\varepsilon) = \left\{ \mathbf{u} \in \mathbb{R}^n : W_2\left(\frac{1}{n} \sum_{i=1}^n \delta_{u_i}, \mathcal{L}_*\right) < \varepsilon \right\}.
$$

For $C_R, C_{\beta_0} \in (0, \infty)$, let $\boldsymbol{\Theta}_c = [-1, 1] \times [0, C_R] \times [-C_{\beta_0}, C_{\beta_0}]$ and let $\boldsymbol{\Theta}_\beta$ be defined as Eq. (91). When $C_R, C_{\beta_0} > 0$ and compact set $\boldsymbol{\Xi}_\mathbf{u} \subset \mathbb{R}^n$ are large enough, by Theorem F.2 we have

$$
\widetilde{M}_n^\varepsilon := M_n(\mathbb{R}^{d+1}, \mathsf{B}_{W_2}^c(\varepsilon)) = M_n(\boldsymbol{\Theta}_\beta, \boldsymbol{\Xi}_\mathbf{u} \setminus \mathsf{B}_{W_2}(\varepsilon)) \;\; (\text{w.h.p.}),
$$

$$
\widetilde{M}_n^{\varepsilon(3)} := M_n^{(3)}(\boldsymbol{\Theta}_c, \mathsf{B}_{W_2}^c(\varepsilon)) = M_n^{(3)}(\boldsymbol{\Theta}_c, \boldsymbol{\Xi}_\mathbf{u} \setminus \mathsf{B}_{W_2}(\varepsilon)).
$$

Combining these with Theorem F.3 and F.4 obtains that for any $\zeta > 0$,

$$
\lim_{n \to \infty} \mathbb{P}\left(\widetilde{M}_n^\varepsilon \leq \widetilde{M}_n^{\varepsilon(3)} - \zeta\right) = 0. \tag{110}
$$

In order to show $W_2(\widehat{\mathcal{L}}_n, \mathcal{L}_*) \xrightarrow{\mathrm{P}} 0$, our goal is to show that

$$
\lim_{n \to \infty} \mathbb{P}\left(\widetilde{M}_n^\varepsilon > M_n\right) = 1.
$$

Then according to Eq. (110) and Theorem F.7, it suffices to show that

$$
\mathrm{p\text{-}\liminf_{n \to \infty}} \widetilde{M}_n^{\varepsilon(3)} > \mathrm{p\text{-}\lim_{n \to \infty}} M_n = M^*. \tag{111}
$$

By Eq. (89) and (90), recall that

$$
\widetilde{M}_n^{\varepsilon(3)} = \min_{(\rho, R, \beta_0) \in \boldsymbol{\Theta}_c} \min_{\mathbf{u} \in \mathsf{N}_n^\delta(\rho, R, \beta_0) \setminus \mathsf{B}_{W_2}(\varepsilon)} \frac{1}{n} \sum_{i=1}^n \ell(u_i),
$$

where we temporarily define

$$
\mathsf{N}_n^\delta(\rho, R, \beta_0) := \left\{ \mathbf{u} \in \mathbb{R}^n : \frac{1}{\sqrt{n}} \big\| \rho \|\boldsymbol{\mu}\|_2 R \mathbf{1}_n + R\boldsymbol{g} + \beta_0 \boldsymbol{y} - \mathbf{u} \big\|_2 \leq \frac{R\sqrt{1 - \rho^2}}{\sqrt{\delta}} \right\}.
$$

Now we split $\widetilde{M}_n^{\varepsilon(3)}$ into two parts by

$$
\widetilde{M}_n^{\varepsilon(3)} = \min\{I, II\} := \min \left\{ \min_{\substack{(\rho, R, \beta_0) \in \boldsymbol{\Theta}_c \setminus \mathsf{B}_{2,c^*}(\eta) \\ \mathbf{u} \in \mathsf{N}_n^\delta(\rho, R, \beta_0) \setminus \mathsf{B}_{W_2}(\varepsilon)}} \frac{1}{n} \sum_{i=1}^n \ell(u_i), \min_{\substack{(\rho, R, \beta_0) \in \boldsymbol{\Theta}_c \cap \mathsf{B}_{2,c^*}(\eta) \\ \mathbf{u} \in \mathsf{N}_n^\delta(\rho, R, \beta_0) \setminus \mathsf{B}_{W_2}(\varepsilon)}} \frac{1}{n} \sum_{i=1}^n \ell(u_i) \right\},
$$

where $\eta > 0$ and

$$
\mathsf{B}_{2,c^*}(\eta) = \left\{ (\rho, R, \beta_0) \in \mathbb{R}^3 : \|(\rho, R, \beta_0) - (\rho^*, R^*, \beta_0^*)\|_2 < \eta \right\}
$$

is a $\eta$-$\mathcal{L}^2$ open ball around the global minimizer $(\rho^*, R^*, \beta_0^*)$.

For the first term, with $\boldsymbol{\Theta}_c$ large enough such that $(\rho^*, R^*, \beta_0^*) \in \boldsymbol{\Theta}_c$, by Theorem F.5 we have

$$
I \geq \min_{(\rho, R, \beta_0) \in \boldsymbol{\Theta}_c \setminus \mathsf{B}_{2,c^*}(\eta)} \min_{\mathbf{u} \in \mathsf{N}_n^\delta(\rho, R, \beta_0)} \frac{1}{n} \sum_{i=1}^n \ell(u_i)
$$

$$
= \min_{\substack{(\rho, R, \beta_0) \in \boldsymbol{\Theta}_c \setminus \mathsf{B}_{2,c^*}(\eta) \\ \xi \in \mathcal{L}^2(\mathbb{Q}_n), \|\xi\|_{\mathbb{Q}_n}^2 \leq 1/\delta}} \mathbb{E}_{\mathbb{Q}_n}\left[\ell\big(\rho \|\boldsymbol{\mu}\|_2 R + RG + \beta_0 Y + R\sqrt{1 - \rho^2} \xi\big)\right]
$$

$$
= M_n^{(3)}\big(\boldsymbol{\Theta}_c \setminus \mathsf{B}_{2,c^*}(\eta)\big) \xrightarrow{\mathrm{P}} M_n^*\big(\boldsymbol{\Theta}_c \setminus \mathsf{B}_{2,c^*}(\eta)\big) > M^*(\boldsymbol{\Theta}_c) = M^*,
$$

where the strict inequality follows from the uniqueness of $(\rho^*, R^*, \beta_0^*)$ according to Theorem F.11(a).

For the second term, we can take $\eta > 0$ small enough, such that $(\rho, R, \beta_0) \in \mathsf{B}_{2,c^*}(\eta)$ implies

$$
\begin{aligned}
&W_2\left(\mathtt{Law}\left(U^*_{\rho,R,\beta_0}\right), \mathcal{L}_*\right) \\
&= W_2\left(\mathtt{Law}\left(U^*_{\rho,R,\beta_0}\right), \mathtt{Law}\left(U^*_{\rho^*,R^*,\beta_0^*}\right)\right) \leq \frac{\varepsilon}{2}, \qquad \forall (\rho, R, \beta_0) \in \Theta_c \cap \mathsf{B}_{2,c^*}(\eta),
\end{aligned}
$$

where $U^*_{\rho,R,\beta_0} := \rho \left\|\boldsymbol{\mu}\right\|_2 R + RG + \beta_0 Y + R\sqrt{1-\rho^2}\xi^*_{\mathbb{Q}_\infty}(\rho, R, \beta_0)$, and $\xi^*_{\mathbb{Q}_\infty}(\rho, R, \beta_0)$ is the unique minimizer of $\mathscr{R}_{\mathbb{Q}}(\xi)$ defined in Eq. (95), with an expression given by Eq. (96). The existence of such $\eta > 0$ is guaranteed by continuity of $W_2$ distance and $(\rho, R, \beta_0) \mapsto U^*_{\rho,R,\beta_0}$ by Theorem F.9. Then $\mathbf{u} \notin \mathsf{B}_{W_2}(\varepsilon)$ implies (by triangle inequality)

$$
W_2\left(\frac{1}{n}\sum_{i=1}^n \delta_{u_i}, \mathtt{Law}\left(U^*_{\rho,R,\beta_0}\right)\right) \geq \frac{\varepsilon}{2}, \qquad \forall (\rho, R, \beta_0) \in \Theta_c \cap \mathsf{B}_{2,c^*}(\eta).
$$

Thus we have

$$
II = \min_{\substack{(\rho,R,\beta_0)\in\Theta_c\cap\mathsf{B}_{2,c^*}(\eta) \\ \mathbf{u}\in\mathsf{N}_n^\delta(\rho,R,\beta_0)\backslash\mathsf{B}_{W_2}(\varepsilon)}} \frac{1}{n}\sum_{i=1}^n \ell(u_i) \geq \min_{\substack{(\rho,R,\beta_0)\in\Theta_c \\ U\in\mathcal{N}_n^\delta(\rho,R,\beta_0)\cap\mathcal{C}_n^\varepsilon(\rho,R,\beta_0)}} \mathbb{E}_{\mathbb{Q}_n}[\ell(U)], \qquad (112)
$$

where denote

$$
\mathcal{C}_n^\varepsilon(\rho, R, \beta_0) := \left\{U \in \mathcal{L}^2(\mathbb{Q}_\infty) : \left\|U - U^*_{\rho,R,\beta_0}\right\|_{\mathbb{Q}_n} \geq \frac{\varepsilon}{2}\right\} \qquad (113)
$$

and recall Eq. (90) that

$$
\mathcal{N}_n^\delta(\rho, R, \beta_0) = \left\{U \in \mathcal{L}^2(\mathbb{Q}_n) : \left\|\rho\left\|\boldsymbol{\mu}\right\|_2 R + RG + \beta_0 Y - U\right\|_{\mathbb{Q}_n} \leq \frac{R\sqrt{1-\rho^2}}{\sqrt{\delta}}\right\}. \qquad (114)
$$

Now, denote $\widehat{U}_{\rho,R,\beta_0} := \rho\left\|\boldsymbol{\mu}\right\|_2 R + RG + \beta_0 Y + R\sqrt{1-\rho^2}\xi^*_{\mathbb{Q}_n}(\rho, R, \beta_0)$. According to Theorem F.9, we know that $\|\xi^*_{\mathbb{Q}_n}(\rho, R, \beta_0)\|^2_{\mathbb{Q}_n} = 1/\delta$, that is,

$$
\left\|\rho\left\|\boldsymbol{\mu}\right\|_2 R + RG + \beta_0 Y - \widehat{U}_{\rho,R,\beta_0}\right\|_{\mathbb{Q}_n} = \frac{R\sqrt{1-\rho^2}}{\sqrt{\delta}}. \qquad (115)
$$

We claim $\left\|U^*_{\rho,R,\beta_0} - \widehat{U}_{\rho,R,\beta_0}\right\|_{\mathbb{Q}_n} \xrightarrow{\text{P}} 0$. Otherwise, there exits a convergent sequence $\{\widehat{\lambda}_m\}_{m\in\mathbb{N}}$ such that p-$\lim_{m\to\infty}\widehat{\lambda}_m \neq \lambda^*$, where $\widehat{\lambda}_m$ satisfies the conditions in Theorem F.9(a) under $\mathbb{Q} = \mathbb{Q}_m$, and $\lambda^*$ satisfies the conditions in Theorem F.9(a) under $\mathbb{Q} = \mathbb{Q}_\infty$. This contradicts the convergence $\arg\max_{\nu>0}\mathscr{R}_{\nu,\mathbb{Q}_m}(A, B, \beta_0) \xrightarrow{\text{P}} \arg\max_{\nu>0}\mathscr{R}_{\nu,\mathbb{Q}_\infty}(A, B, \beta_0)$ by an argmax theorem for the concave process (Liese & Miescke, 2008, Theorem 7.77) according to Theorem F.9(b), and change of variable $\nu = R\sqrt{1-\rho^2}/\lambda$. Hence, for all $n$ large enough, we have

$$
\left\|U^*_{\rho,R,\beta_0} - \widehat{U}_{\rho,R,\beta_0}\right\|_{\mathbb{Q}_n} \leq \frac{\varepsilon}{2}.
$$

Combining this with Eq. (113)—(115) together, by triangle inequality, we obtain

$$
\mathcal{N}_n^\delta(\rho, R, \beta_0) \cap \mathcal{C}_n^\varepsilon(\rho, R, \beta_0) \subseteq \widetilde{\mathcal{N}}_n^{\delta,\varepsilon}(\rho, R, \beta_0) \qquad (116)
$$

where

$$
\widetilde{\mathcal{N}}_n^{\delta,\varepsilon}(\rho, R, \beta_0) := \left\{U \in \mathcal{L}^2(\mathbb{Q}_n) : \left\|\rho\left\|\boldsymbol{\mu}\right\|_2 R + RG + \beta_0 Y - U\right\|_{\mathbb{Q}_n} \leq \frac{R\sqrt{1-\rho^2}}{\sqrt{\delta}} - \varepsilon\right\}.
$$

Recall that $C_R = \max_{(\rho,R,\beta_0)\in\Theta_c} R$. Denote $\delta'_\varepsilon > \delta$ as a constant such that

$$
\frac{1}{\sqrt{\delta'_\varepsilon}} := \frac{1}{\sqrt{\delta}} - \frac{\varepsilon}{C_R}. \qquad (117)
$$

Then following Eq. (112), we have

$$
\begin{aligned}
II &\geq \min_{(\rho,R,\beta_0)\in\Theta_c}\ \min_{U\in\mathcal{N}_n^\delta(\rho,R,\beta_0)\cap\mathcal{C}_n^\varepsilon(\rho,R,\beta_0)} \mathbb{E}_{\mathbb{Q}_n}[\ell(U)] \\
&\overset{(i)}{\geq} \min_{(\rho,R,\beta_0)\in\Theta_c}\ \min_{U\in\widetilde{\mathcal{N}}_n^{\delta,\varepsilon}(\rho,R,\beta_0)} \mathbb{E}_{\mathbb{Q}_n}[\ell(U)] \\
&= \min_{(\rho,R,\beta_0)\in\Theta_c}\ \min_{\xi\in\mathcal{L}^2(\mathbb{Q}_n),\|\xi\|_{\mathbb{Q}_n}\leq\frac{1}{\sqrt{\delta}}-\frac{\varepsilon}{R\sqrt{1-\rho^2}}} \mathbb{E}_{\mathbb{Q}_n}\left[\ell\left(\rho\|\boldsymbol{\mu}\|_2 R + RG + \beta_0 Y + R\sqrt{1-\rho^2}\xi\right)\right] \\
&\overset{(ii)}{\geq} \min_{(\rho,R,\beta_0)\in\Theta_c}\ \min_{\xi\in\mathcal{L}^2(\mathbb{Q}_n),\|\xi\|_{\mathbb{Q}_n}^2\leq 1/\delta_\varepsilon'} \mathbb{E}_{\mathbb{Q}_n}\left[\ell\left(\rho\|\boldsymbol{\mu}\|_2 R + RG + \beta_0 Y + R\sqrt{1-\rho^2}\xi\right)\right] \\
&\overset{\mathrm{P}}{\to} \min_{(\rho,R,\beta_0)\in\Theta_c}\ \min_{\xi\in\mathcal{L}^2(\mathbb{Q}_\infty),\|\xi\|_{\mathbb{Q}_\infty}^2\leq 1/\delta_\varepsilon'} \mathbb{E}\left[\ell\left(\rho\|\boldsymbol{\mu}\|_2 R + RG + \beta_0 Y + R\sqrt{1-\rho^2}\xi\right)\right] \\
&\overset{(iii)}{>} \min_{(\rho,R,\beta_0)\in\Theta_c}\ \min_{\xi\in\mathcal{L}^2(\mathbb{Q}_\infty),\|\xi\|_{\mathbb{Q}_\infty}^2\leq 1/\delta} \mathbb{E}\left[\ell\left(\rho\|\boldsymbol{\mu}\|_2 R + RG + \beta_0 Y + R\sqrt{1-\rho^2}\xi\right)\right] \\
&= M^*(\Theta_c) = M^*,
\end{aligned}
$$

where (i) follows from Eq. (116), (ii) follows from Eq. (117) and the fact that

$$
\frac{1}{\sqrt{\delta}} - \frac{\varepsilon}{R\sqrt{1-\rho^2}} \leq \frac{1}{\sqrt{\delta_\varepsilon'}}, \qquad \forall\,(\rho,R,\beta_0)\in\Theta_c,
$$

the convergence follows from Theorem F.5, and (iii) follows from the uniqueness of $(\rho^*,R^*,\beta_0^*)$ and KKT conditions $\|\xi^*\|_{\mathbb{Q}_\infty}^2 = 1/\delta$ in Theorem F.11.

Finally, combining everthing together, we have

$$
\operatorname*{p-lim\,inf}_{n\to\infty} \widetilde{M}_n^{\varepsilon(3)} \geq \min\left\{\operatorname*{p-lim\,inf}_{n\to\infty} I,\ \operatorname*{p-lim\,inf}_{n\to\infty} II\right\} > M^*.
$$

This shows Eq. (111), and hence completes the proof. □

Using an argument similar to the one at the end of the proof of Theorem E.7, we can show the convergence of empirical logit distribution $W_2(\widehat{\nu}_n,\nu_*)\overset{\mathrm{P}}{\to}0$ from $W_2(\widehat{\mathcal{L}}_n,\mathcal{L}_*)\overset{\mathrm{P}}{\to}0$ given by Theorem F.8.

### F.1.7 COMPLETING THE PROOF OF LEMMA D.3

***Proof of Lemma D.3.*** Consider the ERM problem Eq. (34) with arbitrary $\tau > 0$. Recall that $\widetilde{y}_i = y_i/s(y_i)$ where $s : \{\pm 1\} \to \{1\}\cup\{\tau\}$ is defined as per Eq. (29). $M_n$ is redefined as Eq. (34)

$$
M_n := \min_{\boldsymbol{\beta}\in\mathbb{R}^d,\,\beta_0\in\mathbb{R}} \frac{1}{n}\sum_{i=1}^n \ell\big(\widetilde{y}_i(\langle\boldsymbol{x}_i,\boldsymbol{\beta}\rangle + \beta_0)\big).
$$

Under this modification, $M_n(\Theta_{\boldsymbol{\beta}},\Xi_{\mathbf{u}})$ can be redefined and expressed as

$$
\begin{aligned}
M_n(\Theta_{\boldsymbol{\beta}},\Xi_{\mathbf{u}}) &:= \min_{\substack{(\boldsymbol{\beta},\beta_0)\in\Theta_{\boldsymbol{\beta}} \\ \mathbf{u}\in\Xi_{\mathbf{u}}}}\ \max_{\mathbf{v}\in\mathbb{R}^n}\left\{\frac{1}{n}\sum_{i=1}^n \ell\left(\frac{u_i}{s(y_i)}\right) + \frac{1}{n}\sum_{i=1}^n v_i\left(y_i(\langle\boldsymbol{x}_i,\boldsymbol{\beta}\rangle + \beta_0) - u_i\right)\right\} \\
&= \min_{\substack{(\boldsymbol{\beta},\beta_0)\in\Theta_{\boldsymbol{\beta}} \\ \mathbf{u}\in\Xi_{\mathbf{u}}}}\ \max_{\mathbf{v}\in\mathbb{R}^n}\left\{\frac{1}{n}\sum_{i=1}^n \ell\left(\frac{u_i}{s(y_i)}\right) + \frac{1}{n}\mathbf{v}^\top\mathbf{1}\langle\boldsymbol{\mu},\boldsymbol{\beta}\rangle + \frac{1}{n}\mathbf{v}^\top Z\boldsymbol{\beta} + \frac{1}{n}\beta_0\mathbf{v}^\top\boldsymbol{y} - \frac{1}{n}\mathbf{v}^\top\mathbf{u}\right\}.
\end{aligned}
$$

Consequently, quantities $M_n^{(k)}$, $k = 1, 2, 3$ and $M^*$ used in the proof can be similarly redefined as

$$M_n^{(1)}(\boldsymbol{\Theta}_{\boldsymbol{\beta}}, \boldsymbol{\Xi}_{\mathbf{u}}) := \min_{\substack{(\boldsymbol{\beta}, \beta_0) \in \boldsymbol{\Theta}_{\boldsymbol{\beta}} \\ \mathbf{u} \in \boldsymbol{\Xi}_{\mathbf{u}}}} \max_{\mathbf{v} \in \mathbb{R}^n} \left\{ \frac{1}{n} \sum_{i=1}^n \ell\left( \frac{u_i}{s(y_i)} \right) + \frac{1}{n} \mathbf{v}^{\mathsf{T}} \mathbf{1} \langle \boldsymbol{\mu}, \boldsymbol{\beta} \rangle + \frac{1}{n} \|\mathbf{v}\|_2 \, \boldsymbol{h}^{\mathsf{T}} \boldsymbol{\beta} \right.$$

$$\left. + \frac{1}{n} \|\boldsymbol{\beta}\|_2 \, \boldsymbol{g}^{\mathsf{T}} \mathbf{v} + \frac{1}{n} \beta_0 \mathbf{v}^{\mathsf{T}} \boldsymbol{y} - \frac{1}{n} \mathbf{v}^{\mathsf{T}} \mathbf{u} \right\},$$

$$M_n^{(2)}(\boldsymbol{\Theta}_c, \boldsymbol{\Xi}_{\mathbf{u}}) := \min_{(\rho, R, \beta_0) \in \boldsymbol{\Theta}_c} \min_{U \in \Xi_u \cap \mathcal{N}_n} \mathbb{E}_{\mathbb{Q}_n} \left[ \ell\big( U/s(Y) \big) \right],$$

$$M_n^{(3)}(\boldsymbol{\Theta}_c, \boldsymbol{\Xi}_{\mathbf{u}}) := \min_{(\rho, R, \beta_0) \in \boldsymbol{\Theta}_c} \min_{U \in \Xi_u \cap \mathcal{N}_n^\delta} \mathbb{E}_{\mathbb{Q}_n} \left[ \ell\big( U/s(Y) \big) \right],$$

$$M_n^{(3)}(\boldsymbol{\Theta}_c) := \min_{(\rho, R, \beta_0) \in \boldsymbol{\Theta}_c} \min_{U \in \mathcal{N}_n^\delta} \mathbb{E}_{\mathbb{Q}_n} \left[ \ell\big( U/s(Y) \big) \right],$$

$$= \min_{\substack{(\rho, R, \beta_0) \in \boldsymbol{\Theta}_c \\ \xi \in \mathcal{L}^2(\mathbb{Q}_n), \|\xi\|_{\mathbb{Q}_n} \leq 1/\sqrt{\delta}}} \mathbb{E}_{\mathbb{Q}_n} \left[ \ell\left( \frac{\rho \|\boldsymbol{\mu}\|_2 R + RG + \beta_0 Y + R\sqrt{1 - \rho^2}\xi}{s(Y)} \right) \right],$$

$$M^*(\boldsymbol{\Theta}_c) := \min_{\substack{(\rho, R, \beta_0) \in \boldsymbol{\Theta}_c \\ \xi \in \mathcal{L}^2(\mathbb{Q}_\infty), \|\xi\|_{\mathbb{Q}_\infty} \leq 1/\sqrt{\delta}}} \mathbb{E} \left[ \ell\left( \frac{\rho \|\boldsymbol{\mu}\|_2 R + RG + \beta_0 Y + R\sqrt{1 - \rho^2}\xi}{s(Y)} \right) \right],$$

$$M^* := M^*([-1, 1] \times \mathbb{R}_{\geq 0} \times \mathbb{R}),$$

where $\mathcal{N}_n$, $\mathcal{N}_n^\delta$ are still defined as Eq. (88), (90), and we still apply the change of variable

$$U = \rho \|\boldsymbol{\mu}\|_2 R + RG + \beta_0 Y + R\sqrt{1 - \rho^2}\xi.$$

One can use exactly similar arguments to conclude Theorem F.2—F.5 and Theorem F.6 with definitions above. For Theorem F.9, we can also get similar results, but the KKT condition in Eq. (98) now becomes

$$U + \lambda \ell'(U/s(Y)) = \rho \|\boldsymbol{\mu}\|_2 R + RG + \beta_0 Y,$$

which implies

$$\frac{U}{s(Y)} = \mathsf{prox}_\ell \left( \frac{\rho \|\boldsymbol{\mu}\|_2 R + RG + \beta_0 Y}{s(Y)}; \frac{\lambda}{s(Y)} \right), \tag{118}$$

as a substitute of Eq. (100), and

$$\xi_{\mathbb{Q}}^*(\rho, R, \beta_0) = -\frac{\lambda}{R\sqrt{1 - \rho^2}} \ell' \left( \mathsf{prox}_\ell \left( \frac{\rho \|\boldsymbol{\mu}\|_2 R + RG + \beta_0 Y}{s(Y)}; \frac{\lambda}{s(Y)} \right) \right),$$

as a substitute of Eq. (96).

**(a):** According to the definition above, the KKT conditions Theorem F.11 will become

$$-\frac{R\rho}{\lambda\delta \|\boldsymbol{\mu}\|_2} = \mathbb{E} \left[ \widetilde{\ell}'_Y(U) \right], \tag{119}$$

$$\frac{R}{\lambda\delta} = \mathbb{E} \left[ \widetilde{\ell}'_Y(U)G \right], \tag{120}$$

$$0 = \mathbb{E} \left[ \widetilde{\ell}'_Y(U)Y \right], \tag{121}$$

$$\frac{R^2(1 - \rho^2)}{\lambda^2\delta} = \mathbb{E} \left[ \big( \widetilde{\ell}'_Y(U) \big)^2 \right],$$

where

$$\widetilde{\ell}'_Y(U) := \frac{1}{s(Y)} \ell'\left( \frac{U}{s(Y)} \right) = \frac{1}{s(Y)} \ell' \left( \mathsf{prox}_\ell \left( \frac{\rho \|\boldsymbol{\mu}\|_2 R + RG + \beta_0 Y}{s(Y)}; \frac{\lambda}{s(Y)} \right) \right),$$

and $U$ follows the relation Eq. (118). By Stein's identity, Eq. (120) can be expressed as

$$\frac{R}{\lambda\delta} = \mathbb{E}\left[\widetilde{\ell}'_Y(U)G\right] = \mathbb{E}\left[\frac{1}{s(Y)}\ell''\left(\frac{U}{s(Y)}\right)\cdot\frac{\mathrm{d}(U/s(Y))}{\mathrm{d}G}\right]$$

$$= \mathbb{E}\left[\frac{1}{s(Y)}\ell''\left(\frac{U}{s(Y)}\right)\cdot\frac{1}{1+\dfrac{\lambda}{s(Y)}\ell''\left(\dfrac{U}{s(Y)}\right)}\cdot\frac{R}{s(Y)}\right],$$

which gives the third KKT condition in (a). Besides, Eq. (119) and 121 can be rewritten as

$$-\frac{R\rho}{\lambda\delta\|\boldsymbol{\mu}\|_2} = \pi\mathbb{E}\left[\frac{1}{\tau}\ell'\left(\mathsf{prox}_\ell\left(\frac{\rho\|\boldsymbol{\mu}\|_2 R + RG + \beta_0}{\tau};\frac{\lambda}{\tau}\right)\right)\right]$$
$$+ (1-\pi)\mathbb{E}\left[\ell'\left(\mathsf{prox}_\ell\left(\rho\|\boldsymbol{\mu}\|_2 R + RG - \beta_0;\lambda\right)\right)\right],$$

$$0 = \pi\mathbb{E}\left[\frac{1}{\tau}\ell'\left(\mathsf{prox}_\ell\left(\frac{\rho\|\boldsymbol{\mu}\|_2 R + RG + \beta_0}{\tau};\frac{\lambda}{\tau}\right)\right)\right]$$
$$- (1-\pi)\mathbb{E}\left[\ell'\left(\mathsf{prox}_\ell\left(\rho\|\boldsymbol{\mu}\|_2 R + RG - \beta_0;\lambda\right)\right)\right],$$

which solves the first two KKT conditions in (a). This concludes the proof of part (a).

**(b):** Theorem F.7 still remains valid under arbitrary $\tau > 0$, which concludes the proof.

**(c):** Similar to the proof of Theorem D.1(e), we can show that for any test point $(\boldsymbol{x}_{\mathrm{new}}, y_{\mathrm{new}})$,

$$\widehat{f}(\boldsymbol{x}_{\mathrm{new}}) = \langle\boldsymbol{x}_{\mathrm{new}}, \widehat{\boldsymbol{\beta}}_n\rangle + \widehat{\beta}_{0,n} \xrightarrow{\mathrm{d}} y_{\mathrm{new}} R^*\rho^*\|\boldsymbol{\mu}\|_2 + R^*G + \beta_0^*,$$

where $(y^{\mathrm{new}}, G) \sim P_y \times \mathsf{N}(0,1)$. Therefore, by bounded convergence theorem, the errors have limits

$$\lim_{n\to\infty} \mathrm{Err}_{+,n} = \mathbb{P}\left(+R^*\rho^*\|\boldsymbol{\mu}\|_2 + R^*G + \beta_0^* \leq 0\right) = \Phi\left(-\rho^*\|\boldsymbol{\mu}\|_2 - \frac{\beta_0^*}{R^*}\right),$$

$$\lim_{n\to\infty} \mathrm{Err}_{-,n} = \mathbb{P}\left(-R^*\rho^*\|\boldsymbol{\mu}\|_2 + R^*G + \beta_0^* > 0\right) = \Phi\left(-\rho^*\|\boldsymbol{\mu}\|_2 + \frac{\beta_0^*}{R^*}\right).$$

This concludes the proof of part (c).

**(d):** Based on Eq. (118), we redefine $\mathcal{L}_*$ in Section F.1.6 by

$$\mathcal{L}_* := \mathtt{Law}\left(U^*\right) = \mathtt{Law}\left(s(Y)\,\mathsf{prox}_\ell\left(\frac{\rho^*\|\boldsymbol{\mu}\|_2 R^* + R^*G + \beta_0^*Y}{s(Y)};\frac{\lambda^*}{s(Y)}\right)\right).$$

Then Theorem F.8 and the corresponding convergence of ELD still hold. The convergence of TLD directly comes from the proof of part (c). This concludes the proof of part (d).

Finally, we complete the proof of Theorem D.3. $\qquad\square$

# G   MARGIN REBALANCING IN PROPORTIONAL REGIME: PROOFS FOR SECTION D.2.1

## G.1   PROOFS OF PROPOSITIONS D.4 AND D.5

We show the monotonicity of $\mathrm{Err}_+^*$ for $\tau = 1$ in this subsection by first analyzing the monotonicity of asymptotic parameters $\rho^*, \beta_0^*$, which are the solution to the system of equations in Theorem E.9. We restate these equations here.

$$\pi\delta\cdot g\left(\frac{\rho}{2\pi\|\boldsymbol{\mu}\|_2\delta}\right) + (1-\pi)\delta\cdot g\left(\frac{\rho}{2(1-\pi)\|\boldsymbol{\mu}\|_2\delta}\right) = 1 - \rho^2, \tag{122a}$$

$$-\beta_0 + \kappa\tau = \rho\|\boldsymbol{\mu}\|_2 + g_1^{-1}\left(\frac{\rho}{2\pi\|\boldsymbol{\mu}\|_2\delta}\right), \tag{122b}$$

$$\beta_0 + \kappa = \rho\|\boldsymbol{\mu}\|_2 + g_1^{-1}\left(\frac{\rho}{2(1-\pi)\|\boldsymbol{\mu}\|_2\delta}\right). \tag{122c}$$

The properties of functions $g_1, g_2, g$ therein are summarized below.

**Lemma G.1.** *Recall* $g_1(x) = \mathbb{E}\left[(G + x)_+\right]$, $g_2(x) = \mathbb{E}\left[(G + x)_+^2\right]$, *and* $g = g_2 \circ g_1^{-1}$.

 (a) $g_1$, $g_2$ *are increasing maps from* $\mathbb{R}$ *to* $\mathbb{R}_{>0}$, *and* $g : \mathbb{R}_{>0} \to \mathbb{R}_{>0}$ *is increasing with* $g(0^+) = 0$.

 (b) $g_1$, $g_2$ *have explicit expressions*
$$g_1(x) = x\Phi(x) + \phi(x), \qquad g_2(x) = (x^2 + 1)\Phi(x) + x\phi(x).$$

 (c) $g_1(x) \sim x$, $g_2(x) \sim x^2$, *and* $g(x) \sim x^2$, *as* $x \to +\infty$.

The following preliminary result gives the monotonicity of $\rho^*$. By Theorem D.1(b), $\rho^* \in (0, 1)$ is invariant with respect to $\tau$. Hence $\rho^*$ can be viewed as a function of model parameters $(\pi, \|\boldsymbol{\mu}\|_2, \delta)$ determined by Eq. (122a).

**Lemma G.2** (Monotonicity of $\rho^*$). $\rho^*$ *is an increasing function of* $\pi \in (0, \frac{1}{2})$, $\|\boldsymbol{\mu}\|_2$, *and* $\delta$.

*Proof.* Recall that $\rho^* \in (0, 1)$ as stated in Theorem D.1(b).

**(a) ↑ in** $\|\boldsymbol{\mu}\|_2$**:** This point is obvious from Eq. (122a) and Lemma G.1(a).

**(b) ↑ in** $\delta$**:** Notice that Theorem G.6 implies $x \mapsto x \cdot g(1/x)$ is decreasing in $x$. As a consequence, if we fix $\rho$ and increase $\delta$ on the L.H.S. of Eq. (122a), then the L.H.S. will decrease, and $\rho^*$ have to increase to match the R.H.S.. Therefore, $\rho^*$ is an increasing function of $\delta$.

**(c) ↑ in** $\pi \in (0, \frac{1}{2})$**:** We prove this using a similar strategy. Define
$$x_1 = x_1(\pi) := g_1^{-1}\left(\frac{\rho}{2\pi \|\boldsymbol{\mu}\|_2 \delta}\right), \quad x_2 = x_2(\pi) := g_1^{-1}\left(\frac{\rho}{2(1 - \pi) \|\boldsymbol{\mu}\|_2 \delta}\right),$$
then we know that the L.H.S. of Eq. (122a) (for fixed $\delta$ and $\|\boldsymbol{\mu}\|_2$) is proportional to
$$\rho \cdot \left(\frac{g_2(x_1(\pi))}{g_1(x_1(\pi))} + \frac{g_2(x_2(\pi))}{g_1(x_2(\pi))}\right), \tag{123}$$
with the only constraint on $x_1$ and $x_2$ being
$$\frac{1}{g_1(x_1(\pi))} + \frac{1}{g_1(x_2(\pi))} = C := \frac{2 \|\boldsymbol{\mu}\|_2 \delta}{\rho}.$$
Taking derivative with respect to $\pi$, it follows that
$$-\frac{g_1'(x_1)}{g_1^2(x_1)} \cdot x_1'(\pi) - \frac{g_1'(x_2)}{g_1^2(x_2)} \cdot x_2'(\pi) = 0, \quad \implies \quad x_1'(\pi) = -\frac{g_1^2(x_1)}{g_1'(x_1)} \cdot \frac{g_1'(x_2)}{g_1^2(x_2)} \cdot x_2'(\pi),$$
thus leading to
$$\frac{\mathrm{d}}{\mathrm{d}\pi}\left(\frac{g_2(x_1(\pi))}{g_1(x_1(\pi))} + \frac{g_2(x_2(\pi))}{g_1(x_2(\pi))}\right)$$
$$= \frac{g_2'(x_1)g_1(x_1) - g_2(x_1)g_1'(x_1)}{g_1^2(x_1)} \cdot x_1'(\pi) + \frac{g_2'(x_2)g_1(x_2) - g_2(x_2)g_1'(x_2)}{g_1^2(x_2)} \cdot x_2'(\pi)$$
$$= -\frac{g_1'(x_2)}{g_1^2(x_2)} \cdot x_2'(\pi) \cdot \left(\frac{g_2'(x_1)g_1(x_1) - g_2(x_1)g_1'(x_1)}{g_1'(x_1)} - \frac{g_2'(x_2)g_1(x_2) - g_2(x_2)g_1'(x_2)}{g_1'(x_2)}\right)$$
$$= -\frac{g_1'(x_2)}{g_1^2(x_2)} \cdot x_2'(\pi) \cdot (h(x_1) - h(x_2)),$$
where
$$h(x) := \frac{g_2'(x)g_1(x) - g_2(x)g_1'(x)}{g_1'(x)}, \quad \forall x \in \mathbb{R}$$
is a monotone increasing function according to the proof of Theorem G.6. Therefore, $h(x_1) > h(x_2)$ (since $\pi < 1/2 \implies x_1 > x_2$). By definitions of $x_2$ and $g_1$, we know that $x_2'(\pi) > 0$ and $g_1'(x_2) > 0$. As a consequence,
$$\frac{\mathrm{d}}{\mathrm{d}\pi}\left(\frac{g_2(x_1(\pi))}{g_1(x_1(\pi))} + \frac{g_2(x_2(\pi))}{g_1(x_2(\pi))}\right) < 0.$$
Similar to points (a) and (b), by combining Eq. (122a) and (123), we conclude that $\rho^*$ is an increasing function of $\pi \in (0, \frac{1}{2})$. This completes the proof. $\qquad\square$

As long as $\tau \neq 0$, the linear system Eq. (122b) and (122c) for $(\beta_0, \tau)$ is non-singular, so one can solve for $\beta_0$ and $\kappa$:

$$\beta_0 = \frac{1}{1+\tau}\left(\tau g_1^{-1}\left(\frac{\rho}{2(1-\pi)\|\boldsymbol{\mu}\|_2 \delta}\right) - g_1^{-1}\left(\frac{\rho}{2\pi\|\boldsymbol{\mu}\|_2 \delta}\right) + (\tau - 1)\rho\|\boldsymbol{\mu}\|_2\right), \qquad (124a)$$

$$\kappa = \frac{1}{1+\tau}\left(g_1^{-1}\left(\frac{\rho}{2(1-\pi)\|\boldsymbol{\mu}\|_2 \delta}\right) + g_1^{-1}\left(\frac{\rho}{2\pi\|\boldsymbol{\mu}\|_2 \delta}\right) + 2\rho\|\boldsymbol{\mu}\|_2\right). \qquad (124b)$$

The following lemma establishes the monotonicity of $\beta_0^*$ when $\tau = 1$.

**Lemma G.3** (Monotonicity of $\beta_0^*$). *$\beta_0^*$ is an increasing function of $\pi \in (0, \frac{1}{2})$, $\|\boldsymbol{\mu}\|_2$, and $\delta$, when $\tau = 1$ (without margin rebalancing). Moreover, $\beta_0^* < 0$.*

*Proof.* When $\tau = 1$, the above equations reduce to

$$\beta_0 = \frac{1}{2}\left(g_1^{-1}\left(\frac{\rho}{2(1-\pi)\|\boldsymbol{\mu}\|_2 \delta}\right) - g_1^{-1}\left(\frac{\rho}{2\pi\|\boldsymbol{\mu}\|_2 \delta}\right)\right), \qquad (125)$$

$$\kappa = \frac{1}{2}\left(g_1^{-1}\left(\frac{\rho}{2(1-\pi)\|\boldsymbol{\mu}\|_2 \delta}\right) + g_1^{-1}\left(\frac{\rho}{2\pi\|\boldsymbol{\mu}\|_2 \delta}\right) + 2\rho\|\boldsymbol{\mu}\|_2\right).$$

Clearly $\beta_0^* < 0$, since $g_1^{-1}$ is an increasing function and $\pi < \frac{1}{2}$.

**(a) $\uparrow$ in $\|\mu\|_2$:** Fixing $\pi$ and $\delta$, taking derivative with respect to $\|\boldsymbol{\mu}\|_2$ in Eq. (125), we have

$$\frac{\mathrm{d}\beta_0}{\mathrm{d}\|\boldsymbol{\mu}\|_2} = \frac{1}{2}\left(\frac{1}{2(1-\pi)\delta}\cdot(g_1^{-1})'\left(\frac{\rho}{2(1-\pi)\|\boldsymbol{\mu}\|_2 \delta}\right) - \frac{1}{2\pi\delta}\cdot(g_1^{-1})'\left(\frac{\rho}{2\pi\|\boldsymbol{\mu}\|_2 \delta}\right)\right)\cdot\frac{\mathrm{d}}{\mathrm{d}\|\boldsymbol{\mu}\|_2}\left(\frac{\rho}{\|\boldsymbol{\mu}\|_2}\right).$$

Since $\pi < \frac{1}{2}$, from Theorem G.7 we know that

$$\frac{1}{2(1-\pi)\delta}\cdot(g_1^{-1})'\left(\frac{\rho}{2(1-\pi)\|\boldsymbol{\mu}\|_2 \delta}\right) - \frac{1}{2\pi\delta}\cdot(g_1^{-1})'\left(\frac{\rho}{2\pi\|\boldsymbol{\mu}\|_2 \delta}\right) < 0.$$

According to Theorem G.2, if we increase $\|\boldsymbol{\mu}\|_2$, then $\rho$ will increase, and Eq. (122a) implies that $\rho/\|\boldsymbol{\mu}\|_2$ will decrease. Hence,

$$\frac{\mathrm{d}}{\mathrm{d}\|\boldsymbol{\mu}\|_2}\left(\frac{\rho}{\|\boldsymbol{\mu}\|_2}\right) < 0.$$

Combining the above inequalities, we know that $\mathrm{d}\beta_0/\mathrm{d}\|\boldsymbol{\mu}\|_2 > 0$.

**(b) $\uparrow$ in $\delta$:** Similarly, according to Eq. (122a) and Theorem G.2, for fixed $\pi$ and $\|\boldsymbol{\mu}\|_2$, we can show that $\rho/\delta$ will decrease if $\delta$ increases. By same approach as (a), we can conclude $\mathrm{d}\beta_0/\mathrm{d}\delta > 0$.

**(c) $\uparrow$ in $\pi \in (0, \frac{1}{2})$:** Lastly, we note that if $\pi \in (0, \frac{1}{2})$ increases, then $1 - \pi$ will decrease and $\rho$ will increase. According to Theorem G.6, we know that

$$\frac{(1-\pi)\delta}{\rho}\cdot g\left(\frac{\rho}{2(1-\pi)\|\boldsymbol{\mu}\|_2 \delta}\right)$$

will increase. Since $(1 - \rho^2)/\rho$ will decrease, combining with Eq. (122a), we can show that

$$\frac{\pi\delta}{\rho}\cdot g\left(\frac{\rho}{2\pi\|\boldsymbol{\mu}\|_2 \delta}\right)$$

will decrease. By Theorem G.6 again, we conclude that $\rho/(1-\pi)$ will increase and $\rho/\pi$ will decrease, which implies that $\beta_0$ Eq. (125) will increase. This completes the proof. $\qquad\square$

The monotonicity of minority error is a direct consequence of the two lemmas above.

***Proof of Proposition D.4.*** When $\tau = 1$, according to Theorem G.2 and G.3, both $\rho^*$ and $\beta_0^*$ are increasing in $\pi \in (0, \frac{1}{2})$, $\|\boldsymbol{\mu}\|_2$, and $\delta$. We complete the proof by $\mathrm{Err}_+^* = \Phi\left(-\rho^*\|\boldsymbol{\mu}\|_2 - \beta_0^*\right)$. $\qquad\square$

Now we fix model parameters $\pi \in (0, \frac{1}{2})$, $\delta$, $\|\boldsymbol{\mu}\|_2$, and consider test errors as functions of $\tau$. In order to prove Theorem D.5, we need the following result on the monotonicity of $\rho^*, \beta_0^*$ on $\tau$.

**Lemma G.4** (Dependence of $\tau$). *Fix $\pi \in (0, \frac{1}{2})$, $\|\boldsymbol{\mu}\|_2$, and $\delta$, then we have*

*(a) $\rho_0^*$ does not depend on $\tau$.*

*(b) $\beta_0^*$ is an increasing function of $\tau \in (0, \infty)$.*

*(c) $\kappa^*$ is a decreasing function of $\tau \in (0, \infty)$.*

*As a consequence, $\mathrm{Err}_+^*$ is decreasing in $\tau \in (1, \infty)$, and $\mathrm{Err}_-^*$ is increasing in $\tau \in (1, \infty)$.*

*Proof.* (a) is already proved in Theorem D.1(b). For (c), the conclusion is followed by Eq. (124b), since $\kappa^* \propto (1 + \tau)^{-1}$. For (b), note that $\beta_0^* + \kappa^*$ is a fixed value according to Eq. (122c). Then by using (c), we conclude $\beta_0^*$ is increasing in $\tau$. This concludes the proof. $\square$

These are consistent with the non-asymptotic monotonicity between $(\widehat{\rho}, \widehat{\beta}_0, \widehat{\kappa})$ and $\tau$ in Theorem C.1. Then the monotonicity of test errors is a direct consequence of Theorem G.4.

***Proof of Proposition D.5.*** According to Theorem G.4(a)(b), we know that $-\rho^* \|\boldsymbol{\mu}\|_2 + \beta_0^*$ is increasing in $\tau$ and $-\rho^* \|\boldsymbol{\mu}\|_2 - \beta_0^*$ is decreasing in $\tau$. This completes the proof. $\square$

### G.2   Proofs of Propositions D.6 and D.7

***Proof of Proposition D.6.*** Recall that

$$\mathrm{Err}_b^* = \frac{1}{2} \Big( \Phi\left(-\rho^* \|\boldsymbol{\mu}\|_2 - \beta_0^*\right) + \Phi\left(-\rho^* \|\boldsymbol{\mu}\|_2 + \beta_0^*\right) \Big).$$

Notice that $\rho^*$ does not depend on $\tau$, and $\rho^* \|\boldsymbol{\mu}\|_2 > 0$. We first show that $\tau = \tau^{\mathrm{opt}}$ if and only if $\beta_0^* = 0$. Then is suffices to show that for any fixed $a > 0$, function

$$f(x) := \Phi(-a + x) + \Phi(-a - x), \qquad x \in \mathbb{R}$$

has unique minimizer $x = 0$. This is true by observing $f'(x) = \phi(-a + x) - \phi(-a - x) < 0$ for all $x < 0$, and $f'(x) > 0$ for all $x > 0$. Hence we conclude $\beta_0^* = 0$ and $\mathrm{Err}_+^* = \mathrm{Err}_-^* = \mathrm{Err}_b^*$.

Setting $\beta_0 = 0$ in Eq. (124a) and solving for $\tau$, we get Eq. (38). This completes the proof. $\square$

As stated in Remark D.3, when $\|\boldsymbol{\mu}\|_2$, $\delta$ are fixed and $\pi$ is small, the numerator of $\tau^{\mathrm{opt}}$ scales as $\sqrt{1/\pi}$. We formally prove this in the following lemma.

**Lemma G.5.** *When $\pi = o(1)$, we have*

$$g_1^{-1}\left(\frac{\rho^*}{2\pi \|\boldsymbol{\mu}\|_2 \delta}\right) + \rho^* \|\boldsymbol{\mu}\|_2 \sim \frac{1}{\sqrt{\pi \delta}}.$$

*Proof.* By Theorem G.2, $\rho^*$ is monotone increasing in $\pi \in (0, \frac{1}{2})$. It can be easily shown that $\rho^* \to 0$ as $\pi \to 0$. Otherwise, suppose $\rho^* \to \underline{\rho} > 0$ as $\pi \to 0$, then by Theorem G.1(c)

$$\pi \delta \cdot g\left(\frac{\rho^*}{2\pi \|\boldsymbol{\mu}\|_2 \delta}\right) \sim \pi \delta \cdot \left(\frac{\underline{\rho}}{2\pi \|\boldsymbol{\mu}\|_2 \delta}\right)^2 \propto \frac{1}{\pi} \to \infty,$$

while the other terms in Eq. (122a) are all finite, which is a contradiction. Substitute $\rho^* \to 0$ into Eq. (122a),

$$g\left(\frac{\rho^*}{2\pi \|\boldsymbol{\mu}\|_2 \delta}\right) \sim \frac{1}{\pi \delta} \to \infty \qquad \implies \qquad \frac{\rho^*}{2\pi \|\boldsymbol{\mu}\|_2 \delta} \sim \frac{1}{\sqrt{\pi \delta}}.$$

The proof is complete by using Theorem G.1(c) again. $\square$

**Remark G.1.** *We notice that when $\pi$ is very small or $\|\boldsymbol{\mu}\|_2$, $\delta$ are very large, then $\rho^*$ is close to $0$ and the denominator of $\tau^{\mathrm{opt}}$ can be zero or negative, leading $\tau^{\mathrm{opt}}$ infinity of negative. According to Fig. 14, this happens when the optimal decision boundary (the red solid line) falls on or under the margin of majority class (the black dashed line below with negative support vectors). In such cases, we have $\tau < -1$ and the training error for majority class is nonzero.*

*Actually, our theory remains valid when $\tau < -1$. When $\tau < -1$, one can modify the objective of Eq. (7) to minimizing $\kappa$ (since $\kappa < 0$ and $\tau\kappa > 0$), then the relation Eq. (17) in Theorem C.1 still holds. For the asymptotic problem, one can similarly modify the variational problem Eq. (31). Then one may extend Theorem D.1 to negative $\tau$ by relating Eq. (17) to (81), where Eq. (81) is derived from Eq. (122a)—(122c), which also admits a unique solution when $\tau < -1$.*

Finally, prove the monotonicity of test errors after margin rebalancing.

***Proof of Proposition D.7.*** According to Theorem D.6, $\mathrm{Err}^*_+ = \mathrm{Err}^*_- = \mathrm{Err}^*_{\mathrm{b}} = \Phi(-\rho^* \|\boldsymbol{\mu}\|_2)$. Since $\rho^*$ is increasing in $\pi \in (0, \frac{1}{2})$, $\|\boldsymbol{\mu}\|_2$, and $\delta$ by Theorem G.2, the proof is complete. $\qquad\square$

### G.3 TECHNICAL LEMMAS

Some technical results used in the proof are summarized below.

**Lemma G.6.** *The function $g_2(x)/g_1(x)$ is increasing in $x$. This implies $g(x)/x$ is increasing in $x$, and $x \cdot g(1/x)$ is decreasing in $x$.*

*Proof.* By direct calculation, we have

$$g_2'(x)g_1(x) - g_2(x)g_1'(x) = 2\left(\mathbb{E}[(G + x)_+]\right)^2 - \Phi(x)\mathbb{E}[(G + x)_+^2].$$

It suffices to show that

$$h(x) := \frac{2\left(\mathbb{E}[(G + x)_+]\right)^2}{\Phi(x)} - \mathbb{E}[(G + x)_+^2] > 0, \qquad \forall\, x \in \mathbb{R}.$$

To this end, note that $\lim_{x \to -\infty} h(x) = 0$, and that

$$h'(x) = 2\mathbb{E}[(G + x)_+]\left(1 - \frac{\mathbb{E}[(G + x)_+]\phi(x)}{\Phi(x)^2}\right).$$

Hence, one only need to show that $h'(x) > 0, \forall\, x \in \mathbb{R}$, namely

$$r(x) := \frac{\Phi(x)^2}{\phi(x)} - \mathbb{E}[(G + x)_+] > 0.$$

Notice again that $\lim_{x \to -\infty} r(x) = 0$, and

$$r'(x) = \Phi(x)\left(1 + \frac{x\Phi(x)}{\phi(x)}\right) > 0$$

by Mill's ratio, thus we finally conclude that $r(x) > 0$ for any $x \in \mathbb{R}$. Consequently, $g_2(x)/g_1(x)$ is increasing in $x$.

By change of variable $y = g_1(x)$, we show that $g_2(x)/g_1(x) = g(y)/y$ is increasing in $y$. $\qquad\square$

**Lemma G.7.** *The function $x \mapsto x \cdot (g_1^{-1})'(x)$ is monotone increasing.*

*Proof.* Let $x = g_1(y)$, then we know that

$$x \cdot (g_1^{-1})'(x) = \frac{g_1(y)}{g_1'(y)}.$$

Since $y$ is increasing in $x$, it suffices to show that $g_1(y)/g_1'(y)$ is increasing in $y$. Note that

$$\frac{\mathrm{d}}{\mathrm{d}y}\left(\frac{g_1(y)}{g_1'(y)}\right) = \frac{g_1'(y)^2 - g_1(y)g_1''(y)}{g_1'(y)^2} = \frac{\phi(y)r(y)}{g_1'(y)^2},$$

where the function $r(y)$ is defined in the proof of Theorem G.6, and we know that $r(y) > 0$ for all $y \in \mathbb{R}$. Therefore, $g_1(y)/g_1'(y)$ is increasing. This completes the proof. $\qquad\square$

# H  MARGIN REBALANCING IN HIGH IMBALANCE REGIME: PROOF OF THEOREM D.8

Without loss of generality, we may consider the following case as a substitute of Eq. (8):

$$\pi = d^{-a}, \qquad \|\boldsymbol{\mu}\|_2^2 = d^b, \qquad n = d^{c+1}.$$

Consider a linear classifier based on $f(\boldsymbol{x}) = \langle \boldsymbol{x}, \boldsymbol{\beta} \rangle + \beta_0$ with $\|\boldsymbol{\beta}\|_2 = 1$. Denote projection matrices

$$\mathbf{P}_{\boldsymbol{\mu}} := \frac{1}{\|\boldsymbol{\mu}\|_2^2} \boldsymbol{\mu}\boldsymbol{\mu}^\mathsf{T}, \qquad \mathbf{P}_{\boldsymbol{\mu}}^\perp := \mathbf{I}_d - \frac{1}{\|\boldsymbol{\mu}\|_2^2} \boldsymbol{\mu}\boldsymbol{\mu}^\mathsf{T},$$

where $\mathbf{P}_{\boldsymbol{\mu}}$ is the orthogonal projection onto $\mathrm{span}\{\boldsymbol{\mu}\}$ and $\mathbf{P}_{\boldsymbol{\mu}}^\perp$ is the orthogonal projection onto the orthogonal complement of $\mathrm{span}\{\boldsymbol{\mu}\}$. Then we define auxiliary parameters

$$\rho := \left\langle \boldsymbol{\beta}, \frac{\boldsymbol{\mu}}{\|\boldsymbol{\mu}\|_2} \right\rangle, \qquad \boldsymbol{\theta} := \begin{cases} \dfrac{\mathbf{P}_{\boldsymbol{\mu}}^\perp \boldsymbol{\beta}}{\|\mathbf{P}_{\boldsymbol{\mu}}^\perp \boldsymbol{\beta}\|_2} = \dfrac{\mathbf{P}_{\boldsymbol{\mu}}^\perp \boldsymbol{\beta}}{\sqrt{1-\rho^2}}, & \text{if } |\rho| < 1, \\ \boldsymbol{\mu}_\perp, & \text{if } |\rho| = 1, \end{cases} \tag{126}$$

where $\boldsymbol{\mu}_\perp \in \S^{d-1}$ is some deterministic vector such that $\boldsymbol{\mu}_\perp \perp \boldsymbol{\mu}$. Therefore, we have the following decomposition:

$$\boldsymbol{\beta} = \mathbf{P}_{\boldsymbol{\mu}}\boldsymbol{\beta} + \mathbf{P}_{\boldsymbol{\mu}}^\perp \boldsymbol{\beta} = \rho \frac{\boldsymbol{\mu}}{\|\boldsymbol{\mu}\|_2} + \sqrt{1-\rho^2}\boldsymbol{\theta}.$$

Note that $\|\boldsymbol{\theta}\|_2 = 1$, $\boldsymbol{\theta} \perp \boldsymbol{\mu}$, and there exists a one-to-one correspondence[14] between $\boldsymbol{\beta}$ and $(\rho, \boldsymbol{\theta})$. Therefore, the logit margin of $f(\boldsymbol{x})$ for the $i$-th data point $(\boldsymbol{x}_i, y_i)$ can be reparametrized as

$$\begin{aligned}
\kappa_i = \kappa_i(\boldsymbol{\beta}, \beta_0) &:= \widetilde{y}_i(\langle \boldsymbol{x}_i, \boldsymbol{\beta} \rangle + \beta_0) \\
&= s_i y_i \left( \left\langle y_i \boldsymbol{\mu} + \boldsymbol{z}_i, \rho \frac{\boldsymbol{\mu}}{\|\boldsymbol{\mu}\|_2} + \sqrt{1-\rho^2}\boldsymbol{\theta} \right\rangle + \beta_0 \right) \\
&= s_i \left( \rho \|\boldsymbol{\mu}\|_2 + y_i \beta_0 + \rho y_i g_i + \sqrt{1-\rho^2} y_i \langle \boldsymbol{z}_i, \boldsymbol{\theta} \rangle \right) =: \kappa_i(\rho, \boldsymbol{\theta}, \beta_0),
\end{aligned} \tag{127}$$

where $\boldsymbol{z}_i \sim \mathsf{subG}_{\perp\!\!\!\perp}(\mathbf{0}, \mathbf{I}_n; K)$ according to Definition J.1, $K > 0$ is some absolute constant, and

$$s_i := \begin{cases} \tau^{-1}, & \text{if } y_i = +1, \\ 1, & \text{if } y_i = -1, \end{cases} \qquad g_i := \left\langle \boldsymbol{z}_i, \frac{\boldsymbol{\mu}}{\|\boldsymbol{\mu}\|_2} \right\rangle,$$

where $\boldsymbol{g} := (g_1, \ldots, g_n)^\mathsf{T} \sim \mathsf{subG}_{\perp\!\!\!\perp}(\mathbf{0}, \mathbf{I}_n; K)$ by Theorem J.2(b). Therefore, the margin (in Eq. (20)) of $f(\boldsymbol{x})$ can be viewed as function $(\boldsymbol{\beta}, \beta_0) \mapsto \kappa$ or $(\rho, \boldsymbol{\theta}, \beta_0) \mapsto \kappa$ based on different parametrization:

$$\begin{aligned}
\kappa = \kappa(\boldsymbol{\beta}, \beta_0) &= \min_{i \in [n]} \kappa_i(\boldsymbol{\beta}, \beta_0) \\
&= \kappa(\rho, \boldsymbol{\theta}, \beta_0) = \min_{i \in [n]} \kappa_i(\rho, \boldsymbol{\theta}, \beta_0).
\end{aligned} \tag{128}$$

As a consequence, the max-margin optimization problem Eq. (15) or (7) can be expressed as

$$\begin{aligned}
\underset{\rho, \beta_0 \in \mathbb{R}, \boldsymbol{\theta} \in \mathbb{R}^d}{\text{maximize}} \quad & \kappa(\rho, \boldsymbol{\theta}, \beta_0), \\
\text{subject to} \quad & \rho \in [-1, 1], \\
& \|\boldsymbol{\theta}\|_2 = 1, \ \boldsymbol{\theta} \perp \boldsymbol{\mu},
\end{aligned} \tag{129}$$

where

$$\kappa(\rho, \boldsymbol{\theta}, \beta_0) = \min_{i \in [n]} s_i \left( \rho \|\boldsymbol{\mu}\|_2 + y_i \beta_0 + \rho y_i g_i + \sqrt{1-\rho^2} y_i \langle \boldsymbol{z}_i, \boldsymbol{\theta} \rangle \right).$$

---

[14]In fact, this one-to-one mapping $\boldsymbol{\beta} \mapsto (\rho, \boldsymbol{\theta})$ is restricted to $\S^{d-1} \to \Theta_{\rho, \boldsymbol{\theta}}$, where the range is $\Theta_{\rho, \boldsymbol{\theta}} := \{(\rho, \boldsymbol{\theta}) : \rho \in (-1, 1), \|\boldsymbol{\theta}\|_2 = 1, \boldsymbol{\theta} \perp \boldsymbol{\mu}\} \cup \{(\rho, \boldsymbol{\theta}) : \rho = \pm 1, \boldsymbol{\theta} = \boldsymbol{\mu}_\perp\}$. However, for simplicity, we can expand the parameter space of $(\rho, \boldsymbol{\theta})$ into $\{(\rho, \boldsymbol{\theta}) : \rho \in [-1, 1], \|\boldsymbol{\theta}\|_2 = 1, \boldsymbol{\theta} \perp \boldsymbol{\mu}\}$. This is because if $\rho = \pm 1$, we have $\mathbf{P}_{\boldsymbol{\mu}}^\perp \boldsymbol{\beta} = \mathbf{0}$, and $\sqrt{1-\rho^2}\boldsymbol{\theta} = \mathbf{0}$ for any $\boldsymbol{\theta}$. We will see that $\boldsymbol{\theta}$ always appears in the form of $\sqrt{1-\rho^2}\boldsymbol{\theta}$ (for example, in the decomposition of $\boldsymbol{\beta}$, and the expression of $\kappa_i$ and $\kappa$). That also explains why we can take $\boldsymbol{\mu}_\perp$ arbitrarily in Eq. (126).

Recall that $(\widehat{\boldsymbol{\beta}}, \widehat{\beta}_0)$ is the max-margin solution to Eq. (15), and the maximum margin is given by

$$\widehat{\kappa} = \kappa(\widehat{\boldsymbol{\beta}}, \widehat{\beta}_0) = \min_{i \in [n]} \kappa_i(\widehat{\boldsymbol{\beta}}, \widehat{\beta}_0). \tag{130}$$

Similarly, we can also reparametrize $\widehat{\boldsymbol{\beta}}$ as in Eq. (126):

$$\widehat{\rho} := \left\langle \widehat{\boldsymbol{\beta}}, \frac{\boldsymbol{\mu}}{\|\boldsymbol{\mu}\|_2} \right\rangle, \qquad \widehat{\boldsymbol{\theta}} := \begin{cases} \dfrac{\mathbf{P}_{\boldsymbol{\mu}}^{\perp} \widehat{\boldsymbol{\beta}}}{\|\mathbf{P}_{\boldsymbol{\mu}}^{\perp} \widehat{\boldsymbol{\beta}}\|_2} = \dfrac{\mathbf{P}_{\boldsymbol{\mu}}^{\perp} \widehat{\boldsymbol{\beta}}}{\sqrt{1 - \widehat{\rho}^2}}, & \text{if } |\widehat{\rho}| < 1, \\ \boldsymbol{\mu}_{\perp}, & \text{if } |\widehat{\rho}| = 1. \end{cases} \tag{131}$$

Then, $(\widehat{\rho}, \widehat{\boldsymbol{\theta}}, \widehat{\beta}_0)$ is the optimal solution to Eq. (129)[15]. Combining Eq. (128) and (130), the maximum margin can be rewritten as

$$\widehat{\kappa} = \kappa(\widehat{\rho}, \widehat{\boldsymbol{\theta}}, \widehat{\beta}_0) = \min_{i \in [n]} \kappa_i(\widehat{\rho}, \widehat{\boldsymbol{\theta}}, \widehat{\beta}_0), \tag{132}$$

which is also the optimal objective value of Eq. (129). Finally, we define a few quantities:

$$\overline{g}_+ := \frac{1}{n_+} \sum_{i \in \mathcal{I}_+} g_i, \qquad \overline{g}_- := \frac{1}{n_-} \sum_{i \in \mathcal{I}_-} g_i, \qquad \widetilde{g} := \frac{\overline{g}_+ - \overline{g}_-}{2},$$

$$\overline{\boldsymbol{z}}_+ := \frac{1}{n_+} \sum_{i \in \mathcal{I}_+} \boldsymbol{z}_i, \qquad \overline{\boldsymbol{z}}_- := \frac{1}{n_-} \sum_{i \in \mathcal{I}_-} \boldsymbol{z}_i, \qquad \widetilde{\boldsymbol{z}} := \frac{\overline{\boldsymbol{z}}_+ - \overline{\boldsymbol{z}}_-}{2}.$$

The proof structure of Theorem D.8 is as follows:

1. In Section H.1, we provide a (stochastic) tight upper bound for the maximum margin $\widehat{\kappa}$, and a constructed solution $(\widetilde{\rho}, \widetilde{\boldsymbol{\theta}}, \widetilde{\beta}_0)$ which approximates $(\widehat{\rho}, \widehat{\boldsymbol{\theta}}, \widehat{\beta}_0)$ well.

2. In Section H.2, we derive the asymptotic orders of $(\widehat{\rho}, \widehat{\boldsymbol{\theta}}, \widehat{\beta}_0)$ by using $(\widetilde{\rho}, \widetilde{\boldsymbol{\theta}}, \widetilde{\beta}_0)$.

3. In Section H.3, we use these asymptotics to analyze test errors and conclude Theorem D.8.

## H.1 A TIGHT UPPER BOUND ON MAXIMUM MARGIN: PROOF OF LEMMA H.1

The following Lemma provides a data-dependent upper bound on the margin $\kappa(\boldsymbol{\beta}, \beta_0)$ which holds for all linear classifiers with $\|\boldsymbol{\beta}\|_2 = 1$. The bound is tight in sense that it can be (almost) achieved by a constructed solution. Therefore, such tightness ensures the optimal margin $\widehat{\kappa}$ should have the same asymptotics given by its upper bound, which also deduces the data is linearly separable with probability tending to one (as $d \to \infty$). Notably, Theorem C.1 implies that $\tau$ has no effect on $\widehat{\boldsymbol{\beta}}$, and $\widehat{\kappa} \propto (1 + \tau)^{-1}$ in a fixed dataset. Hence, $\tau$ simply scales the magnitude of $\widehat{\kappa}$, and it suffices to consider $\tau = 1$ in the following lemma.

**Lemma H.1.** *Fix $\tau = 1$. Denote*

$$\overline{\kappa} := \sqrt{(\|\boldsymbol{\mu}\|_2 + \widetilde{g})^2 + \|\mathbf{P}_{\boldsymbol{\mu}}^{\perp} \widetilde{\boldsymbol{z}}\|_2^2}. \tag{133}$$

*(a) (Upper bound) $\kappa(\rho, \boldsymbol{\theta}, \beta_0) \leq \overline{\kappa}$, for any $\rho \in [-1, 1]$, $\boldsymbol{\theta} \in \S^{d-1}$, $\boldsymbol{\theta} \perp \boldsymbol{\mu}$, $\beta_0 \in \mathbb{R}$. Moreover,*

$$\overline{\kappa} = \left(1 + o_{\mathbb{P}}(1)\right) \sqrt{d^b + \frac{1}{4} d^{a-c}},$$

*as $d \to \infty$.*

*(b) (Tightness) $\kappa(\widetilde{\rho}, \widetilde{\boldsymbol{\theta}}, \widetilde{\beta}_0) \geq \overline{\kappa} - \widetilde{O}_{\mathbb{P}}(1)$, where*

$$\widetilde{\rho} := \frac{\|\boldsymbol{\mu}\|_2 + \widetilde{g}}{\sqrt{(\|\boldsymbol{\mu}\|_2 + \widetilde{g})^2 + \|\mathbf{P}_{\boldsymbol{\mu}}^{\perp} \widetilde{\boldsymbol{z}}\|_2^2}}, \qquad \widetilde{\boldsymbol{\theta}} := \frac{\mathbf{P}_{\boldsymbol{\mu}}^{\perp} \widetilde{\boldsymbol{z}}}{\|\mathbf{P}_{\boldsymbol{\mu}}^{\perp} \widetilde{\boldsymbol{z}}\|_2},$$

$$\widetilde{\beta}_0 := -\widetilde{\rho} \cdot \frac{\overline{g}_+ + \overline{g}_-}{2} - \sqrt{1 - \widetilde{\rho}^2} \cdot \left\langle \frac{\overline{\boldsymbol{z}}_+ + \overline{\boldsymbol{z}}_-}{2}, \widetilde{\boldsymbol{\theta}} \right\rangle \tag{134}$$

*is a feasible solution to Eq. (129).*

---

[15] According to Eq. (131) and Theorem C.1, for linearly separable data, $(\widehat{\rho}, \widehat{\beta}_0)$ is the unique solution to Eq. (129). If $|\widehat{\rho}| < 1$, then $\widehat{\boldsymbol{\theta}}$ is also the unique solution to Eq. (129). Otherwise, if $\widehat{\rho} = \pm 1$, then $\sqrt{1 - \widehat{\rho}^2} y_i \langle \boldsymbol{z}_i, \boldsymbol{\theta} \rangle \equiv 0$ and thus any feasible $\boldsymbol{\theta}$ could solve Eq. (129).

*(c) (Asymptotics of $\widehat{\kappa}$) As a consequence, the data is linearly separable with high probability, and the maximum margin satisfies $\overline{\kappa} - \widetilde{O}_{\mathbb{P}}(1) \le \widehat{\kappa} \le \overline{\kappa}$.*

*Proof.* **(a):** We reparametrize $\kappa(\boldsymbol{\beta}, \beta_0) = \kappa(\rho, \boldsymbol{\theta}, \beta_0)$ by using Eq. (126) and (127). Then, the upper bound is established by calculating the *average logit margin* for each class. Let

$$
\begin{aligned}
\overline{\kappa}_+(\rho, \boldsymbol{\theta}, \beta_0) &:= \frac{1}{n_+} \sum_{i \in \mathcal{I}_+} \kappa_i(\rho, \boldsymbol{\theta}, \beta_0) = \rho \left\| \boldsymbol{\mu} \right\|_2 + \beta_0 + \rho \overline{g}_+ + \sqrt{1 - \rho^2} \langle \overline{\boldsymbol{z}}_+, \boldsymbol{\theta} \rangle, \\
\overline{\kappa}_-(\rho, \boldsymbol{\theta}, \beta_0) &:= \frac{1}{n_-} \sum_{i \in \mathcal{I}_-} \kappa_i(\rho, \boldsymbol{\theta}, \beta_0) = \rho \left\| \boldsymbol{\mu} \right\|_2 - \beta_0 - \rho \overline{g}_- - \sqrt{1 - \rho^2} \langle \overline{\boldsymbol{z}}_-, \boldsymbol{\theta} \rangle.
\end{aligned}
\tag{135}
$$

Clearly, $\kappa(\rho, \boldsymbol{\theta}, \beta_0) \le \overline{\kappa}_+(\rho, \boldsymbol{\theta}, \beta_0)$ and $\kappa(\rho, \boldsymbol{\theta}, \beta_0) \le \overline{\kappa}_-(\rho, \boldsymbol{\theta}, \beta_0)$. By averaging these two bounds,

$$
\begin{aligned}
\kappa(\rho, \boldsymbol{\theta}, \beta_0) &\le \frac{\overline{\kappa}_+(\rho, \boldsymbol{\theta}, \beta_0) + \overline{\kappa}_-(\rho, \boldsymbol{\theta}, \beta_0)}{2} \\
&= \rho \left\| \boldsymbol{\mu} \right\|_2 + \rho \cdot \frac{\overline{g}_+ - \overline{g}_-}{2} + \sqrt{1 - \rho^2} \cdot \left\langle \frac{\overline{\boldsymbol{z}}_+ - \overline{\boldsymbol{z}}_-}{2}, \boldsymbol{\theta} \right\rangle \\
&= \rho \left( \left\| \boldsymbol{\mu} \right\|_2 + \widetilde{g} \right) + \sqrt{1 - \rho^2} \langle \widetilde{\boldsymbol{z}}, \boldsymbol{\theta} \rangle \\
&\overset{\text{(i)}}{\le} \rho \left( \left\| \boldsymbol{\mu} \right\|_2 + \widetilde{g} \right) + \sqrt{1 - \rho^2} \| \mathbf{P}_{\boldsymbol{\mu}}^{\perp} \widetilde{\boldsymbol{z}} \|_2 \\
&\overset{\text{(ii)}}{\le} \sqrt{(\left\| \boldsymbol{\mu} \right\|_2 + \widetilde{g})^2 + \| \mathbf{P}_{\boldsymbol{\mu}}^{\perp} \widetilde{\boldsymbol{z}} \|_2^2} = \overline{\kappa},
\end{aligned}
\tag{136}
$$

which leads to $\overline{\kappa}$ defined in Eq. (133). Here, (i) is based on the fact that

$$
\underset{\boldsymbol{\theta} \in \mathbb{R}^d : \|\boldsymbol{\theta}\|_2 = 1, \langle \boldsymbol{\mu}, \boldsymbol{\theta} \rangle = 0}{\arg \max} \langle \widetilde{\boldsymbol{z}}, \boldsymbol{\theta} \rangle = \frac{\mathbf{P}_{\boldsymbol{\mu}}^{\perp} \widetilde{\boldsymbol{z}}}{\| \mathbf{P}_{\boldsymbol{\mu}}^{\perp} \widetilde{\boldsymbol{z}} \|_2}, \qquad \underset{\boldsymbol{\theta} \in \mathbb{R}^d : \|\boldsymbol{\theta}\|_2 = 1, \langle \boldsymbol{\mu}, \boldsymbol{\theta} \rangle = 0}{\max} \langle \widetilde{\boldsymbol{z}}, \boldsymbol{\theta} \rangle = \| \mathbf{P}_{\boldsymbol{\mu}}^{\perp} \widetilde{\boldsymbol{z}} \|_2,
\tag{137}
$$

and recall that the optimal $\boldsymbol{\theta}$ equals $\widetilde{\boldsymbol{\theta}}$ defined in Eq. (134). Moreover, (ii) is a consequence of Cauchy–Schwarz inequality ($A \in \mathbb{R}, B > 0$)

$$
\begin{aligned}
\underset{\rho \in [-1, 1]}{\max} \left\{ \rho A + \sqrt{1 - \rho^2} B \right\} &= \underset{\rho \in [-1, 1]}{\max} \left\langle \begin{pmatrix} \rho \\ \sqrt{1 - \rho^2} \end{pmatrix}, \begin{pmatrix} A \\ B \end{pmatrix} \right\rangle = \sqrt{A^2 + B^2}, \\
\underset{\rho \in [-1, 1]}{\arg \max} \left\{ \rho A + \sqrt{1 - \rho^2} B \right\} &= \frac{A}{\sqrt{A^2 + B^2}},
\end{aligned}
\tag{138}
$$

and also note that the optimal $\rho$ in (ii) equals $\widetilde{\rho}$ defined in Eq. (134).

To study the asymptotics of $\overline{\kappa}$, recall that $\pi = n_+/n = o(1)$, $n_- = n - n_+ = n(1 - o(1))$. Then

$$
\frac{1}{n_+} + \frac{1}{n_-} = \frac{1}{\pi n} + \frac{1}{n(1 - o(1))} = \frac{1}{\pi n} \left( 1 + o(1) \right).
$$

Denote

$$
\alpha_d := \frac{1}{2} \sqrt{\frac{1}{n_+} + \frac{1}{n_-}} = \frac{1}{2\sqrt{\pi n}} \left( 1 + o(1) \right).
\tag{139}
$$

Theorem J.2(b) implies $\widetilde{\boldsymbol{z}}/\alpha_d \sim \mathsf{subG}_{\perp\!\!\!\perp}(\mathbf{0}, \mathbf{I}_d; K)$. Then according to Theorem J.3(b),

$$
\mathbb{P}\left( \left| \frac{\| \mathbf{P}_{\boldsymbol{\mu}}^{\perp} \widetilde{\boldsymbol{z}} \|_2}{\alpha_d \| \mathbf{P}_{\boldsymbol{\mu}}^{\perp} \|_{\mathrm{F}}} - 1 \right| > t \right) \le 2 \exp\left( -\frac{ct^2}{K^4} \frac{\| \mathbf{P}_{\boldsymbol{\mu}}^{\perp} \|_{\mathrm{F}}^2}{\| \mathbf{P}_{\boldsymbol{\mu}}^{\perp} \|_{\mathrm{op}}^2} \right) = 2 \exp\left( -\frac{ct^2(d - 1)}{K^4} \right),
$$

where $\| \mathbf{P}_{\boldsymbol{\mu}}^{\perp} \|_{\mathrm{F}} = \sqrt{d - 1}$, $\| \mathbf{P}_{\boldsymbol{\mu}}^{\perp} \|_{\mathrm{op}} = 1$, and $c$ is an absolute constant. Therefore,

$$
\| \mathbf{P}_{\boldsymbol{\mu}}^{\perp} \widetilde{\boldsymbol{z}} \|_2 = \alpha_d \| \mathbf{P}_{\boldsymbol{\mu}}^{\perp} \|_{\mathrm{F}} \left( 1 + o_{\mathbb{P}}(1) \right) = \frac{1}{2\sqrt{\pi n}} \left( 1 + o(1) \right) \cdot \sqrt{d - 1} \left( 1 + o_{\mathbb{P}}(1) \right) = \frac{1}{2} \sqrt{\frac{d}{\pi n}} \left( 1 + o_{\mathbb{P}}(1) \right).
\tag{140}
$$

In addition, by Theorem J.3(c),

$$\widetilde{g} = O_{\mathbb{P}}(\alpha_d) = O_{\mathbb{P}}\left(\frac{1}{\sqrt{\pi n}}\right).$$

Recall that $a - c - 1 < 0$. Finally, we have

$$
\begin{aligned}
\overline{\kappa} &= \sqrt{(\|\boldsymbol{\mu}\|_2 + \widetilde{g})^2 + \|\mathbf{P}_{\boldsymbol{\mu}}^\perp \widetilde{z}\|_2^2} \\
&= \sqrt{\left(\|\boldsymbol{\mu}\|_2 + O_{\mathbb{P}}(1/\sqrt{\pi n})\right)^2 + \frac{d}{4\pi n}(1 + o_{\mathbb{P}}(1))} \\
&= \sqrt{\left(d^{b/2} + O_{\mathbb{P}}(d^{(a-c-1)/2})\right)^2 + \frac{1}{4}d^{a-c}(1 + o_{\mathbb{P}}(1))} \\
&= \sqrt{d^b + \frac{1}{4}d^{a-c}}(1 + o_{\mathbb{P}}(1)).
\end{aligned}
\tag{141}
$$

This concludes the proof of part (a).

**(b):** Next we show that the upper bound $\overline{\kappa}$ is nearly attainable, by a constructed solution $(\widetilde{\rho}, \widetilde{\boldsymbol{\theta}}, \widetilde{\beta}_0)$ defined in Eq. (134). Clearly, $(\widetilde{\rho}, \widetilde{\boldsymbol{\theta}}, \widetilde{\beta}_0)$ satisfies the constraints in Eq. (129). This candidate solution is motivated by the optimal $(\rho, \boldsymbol{\theta})$ that makes (i) and (ii) equal in Eq. (136), i.e.,

$$\overline{\kappa} = \widetilde{\rho}(\|\boldsymbol{\mu}\|_2 + \widetilde{g}) + \sqrt{1 - \widetilde{\rho}^2}\langle \widetilde{z}, \widetilde{\boldsymbol{\theta}}\rangle,$$

and $\beta_0$ that balances the magnitude of average logit margins from the two classes, i.e., we choose $\beta_0$ such that $\overline{\kappa}_+ = \overline{\kappa}_-$ in Eq. (135). Substituting $(\widetilde{\rho}, \widetilde{\boldsymbol{\theta}}, \widetilde{\beta}_0)$ back into Eq. (127), we obtain

$$
\begin{aligned}
\kappa_i(\widetilde{\rho}, \widetilde{\boldsymbol{\theta}}, \widetilde{\beta}_0) &= \widetilde{\rho}\|\boldsymbol{\mu}\|_2 + y_i\widetilde{\beta}_0 + \widetilde{\rho}y_i g_i + \sqrt{1 - \widetilde{\rho}^2}y_i\langle z_i, \widetilde{\boldsymbol{\theta}}\rangle \\
&= \widetilde{\rho}\left(\|\boldsymbol{\mu}\|_2 + y_i g_i - y_i\frac{\overline{g}_+ + \overline{g}_-}{2}\right) + \sqrt{1 - \widetilde{\rho}^2}\left\langle y_i z_i - y_i\frac{\overline{z}_+ + \overline{z}_-}{2}, \widetilde{\boldsymbol{\theta}}\right\rangle.
\end{aligned}
$$

Therefore, the difference between each logit margin and the upper bound can be expressed as

$$
\begin{aligned}
\overline{\kappa} - \kappa_i(\widetilde{\rho}, \widetilde{\boldsymbol{\theta}}, \widetilde{\beta}_0) &= \widetilde{\rho}\left(\widetilde{g} + y_i\frac{\overline{g}_+ + \overline{g}_-}{2} - y_i g_i\right) + \sqrt{1 - \widetilde{\rho}^2}\left\langle \widetilde{z} + y_i\frac{\overline{z}_+ + \overline{z}_-}{2} - y_i z_i, \widetilde{\boldsymbol{\theta}}\right\rangle \\
&= \begin{cases} \widetilde{\rho}(\overline{g}_+ - g_i) + \sqrt{1 - \widetilde{\rho}^2}\langle \overline{z}_+ - z_i, \widetilde{\boldsymbol{\theta}}\rangle, & \text{if } y_i = +1, \\ \widetilde{\rho}(g_i - \overline{g}_-) + \sqrt{1 - \widetilde{\rho}^2}\langle z_i - \overline{z}_-, \widetilde{\boldsymbol{\theta}}\rangle, & \text{if } y_i = -1, \end{cases}
\end{aligned}
\tag{142}
$$

where the leading terms $\rho\|\boldsymbol{\mu}\|_2$, $\langle \overline{z}_-, \widetilde{\boldsymbol{\theta}}\rangle$ (for $i = +1$), $\langle \overline{z}_+, \widetilde{\boldsymbol{\theta}}\rangle$ (for $i = -1$) are all cancelled out. Our goal is to bound the maximum difference over all data points. Note that

$$
\max_{i \in [n]}\left|\overline{\kappa} - \kappa_i(\widetilde{\rho}, \widetilde{\boldsymbol{\theta}}, \widetilde{\beta}_0)\right| = \max_{i \in \mathcal{I}_+}\left|\overline{\kappa} - \kappa_i(\widetilde{\rho}, \widetilde{\boldsymbol{\theta}}, \widetilde{\beta}_0)\right| \vee \max_{i \in \mathcal{I}_-}\left|\overline{\kappa} - \kappa_i(\widetilde{\rho}, \widetilde{\boldsymbol{\theta}}, \widetilde{\beta}_0)\right|
$$

$$
\leq \max_{i \in \mathcal{I}_+}\left\{\left|g_i - \overline{g}_+\right| + \left|\langle z_i - \overline{z}_+, \widetilde{\boldsymbol{\theta}}\rangle\right|\right\} \vee \max_{i \in \mathcal{I}_-}\left\{\left|g_i - \overline{g}_-\right| + \left|\langle z_i - \overline{z}_-, \widetilde{\boldsymbol{\theta}}\rangle\right|\right\}
\tag{143}
$$

$$
\leq \left\{\max_{i \in \mathcal{I}_+}\left|g_i - \overline{g}_+\right| + \max_{i \in \mathcal{I}_+}\left|\langle z_i - \overline{z}_+, \widetilde{\boldsymbol{\theta}}\rangle\right|\right\} \vee \left\{\max_{i \in \mathcal{I}_-}\left|g_i - \overline{g}_-\right| + \max_{i \in \mathcal{I}_-}\left|\langle z_i - \overline{z}_-, \widetilde{\boldsymbol{\theta}}\rangle\right|\right\}.
$$

For the first term involving $g_i$'s, recall that $\max_{i \in [n]}\|g_i\|_{\psi_2} \lesssim K$. Therefore, as per Theorem J.2(c) and Theorem J.3(c), $g_i, \overline{g}_\pm$ are sub-gaussian, and

$$
\begin{aligned}
\max_{i \in \mathcal{I}_+}\left|g_i - \overline{g}_+\right| &\leq \max_{i \in \mathcal{I}_+}|g_i| + \left|\overline{g}_+\right| = O_{\mathbb{P}}(\sqrt{\log n_+}) + O_{\mathbb{P}}\left(\frac{1}{\sqrt{n_+}}\right) = O_{\mathbb{P}}(\sqrt{\log d}), \\
\max_{i \in \mathcal{I}_-}\left|g_i - \overline{g}_-\right| &\leq \max_{i \in \mathcal{I}_-}|g_i| + \left|\overline{g}_-\right| = O_{\mathbb{P}}(\sqrt{\log n_-}) + O_{\mathbb{P}}\left(\frac{1}{\sqrt{n_-}}\right) = O_{\mathbb{P}}(\sqrt{\log d}).
\end{aligned}
\tag{144}
$$

For the second term involving $z_i$'s, note that

$$
\begin{aligned}
\max_{i \in \mathcal{I}_+}\left|\langle z_i - \overline{z}_+, \widetilde{\boldsymbol{\theta}}\rangle\right| &\leq \max_{i \in \mathcal{I}_+}\frac{1}{n_+}\sum_{j \in \mathcal{I}_+}\left|\langle z_i - z_j, \widetilde{\boldsymbol{\theta}}\rangle\right| \leq \frac{1}{\|\mathbf{P}_{\boldsymbol{\mu}}^\perp \widetilde{z}\|_2}\max_{i,j \in \mathcal{I}_+}\left|\langle z_i - z_j, \mathbf{P}_{\boldsymbol{\mu}}^\perp \widetilde{z}\rangle\right|, \\
\max_{i \in \mathcal{I}_-}\left|\langle z_i - \overline{z}_-, \widetilde{\boldsymbol{\theta}}\rangle\right| &\leq \max_{i \in \mathcal{I}_-}\frac{1}{n_-}\sum_{j \in \mathcal{I}_-}\left|\langle z_i - z_j, \widetilde{\boldsymbol{\theta}}\rangle\right| \leq \frac{1}{\|\mathbf{P}_{\boldsymbol{\mu}}^\perp \widetilde{z}\|_2}\max_{i,j \in \mathcal{I}_-}\left|\langle z_i - z_j, \mathbf{P}_{\boldsymbol{\mu}}^\perp \widetilde{z}\rangle\right|.
\end{aligned}
\tag{145}
$$

So it remains to bound $\langle \boldsymbol{z}_i - \boldsymbol{z}_j, \mathbf{P}_{\boldsymbol{\mu}}^{\perp} \widetilde{\boldsymbol{z}} \rangle$ uniformly. We decompose it as

$$\langle \boldsymbol{z}_i - \boldsymbol{z}_j, \mathbf{P}_{\boldsymbol{\mu}}^{\perp} \widetilde{\boldsymbol{z}} \rangle = \langle \boldsymbol{z}_i - \boldsymbol{z}_j, \widetilde{\boldsymbol{z}} \rangle - \left\langle \boldsymbol{z}_i - \boldsymbol{z}_j, \frac{\boldsymbol{\mu}}{\|\boldsymbol{\mu}\|_2} \right\rangle \left\langle \widetilde{\boldsymbol{z}}, \frac{\boldsymbol{\mu}}{\|\boldsymbol{\mu}\|_2} \right\rangle := I - II.$$

We will show that both $I$ and $II$ are sub-exponential. To bound $\|I\|_{\psi_1}$ via Theorem J.4(b), we claim the inner product

$$I = \langle \boldsymbol{z}_i - \boldsymbol{z}_j, \widetilde{\boldsymbol{z}} \rangle = \sum_{k=1}^{d} (\boldsymbol{z}_i - \boldsymbol{z}_j)_k (\widetilde{\boldsymbol{z}})_k$$

is the sum of $d$ mean-zero random variables, i.e., $\mathbb{E}[(\boldsymbol{z}_i - \boldsymbol{z}_j)_k (\widetilde{\boldsymbol{z}})_k] = 0, \forall k \in [d]$, where we write $(\mathbf{a})_k$ as the $k$-th entry of vector $\mathbf{a}$. To see this, we decompose $\widetilde{\boldsymbol{z}}$ into terms that are independent or dependent of $(\boldsymbol{z}_i, \boldsymbol{z}_j)$.

- If $y_i = y_j = +1$ and $i \neq j$, then

$$\mathbb{E}[(\boldsymbol{z}_i - \boldsymbol{z}_j) \odot \widetilde{\boldsymbol{z}}] = \mathbb{E}\left[ (\boldsymbol{z}_i - \boldsymbol{z}_j) \odot \left( \underbrace{\frac{1}{2n_+} \sum_{\substack{k: k \neq i,j \\ y_k = +1}} \boldsymbol{z}_k + \frac{\overline{\boldsymbol{z}}_-}{2}}_{=: \widetilde{\boldsymbol{z}}_{-ij}^+} \right) \right] + \frac{1}{2n_+} \mathbb{E}[(\boldsymbol{z}_i - \boldsymbol{z}_j) \odot (\boldsymbol{z}_i + \boldsymbol{z}_j)]$$

$$= \mathbb{E}[\boldsymbol{z}_i - \boldsymbol{z}_j] \odot \mathbb{E}[\widetilde{\boldsymbol{z}}_{-ij}^+] + \frac{1}{2n_+} \left( \mathbb{E}[\boldsymbol{z}_i \odot \boldsymbol{z}_i] - \mathbb{E}[\boldsymbol{z}_j \odot \boldsymbol{z}_j] \right) \qquad (\widetilde{\boldsymbol{z}}_{-ij}^+ \perp\!\!\!\perp \boldsymbol{z}_i, \boldsymbol{z}_j)$$

$$= \mathbf{0} \odot \mathbf{0} + \frac{1}{2n_+} (\mathbf{1} - \mathbf{1}) = \mathbf{0}.$$

- If $y_i = y_j = -1$ and $i \neq j$, similarly

$$\mathbb{E}[(\boldsymbol{z}_i - \boldsymbol{z}_j) \odot \widetilde{\boldsymbol{z}}] = \mathbb{E}\left[ (\boldsymbol{z}_i - \boldsymbol{z}_j) \odot \left( \underbrace{\frac{1}{2n_-} \sum_{\substack{k: k \neq i,j \\ y_k = -1}} \boldsymbol{z}_k + \frac{\overline{\boldsymbol{z}}_+}{2}}_{=: \widetilde{\boldsymbol{z}}_{-ij}^-} \right) \right] + \frac{1}{2n_-} \mathbb{E}[(\boldsymbol{z}_i - \boldsymbol{z}_j) \odot (\boldsymbol{z}_i + \boldsymbol{z}_j)]$$

$$= \mathbb{E}[\boldsymbol{z}_i - \boldsymbol{z}_j] \odot \mathbb{E}[\widetilde{\boldsymbol{z}}_{-ij}^-] + \frac{1}{2n_-} \left( \mathbb{E}[\boldsymbol{z}_i \odot \boldsymbol{z}_i] - \mathbb{E}[\boldsymbol{z}_j \odot \boldsymbol{z}_j] \right) \qquad (\widetilde{\boldsymbol{z}}_{-ij}^- \perp\!\!\!\perp \boldsymbol{z}_i, \boldsymbol{z}_j)$$

$$= \mathbf{0}.$$

Therefore, when $d$ is large enough, we have

$$\begin{aligned}
\|I\|_{\psi_1} = \|\langle \boldsymbol{z}_i - \boldsymbol{z}_j, \widetilde{\boldsymbol{z}} \rangle\|_{\psi_1} &= \left\| \sum_{k=1}^{d} (\boldsymbol{z}_i - \boldsymbol{z}_j)_k (\widetilde{\boldsymbol{z}})_k \right\|_{\psi_1} \\
&\overset{(i)}{\lesssim} \sqrt{d} \max_{1 \leq k \leq d} \|(\boldsymbol{z}_i - \boldsymbol{z}_j)_k (\widetilde{\boldsymbol{z}})_k\|_{\psi_1} \\
&\overset{(ii)}{\leq} \sqrt{d} \max_{1 \leq k \leq d} \|(\boldsymbol{z}_i - \boldsymbol{z}_j)_k\|_{\psi_2} \max_{1 \leq k \leq d} \|(\widetilde{\boldsymbol{z}})_k\|_{\psi_2} \\
&\overset{(iii)}{\lesssim} \sqrt{d} K \cdot \alpha_d K \lesssim \sqrt{\frac{d}{\pi n}} K^2,
\end{aligned}$$

where (i) results from coordinate independence and Theorem J.4(b), (ii) is from Theorem J.4(d), and (iii) is based on $\widetilde{\boldsymbol{z}}/\alpha_d, (\boldsymbol{z}_i - \boldsymbol{z}_j)/\sqrt{2} \sim \mathsf{subG}_{\perp\!\!\!\perp}(\mathbf{0}, \mathbf{I}_d; K)$. For the term $II$, we have

$$\begin{aligned}
\|II\|_{\psi_2} &= \left\| \left\langle \boldsymbol{z}_i - \boldsymbol{z}_j, \frac{\boldsymbol{\mu}}{\|\boldsymbol{\mu}\|_2} \right\rangle \left\langle \widetilde{\boldsymbol{z}}, \frac{\boldsymbol{\mu}}{\|\boldsymbol{\mu}\|_2} \right\rangle \right\|_{\psi_1} \\
&\overset{(i)}{\leq} \left\| \left\langle \boldsymbol{z}_i - \boldsymbol{z}_j, \frac{\boldsymbol{\mu}}{\|\boldsymbol{\mu}\|_2} \right\rangle \right\|_{\psi_2} \left\| \left\langle \widetilde{\boldsymbol{z}}, \frac{\boldsymbol{\mu}}{\|\boldsymbol{\mu}\|_2} \right\rangle \right\|_{\psi_2} \\
&\overset{(ii)}{\lesssim} \max_{1 \leq k \leq d} \|(\boldsymbol{z}_i - \boldsymbol{z}_j)_k\|_{\psi_2} \max_{1 \leq k \leq d} \|(\widetilde{\boldsymbol{z}})_k\|_{\psi_2} \\
&\lesssim K \cdot \alpha_d K \lesssim \frac{1}{\sqrt{\pi n}} K^2,
\end{aligned}$$

where (i) is from Theorem J.4(d), and (ii) is from Theorem J.2(b). Hence,

$$\left\| \frac{\langle \boldsymbol{z}_i - \boldsymbol{z}_j, \mathbf{P}_{\boldsymbol{\mu}}^{\perp} \widetilde{\boldsymbol{z}} \rangle}{\sqrt{d/\pi n}} \right\|_{\psi_1} \leq \sqrt{\frac{\pi n}{d}} \big( \|I\|_{\psi_1} + \|II\|_{\psi_1} \big) \lesssim K^2.$$

Substituting this back into Eq. (145), referring to Eq. (140) and Theorem J.4(c), we obtain

$$
\begin{aligned}
\max_{i \in \mathcal{I}_+} \big| \langle \boldsymbol{z}_i - \overline{\boldsymbol{z}}_+, \widetilde{\boldsymbol{\theta}} \rangle \big| &\leq \frac{1}{\|\mathbf{P}_{\boldsymbol{\mu}}^{\perp} \widetilde{\boldsymbol{z}}\|_2} \max_{\substack{i \in \mathcal{I}_+ \\ j \in \mathcal{I}_+}} \big| \langle \boldsymbol{z}_i - \boldsymbol{z}_j, \mathbf{P}_{\boldsymbol{\mu}}^{\perp} \widetilde{\boldsymbol{z}} \rangle \big| = \big(1 + o_{\mathbb{P}}(1)\big) \max_{\substack{i \in \mathcal{I}_+ \\ j \in \mathcal{I}_+}} \left| \frac{\langle \boldsymbol{z}_i - \boldsymbol{z}_j, \mathbf{P}_{\boldsymbol{\mu}}^{\perp} \widetilde{\boldsymbol{z}} \rangle}{\sqrt{d/\pi n}} \right| \\
&= O_{\mathbb{P}}(\log n_+^2) = O_{\mathbb{P}}(\log d), \\
\max_{i \in \mathcal{I}_-} \big| \langle \boldsymbol{z}_i - \overline{\boldsymbol{z}}_-, \widetilde{\boldsymbol{\theta}} \rangle \big| &\leq \frac{1}{\|\mathbf{P}_{\boldsymbol{\mu}}^{\perp} \widetilde{\boldsymbol{z}}\|_2} \max_{\substack{i \in \mathcal{I}_- \\ j \in \mathcal{I}_-}} \big| \langle \boldsymbol{z}_i - \boldsymbol{z}_j, \mathbf{P}_{\boldsymbol{\mu}}^{\perp} \widetilde{\boldsymbol{z}} \rangle \big| = \big(1 + o_{\mathbb{P}}(1)\big) \max_{\substack{i \in \mathcal{I}_- \\ j \in \mathcal{I}_-}} \left| \frac{\langle \boldsymbol{z}_i - \boldsymbol{z}_j, \mathbf{P}_{\boldsymbol{\mu}}^{\perp} \widetilde{\boldsymbol{z}} \rangle}{\sqrt{d/\pi n}} \right| \\
&= O_{\mathbb{P}}(\log n_-^2) = O_{\mathbb{P}}(\log d).
\end{aligned}
\tag{146}
$$

Finally, incorporating Eq. (144) and Eq. (146) into Eq. (143), we have

$$
\begin{aligned}
&\max_{i \in [n]} \big| \overline{\kappa} - \kappa_i(\widetilde{\rho}, \widetilde{\boldsymbol{\theta}}, \widetilde{\beta}_0) \big| \\
&\leq \left\{ \max_{i \in \mathcal{I}_+} |g_i - \overline{g}_+| + \max_{i \in \mathcal{I}_+} \big| \langle \boldsymbol{z}_i - \overline{\boldsymbol{z}}_+, \widetilde{\boldsymbol{\theta}} \rangle \big| \right\} \vee \left\{ \max_{i \in \mathcal{I}_-} |g_i - \overline{g}_-| + \max_{i \in \mathcal{I}_-} \big| \langle \boldsymbol{z}_i - \overline{\boldsymbol{z}}_-, \widetilde{\boldsymbol{\theta}} \rangle \big| \right\} \\
&\leq \left\{ O_{\mathbb{P}}(\sqrt{\log d}) + O_{\mathbb{P}}(\log d) \right\} \vee \left\{ O_{\mathbb{P}}(\sqrt{\log d}) + O_{\mathbb{P}}(\log d) \right\} = O_{\mathbb{P}}(\log d).
\end{aligned}
$$

Therefore, the difference between the margin of classifier characterized by $(\widetilde{\rho}, \widetilde{\boldsymbol{\theta}}, \widetilde{\beta}_0)$ and its upper bound $\overline{\kappa}$ is bounded by

$$\overline{\kappa} - \kappa(\widetilde{\rho}, \widetilde{\boldsymbol{\theta}}, \widetilde{\beta}_0) = \overline{\kappa} - \min_{i \in [n]} \kappa_i(\widetilde{\rho}, \widetilde{\boldsymbol{\theta}}, \widetilde{\beta}_0) = \max_{i \in [n]} \big| \overline{\kappa} - \kappa_i(\widetilde{\rho}, \widetilde{\boldsymbol{\theta}}, \widetilde{\beta}_0) \big| = O_{\mathbb{P}}(\log d) = \widetilde{O}_{\mathbb{P}}(1).$$

This concludes the proof of part (b).

**(c):** According to max-margin optimization problem Eq. (129), note that

$$\widehat{\kappa} = \max_{\substack{\rho \in [-1,1], \beta_0 \in \mathbb{R} \\ \boldsymbol{\theta} \in \S^{d-1}, \boldsymbol{\theta} \perp \boldsymbol{\mu}}} \kappa(\rho, \boldsymbol{\theta}, \beta_0) \geq \kappa(\widetilde{\rho}, \widetilde{\boldsymbol{\theta}}, \widetilde{\beta}_0),$$

hence the asymptotics of $\widehat{\kappa}$ is followed by (a) and (b). As $d \to \infty$, note that

$$\widehat{\kappa} \geq \overline{\kappa} - \widetilde{O}_{\mathbb{P}}(1) = \big(1 + o_{\mathbb{P}}(1)\big) \sqrt{d^b + \frac{1}{4} d^{a-c}}, \qquad \sqrt{d^b + \frac{1}{4} d^{a-c}} \geq d^{b/2} \to +\infty,$$

which implies $\widehat{\kappa}$ diverges with high probability, i.e., $\lim_{d \to \infty} \mathbb{P}(\widehat{\kappa} > C) = 1, \forall C \in \mathbb{R}$. As the result, $\mathbb{P}\{\text{linearly separable}\} = \mathbb{P}(\widehat{\kappa} > 0) \to 1$ as $d \to \infty$, deducing $\|\widehat{\boldsymbol{\beta}}\|_2 = 1$ with high probability. This concludes the proof of part (c). $\qquad \square$

## H.2 ASYMPTOTICS OF OPTIMAL PARAMETERS: PROOFS OF LEMMAS H.3, H.4, H.5

Followed by tightness of the upper bound $\overline{\kappa}$, we show that the optimal parameters $\widehat{\rho}, \widehat{\boldsymbol{\theta}}$ should be very "close" to the constructed solution $\widetilde{\rho}, \widetilde{\boldsymbol{\theta}}$ defined in Eq. (134) in some sense. On the event that the data is linearly separable, we have showed that $\widehat{\boldsymbol{\beta}}$, and therefore both $\widehat{\rho}$ and $\widehat{\boldsymbol{\theta}}$, do not depend on $\tau$ in Theorem C.1. Hence, it still suffices to consider $\tau = 1$ in our proof.

### H.2.1 ASYMPTOTIC ORDER OF $\widehat{\rho}$: PROOF OF LEMMA H.3

The following technical Lemma is important for deriving the asymptotics of $\widehat{\rho}$, which introduces a function of $\rho$ used implicitly in Eq. (136) and (138) for optimization.

**Lemma H.2.** *Define $F_{A,B}(\rho) = \rho A + \sqrt{1 - \rho^2}B$, $\rho \in [-1, 1]$, with $A \in \mathbb{R}$, $B > 0$. Then*

$$F'_{A,B}(\rho) = A - \frac{\rho}{\sqrt{1 - \rho^2}}B, \qquad F''_{A,B}(\rho) = -\frac{1}{(1 - \rho^2)^{3/2}}B,$$

*which implies $F_{A,B}$ is $B$-strongly concave, that is, for all $\rho_1, \rho_2 \in [-1, 1]$,*

$$F_{A,B}(\rho_2) \leq F_{A,B}(\rho_1) + F'_{A,B}(\rho_1)(\rho_2 - \rho_1) - \frac{1}{2}B(\rho_2 - \rho_1)^2.$$

*Moreover,*

$$\arg\max_{\rho \in [-1,1]} F_{A,B}(\rho) = \frac{A}{\sqrt{A^2 + B^2}}, \qquad \max_{\rho \in [-1,1]} F_{A,B}(\rho) = \sqrt{A^2 + B^2}.$$

*Proof.* Strongly concavity is given by direct calculation and the fact that

$$\sup_{\rho \in [-1,1]} F''_{A,B}(\rho) = -B.$$

The optimality condition is already derived in Eq. (138). This concludes the proof. $\qquad\square$

In the rest of this section, the (stochastic) parameters $A$, $B$ are defined as

$$A := \|\boldsymbol{\mu}\|_2 + \widetilde{g} = d^{b/2}\big(1 + o_{\mathbb{P}}(1)\big), \qquad B := \|\mathbf{P}_{\boldsymbol{\mu}}^{\perp}\widetilde{\boldsymbol{z}}\|_2 = \frac{1}{2}d^{(a-c)/2}\big(1 + o_{\mathbb{P}}(1)\big). \tag{147}$$

Then followed by Theorem H.2, we have $\widetilde{\rho} = \arg\max_{\rho \in [-1,1]} F_{A,B}(\rho)$ and $F'_{A,B}(\widetilde{\rho}) = 0$, where $\widetilde{\rho}$ is defined in Eq. (134). The following Lemma describes the asymptotics of $\widehat{\rho}$ with respect to $\widetilde{\rho}$.

**Lemma H.3** (Asymptotics of $\widehat{\rho}$ and $\widetilde{\rho}$)**.** *Suppose that $a < c + 1$.*

(a) *If $a < b + c$, then $\widetilde{\rho} = 1 - o_{\mathbb{P}}(1)$, $\widehat{\rho} = 1 - o_{\mathbb{P}}(1)$, and*

$$\sqrt{1 - \widetilde{\rho}^2} = \frac{1}{2}d^{(a-b-c)/2}\big(1 + o_{\mathbb{P}}(1)\big).$$

*Moreover, we further assume:*

  i. *If $a > \frac{b}{2} + c$, then $\sqrt{1 - \widehat{\rho}^2} = \sqrt{1 - \widetilde{\rho}^2}\big(1 + o_{\mathbb{P}}(1)\big)$.*
  ii. *If $a \leq \frac{b}{2} + c$, then $\sqrt{1 - \widehat{\rho}^2} = \widetilde{O}_{\mathbb{P}}(d^{-b/4})$ and thus $\sqrt{1 - \widehat{\rho}^2}\sqrt{d/\pi n} = \widetilde{O}_{\mathbb{P}}(1)$.*

(b) *If $a > b + c$, then $\widetilde{\rho} = o_{\mathbb{P}}(1)$, $\widehat{\rho} = o_{\mathbb{P}}(1)$, and*

$$\widetilde{\rho} = 2d^{(b-a+c)/2}\big(1 + o_{\mathbb{P}}(1)\big).$$

*Moreover, we further assume:*

  i. *If $a < 2b + c$, then $\widehat{\rho} = \widetilde{\rho}\big(1 + o_{\mathbb{P}}(1)\big)$.*
  ii. *If $a > 2b + c$, then $\widehat{\rho} = \widetilde{O}_{\mathbb{P}}(d^{-(a-c)/4})$ and thus $\widehat{\rho}\|\boldsymbol{\mu}\|_2 = o_{\mathbb{P}}(1)$.*

*Proof.* According to Eq. (134) and (147), an explicit expression of $\widetilde{\rho}$ is given by

$$\widetilde{\rho} = \frac{A}{\sqrt{A^2 + B^2}} = \frac{\|\boldsymbol{\mu}\|_2 + \widetilde{g}}{\sqrt{(\|\boldsymbol{\mu}\|_2 + \widetilde{g})^2 + \|\mathbf{P}_{\boldsymbol{\mu}}^{\perp}\widetilde{\boldsymbol{z}}\|_2^2}} = \frac{d^{b/2}}{\sqrt{d^b + \frac{1}{4}d^{a-c}}}\big(1 + o_{\mathbb{P}}(1)\big). \tag{148}$$

In order to connect $\widehat{\rho}$ with $\widetilde{\rho}$, recall Eq. (136) that

$$\kappa(\rho, \boldsymbol{\theta}, \beta_0) \leq F_{A,B}(\rho) \leq F_{A,B}(\widetilde{\rho}) = \overline{\kappa}, \qquad \forall \rho \in [-1, 1], \ \boldsymbol{\theta} \in \S^{d-1}, \boldsymbol{\theta} \perp \boldsymbol{\mu}, \ \beta_0 \in \mathbb{R}.$$

Apply this to $\widehat{\kappa} = \kappa(\widehat{\rho}, \widehat{\boldsymbol{\theta}}, \widehat{\beta}_0)$ and use Theorem H.1, we have

$$0 \leq F_{A,B}(\widetilde{\rho}) - F_{A,B}(\widehat{\rho}) \leq \widetilde{O}_{\mathbb{P}}(1). \tag{149}$$

Since $\widetilde{O}_{\mathbb{P}}(1)/F_{A,B}(\widetilde{\rho}) = \widetilde{O}_{\mathbb{P}}(1)/\sqrt{A^2 + B^2} \leq \widetilde{O}_{\mathbb{P}}(d^{-b/2}) = o_{\mathbb{P}}(1)$, it implies

$$1 - o_{\mathbb{P}}(1) = \frac{F_{A,B}(\widehat{\rho})}{F_{A,B}(\widetilde{\rho})} = \frac{\widehat{\rho}A}{\sqrt{A^2 + B^2}} + \frac{\sqrt{1 - \widehat{\rho}^2}B}{\sqrt{A^2 + B^2}} = \widehat{\rho}\widetilde{\rho} + \sqrt{1 - \widehat{\rho}^2}\sqrt{1 - \widetilde{\rho}^2}.$$

Therefore,

$$\begin{aligned} \widetilde{\rho} = 1 - o_{\mathbb{P}}(1) &\implies \widehat{\rho} = 1 - o_{\mathbb{P}}(1) \\ \widetilde{\rho} = o_{\mathbb{P}}(1) &\implies \widehat{\rho} = o_{\mathbb{P}}(1) \end{aligned} \tag{150}$$

**(a):** If $a - c < b$, then $d^{b/2} \gg d^{(a-c)/2}$ and by Eq. (148) and (150) we have $\widetilde{\rho}, \widehat{\rho} = 1 - o_{\mathbb{P}}(1)$. Also,

$$\sqrt{1 - \widetilde{\rho}^2} = \frac{B}{\sqrt{A^2 + B^2}} = \frac{\frac{1}{2}d^{(a-b-c)/2}}{\sqrt{1 + \frac{1}{4}d^{a-b-c}}}\big(1 + o_{\mathbb{P}}(1)\big) = \frac{1}{2}d^{(a-b-c)/2}\big(1 + o_{\mathbb{P}}(1)\big).$$

To derive the precise order of $\sqrt{1 - \widehat{\rho}^2}$, we define $r := \sqrt{1 - \rho^2}$ and $F_{B,A}(r) := rB + \sqrt{1 - r^2}A$. Then $F_{A,B}(\rho) = F_{B,A}(r)$ for any $\rho \in [0, 1]$. We similarly define $\widehat{r} := \sqrt{1 - \widehat{\rho}^2}$ and $\widetilde{r} := \sqrt{1 - \widetilde{\rho}^2}$. On the event $\mathcal{E} = \{A > 0, \widetilde{\rho} > 0, \widehat{\rho} > 0\}$, by Theorem H.2, we have

$$F_{A,B}(\widehat{\rho}) - F_{A,B}(\widetilde{\rho}) = F_{B,A}(\widehat{r}) - F_{B,A}(\widetilde{r}) \leq -\frac{1}{2}A(\widehat{r} - \widetilde{r})^2.$$

Combined with Eq. (149), it implies

$$(\widehat{r} - \widetilde{r})^2 \leq \frac{2}{A}\big(F_{A,B}(\widetilde{\rho}) - F_{A,B}(\widehat{\rho})\big) \leq \widetilde{O}_{\mathbb{P}}(d^{-b/2}),$$

so $|\widehat{r} - \widetilde{r}| = \widetilde{O}_{\mathbb{P}}(d^{-b/4})$. Now consider different scenarios. Recall that $\widetilde{r} = \frac{1}{2}d^{(a-b-c)/2}\big(1 + o_{\mathbb{P}}(1)\big)$.

- If $a - c > b/2$, then $|\widehat{r} - \widetilde{r}|/\widetilde{r} = \widetilde{O}_{\mathbb{P}}(d^{(-2a+b+2c)/4}) = o_{\mathbb{P}}(1)$, deduces $\widehat{r} = \widetilde{r}\big(1 + o_{\mathbb{P}}(1)\big)$.

- If $a - c \leq b/2$, then we only get $\widehat{r} = \widetilde{O}_{\mathbb{P}}(d^{-b/4})$, and $\widehat{r}\sqrt{d/\pi n} = \widetilde{O}_{\mathbb{P}}(d^{(2a-b-2c)/4}) \leq \widetilde{O}_{\mathbb{P}}(1)$.

Recall that these hold on event $\mathcal{E}$. Since $\mathbb{P}(\mathcal{E}) \to 1$ as $d \to \infty$, these asymptotic results involving $o_{\mathbb{P}}(\cdot)$ and $\widetilde{O}_{\mathbb{P}}(\cdot)$ also hold on the whole sample space $\Omega$. This concludes the proof of part (a).

**(b):** If $a - c > b$, then $d^{b/2} \ll d^{(a-c)/2}$ and by Eq. (148) and (150) we have $\widetilde{\rho}, \widehat{\rho} = o_{\mathbb{P}}(1)$. Also,

$$\widetilde{\rho} = \frac{A}{\sqrt{A^2 + B^2}} = \frac{d^{(b-a+c)/2}}{\sqrt{d^{b-a+c} + \frac{1}{4}}}\big(1 + o_{\mathbb{P}}(1)\big) = 2d^{(b-a+c)/2}\big(1 + o_{\mathbb{P}}(1)\big).$$

Again, by Theorem H.2,

$$F_{A,B}(\widehat{\rho}) - F_{A,B}(\widetilde{\rho}) \leq -\frac{1}{2}B(\widehat{\rho} - \widetilde{\rho})^2.$$

Combined with Eq. (149), it implies

$$(\widehat{\rho} - \widetilde{\rho})^2 \leq \frac{2}{B}\big(F_{A,B}(\widetilde{\rho}) - F_{A,B}(\widehat{\rho})\big) \leq \widetilde{O}_{\mathbb{P}}(d^{-(a-c)/2}),$$

so $|\widehat{\rho} - \widetilde{\rho}| = \widetilde{O}_{\mathbb{P}}(d^{-(a-c)/4})$. Now consider different scenarios.

- If $a - c < 2b$, then $|\widehat{\rho} - \widetilde{\rho}|/\widetilde{\rho} = \widetilde{O}_{\mathbb{P}}(d^{(a-2b-c)/4}) = o_{\mathbb{P}}(1)$, deduces $\widehat{\rho} = \widetilde{\rho}\big(1 + o_{\mathbb{P}}(1)\big)$.

- If $a - c > 2b$, then we only get $\widehat{\rho} = \widetilde{O}_{\mathbb{P}}(d^{-(a-c)/4})$, and $\widehat{\rho}\|\boldsymbol{\mu}\|_2 = \widetilde{O}_{\mathbb{P}}(d^{(2b-a+c)/4}) = o_{\mathbb{P}}(1)$.

This concludes the proof of part (b). $\qquad\square$

**Remark H.1.** *In each part i. of Theorem H.3(a) and (b), we can derive the precise asymptotic of $\widehat{\rho}$, which is same as $\widetilde{\rho}$. It is difficult to do so in part ii. of (a) and (b). However, as we will show in Theorem H.4 and H.5, in case ii. the corresponding term ($\sqrt{1 - \widehat{\rho}}$ or $\widehat{\rho}$) is negligible, which won't affect the asymptotics of test errors.*

### H.2.2 Asymptotic order of $\langle z_i, \widehat{\theta} \rangle$'s on the margin: Proof of Lemma H.4

Next, we discuss the asymptotics of $\widehat{\theta}$ and $\widetilde{\theta}$. In fact, it suffices to consider the magnitude of their projection on some "important" $z_i$, which is related to the *support vectors*, defined in Eq. (22). As we mentioned, $\mathcal{SV}_+(\beta), \mathcal{SV}_-(\beta)$ only depend on $\beta$ and $(X, y)$, not $\beta_0$ or $\tau$. If we fix $\rho = \widehat{\rho}$, then the dependency of $\mathcal{SV}_\pm$ on $\beta$ only comes from $\theta$. So, recalling Eq. (127),

$$\kappa_i(\rho, \theta, \beta_0) = s_i \left( \rho \|\mu\|_2 + y_i \beta_0 + \rho y_i g_i + \sqrt{1 - \rho^2} y_i \langle z_i, \theta \rangle \right),$$

we can rewrite Eq. (22) in terms of $\theta$:

$$
\begin{aligned}
\mathcal{SV}_+ = \mathcal{SV}_+(\theta) &:= \underset{i \in \mathcal{I}_+}{\arg\min}\, \kappa_i(\widehat{\rho}, \theta, \beta_0) = \underset{i \in \mathcal{I}_+}{\arg\min} \left\{ \widehat{\rho} g_i + \sqrt{1 - \widehat{\rho}^2} \langle z_i, \theta \rangle \right\}, \\
\mathcal{SV}_- = \mathcal{SV}_-(\theta) &:= \underset{i \in \mathcal{I}_-}{\arg\min}\, \kappa_i(\widehat{\rho}, \theta, \beta_0) = \underset{i \in \mathcal{I}_-}{\arg\min} \left\{ -\widehat{\rho} g_i - \sqrt{1 - \widehat{\rho}^2} \langle z_i, \theta \rangle \right\}.
\end{aligned}
\tag{151}
$$

As before, let $\mathsf{sv}_+(\theta), \mathsf{sv}_-(\theta)$ be (the indices of) any positive and negative support vectors, i.e.,

$$\mathsf{sv}_+(\theta) \in \mathcal{SV}_+(\theta), \qquad \mathsf{sv}_-(\theta) \in \mathcal{SV}_-(\theta).$$

Now, recall that whenever a slope parameter $\beta$ is given, the optimal intercept $\check{\beta}_0 := \check{\beta}_0(\beta)$ (defined in Eq. (26)) must satisfy the *margin-balancing* condition Eq. (27), according to Theorem C.3. Hence, fixing $\rho = \widehat{\rho}$ and considering arbitrary $\theta$, we can rewrite Eq. (25) and (27) as

$$
\begin{aligned}
\kappa(\widehat{\rho}, \theta, \check{\beta}_0) &= \kappa_{\mathsf{sv}_+(\theta)}(\widehat{\rho}, \theta, \check{\beta}_0) = \widehat{\rho} \|\mu\|_2 + \check{\beta}_0 + \widehat{\rho} g_{\mathsf{sv}_+(\theta)} + \sqrt{1 - \widehat{\rho}^2} \langle z_{\mathsf{sv}_+(\theta)}, \theta \rangle \\
&= \kappa_{\mathsf{sv}_-(\theta)}(\widehat{\rho}, \theta, \check{\beta}_0) = \widehat{\rho} \|\mu\|_2 - \check{\beta}_0 - \widehat{\rho} g_{\mathsf{sv}_-(\theta)} - \sqrt{1 - \widehat{\rho}^2} \langle z_{\mathsf{sv}_-(\theta)}, \theta \rangle.
\end{aligned}
\tag{152}
$$

In particular, if $\theta = \widehat{\theta}$, we denote $\mathsf{sv}_+(\widehat{\theta}) \in \mathcal{SV}_+(\widehat{\theta})$, $\mathsf{sv}_-(\widehat{\theta}) \in \mathcal{SV}_-(\widehat{\theta})$ as the support vectors of max-margin classifier. The Lemma below describes the magnitude of $\langle z_{\mathsf{sv}_+(\widehat{\theta})}, \widehat{\theta} \rangle$ and $\langle z_{\mathsf{sv}_-(\widehat{\theta})}, \widehat{\theta} \rangle$.

**Lemma H.4** (Asymptotics of $\langle z_i, \widehat{\theta} \rangle$'s for support vectors). *Suppose that $a < c + 1$.*

*(a) If $a < b + c$, then*

$$\sqrt{1 - \widehat{\rho}^2} \langle z_{\mathsf{sv}_+(\widehat{\theta})}, \widehat{\theta} \rangle = \widetilde{O}_{\mathbb{P}}(d^{a - \frac{b}{2} - c} \vee 1), \qquad \sqrt{1 - \widehat{\rho}^2} \langle z_{\mathsf{sv}_-(\widehat{\theta})}, \widehat{\theta} \rangle = \widetilde{O}_{\mathbb{P}}(1).$$

*(b) If $a > b + c$, then*

$$\langle z_{\mathsf{sv}_+(\widehat{\theta})}, \widehat{\theta} \rangle = \sqrt{\frac{d}{\pi n}} (1 + o_{\mathbb{P}}(1)), \qquad \langle z_{\mathsf{sv}_-(\widehat{\theta})}, \widehat{\theta} \rangle = \widetilde{O}_{\mathbb{P}}(1).$$

*Proof.* $\mathcal{SV}_\pm(\theta)$ may not be tractable, since it involves a nuisance term $\widehat{\rho} g_i$ as defined in Eq. (151). Therefore, we introduce a proxy of support vectors, which is easier to work with. Formally, let

$$
\begin{aligned}
\mathcal{V}_+ = \mathcal{V}_+(\theta) &:= \underset{i \in \mathcal{I}_+}{\arg\min} +\langle z_i, \theta \rangle, \\
\mathcal{V}_- = \mathcal{V}_-(\theta) &:= \underset{i \in \mathcal{I}_-}{\arg\min} -\langle z_i, \theta \rangle,
\end{aligned}
\tag{153}
$$

where $\mathcal{V}_+, \mathcal{V}_-$ are sets of (the indices of) the smallest $y_i \langle z_i, \theta \rangle$ from each class. Similarly, let

$$\mathsf{v}_+(\theta) \in \mathcal{V}_+(\theta), \qquad \mathsf{v}_-(\theta) \in \mathcal{V}_-(\theta),$$

which are arbitrary elements in $\mathcal{V}_+(\theta)$ and $\mathcal{V}_-(\theta)$. Note that $\mathcal{V}_\pm$ is simply $\mathcal{SV}_\pm$ but ignoring term $\widehat{\rho} g_i$. Indeed, as we will show later, the impact of $\widehat{\rho} g_i = O_{\mathbb{P}}(1)$ is almost negligible.

We are going to prove Theorem H.4 by deriving tight upper bounds and lower bounds for both $\pm \sqrt{1 - \widehat{\rho}^2} \langle z_{\mathsf{sv}_\pm(\widehat{\theta})}, \widehat{\theta} \rangle$. Then we conclude the precise asymptotics by verifying the upper and lower bounds are matched.

**Upper bounds** Applying the same idea as Eq. (135), we can bound $\pm\langle z_{\mathsf{v}_{\pm}(\boldsymbol{\theta})}, \boldsymbol{\theta}\rangle$ via averaging:

$$\langle z_{\mathsf{v}_{+}(\boldsymbol{\theta})}, \boldsymbol{\theta}\rangle \leq \langle \overline{z}_{+}, \boldsymbol{\theta}\rangle \leq \|\mathbf{P}_{\boldsymbol{\mu}}^{\perp}\overline{z}_{+}\|_2, \qquad -\langle z_{\mathsf{v}_{-}(\boldsymbol{\theta})}, \boldsymbol{\theta}\rangle \leq -\langle \overline{z}_{-}, \boldsymbol{\theta}\rangle \leq \|\mathbf{P}_{\boldsymbol{\mu}}^{\perp}\overline{z}_{-}\|_2, \qquad (154)$$

where the second inequality for each comes from Eq. (137). To create a connection between $\mathsf{sv}_{\pm}(\widehat{\boldsymbol{\theta}})$ and $\mathsf{v}_{\pm}(\widehat{\boldsymbol{\theta}})$, note that by definition Eq. (151)

$$\widehat{\rho}g_{\mathsf{sv}_{+}(\widehat{\boldsymbol{\theta}})} + \sqrt{1-\widehat{\rho}^2}\langle z_{\mathsf{sv}_{+}(\widehat{\boldsymbol{\theta}})}, \widehat{\boldsymbol{\theta}}\rangle \leq \ \widehat{\rho}g_{\mathsf{v}_{+}(\widehat{\boldsymbol{\theta}})} + \sqrt{1-\widehat{\rho}^2}\langle z_{\mathsf{v}_{+}(\widehat{\boldsymbol{\theta}})}, \widehat{\boldsymbol{\theta}}\rangle,$$

$$-\widehat{\rho}g_{\mathsf{sv}_{-}(\widehat{\boldsymbol{\theta}})} - \sqrt{1-\widehat{\rho}^2}\langle z_{\mathsf{sv}_{-}(\widehat{\boldsymbol{\theta}})}, \widehat{\boldsymbol{\theta}}\rangle \leq -\widehat{\rho}g_{\mathsf{v}_{-}(\widehat{\boldsymbol{\theta}})} - \sqrt{1-\widehat{\rho}^2}\langle z_{\mathsf{v}_{-}(\widehat{\boldsymbol{\theta}})}, \widehat{\boldsymbol{\theta}}\rangle.$$

Using Eq. (154), therefore we obtain the following non-asymptotic upper bounds on $\langle z_{\mathsf{sv}_{\pm}(\widehat{\boldsymbol{\theta}})}, \widehat{\boldsymbol{\theta}}\rangle$:

$$
\begin{aligned}
\sqrt{1-\widehat{\rho}^2}\langle z_{\mathsf{sv}_{+}(\widehat{\boldsymbol{\theta}})}, \widehat{\boldsymbol{\theta}}\rangle &\leq \ \sqrt{1-\widehat{\rho}^2}\langle z_{\mathsf{v}_{+}(\widehat{\boldsymbol{\theta}})}, \widehat{\boldsymbol{\theta}}\rangle + \widehat{\rho}\big(g_{\mathsf{v}_{+}(\widehat{\boldsymbol{\theta}})} - g_{\mathsf{sv}_{+}(\widehat{\boldsymbol{\theta}})}\big) \\
&\leq \ \sqrt{1-\widehat{\rho}^2}\|\mathbf{P}_{\boldsymbol{\mu}}^{\perp}\overline{z}_{+}\|_2 + 2\widehat{\rho}\max_{i\in[n]}|g_i|, \\
-\sqrt{1-\widehat{\rho}^2}\langle z_{\mathsf{sv}_{-}(\widehat{\boldsymbol{\theta}})}, \widehat{\boldsymbol{\theta}}\rangle &\leq -\sqrt{1-\widehat{\rho}^2}\langle z_{\mathsf{v}_{-}(\widehat{\boldsymbol{\theta}})}, \widehat{\boldsymbol{\theta}}\rangle - \widehat{\rho}\big(g_{\mathsf{v}_{-}(\widehat{\boldsymbol{\theta}})} - g_{\mathsf{sv}_{-}(\widehat{\boldsymbol{\theta}})}\big) \\
&\leq \ \sqrt{1-\widehat{\rho}^2}\|\mathbf{P}_{\boldsymbol{\mu}}^{\perp}\overline{z}_{-}\|_2 + 2\widehat{\rho}\max_{i\in[n]}|g_i|.
\end{aligned}
\qquad (155)
$$

To compute its asymptotics, recall that $\sqrt{n_+}\cdot\overline{z}_{+}\sim\mathsf{subG}_{\perp\!\!\!\perp}(\mathbf{0},\mathbf{I}_d;K)$, $\sqrt{n_-}\cdot\overline{z}_{-}\sim\mathsf{subG}_{\perp\!\!\!\perp}(\mathbf{0},\mathbf{I}_d;K)$, and $\|\mathbf{P}_{\boldsymbol{\mu}}^{\perp}\|_{\mathrm{F}}=\sqrt{d-1}$. Then by Theorem J.3(b),

$$\|\mathbf{P}_{\boldsymbol{\mu}}^{\perp}\overline{z}_{+}\|_2 = \frac{1}{\sqrt{n_+}}\|\mathbf{P}_{\boldsymbol{\mu}}^{\perp}\|_{\mathrm{F}}\big(1+o_{\mathbb{P}}(1)\big) = \sqrt{\frac{d}{\pi n}}\big(1+o_{\mathbb{P}}(1)\big),$$

$$\|\mathbf{P}_{\boldsymbol{\mu}}^{\perp}\overline{z}_{-}\|_2 = \frac{1}{\sqrt{n_-}}\|\mathbf{P}_{\boldsymbol{\mu}}^{\perp}\|_{\mathrm{F}}\big(1+o_{\mathbb{P}}(1)\big) = \ \sqrt{\frac{d}{n}}\big(1+o_{\mathbb{P}}(1)\big) = o_{\mathbb{P}}(1). \qquad (156)$$

While, by maximal inequality Theorem J.2(c) or Eq. (144), we have

$$\max_{i\in[n]}|g_i| = O_{\mathbb{P}}(\log n) = \widetilde{O}_{\mathbb{P}}(1), \qquad (157)$$

Plugging Eq. (156) and (157) into Eq. (155) gives the asymptotic upper bounds (involving $\widehat{\rho}$):

$$
\begin{aligned}
\sqrt{1-\widehat{\rho}^2}\langle z_{\mathsf{sv}_{+}(\widehat{\boldsymbol{\theta}})}, \widehat{\boldsymbol{\theta}}\rangle &\leq \sqrt{1-\widehat{\rho}^2}\sqrt{\frac{d}{\pi n}}\big(1+o_{\mathbb{P}}(1)\big) + \widehat{\rho}\cdot\widetilde{O}_{\mathbb{P}}(1), \\
-\sqrt{1-\widehat{\rho}^2}\langle z_{\mathsf{sv}_{-}(\widehat{\boldsymbol{\theta}})}, \widehat{\boldsymbol{\theta}}\rangle &\leq \sqrt{1-\widehat{\rho}^2}\cdot o_{\mathbb{P}}(1) + \widehat{\rho}\cdot\widetilde{O}_{\mathbb{P}}(1).
\end{aligned}
\qquad (158)
$$

**Lower bounds** Similar as the proof of Theorem H.1, a lower bound can be obtained by plugging our constructed solution $\widetilde{\boldsymbol{\theta}} = \mathbf{P}_{\boldsymbol{\mu}}^{\perp}\widetilde{z}/\|\mathbf{P}_{\boldsymbol{\mu}}^{\perp}\widetilde{z}\|_2$, which can be a good "proxy" of $\boldsymbol{\theta}$. Again, by margin-balancing condition Eq. (152), we can express the optimal $\boldsymbol{\theta}$ as[16]

$$
\begin{aligned}
\widehat{\boldsymbol{\theta}} &\in \operatorname*{arg\,max}_{\boldsymbol{\theta}\in\S^{d-1},\boldsymbol{\theta}\perp\boldsymbol{\mu}} \kappa(\widehat{\rho},\boldsymbol{\theta},\widehat{\beta}_0) = \operatorname*{arg\,max}_{\boldsymbol{\theta}\in\S^{d-1},\boldsymbol{\theta}\perp\boldsymbol{\mu}} \frac{\kappa_{\mathsf{sv}_{+}(\boldsymbol{\theta})}(\widehat{\rho},\boldsymbol{\theta},\widehat{\beta}_0) + \kappa_{\mathsf{sv}_{-}(\boldsymbol{\theta})}(\widehat{\rho},\boldsymbol{\theta},\widehat{\beta}_0)}{2} \\
&= \operatorname*{arg\,max}_{\boldsymbol{\theta}\in\S^{d-1},\boldsymbol{\theta}\perp\boldsymbol{\mu}} \left\{\widehat{\rho}\|\boldsymbol{\mu}\|_2 + \widehat{\rho}\frac{g_{\mathsf{sv}_{+}(\boldsymbol{\theta})} - g_{\mathsf{sv}_{-}(\boldsymbol{\theta})}}{2} + \sqrt{1-\widehat{\rho}^2}\frac{\langle z_{\mathsf{sv}_{+}(\boldsymbol{\theta})}, \boldsymbol{\theta}\rangle - \langle z_{\mathsf{sv}_{-}(\boldsymbol{\theta})}, \boldsymbol{\theta}\rangle}{2}\right\} \\
&= \operatorname*{arg\,max}_{\boldsymbol{\theta}\in\S^{d-1},\boldsymbol{\theta}\perp\boldsymbol{\mu}} \left\{\widehat{\rho}\big(g_{\mathsf{sv}_{+}(\boldsymbol{\theta})} - g_{\mathsf{sv}_{-}(\boldsymbol{\theta})}\big) + \sqrt{1-\widehat{\rho}^2}\big(\langle z_{\mathsf{sv}_{+}(\boldsymbol{\theta})}, \boldsymbol{\theta}\rangle + \langle z_{\mathsf{sv}_{-}(\boldsymbol{\theta})}, \boldsymbol{\theta}\rangle\big)\right\}.
\end{aligned}
$$

---

[16]Notice that if $|\widehat{\rho}| < 1$, then $\arg\max_{\boldsymbol{\theta}\in\S^{d-1},\boldsymbol{\theta}\perp\boldsymbol{\mu}}\kappa(\widehat{\rho},\boldsymbol{\theta},\widehat{\beta}_0)$ is unique (on the event of $\{\widehat{\kappa} > 0\}$), and we could write $\widehat{\boldsymbol{\theta}} = \arg\max_{\boldsymbol{\theta}\in\S^{d-1},\boldsymbol{\theta}\perp\boldsymbol{\mu}}\kappa(\widehat{\rho},\boldsymbol{\theta},\widehat{\beta}_0)$. However, if $|\widehat{\rho}| = 1$, then according to our construction Eq. (126), the arguments of the maxima can be any $\boldsymbol{\theta}\in\S^{d-1}$ such that $\boldsymbol{\theta}\perp\boldsymbol{\mu}$, while $\widehat{\boldsymbol{\theta}} = \boldsymbol{\mu}_{\perp}$ as defined in Eq. (131).

Therefore, recalling Eq. (153), we have

$$\sqrt{1 - \widehat{\rho}^2}\big(\langle z_{\mathsf{sv}_+(\widehat{\theta})}, \widehat{\theta}\rangle - \langle z_{\mathsf{sv}_-(\widehat{\theta})}, \widehat{\theta}\rangle\big)$$
$$\geq \sqrt{1 - \widehat{\rho}^2}\big(\langle z_{\mathsf{sv}_+(\widetilde{\theta})}, \widehat{\theta}\rangle - \langle z_{\mathsf{sv}_-(\widetilde{\theta})}, \widehat{\theta}\rangle\big) + \widehat{\rho}\big(g_{\mathsf{sv}_+(\widetilde{\theta})} - g_{\mathsf{sv}_-(\widetilde{\theta})} - g_{\mathsf{sv}_+(\widehat{\theta})} + g_{\mathsf{sv}_-(\widehat{\theta})}\big)$$
$$\geq \sqrt{1 - \widehat{\rho}^2}\big(\langle z_{\mathsf{v}_+(\widetilde{\theta})}, \widehat{\theta}\rangle - \langle z_{\mathsf{v}_-(\widetilde{\theta})}, \widehat{\theta}\rangle\big) - 4\widehat{\rho}\max_{i\in[n]}|g_i|.$$

Combining it with Eq. (155), we can obtain a lower bound for each term using $\widetilde{\theta}$:

$$\sqrt{1 - \widehat{\rho}^2}\langle z_{\mathsf{sv}_+(\widehat{\theta})}, \widehat{\theta}\rangle \geq \sqrt{1 - \widehat{\rho}^2}\big(\langle z_{\mathsf{v}_+(\widetilde{\theta})}, \widetilde{\theta}\rangle - \langle z_{\mathsf{v}_-(\widetilde{\theta})}, \widetilde{\theta}\rangle\big) - 4\widehat{\rho}\max_{i\in[n]}|g_i| + \sqrt{1 - \widehat{\rho}^2}\langle z_{\mathsf{sv}_-(\widehat{\theta})}, \widehat{\theta}\rangle$$
$$\geq \sqrt{1 - \widehat{\rho}^2}\big(-\langle z_{\mathsf{v}_-(\widetilde{\theta})}, \widetilde{\theta}\rangle - \|\mathbf{P}_{\boldsymbol{\mu}}^{\perp}\overline{z}_-\|_2\big) - 6\widehat{\rho}\max_{i\in[n]}|g_i| + \sqrt{1 - \widehat{\rho}^2}\langle z_{\mathsf{v}_+(\widetilde{\theta})}, \widetilde{\theta}\rangle,$$
$$-\sqrt{1 - \widehat{\rho}^2}\langle z_{\mathsf{sv}_-(\widehat{\theta})}, \widehat{\theta}\rangle \geq \sqrt{1 - \widehat{\rho}^2}\big(\langle z_{\mathsf{v}_+(\widetilde{\theta})}, \widetilde{\theta}\rangle - \langle z_{\mathsf{v}_-(\widetilde{\theta})}, \widetilde{\theta}\rangle\big) - 4\widehat{\rho}\max_{i\in[n]}|g_i| - \sqrt{1 - \widehat{\rho}^2}\langle z_{\mathsf{sv}_+(\widehat{\theta})}, \widehat{\theta}\rangle$$
$$\geq \sqrt{1 - \widehat{\rho}^2}\big(+\langle z_{\mathsf{v}_+(\widetilde{\theta})}, \widetilde{\theta}\rangle - \|\mathbf{P}_{\boldsymbol{\mu}}^{\perp}\overline{z}_+\|_2\big) - 6\widehat{\rho}\max_{i\in[n]}|g_i| - \sqrt{1 - \widehat{\rho}^2}\langle z_{\mathsf{v}_-(\widetilde{\theta})}, \widetilde{\theta}\rangle. \tag{159}$$

To derive its asymptotic order, we first define two statistics that are closely related to $\widetilde{\theta}$:

$$\widetilde{\theta}_+ := \frac{\mathbf{P}_{\boldsymbol{\mu}}^{\perp}\overline{z}_+}{\|\mathbf{P}_{\boldsymbol{\mu}}^{\perp}\overline{z}_+\|_2}, \qquad \widetilde{\theta}_- := \frac{-\mathbf{P}_{\boldsymbol{\mu}}^{\perp}\overline{z}_-}{\|\mathbf{P}_{\boldsymbol{\mu}}^{\perp}\overline{z}_-\|_2}. \tag{160}$$

Then, the difference terms inside the parentheses in Eq. (159) can be expressed as

$$\langle z_{\mathsf{v}_+(\widetilde{\theta})}, \widetilde{\theta}\rangle - \|\mathbf{P}_{\boldsymbol{\mu}}^{\perp}\overline{z}_+\|_2 = \min_{i\in\mathcal{I}_+}\langle +z_i, \widetilde{\theta}\rangle - \langle \overline{z}_+, \widetilde{\theta}_+\rangle = \min_{i\in\mathcal{I}_+}\langle z_i - \overline{z}_+, \widetilde{\theta}\rangle + \langle \overline{z}_+, \widetilde{\theta} - \widetilde{\theta}_+\rangle,$$
$$-\langle z_{\mathsf{v}_-(\widetilde{\theta})}, \widetilde{\theta}\rangle - \|\mathbf{P}_{\boldsymbol{\mu}}^{\perp}\overline{z}_-\|_2 = \min_{i\in\mathcal{I}_-}\langle -z_i, \widetilde{\theta}\rangle + \langle \overline{z}_-, \widetilde{\theta}_-\rangle = \min_{i\in\mathcal{I}_-}\langle \overline{z}_- - z_i, \widetilde{\theta}\rangle - \langle \overline{z}_-, \widetilde{\theta} - \widetilde{\theta}_-\rangle. \tag{161}$$

Now we study the two terms on the R.H.S. of Eq. (161). For the first term, based on Eq. (146),

$$\min_{i\in\mathcal{I}_+}\langle z_i - \overline{z}_+, \widetilde{\theta}\rangle \geq -\max_{i\in\mathcal{I}_+}\big|\langle z_i - \overline{z}_+, \widetilde{\theta}\rangle\big| = \widetilde{O}_{\mathbb{P}}(1),$$
$$\min_{i\in\mathcal{I}_-}\langle \overline{z}_- - z_i, \widetilde{\theta}\rangle \geq -\max_{i\in\mathcal{I}_-}\big|\langle z_i - \overline{z}_-, \widetilde{\theta}\rangle\big| = \widetilde{O}_{\mathbb{P}}(1). \tag{162}$$

For the second term,

$$\langle \overline{z}_+, \widetilde{\theta} - \widetilde{\theta}_+\rangle = \frac{1}{\|\mathbf{P}_{\boldsymbol{\mu}}^{\perp}\widetilde{z}\|_2}\langle \overline{z}_+, \mathbf{P}_{\boldsymbol{\mu}}^{\perp}\widetilde{z}\rangle - \frac{1}{\|\mathbf{P}_{\boldsymbol{\mu}}^{\perp}\overline{z}_+\|_2}\langle \overline{z}_+, \mathbf{P}_{\boldsymbol{\mu}}^{\perp}\overline{z}_+\rangle$$
$$= \frac{1}{\|\mathbf{P}_{\boldsymbol{\mu}}^{\perp}\widetilde{z}\|_2}\left\{\langle \overline{z}_+, \mathbf{P}_{\boldsymbol{\mu}}^{\perp}\widetilde{z}\rangle - \langle \overline{z}_+, \mathbf{P}_{\boldsymbol{\mu}}^{\perp}\overline{z}_+\rangle \cdot \frac{\|\mathbf{P}_{\boldsymbol{\mu}}^{\perp}\widetilde{z}\|_2}{\|\mathbf{P}_{\boldsymbol{\mu}}^{\perp}\overline{z}_+\|_2}\right\}$$
$$\stackrel{(i)}{\geq} \frac{1}{\|\mathbf{P}_{\boldsymbol{\mu}}^{\perp}\widetilde{z}\|_2}\left\{\langle \overline{z}_+, \mathbf{P}_{\boldsymbol{\mu}}^{\perp}\widetilde{z}\rangle - \langle \overline{z}_+, \mathbf{P}_{\boldsymbol{\mu}}^{\perp}\overline{z}_+\rangle \cdot \frac{1}{2}\left(1 + \frac{\|\mathbf{P}_{\boldsymbol{\mu}}^{\perp}\overline{z}_-\|_2}{\|\mathbf{P}_{\boldsymbol{\mu}}^{\perp}\overline{z}_+\|_2}\right)\right\} \tag{163}$$
$$\stackrel{(ii)}{=} -\frac{1}{2\|\mathbf{P}_{\boldsymbol{\mu}}^{\perp}\widetilde{z}\|_2}\big(\langle \overline{z}_+, \mathbf{P}_{\boldsymbol{\mu}}^{\perp}\overline{z}_-\rangle + \|\mathbf{P}_{\boldsymbol{\mu}}^{\perp}\overline{z}_+\|_2\|\mathbf{P}_{\boldsymbol{\mu}}^{\perp}\overline{z}_-\|_2\big)$$
$$\stackrel{(iii)}{=} -\sqrt{\frac{\pi n}{d}}\big(1 + o_{\mathbb{P}}(1)\big)\left\{O_{\mathbb{P}}\left(\sqrt{\frac{d}{\pi n^2}}\right) + \sqrt{\frac{d}{\pi n}}\sqrt{\frac{d}{n}}\big(1 + o_{\mathbb{P}}(1)\big)\right\}$$
$$= -\sqrt{\frac{d}{n}}\big(1 + o_{\mathbb{P}}(1)\big) = o_{\mathbb{P}}(1),$$

where (i) is from triangular inequality $2\|\mathbf{P}_{\boldsymbol{\mu}}^{\perp}\widetilde{z}\|_2 \leq \|\mathbf{P}_{\boldsymbol{\mu}}^{\perp}\overline{z}_+\|_2 + \|\mathbf{P}_{\boldsymbol{\mu}}^{\perp}\overline{z}_-\|_2$, (ii) uses $2\widetilde{z} - \overline{z}_+ = -\overline{z}_-$, and (iii) applies the asymptotic results Eq. (140), (156), and the fact that $\overline{z}_+ \perp\!\!\!\perp \overline{z}_-$,

$$\langle \overline{z}_+, \mathbf{P}_{\boldsymbol{\mu}}^{\perp}\overline{z}_-\rangle = \frac{1}{\sqrt{n_+ n_-}}O_{\mathbb{P}}(\|\mathbf{P}_{\boldsymbol{\mu}}^{\perp}\|_{\mathrm{F}}) = O_{\mathbb{P}}\left(\sqrt{\frac{d}{\pi n^2}}\right),$$

by Theorem J.3(d). Similarly, we also have

$$
\begin{aligned}
-\langle \overline{z}_-, \widetilde{\theta} - \widetilde{\theta}_- \rangle &= -\frac{1}{\|\mathbf{P}_{\boldsymbol{\mu}}^{\perp}\widetilde{z}\|_2}\langle \overline{z}_-, \mathbf{P}_{\boldsymbol{\mu}}^{\perp}\widetilde{z} \rangle - \frac{1}{\|\mathbf{P}_{\boldsymbol{\mu}}^{\perp}\overline{z}_-\|_2}\langle \overline{z}_-, \mathbf{P}_{\boldsymbol{\mu}}^{\perp}\overline{z}_- \rangle \\
&\geq -\frac{1}{2\|\mathbf{P}_{\boldsymbol{\mu}}^{\perp}\widetilde{z}\|_2}\left(\langle \overline{z}_+, \mathbf{P}_{\boldsymbol{\mu}}^{\perp}\overline{z}_- \rangle + \|\mathbf{P}_{\boldsymbol{\mu}}^{\perp}\overline{z}_+\|_2\|\mathbf{P}_{\boldsymbol{\mu}}^{\perp}\overline{z}_-\|_2\right) \\
&= o_{\mathbb{P}}(1).
\end{aligned}
\tag{164}
$$

Substituting Eq. (162), (163), and (164) into Eq. (161), we get

$$
\langle z_{\mathsf{v}_+(\widetilde{\theta})}, \widetilde{\theta} \rangle - \|\mathbf{P}_{\boldsymbol{\mu}}^{\perp}\overline{z}_+\|_2 \geq \widetilde{O}_{\mathbb{P}}(1), \qquad -\langle z_{\mathsf{v}_-(\widetilde{\theta})}, \widetilde{\theta} \rangle - \|\mathbf{P}_{\boldsymbol{\mu}}^{\perp}\overline{z}_-\|_2 \geq \widetilde{O}_{\mathbb{P}}(1).
\tag{165}
$$

And combining this with Eq. (156), we have

$$
\langle z_{\mathsf{v}_+(\widetilde{\theta})}, \widetilde{\theta} \rangle \geq \sqrt{\frac{d}{\pi n}}\left(1 + o_{\mathbb{P}}(1)\right) + \widetilde{O}_{\mathbb{P}}(1), \qquad -\langle z_{\mathsf{v}_-(\widetilde{\theta})}, \widetilde{\theta} \rangle \geq \widetilde{O}_{\mathbb{P}}(1).
\tag{166}
$$

Plugging Eq. (165), (166), and (157) into Eq. (159) gives the asymptotic lower bounds (involving $\widehat{\rho}$):

$$
\begin{aligned}
\sqrt{1 - \widehat{\rho}^2}\langle z_{\mathsf{sv}_+(\widehat{\theta})}, \widehat{\theta} \rangle &\geq \sqrt{1 - \widehat{\rho}^2}\left(\sqrt{\frac{d}{\pi n}}\left(1 + o_{\mathbb{P}}(1)\right) + \widetilde{O}_{\mathbb{P}}(1)\right) + \widehat{\rho} \cdot \widetilde{O}_{\mathbb{P}}(1), \\
-\sqrt{1 - \widehat{\rho}^2}\langle z_{\mathsf{sv}_-(\widehat{\theta})}, \widehat{\theta} \rangle &\geq \sqrt{1 - \widehat{\rho}^2} \cdot \widetilde{O}_{\mathbb{P}}(1) + \widehat{\rho} \cdot \widetilde{O}_{\mathbb{P}}(1).
\end{aligned}
\tag{167}
$$

Finally, combining upper bounds Eq. (158) and lower bounds Eq. (167), we obtain the exact order

$$
\begin{aligned}
\sqrt{1 - \widehat{\rho}^2}\langle z_{\mathsf{sv}_+(\widehat{\theta})}, \widehat{\theta} \rangle &= \sqrt{1 - \widehat{\rho}^2}\left(\sqrt{\frac{d}{\pi n}}\left(1 + o_{\mathbb{P}}(1)\right) + \widetilde{O}_{\mathbb{P}}(1)\right) + \widehat{\rho} \cdot \widetilde{O}_{\mathbb{P}}(1) \\
&= \sqrt{1 - \widehat{\rho}^2}\sqrt{\frac{d}{\pi n}}\left(1 + o_{\mathbb{P}}(1)\right) + \widetilde{O}_{\mathbb{P}}(1), \\
-\sqrt{1 - \widehat{\rho}^2}\langle z_{\mathsf{sv}_-(\widehat{\theta})}, \widehat{\theta} \rangle &= \sqrt{1 - \widehat{\rho}^2} \cdot \widetilde{O}_{\mathbb{P}}(1) + \widehat{\rho} \cdot \widetilde{O}_{\mathbb{P}}(1) \\
&= \widetilde{O}_{\mathbb{P}}(1).
\end{aligned}
$$

**(a):** If $a < b + c$, according to Theorem H.3(a), $\widehat{\rho} = 1 - o_{\mathbb{P}}(1)$. It is clear that Theorem H.4 holds for $\widehat{\rho} = \pm 1$. Now, restrict on the event $\{|\widehat{\rho}| < 1\}$.

- If $a > \frac{b}{2} + c$, then $\sqrt{1 - \widehat{\rho}^2} = \frac{1}{2}d^{(a-b-c)/2}\left(1 + o_{\mathbb{P}}(1)\right)$, hence

$$
\sqrt{1 - \widehat{\rho}^2}\langle z_{\mathsf{sv}_+(\widehat{\theta})}, \widehat{\theta} \rangle = \frac{1}{2}d^{a - \frac{b}{2} - c}\left(1 + o_{\mathbb{P}}(1)\right).
$$

- If $a \leq \frac{b}{2} + c$, then $\sqrt{1 - \widehat{\rho}^2}\sqrt{d/\pi n} = \widetilde{O}_{\mathbb{P}}(1)$, hence

$$
\sqrt{1 - \widehat{\rho}^2}\langle z_{\mathsf{sv}_+(\widehat{\theta})}, \widehat{\theta} \rangle = \widetilde{O}_{\mathbb{P}}(1).
$$

**(b):** If $a > b + c$, according to Theorem H.3(b), $\widehat{\rho} = o_{\mathbb{P}}(1)$. Hence, on the event $\{|\widehat{\rho}| < 1\}$,

$$
\langle z_{\mathsf{sv}_+(\widehat{\theta})}, \widehat{\theta} \rangle = \sqrt{\frac{d}{\pi n}}\left(1 + o_{\mathbb{P}}(1)\right).
$$

This also holds regardless of $\widehat{\rho}$, since $\mathbb{P}(|\widehat{\rho}| < 1) \to 1$ as $d \to \infty$. Then we complete the proof. $\quad\square$

### H.2.3 ASYMPTOTIC EXPRESSION OF $\widehat{\beta}_0$: PROOF OF LEMMA H.5

Finally, we consider arbitrary $\tau \geq 1$ and give an explicit expression for $\widehat{\beta}_0$ with its asymptotics. Be aware that $\tau = \tau_d$ may depend on $d$.

**Lemma H.5** (Asymptotics of $\widehat{\beta}_0$). *Suppose that $a < c + 1$ and $\tau \geq 1$. Then we have*

$$\widehat{\beta}_0 = \left(1 - \frac{2}{\tau+1}\right)\widehat{\rho}\,\|\boldsymbol{\mu}\|_2 - \widehat{\rho}\frac{\tau g_{\mathsf{sv}-(\widehat{\boldsymbol{\theta}})} + g_{\mathsf{sv}+(\widehat{\boldsymbol{\theta}})}}{\tau+1} - \sqrt{1-\widehat{\rho}^2}\frac{\tau\langle \boldsymbol{z}_{\mathsf{sv}-(\widehat{\boldsymbol{\theta}})}, \widehat{\boldsymbol{\theta}}\rangle + \langle \boldsymbol{z}_{\mathsf{sv}+(\widehat{\boldsymbol{\theta}})}, \widehat{\boldsymbol{\theta}}\rangle}{\tau+1}$$

$$= \left(1 - \frac{2}{\tau+1}\right)\widehat{\rho}\,\|\boldsymbol{\mu}\|_2 - \frac{1}{\tau+1}\sqrt{1-\widehat{\rho}^2}\langle \boldsymbol{z}_{\mathsf{sv}+(\widehat{\boldsymbol{\theta}})}, \widehat{\boldsymbol{\theta}}\rangle + \widetilde{O}_{\mathbb{P}}(1).$$

*(a) If $a < b + c$, then*

$$\widehat{\beta}_0 = \left(1 - \frac{2}{\tau+1}\right)\widehat{\rho}\,\|\boldsymbol{\mu}\|_2 - \frac{1}{\tau+1}\widetilde{O}_{\mathbb{P}}(d^{a-\frac{b}{2}-c} \vee 1) + \widetilde{O}_{\mathbb{P}}(1)$$

$$= \left(1 - \frac{2}{\tau+1}\right)d^{b/2}\big(1 + o_{\mathbb{P}}(1)\big) - \frac{1}{\tau+1}\widetilde{O}_{\mathbb{P}}(d^{a-\frac{b}{2}-c} \vee 1) + \widetilde{O}_{\mathbb{P}}(1).$$

*(b) If $a > b + c$, then*

$$\widehat{\beta}_0 = \left(1 - \frac{2}{\tau+1}\right)\widehat{\rho}\,\|\boldsymbol{\mu}\|_2 - \frac{1}{\tau+1}\sqrt{\frac{d}{\pi n}}\big(1 + o_{\mathbb{P}}(1)\big) + \widetilde{O}_{\mathbb{P}}(1)$$

$$= \begin{cases} \left(1 - \dfrac{2}{\tau+1}\right)2d^{(2b-a+c)/2}\big(1 + o_{\mathbb{P}}(1)\big) - \dfrac{1}{\tau+1}d^{(a-c)/2}\big(1 + o_{\mathbb{P}}(1)\big) + \widetilde{O}_{\mathbb{P}}(1), & \text{if } a < 2b + c, \\[2mm] \qquad\qquad\qquad\qquad - \dfrac{1}{\tau+1}d^{(a-c)/2}\big(1 + o_{\mathbb{P}}(1)\big) + \widetilde{O}_{\mathbb{P}}(1), & \text{if } a > 2b + c. \end{cases}$$

*Proof.* We rewrite the *margin-balancing* condition Eq. (25), (27) in terms of $\widehat{\rho}, \widehat{\boldsymbol{\theta}}, \widehat{\beta}_0$, which generalizes Eq. (152) to arbitrary $\tau \geq 1$:

$$\kappa(\widehat{\rho}, \widehat{\boldsymbol{\theta}}, \widehat{\beta}_0) = \kappa_{\mathsf{sv}+(\widehat{\boldsymbol{\theta}})}(\widehat{\rho}, \widehat{\boldsymbol{\theta}}, \widehat{\beta}_0) = \tau^{-1}\left(\widehat{\rho}\,\|\boldsymbol{\mu}\|_2 + \widehat{\beta}_0 + \widehat{\rho}g_{\mathsf{sv}+(\widehat{\boldsymbol{\theta}})} + \sqrt{1-\widehat{\rho}^2}\langle \boldsymbol{z}_{\mathsf{sv}+(\widehat{\boldsymbol{\theta}})}, \widehat{\boldsymbol{\theta}}\rangle\right)$$

$$= \kappa_{\mathsf{sv}-(\widehat{\boldsymbol{\theta}})}(\widehat{\rho}, \widehat{\boldsymbol{\theta}}, \widehat{\beta}_0) = \widehat{\rho}\,\|\boldsymbol{\mu}\|_2 - \widehat{\beta}_0 - \widehat{\rho}g_{\mathsf{sv}-(\widehat{\boldsymbol{\theta}})} - \sqrt{1-\widehat{\rho}^2}\langle \boldsymbol{z}_{\mathsf{sv}-(\widehat{\boldsymbol{\theta}})}, \widehat{\boldsymbol{\theta}}\rangle.$$

Then we can solve the expression for $\widehat{\beta}_0$ (this equals Eq. (24) in Theorem C.3 with parametrization Eq. (131)). Its asymptotic simplification is followed by Eq. (157):

$$\left|\widehat{\rho}\frac{\tau g_{\mathsf{sv}-(\widehat{\boldsymbol{\theta}})} + g_{\mathsf{sv}+(\widehat{\boldsymbol{\theta}})}}{\tau+1}\right| \leq |\widehat{\rho}|\frac{\tau|g_{\mathsf{sv}-(\widehat{\boldsymbol{\theta}})}| + |g_{\mathsf{sv}+(\widehat{\boldsymbol{\theta}})}|}{\tau+1} \leq \max_{i\in[n]}|g_i| = \widetilde{O}_{\mathbb{P}}(1),$$

and Theorem H.4:

$$\left|\frac{\tau}{\tau+1}\sqrt{1-\widehat{\rho}^2}\langle \boldsymbol{z}_{\mathsf{sv}-(\widehat{\boldsymbol{\theta}})}, \widehat{\boldsymbol{\theta}}\rangle\right| = \widetilde{O}_{\mathbb{P}}(1).$$

For **(a)**, plugging $\widehat{\rho} = 1 - o_{\mathbb{P}}(1)$ by Theorem H.3(a) and asymptotics of $\langle \boldsymbol{z}_{\mathsf{sv}+(\widehat{\boldsymbol{\theta}})}, \widehat{\boldsymbol{\theta}}\rangle$ by Theorem H.4(a). For **(b)**, plugging $\widehat{\rho} = 2d^{(b-a+c)/2}\big(1 + o_{\mathbb{P}}(1)\big)$ by Theorem H.3(b) from i., while $\widehat{\rho}\,\|\boldsymbol{\mu}\|_2 = o_{\mathbb{P}}(1)$ from ii., and asymptotics of $\langle \boldsymbol{z}_{\mathsf{sv}+(\widehat{\boldsymbol{\theta}})}, \widehat{\boldsymbol{\theta}}\rangle$ by Theorem H.4(b). This completes the proof. $\square$

### H.3 CLASSIFICATION ERROR: COMPLETING THE PROOF OF THEOREM D.8

***Proof of Theorem D.8.*** Let $(\boldsymbol{x}_{\text{new}}, y_{\text{new}})$ be a test data point independent of the training set $\{(\boldsymbol{x}_i, y_i)\}_{i=1}^n$, such that $\boldsymbol{x}_{\text{new}} = y_{\text{new}}\boldsymbol{\mu} + \boldsymbol{z}_{\text{new}}$, and $\boldsymbol{z}_{\text{new}} \sim \mathsf{subG}_{\perp\!\!\!\perp}(\boldsymbol{0}, \mathbf{I}_d; K)$. Recall $\widehat{f}(\boldsymbol{x}) = \langle \boldsymbol{x}, \widehat{\boldsymbol{\beta}}\rangle + \widehat{\beta}_0$. Following the same decomposition as Eq. (127),

$$y_{\text{new}}\widehat{f}(\boldsymbol{x}_{\text{new}}) = y_{\text{new}}(\langle \boldsymbol{x}_{\text{new}}, \widehat{\boldsymbol{\beta}}\rangle + \widehat{\beta}_0)$$

$$= \widehat{\rho}\,\|\boldsymbol{\mu}\|_2 + y_{\text{new}}\widehat{\beta}_0 + y_{\text{new}}\big(\widehat{\rho}g_{\text{new}} + \sqrt{1-\widehat{\rho}^2}\langle \boldsymbol{z}_{\text{new}}, \widehat{\boldsymbol{\theta}}\rangle\big)$$

$$= \widehat{\rho}\,\|\boldsymbol{\mu}\|_2 + y_{\text{new}}\widehat{\beta}_0 + y_{\text{new}}G_d,$$

where
$$g_{\text{new}} := \left\langle \boldsymbol{z}_{\text{new}}, \frac{\boldsymbol{\mu}}{\|\boldsymbol{\mu}\|_2} \right\rangle, \qquad G_d := \widehat{\rho} g_{\text{new}} + \sqrt{1 - \widehat{\rho}^2} \langle \boldsymbol{z}_{\text{new}}, \widehat{\boldsymbol{\theta}} \rangle.$$

Therefore, the minority and majority test errors are
$$\text{Err}_+ = \mathbb{P}\left( \widehat{f}(\boldsymbol{x}_{\text{new}}) \le 0 \,\middle|\, y_{\text{new}} = +1 \right) = \mathbb{P}\left( \widehat{\rho} \|\boldsymbol{\mu}\|_2 + \widehat{\beta}_0 + G_d \le 0 \right),$$
$$\text{Err}_- = \mathbb{P}\left( \widehat{f}(\boldsymbol{x}_{\text{new}}) > 0 \,\middle|\, y_{\text{new}} = -1 \right) = \mathbb{P}\left( \widehat{\rho} \|\boldsymbol{\mu}\|_2 - \widehat{\beta}_0 - G_d < 0 \right).$$

By Theorem J.3(c), we have $\|g_{\text{new}}\|_{\psi_2}, \|\langle \boldsymbol{z}_{\text{new}}, \widehat{\boldsymbol{\theta}} \rangle\|_{\psi_2} \lesssim K$, since $\boldsymbol{z}_{\text{new}} \perp\!\!\!\perp (\widehat{\rho}, \widehat{\boldsymbol{\theta}})$ and then $\forall\, t > 0$,
$$\mathbb{P}\left( |\langle \boldsymbol{z}_{\text{new}}, \widehat{\boldsymbol{\theta}} \rangle| > t \right) = \mathbb{E}\left[ \mathbb{P}\left( |\langle \boldsymbol{z}_{\text{new}}, \widehat{\boldsymbol{\theta}} \rangle| > t \,\middle|\, \widehat{\boldsymbol{\theta}} \right) \right] \le 2e^{-ct^2/K^2}, \qquad \text{for some } c > 0.$$

Then by Theorem J.2(a),
$$\|G_d\|_{\psi_2} \le \|\widehat{\rho} g_{\text{new}}\|_{\psi_2} + \|\sqrt{1 - \widehat{\rho}^2} \langle \boldsymbol{z}_{\text{new}}, \widehat{\boldsymbol{\theta}} \rangle\|_{\psi_2} \le \|g_{\text{new}}\|_{\psi_2} + \|\langle \boldsymbol{z}_{\text{new}}, \widehat{\boldsymbol{\theta}} \rangle\|_{\psi_2} \lesssim K,$$

which implies $G_d = O_{\mathbb{P}}(1)$.

**1. High signal:** If $a < b + c$, then we have $\widehat{\rho} = 1 - o_{\mathbb{P}}(1)$ by Theorem H.3(a). Therefore, according to Theorem H.5(a), for all $\tau_d \ge 1$, we have
$$\begin{aligned}
\widehat{\rho} \|\boldsymbol{\mu}\|_2 + \widehat{\beta}_0 &= \left( 2 - \frac{2}{\tau_d + 1} \right) \widehat{\rho} \|\boldsymbol{\mu}\|_2 - \frac{1}{\tau_d + 1} \widetilde{O}_{\mathbb{P}}(d^{a - \frac{b}{2} - c} \vee 1) + \widetilde{O}_{\mathbb{P}}(1) \\
&\ge d^{b/2}\big(1 + o_{\mathbb{P}}(1)\big) - \widetilde{O}_{\mathbb{P}}(d^{a - \frac{b}{2} - c} \vee 1) \\
&\stackrel{\text{(i)}}{=} d^{b/2}\big(1 + o_{\mathbb{P}}(1)\big), \qquad \lim_{d \to \infty} d^{b/2} = +\infty,
\end{aligned}$$

where (i) is because $d^{b/2} \gg d^{a - \frac{b}{2} - c}$, as $d \to \infty$. If $1 \le \tau_d \ll d^{b/2}$, we also have
$$\begin{aligned}
\widehat{\rho} \|\boldsymbol{\mu}\|_2 - \widehat{\beta}_0 &= \frac{2}{\tau_d + 1} \widehat{\rho} \|\boldsymbol{\mu}\|_2 + \frac{1}{\tau_d + 1} \widetilde{O}_{\mathbb{P}}(d^{a - \frac{b}{2} - c} \vee 1) + \widetilde{O}_{\mathbb{P}}(1) \\
&= \frac{2}{\tau_d + 1} d^{b/2} + \frac{1}{\tau_d + 1} \widetilde{O}_{\mathbb{P}}(d^{a - \frac{b}{2} - c} \vee 1) + \widetilde{O}_{\mathbb{P}}(1) \\
&\stackrel{\text{(ii)}}{=} \frac{2}{\tau_d + 1} d^{b/2}\big(1 + o_{\mathbb{P}}(1)\big) + \widetilde{O}_{\mathbb{P}}(1) \\
&\ge \tau_d^{-1} d^{b/2}\big(1 + o_{\mathbb{P}}(1)\big) + \widetilde{O}_{\mathbb{P}}(1), \qquad \lim_{d \to \infty} \tau_d^{-1} d^{b/2} = +\infty,
\end{aligned}$$

where (ii) is because $(\tau_d + 1)^{-1} d^{b/2} \gg (\tau_d + 1)^{-1} d^{a - \frac{b}{2} - c}$ and $(\tau_d + 1)^{-1} d^{b/2} \gg (\log d)^k, \forall\, k \ge 0$, as $d \to \infty$. Under these conditions, both $\widehat{\rho} \|\boldsymbol{\mu}\|_2 \pm \widehat{\beta}_0$ diverges to $+\infty$ with high probability, i.e.,
$$\lim_{d \to \infty} \mathbb{P}\left( \widehat{\rho} \|\boldsymbol{\mu}\|_2 + \widehat{\beta}_0 + G_d > C \right) = \lim_{d \to \infty} \mathbb{P}\left( \widehat{\rho} \|\boldsymbol{\mu}\|_2 - \widehat{\beta}_0 - G_d > C \right) = 1, \qquad \forall\, C \in \mathbb{R}.$$

Hence
$$\text{Err}_+ = o(1), \qquad \text{Err}_- = o(1).$$

This concludes the proof for high signal regime.

**2. Moderate signal:** If $b + c < a < 2b + c$, then $\widehat{\rho} = 2d^{(b - a + c)/2}\big(1 + o_{\mathbb{P}}(1)\big)$ by Theorem H.3(b). Therefore, according to Theorem H.5(b), if $\tau_d \gg d^{a - b - c}$, then
$$\begin{aligned}
\widehat{\rho} \|\boldsymbol{\mu}\|_2 + \widehat{\beta}_0 &= \left( 2 - \frac{2}{\tau_d + 1} \right) \widehat{\rho} \|\boldsymbol{\mu}\|_2 - \frac{1}{\tau_d + 1} \sqrt{\frac{d}{\pi n}} \big(1 + o_{\mathbb{P}}(1)\big) + \widetilde{O}_{\mathbb{P}}(1) \\
&= 4d^{(2b - a + c)/2}\big(1 + o_{\mathbb{P}}(1)\big) - \tau_d^{-1} d^{(a - c)/2}\big(1 + o_{\mathbb{P}}(1)\big) + \widetilde{O}_{\mathbb{P}}(1) \\
&\stackrel{\text{(iii)}}{=} 4d^{(2b - a + c)/2}\big(1 + o_{\mathbb{P}}(1)\big), \qquad \lim_{d \to \infty} d^{(2b - a + c)/2} = +\infty,
\end{aligned}$$

where (iii) is because $d^{(2b-a+c)/2} \gg \tau_d^{-1} d^{(a-c)/2}$ and $d^{(2b-a+c)/2} \gg (\log d)^k, \forall k \geq 0$, as $d \to \infty$. If $1 \leq \tau_d \ll d^{(a-c)/2}$, we also have

$$
\begin{aligned}
\widehat{\rho} \|\boldsymbol{\mu}\|_2 - \widehat{\beta}_0 &= \frac{2}{\tau_d + 1} \widehat{\rho} \|\boldsymbol{\mu}\|_2 + \frac{1}{\tau_d + 1} \sqrt{\frac{d}{\pi n}} \left(1 + o_{\mathbb{P}}(1)\right) + \widetilde{O}_{\mathbb{P}}(1) \\
&= \frac{4}{\tau_d + 1} d^{(2b-a+c)/2} \left(1 + o_{\mathbb{P}}(1)\right) + \frac{1}{\tau_d + 1} d^{(a-c)/2} \left(1 + o_{\mathbb{P}}(1)\right) + \widetilde{O}_{\mathbb{P}}(1) \\
&\overset{(iv)}{=} \frac{1}{\tau_d + 1} d^{(a-c)/2} \left(1 + o_{\mathbb{P}}(1)\right) + \widetilde{O}_{\mathbb{P}}(1) \\
&\geq \frac{1}{2} \tau_d^{-1} d^{(a-c)/2} \left(1 + o_{\mathbb{P}}(1)\right) + \widetilde{O}_{\mathbb{P}}(1), \qquad \lim_{d \to \infty} \tau_d^{-1} d^{(a-c)/2} = +\infty,
\end{aligned}
$$

where (iv) is from $(\tau_d + 1)^{-1} d^{(2b-a+c)/2} \ll (\tau_d + 1)^{-1} d^{(a-c)/2}$. Under these conditions on $\tau_d$, both $\widehat{\rho} \|\boldsymbol{\mu}\|_2 \pm \widehat{\beta}_0$ diverges to $+\infty$ with high probability. Using the same approach, we can show that

$$
\text{Err}_+ = o(1), \qquad \text{Err}_- = o(1).
$$

Now suppose $\tau_d \asymp 1$, then again $\widehat{\rho} \|\boldsymbol{\mu}\|_2 - \widehat{\beta}_0 \to +\infty$ and hence $\text{Err}_- = o_{\mathbb{P}}(1)$ still holds. However,

$$
\begin{aligned}
\widehat{\rho} \|\boldsymbol{\mu}\|_2 + \widehat{\beta}_0 &= \left(2 - \frac{2}{\tau_d + 1}\right) \widehat{\rho} \|\boldsymbol{\mu}\|_2 - \frac{1}{\tau_d + 1} \sqrt{\frac{d}{\pi n}} \left(1 + o_{\mathbb{P}}(1)\right) + \widetilde{O}_{\mathbb{P}}(1) \\
&\leq 2 d^{(2b-a+c)/2} \left(1 + o_{\mathbb{P}}(1)\right) - C d^{(a-c)/2} \left(1 + o_{\mathbb{P}}(1)\right) + \widetilde{O}_{\mathbb{P}}(1), \\
&\overset{(v)}{=} -C d^{(a-c)/2} \left(1 + o_{\mathbb{P}}(1)\right), \qquad \lim_{d \to \infty} -d^{(a-c)/2} = -\infty,
\end{aligned}
$$

where (v) is because $d^{(2b-a+c)/2} \ll d^{(a-c)/2}$, and $C \in (0, \infty)$ is an absolute constant. As the result, $-\widehat{\rho} \|\boldsymbol{\mu}\|_2 - \widehat{\beta}_0$ diverges to $+\infty$ with high probability. Using the same approach, we have

$$
\text{Err}_+ = 1 - o(1).
$$

This concludes the proof for moderate signal regime.

**3. Low signal:** If $a > 2b + c$, then $\widehat{\rho} \|\boldsymbol{\mu}\|_2 = o_{\mathbb{P}}(1) > 0$ by Theorem H.3(b). Therefore,

$$
\begin{aligned}
\text{Err}_+ + \text{Err}_- &= \mathbb{P}\left(\widehat{\rho} \|\boldsymbol{\mu}\|_2 + \widehat{\beta}_0 + G_d \leq 0\right) + \mathbb{P}\left(\widehat{\rho} \|\boldsymbol{\mu}\|_2 - \widehat{\beta}_0 - G_d < 0\right) \\
&= 1 - \mathbb{P}\left(-\widehat{\rho} \|\boldsymbol{\mu}\|_2 \leq \widehat{\beta}_0 + G_d < \widehat{\rho} \|\boldsymbol{\mu}\|_2\right) \\
&= 1 - o(1).
\end{aligned}
$$

Hence, we have $\text{Err}_b \geq \frac{1}{2} - o(1)$. This concludes the proof for low signal regime.

Finally, we complete the proof of Theorem D.8. $\qquad \square$

# I    CONFIDENCE ESTIMATION AND CALIBRATION: PROOFS FOR SECTION D.3

## I.1    PROOF OF PROPOSITION D.9

The following preliminary result summarizes the precise asymptotics of three quantities: $\widehat{p}(\boldsymbol{x})$ (max-margin confidence), $p^*(\boldsymbol{x})$ (Bayes optimal probability), and $\widehat{p}_0(\boldsymbol{x})$ (true posterior probability).

**Lemma I.1.** *Consider 2-GMM and proportional settings in Section D.1.1 on separable dataset $(\delta < \delta^*(0))$. Let $(\rho^*, \beta_0^*)$ be defined as per Theorem D.1, and $(Y, G, H) \sim P_y \times \mathsf{N}(0, 1) \times \mathsf{N}(0, 1)$. Let $G' := \rho^* G + \sqrt{1 - \rho^{*2}} H$. Then for any test point $(\boldsymbol{x}, y) \sim P_{\boldsymbol{x}, y}$ independent of $\widehat{p}$, as $n \to \infty$,*

$$
\begin{pmatrix} y \\ \widehat{p}(\boldsymbol{x}) \\ p^*(\boldsymbol{x}) \\ \widehat{p}_0(\boldsymbol{x}) \end{pmatrix} \overset{d}{\to} \begin{pmatrix} Y \\ \sigma\left(\rho^* \|\boldsymbol{\mu}\|_2 Y + G + \beta_0^*\right) \\ \sigma\left(2\|\boldsymbol{\mu}\|_2 (\|\boldsymbol{\mu}\|_2 Y + G') + \log \frac{\pi}{1-\pi}\right) \\ \sigma\left(2\rho^* \|\boldsymbol{\mu}\|_2 (\rho^* \|\boldsymbol{\mu}\|_2 Y + G) + \log \frac{\pi}{1-\pi}\right) \end{pmatrix}. \tag{168}
$$

*Proof.* Rewrite $\boldsymbol{x} = y\boldsymbol{\mu} + \boldsymbol{z}$ where $\boldsymbol{z} \sim \mathsf{N}(\mathbf{0}, \mathbf{I}_d)$. By direct calculation, the three quantities $\widehat{p}(\boldsymbol{x})$, $p^*(\boldsymbol{x})$, and $\widehat{p}_0(\boldsymbol{x})$ can be expressed by

$$\widehat{p}(\boldsymbol{x}) = \sigma\big(\langle \boldsymbol{x}, \widehat{\boldsymbol{\beta}} \rangle + \widehat{\beta}_0\big) = \sigma\left(\widehat{\rho}\|\boldsymbol{\mu}\|_2\, y + \langle \boldsymbol{z}, \widehat{\boldsymbol{\beta}} \rangle + \beta_0\right), \tag{169}$$

$$p^*(\boldsymbol{x}) = \mathbb{P}(y = 1 \,|\, \boldsymbol{x}) = \frac{\pi e^{-\frac{1}{2}\|\boldsymbol{x}-\boldsymbol{\mu}\|_2^2}}{\pi e^{-\frac{1}{2}\|\boldsymbol{x}-\boldsymbol{\mu}\|_2^2} + (1-\pi)e^{-\frac{1}{2}\|\boldsymbol{x}+\boldsymbol{\mu}\|_2^2}} \tag{170}$$

$$= \sigma\left(2\langle \boldsymbol{x}, \boldsymbol{\mu} \rangle + \log \frac{\pi}{1-\pi}\right)$$

$$= \sigma\left(2\|\boldsymbol{\mu}\|_2\Big(\|\boldsymbol{\mu}\|_2\, y + \langle \boldsymbol{z}, \boldsymbol{\mu}/\|\boldsymbol{\mu}\|_2 \rangle\Big) + \log \frac{\pi}{1-\pi}\right), \tag{171}$$

$$\widehat{p}_0(\boldsymbol{x}) = \mathbb{P}\big(y = 1 \,|\, \widehat{p}(\boldsymbol{x})\big) = \frac{\pi e^{-\frac{1}{2}(\widehat{f}(\boldsymbol{x})-\widehat{\rho}\|\boldsymbol{\mu}\|_2-\widehat{\beta}_0)^2}}{\pi e^{-\frac{1}{2}(\widehat{f}(\boldsymbol{x})-\widehat{\rho}\|\boldsymbol{\mu}\|_2-\widehat{\beta}_0)^2} + (1-\pi)e^{-\frac{1}{2}(\widehat{f}(\boldsymbol{x})+\widehat{\rho}\|\boldsymbol{\mu}\|_2-\widehat{\beta}_0)^2}} \tag{172}$$

$$= \sigma\left(2\widehat{\rho}\|\boldsymbol{\mu}\|_2\langle \boldsymbol{x}, \widehat{\boldsymbol{\beta}} \rangle + \log \frac{\pi}{1-\pi}\right)$$

$$= \sigma\left(2\widehat{\rho}\|\boldsymbol{\mu}\|_2\Big(\widehat{\rho}\|\boldsymbol{\mu}\|_2\, y + \langle \boldsymbol{z}, \widehat{\boldsymbol{\beta}} \rangle\Big) + \log \frac{\pi}{1-\pi}\right), \tag{173}$$

where the Bayes' law is applied in Eq. (170) and (172).

Next, it suffices to obtain the joint asymptotics of $\langle \boldsymbol{z}, \widehat{\boldsymbol{\beta}} \rangle$ and $\langle \boldsymbol{z}, \boldsymbol{\mu}/\|\boldsymbol{\mu}\|_2 \rangle$, which appear in the expressions of Eq. (169), (171), (173). Note that $\langle \boldsymbol{z}, \widehat{\boldsymbol{\beta}} \rangle \xrightarrow{\mathrm{d}} \mathsf{N}(0, 1)$ (since $\boldsymbol{z} \perp\!\!\!\perp \widehat{\boldsymbol{\beta}}$, $\mathbb{P}(\|\widehat{\boldsymbol{\beta}}\|_2 = 1) \to 1$), $\langle \boldsymbol{z}, \boldsymbol{\mu}/\|\boldsymbol{\mu}\|_2 \rangle \sim \mathsf{N}(0, 1)$. Moreover, $\mathbb{E}[\langle \boldsymbol{z}, \widehat{\boldsymbol{\beta}} \rangle \langle \boldsymbol{z}, \boldsymbol{\mu}/\|\boldsymbol{\mu}\|_2 \rangle] = \mathbb{E}[\widehat{\rho}] \to \rho^*$ by Theorem D.1 and bounded convergence. These implies

$$\begin{pmatrix} \langle \boldsymbol{z}, \widehat{\boldsymbol{\beta}} \rangle \\ \langle \boldsymbol{z}, \boldsymbol{\mu}/\|\boldsymbol{\mu}\|_2 \rangle \end{pmatrix} \xrightarrow{\mathrm{d}} \mathsf{N}\left(\begin{pmatrix} 0 \\ 0 \end{pmatrix}, \begin{pmatrix} 1 & \rho^* \\ \rho^* & 1 \end{pmatrix}\right) \overset{\mathrm{d}}{=} \begin{pmatrix} G \\ G' \end{pmatrix}.$$

Since $y \perp\!\!\!\perp (\boldsymbol{z}, \widehat{\boldsymbol{\beta}})$ and $(\widehat{\rho}, \widehat{\beta}_0) \xrightarrow{\mathrm{P}} (\rho^*, \beta_0^*)$, we conclude Eq. (168) by Eq. (169), (171), (173) and then using the Slutsky's theorem. This completes the proof. $\qquad\square$

The proof of Theorem D.9 is primarily based on asymptotics in Theorem I.1.

***Proof of Proposition D.9.*** **(a):** For MSE, by directly using the asymptotics in Theorem I.1 and bounded convergence theorem, we have

$$\lim_{n\to\infty} \mathrm{MSE}(\widehat{p}) = \lim_{n\to\infty} \mathbb{E}\left[\big(\mathbb{1}\{y = 1\} - \widehat{p}(\boldsymbol{x})\big)^2\right]$$

$$= \mathbb{E}\left[\big(\mathbb{1}\{Y = 1\} - \sigma\left(\rho^*\|\boldsymbol{\mu}\|_2 Y + G + \beta_0^*\right)\big)^2\right]$$

$$= \mathbb{E}\left[\sigma\big(-\rho^*\|\boldsymbol{\mu}\|_2 - \beta_0^* Y + G\big)^2\right] = \mathrm{MSE}^*,$$

$$\lim_{n\to\infty} \mathrm{mMSE}(\widehat{p}) = \mathrm{MSE}^* - \mathrm{Var}\left[\mathbb{1}\{y = 1\}\right] = \mathrm{MSE}^* - \pi(1-\pi).$$

For CalErr, we similarly get

$$\lim_{n\to\infty} \mathrm{CalErr}(\widehat{p}) = \lim_{n\to\infty} \mathbb{E}\left[\big(\widehat{p}(\boldsymbol{x}) - \widehat{p}_0(\boldsymbol{x})\big)^2\right]$$

$$= \mathbb{E}\left[\left(\sigma\big(\rho^*\|\boldsymbol{\mu}\|_2 Y + G + \beta_0^*\big) - \sigma\Big(2\rho^*\|\boldsymbol{\mu}\|_2 \big(\rho^*\|\boldsymbol{\mu}\|_2 Y + G\big) + \log \frac{\pi}{1-\pi}\Big)\right)^2\right] = \mathrm{CalErr}^*.$$

For ConfErr, we can first obtain

$$
\begin{aligned}
\lim_{n \to \infty} \mathbb{E}\left[p^*(\boldsymbol{x})\big(1 - p^*(\boldsymbol{x})\big)\right] &= \lim_{n \to \infty} \mathbb{E}\left[\mathrm{Var}\left[\mathbb{1}\{y = 1\} \mid \boldsymbol{x}\right]\right] \\
&= \lim_{n \to \infty} \mathbb{E}\left[\big(\mathbb{1}\{y = 1\} - p^*(\boldsymbol{x})\big)^2\right] \\
&= \mathbb{E}\left[\left(\mathbb{1}\{Y = 1\} - \sigma\Big(2\|\boldsymbol{\mu}\|_2(\|\boldsymbol{\mu}\|_2 Y + G) + \log\frac{\pi}{1 - \pi}\Big)\right)^2\right] \\
&= \mathbb{E}\left[\sigma\Big(-2\|\boldsymbol{\mu}\|_2(\|\boldsymbol{\mu}\|_2 + G) - \log\frac{\pi}{1 - \pi}Y\Big)^2\right] = V^*_{y|\boldsymbol{x}},
\end{aligned}
$$

and then by relation between ConfErr and MSE Eq. (39)

$$
\lim_{n \to \infty} \mathrm{ConfErr}(\widehat{p}) = \lim_{n \to \infty} \mathrm{MSE}(\widehat{p}) - \lim_{n \to \infty} \mathbb{E}\left[p^*(\boldsymbol{x})\big(1 - p^*(\boldsymbol{x})\big)\right] = \mathrm{MSE}^* - V^*_{y|\boldsymbol{x}}.
$$

This concludes the proof of part (a).

**(b):** When $\tau = \tau^{\mathrm{opt}}$, by Theorem 3.1 $\beta_0^* = 0$. Then we can simplify

$$
\mathrm{MSE}^* = \mathbb{E}\left[\big(1 + \exp(\rho^*\|\boldsymbol{\mu}\|_2 + G)\big)^{-2}\right].
$$

According to Theorem G.2, we know that $\rho^*\|\boldsymbol{\mu}\|_2$ is increasing in $\pi \in (0, \frac{1}{2})$, $\|\boldsymbol{\mu}\|_2$, and $\delta$. It suffices to show that $\mathrm{MSE}^*$ is decreasing in $\rho^*\|\boldsymbol{\mu}\|_2$, which is obvious by noticing $t \mapsto (1 + \exp(t))^{-2}$ is a strictly decreasing function.

For $\mathrm{mMSE}^*$, note that $\pi(1 - \pi)$ is a increasing function of $\pi \in (0, \frac{1}{2})$, and it does not depend on $\|\boldsymbol{\mu}\|_2$, $\delta$. These shows the monotonicity of $\mathrm{mMSE}^* = \mathrm{MSE}^* - \pi(1 - \pi)$.

For $\mathrm{ConfErr}^*$, note that $V^*_{y|\boldsymbol{x}}$ does not depend on $\delta$. This implies that $\mathrm{ConfErr}^*$ has the same monotonicity in $\delta$ as $\mathrm{MSE}^*$, which concludes the proof of part (b). $\qquad\square$

## I.2 VERIFICATION OF CLAIM D.10

The analytical dependence of $\mathrm{CalErr}^*$ and $\mathrm{ConfErr}^*$ on model parameters is more complicated. We provide a numerical verification of Theorem D.10.

***Verification of Claim D.10.*** For $\mathrm{CalErr}^*$, denote

$$
\begin{aligned}
h_1(t) &:= \mathbb{E}\left[\left(\sigma\big(2t(G + t) + c\big) - \sigma(G + t)\right)^2\right] \\
h_2(t) &:= \mathbb{E}\left[\left(\sigma\big(2t(G - t) + c\big) - \sigma(G - t)\right)^2\right]
\end{aligned}
$$

where $c < 0$ is a constant. When $\tau = \tau^{\mathrm{opt}}$, we have $\beta_0^* = 0$ and

$$
\mathrm{CalErr}^* = \pi h_1(\rho^*\|\boldsymbol{\mu}\|_2) + (1 - \pi)h_2(\rho^*\|\boldsymbol{\mu}\|_2), \qquad \text{where } c = \log\frac{\pi}{1 - \pi}.
$$

According to Fig. 15, we can numerically show that $h(t) := \pi h_1(t) + (1 - \pi)h_2(t)$ is a decreasing function when $\pi \leq \overline{\pi} \approx 0.25$ is fixed. Under this condition, $\mathrm{CalErr}^*$ is decreasing in $\rho^*\|\boldsymbol{\mu}\|_2$. Then by using Theorem G.2 and similar arguments in the proof of Theorem D.9(b), we can conclude the monotonicity of $\mathrm{CalErr}^*$ in $\|\boldsymbol{\mu}\|_2$ and $\delta$.

For $\mathrm{ConfErr}^*$, in Fig. 15 we numerically show that $V^*_{y|\boldsymbol{x}}$ is increasing in $\pi$ when $\|\boldsymbol{\mu}\|_2$ is fixed. Since $\mathrm{ConfErr}^* = \mathrm{MSE}^* - V^*_{y|\boldsymbol{x}}$ and we have shown in Theorem D.9(b) that $\mathrm{MSE}^*$ is decreasing in $\pi$, we conclude $\mathrm{ConfErr}^*$ is also decreasing in $\pi$. $\qquad\square$

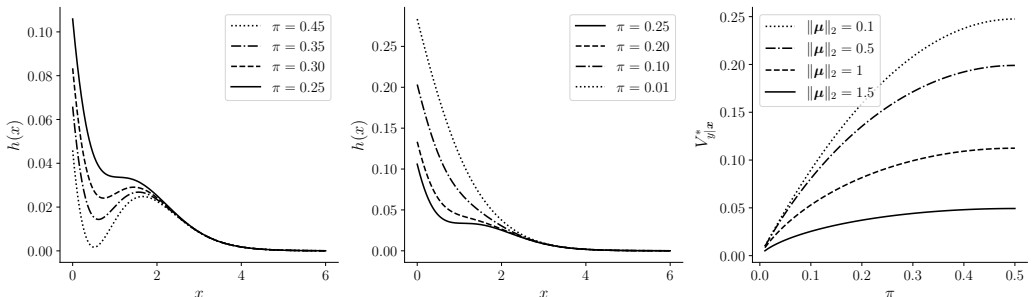

Figure 15: **Monotonicity of** $x \mapsto h(x)$ **and** $\pi \mapsto V_{y|\boldsymbol{x}}^*$. **Left:** $h$ is not monotone when $\pi > \overline{\pi} \approx 0.25$. **Middle:** $h$ is monotone decreasing when $\pi \leq \overline{\pi} \approx 0.25$. **Right:** $V_{y|\boldsymbol{x}}^*$ is monotone increasing in $\pi$ for different values of $\|\boldsymbol{\mu}\|_2$.

## J  TECHNICAL LEMMAS

### J.1  PROPERTIES OF GAUSSIAN RANDOM VARIABLES

We need the following variant of Gordon's comparison theorem for Gaussian processes.

**Lemma J.1** (CGMT). *Let* $D_{\mathbf{u}} \subset \mathbb{R}^{n_1+n_2}$, $D_{\mathbf{v}} \subset \mathbb{R}^{m_1+m_2}$ *be compact sets and let* $Q : D_{\mathbf{u}} \times D_{\mathbf{v}} \to \mathbb{R}$ *be a continuous function. Let* $\boldsymbol{G} = (G_{i,j}) \overset{\text{i.i.d.}}{\sim} \mathsf{N}(0,1)$, $\boldsymbol{g} \sim \mathsf{N}(\boldsymbol{0}, \boldsymbol{1}_{n_1})$, $\boldsymbol{h} \sim \mathsf{N}(\boldsymbol{0}, \boldsymbol{1}_{m_1})$ *be independent standard Gaussian vectors. For any* $\mathbf{u} \in \mathbb{R}^{n_1+n_2}$ *and* $\mathbf{v} \in \mathbb{R}^{m_1+m_2}$ *we define* $\widetilde{\mathbf{u}} = (u_1, \ldots, u_{n_1})$ *and* $\widetilde{\mathbf{v}} = (v_1, \ldots, v_{m_1})$. *Define*

$$C^*(\boldsymbol{G}) = \min_{\mathbf{u} \in D_{\mathbf{u}}} \max_{\mathbf{v} \in D_{\mathbf{v}}} \widetilde{\mathbf{v}}^\mathsf{T} \boldsymbol{G} \widetilde{\mathbf{u}} + Q(\mathbf{u}, \mathbf{v}),$$

$$L^*(\boldsymbol{g}, \boldsymbol{h}) = \min_{\mathbf{u} \in D_{\mathbf{u}}} \max_{\mathbf{v} \in D_{\mathbf{v}}} \|\widetilde{\mathbf{v}}\|_2 \boldsymbol{g}^\mathsf{T} \widetilde{\mathbf{u}} + \|\widetilde{\mathbf{u}}\|_2 \boldsymbol{h}^\mathsf{T} \widetilde{\mathbf{v}} + Q(\mathbf{u}, \mathbf{v}).$$

*Then we have:*

*(a) For all* $t \in \mathbb{R}$,

$$\mathbb{P}\left(C^*(\boldsymbol{G}) \leq t\right) \leq 2\,\mathbb{P}\left(L^*(\boldsymbol{g}, \boldsymbol{h}) \leq t\right).$$

*(b) If* $D_{\mathbf{u}}$ *and* $D_{\mathbf{v}}$ *are convex and if* $Q$ *is convex concave, then for all* $t \in \mathbb{R}$,

$$\mathbb{P}\left(C^*(\boldsymbol{G}) \geq t\right) \leq 2\,\mathbb{P}\left(L^*(\boldsymbol{g}, \boldsymbol{h}) \geq t\right).$$

*Proof.* See (Miolane & Montanari, 2018, Corollary G.1). $\square$

### J.2  PROPERTIES OF SUB-GAUSSIAN AND SUB-EXPONENTIAL RANDOM VARIABLES

**Definition J.1** (Sub-gaussianity). *The sub-gaussian norm of random variable* $X$ *is defined as*

$$\|X\|_{\psi_2} := \inf\left\{K > 0 : \mathbb{E}[\exp(X^2/K^2)] \leq 2\right\}.$$

- *A random variable* $X \in \mathbb{R}$ *is called sub-gaussian if* $\|X\|_{\psi_2} < \infty$.

- *A random vector* $\boldsymbol{x} = (X_1, \ldots, X_d)^\mathsf{T} \in \mathbb{R}^d$ *is called sub-gaussian if* $\sup_{\mathbf{v} \in \S^{d-1}} \|\langle \boldsymbol{x}, \mathbf{v} \rangle\|_{\psi_2} < \infty$. *Specifically, write* $\boldsymbol{x} \sim \mathsf{subG}_{\perp\!\!\!\perp}(\boldsymbol{0}, \mathbf{I}_d; K)$ *if* $X_1, \ldots, X_d$ *are independent random variables with* $\mathbb{E}[X_i] = 0$, $\mathrm{Var}(X_i) = 1$, *and* $\max_{1 \leq i \leq d} \|X_i\|_{\psi_2} \lesssim K$.

Theorem J.2 and J.3 summarize some basic facts and concentration inequalities about sub-gaussian random variables and vectors.

**Lemma J.2.** *Some facts about sub-gaussian random variables.*

*(a)* $\|\cdot\|_{\psi_2}$ *is a norm on the space of sub-gaussian random variables.*

(b) *Let $X_1, \ldots, X_N$ be independent mean-zero sub-gaussian random variables. Then $\sum_{i=1}^{N} X_i$ is also a sub-gaussian random variable, and*

$$\left\| \sum_{i=1}^{N} X_i \right\|_{\psi_2}^2 \leq C \sum_{i=1}^{N} \|X_i\|_{\psi_2}^2 \,,$$

*where $C$ is an absolute constant.*

(c) *(Maximum) Let $X_1, \ldots, X_N$ be sub-gaussian random variables (not necessarily independent) with $K := \max_{1 \leq i \leq N} \|X_i\|_{\psi_2}$. Then*

$$\mathbb{E}\left[ \max_{1 \leq i \leq N} |X_i| \right] \leq CK\sqrt{\log N}, \qquad (N \geq 2),$$

*where $C$ is an absolute constant.*

*Proof.* See (Vershynin, 2018, Exercise 2.5.7, Proposition 2.6.1, Exercise 2.5.10). □

**Lemma J.3** (Concentration). *Suppose $\boldsymbol{x}, \boldsymbol{y} \sim \mathsf{subG}_{\perp\!\!\!\perp}(\mathbf{0}, \mathbf{I}_d; K)$ and $\boldsymbol{x} \perp\!\!\!\perp \boldsymbol{y}$.*

(a) *(Hanson-Wright inequality I) Let $\mathbf{A} \in \mathbb{R}^{d \times d}$ be a matrix. Then, for every $t \geq 0$,*

$$\mathbb{P}\left( |\boldsymbol{x}^\mathsf{T} \mathbf{A} \boldsymbol{x} - \mathbb{E}[\boldsymbol{x}^\mathsf{T} \mathbf{A} \boldsymbol{x}]| \geq t \right) \leq 2\exp\left( -c\min\left\{ \frac{t^2}{K^4 \|\mathbf{A}\|_{\mathrm{F}}^2}, \frac{t}{K^2 \|\mathbf{A}\|_{\mathrm{op}}} \right\} \right),$$

*where $c$ is an absolute constant.*

(b) *(Hanson-Wright inequality II) Let $\mathbf{B} \in \mathbb{R}^{d' \times d}$ be a matrix. Then, for every $t \geq 0$,*

$$\mathbb{P}\left( \left| \frac{\|\mathbf{B}\boldsymbol{x}\|_2}{\|\mathbf{B}\|_{\mathrm{F}}} - 1 \right| > t \right) \leq 2\exp\left( -\frac{ct^2 \|\mathbf{B}\|_{\mathrm{F}}^2}{K^4 \|\mathbf{B}\|_{\mathrm{op}}^2} \right),$$

*where $c$ is an absolute constant. In particular, when $\mathbf{B} = \mathbf{I}_d$,*

$$\mathbb{P}\left( \left| \frac{\|\boldsymbol{x}\|_2}{\sqrt{d}} - 1 \right| > t \right) \leq 2\exp\left( -\frac{ct^2 d}{K^4} \right).$$

(c) *(Hoeffding's inequality) Let $\mathbf{a} \in \mathbb{R}^d$ be a vector. Then, for every $t \geq 0$,*

$$\mathbb{P}\left( \frac{|\langle \boldsymbol{x}, \mathbf{a} \rangle|}{\|\mathbf{a}\|_2} > t \right) \leq 2\exp\left( -\frac{ct^2}{K^2} \right),$$

*where $c$ is an absolute constant.*

(d) *(Bernstein's inequality) Let $\mathbf{B} \in \mathbb{R}^{d \times d}$ be a matrix. Then, for every $t \geq 0$,*

$$\mathbb{P}\left( \frac{|\boldsymbol{x}^\mathsf{T} \mathbf{B} \boldsymbol{y}|}{\|\mathbf{B}\|_{\mathrm{F}}} > t \right) \leq 2\exp\left( -c\min\left\{ \frac{t^2}{K^4}, \frac{t\|\mathbf{B}\|_{\mathrm{F}}}{K^2 \|\mathbf{B}\|_{\mathrm{op}}} \right\} \right),$$

*where $c$ is an absolute constant. In particular, when $\mathbf{B} = \mathbf{I}_d$,*

$$\mathbb{P}\left( \frac{|\langle \boldsymbol{x}, \boldsymbol{y} \rangle|}{\sqrt{d}} > t \right) \leq 2\exp\left( -c\min\left\{ \frac{t^2}{K^4}, \frac{t\sqrt{d}}{K^2} \right\} \right).$$

*Proof.* For (a), (b) and (c), see (Vershynin, 2018, Theorem 6.2.1, Theorem 6.3.2, Theorem 2.6.3). For (d), let $\overline{\boldsymbol{x}} = \begin{pmatrix} \boldsymbol{x} \\ \boldsymbol{y} \end{pmatrix}, \overline{\mathbf{A}} = \frac{1}{2}\begin{pmatrix} \mathbf{0} & \mathbf{B} \\ \mathbf{B} & \mathbf{0} \end{pmatrix}$, then apply $\overline{\boldsymbol{x}}^\mathsf{T} \overline{\mathbf{A}} \overline{\boldsymbol{x}} = \boldsymbol{x}^\mathsf{T} \mathbf{B} \boldsymbol{y}$ to (a) and simplify. □

**Definition J.2** (Sub-exponentiality). *The sub-exponential norm of random variable $X$ is defined as*

$$\|X\|_{\psi_1} = \inf \left\{ K > 0 : \mathbb{E}[\exp(|X|/K)] \leq 2 \right\}.$$

- *A random variable $X \in \mathbb{R}$ is called sub-exponential if $\|X\|_{\psi_1} < \infty$.*

Theorem J.4 summarizes some basic facts about sub-exponential random variables.

**Lemma J.4.** *Some facts about sub-exponential random variables.*

(a) $\|\cdot\|_{\psi_1}$ *is a norm on the space of sub-exponential random variables.*

(b) *Let $X_1, \ldots, X_N$ be independent mean-zero sub-exponential random variables. Then $\sum_{i=1}^{N} X_i$ is also a sub-exponential random variable. If $K := \max_{1 \leq i \leq N} \|X_i\|_{\psi_1}$ and $N \geq C$, then*

$$\left\| \sum_{i=1}^{N} X_i \right\|_{\psi_1} \leq C' K \sqrt{N},$$

*where $C, C'$ are absolute constants.*

(c) *(Maximum) Let $X_1, \ldots, X_N$ be sub-exponential random variables (not necessarily independent) with $K := \max_{1 \leq i \leq N} \|X_i\|_{\psi_1}$. Then*

$$\mathbb{E}\left[ \max_{1 \leq i \leq N} |X_i| \right] \leq CK \log N, \qquad (N \geq 2),$$

*where $C$ is an absolute constant.*

(d) *Let $X$ and $Y$ be sub-gaussian random variables. Then $XY$ is sub-exponential. Moreover,*

$$\|XY\|_{\psi_1} \leq \|X\|_{\psi_2} \|Y\|_{\psi_2}.$$

*In particular, $X^2$ is sub-exponential, and*

$$\|X^2\|_{\psi_1} \leq \|X\|_{\psi_2}^2.$$

*Proof.* For (a) and (d), see (Vershynin, 2018, Exercise 2.7.11, Lemma 2.7.6, Lemma 2.7.7). For (b), the proof is analogous to (Vershynin, 2018, Proposition 2.6.1). For any $|\lambda| \leq 1/K$, we have

$$\mathbb{E}\left[ \exp\left( \lambda \sum_{i=1}^{N} X_i \right) \right] = \prod_{i=1}^{N} \mathbb{E}[\exp(\lambda X_i)] \leq \prod_{i=1}^{N} \exp\left( C\lambda^2 \|X_i\|_{\psi_1}^2 \right) \leq \exp\left( C\lambda^2 N K^2 \right),$$

where sub-exponential properties (Vershynin, 2018, Proposition 2.7.1 (iv)(v)) are used, and $C$ is an absolute constant. If $N \geq 1/C$, then $1/\sqrt{CNK^2} \leq 1/K$ and therefore

$$\mathbb{E}\left[ \exp\left( \lambda \sum_{i=1}^{N} X_i \right) \right] \leq \exp\left( \lambda^2 CNK^2 \right), \quad \text{for all } \lambda \text{ such that } |\lambda| \leq \frac{1}{\sqrt{CN}K}.$$

Then the proof is completed by using (Vershynin, 2018, Proposition 2.7.1 (iv)(v)) again.

For (d), the proof is analogous to (Vershynin, 2018, Exercise 2.5.10). By (Vershynin, 2018, Proposition 2.7.1 (i)(iv)), $\mathbb{P}(|X_i| \geq t) \leq 2 \exp(-ct/\|X\|_{\psi_1}) \leq 2 \exp(-ct/K), \forall t \geq 0$, where $c$ is an absolute constant. Denote $t_0 := 2K/c$, then

$$\mathbb{E}\left[ \max_{i \geq 1} \frac{|X_i|}{1 + \log i} \right] \leq t_0 + \int_{t_0}^{\infty} \mathbb{P}\left( \max_{i \geq 1} \frac{|X_i|}{1 + \log i} > t \right) dt \leq \frac{2K}{c} + \int_{t_0}^{\infty} \sum_{i=1}^{\infty} \mathbb{P}\left( \frac{|X_i|}{1 + \log i} > t \right) dt$$

$$= \frac{2K}{c} + \sum_{i=1}^{\infty} \int_{t_0}^{\infty} \mathbb{P}\left( |X_i| > t(1 + \log i) \right) dt \leq \frac{2K}{c} + \sum_{i=1}^{\infty} \int_{t_0}^{\infty} 2 \exp\left( -ct(1 + \log i)/K \right) dt$$

$$\leq \frac{2K}{c} + \sum_{i=1}^{\infty} \int_{t_0}^{\infty} \exp\left( -(\log i)ct_0/K \right) \cdot 2 \exp(-ct/K) dt \leq \frac{2K}{c} + \sum_{i=1}^{\infty} i^{-2} \int_{0}^{\infty} 2 \exp(-ct/K) dt$$

$$= \frac{2K}{c} + C_0 \cdot \frac{2K}{c} \leq CK,$$

where $C_0, C$ are absolute constants. Hence, for any $N \geq 2$,

$$\mathbb{E}\left[ \max_{1 \leq i \leq N} |X_i| \right] \leq (1 + \log N) \cdot \mathbb{E}\left[ \max_{1 \leq i \leq N} \frac{|X_i|}{1 + \log i} \right] \lesssim K \log N.$$

This concludes the proof. □

### J.3 PROPERTIES OF THE MOREAU ENVELOPE AND PROXIMAL OPERATOR

Let $\ell : \mathbb{R} \to \mathbb{R}_{\geq 0}$ be a continuous convex function. For any $x \in \mathbb{R}$ and $\lambda > 0$, the Moreau envelope of $\ell$ is defined as

$$\mathsf{e}_\ell(x;\lambda) = \mathsf{e}_{\lambda\ell}(x) := \min_{t \in \mathbb{R}} \left\{ \ell(t) + \frac{1}{2\lambda}(t-x)^2 \right\}, \tag{174}$$

and the proximal operator of $\ell$ is defined as

$$\mathsf{prox}_\ell(x;\lambda) = \mathsf{prox}_{\lambda\ell}(x) := \arg\min_{t \in \mathbb{R}} \left\{ \ell(t) + \frac{1}{2\lambda}(t-x)^2 \right\}.$$

**Lemma J.5.** *For any $x \in \mathbb{R}, \lambda > 0$, $\mathsf{prox}_\ell(x;\lambda)$ is uniquely determined by stationarity condition*

$$\mathsf{prox}_\ell(x;\lambda) + \lambda\ell'\big(\mathsf{prox}_\ell(x;\lambda)\big) - x = 0.$$

*(a)* $\mathsf{e}_\ell(x;\lambda)$ *is continuous and convex in* $(x,\lambda)$*. If $\ell$ is differentiable, then $\mathsf{e}_\ell(x;\lambda)$ is also differentiable in its domain, with partial derivatives*

$$\frac{\partial \mathsf{e}_\ell(x;\lambda)}{\partial x} = \frac{1}{\lambda}\big(x - \mathsf{prox}_\ell(x;\lambda)\big) = \ell'(z)\big|_{z=\mathsf{prox}_\ell(x;\lambda)},$$

$$\frac{\partial \mathsf{e}_\ell(x;\lambda)}{\partial \lambda} = -\frac{1}{2\lambda^2}\big(x - \mathsf{prox}_\ell(x;\lambda)\big)^2 = -\frac{1}{2}\big(\ell'(z)\big)^2\big|_{z=\mathsf{prox}_\ell(x;\lambda)}.$$

*Moreover, $\mathsf{e}_\ell(x;\lambda)$ is non-increasing in $\lambda$ and $\mathsf{e}_\ell(x;\lambda) \to \ell(x)$ when $\lambda \to 0^+$.*

*(b)* $\mathsf{prox}_\ell(x;\lambda)$ *is continuous in* $(x,\lambda)$*. If $\ell$ is twice differentiable, then $\mathsf{prox}_\ell(x;\lambda)$ is also differentiable in its domain, with partial derivatives*

$$\frac{\partial \mathsf{prox}_\ell(x;\lambda)}{\partial x} = \frac{1}{1+\lambda\ell''(z)}\bigg|_{z=\mathsf{prox}_\ell(x;\lambda)} \qquad \frac{\partial \mathsf{prox}_\ell(x;\lambda)}{\partial \lambda} = -\frac{\ell'(z)}{1+\lambda\ell''(z)}\bigg|_{z=\mathsf{prox}_\ell(x;\lambda)}.$$

*Moreover, $\mathsf{prox}_\ell(x;\lambda) \to x$ when $\lambda \to 0^+$.*

*Proof.* See (Thrampoulidis et al., 2018, Lemma 15), (Donoho & Montanari, 2016, Proposition A.1), (Salehi et al., 2019, Lemma 2, Lemma 4), and relevant references therein. □

## K MISCELLANEOUS

Let $\widehat{\kappa}$ be the optimal objective value in Eq. (15), which is the *maximum margin* for data $(\boldsymbol{X}, \boldsymbol{y})$. Moreover, $(\widehat{\boldsymbol{\beta}}, \widehat{\beta}_0, \widehat{\kappa})$ is also the optimal solution to Eq. (7). Notice $\widehat{\kappa} \geq 0$ always holds (by taking $\boldsymbol{\beta} = 0, \beta_0 = 0$ in Eq. (15)), and we can observe the following relation.

(linearly separable) $\exists \boldsymbol{\beta} \neq \mathbf{0}, \beta_0 \in \mathbb{R}$, such that $y_i(\langle \boldsymbol{x}_i, \boldsymbol{\beta}\rangle + \beta_0) > 0, \forall i \in [n]$,

$$\iff \quad \widehat{\kappa} > 0, \quad \implies \quad \|\widehat{\boldsymbol{\beta}}\|_2 = 1,$$

(not linearly separable) $\forall \boldsymbol{\beta} \neq \mathbf{0}, \beta_0 \in \mathbb{R}$, such that $y_i(\langle \boldsymbol{x}_i, \boldsymbol{\beta}\rangle + \beta_0) \overset{(*)}{\leq} 0, \forall i \in [n]$,

$$\iff \quad \widehat{\kappa} = 0, \quad \implies \quad \widehat{\boldsymbol{\beta}} = \mathbf{0}, \widehat{\beta}_0 = 0 \text{ is a solution.}[17]$$

When data is linearly separable, it turns out Eq. (7) also has the following equivalent form:

$$\begin{aligned} \underset{\mathbf{w}\in\mathbb{R}^d, w_0\in\mathbb{R}}{\text{minimize}} \quad & \|\mathbf{w}\|_2^2, \\ \text{subject to} \quad & \widetilde{y}_i(\langle \boldsymbol{x}_i, \mathbf{w}\rangle + w_0) \geq 1, \quad \forall i \in [n]. \end{aligned} \tag{175}$$

The parameters in Eq. (7) and (175) have one-to-one relation $(\kappa, \boldsymbol{\beta}, \beta_0) = (1, \mathbf{w}, w_0)/\|\mathbf{w}\|_2$. Notably, Eq. (175) is known as the hard-margin SVM (Vapnik, 1998) if $\tau = 1$.

**LLM usage statement** We used a large language model (LLM) solely for polishing the writing of this paper (e.g., grammar, wording). All edits were checked and revised by the authors.

---

[20]If $(*)$ is strict $(<)$, then $\widehat{\boldsymbol{\beta}} = \mathbf{0}, \widehat{\beta}_0 = 0$ is the *unique* solution.

## L ADDITIONAL EXPERIMENTS

### L.1 LOGIT DISTRIBUTION FOR NON-GAUSSIAN DATA

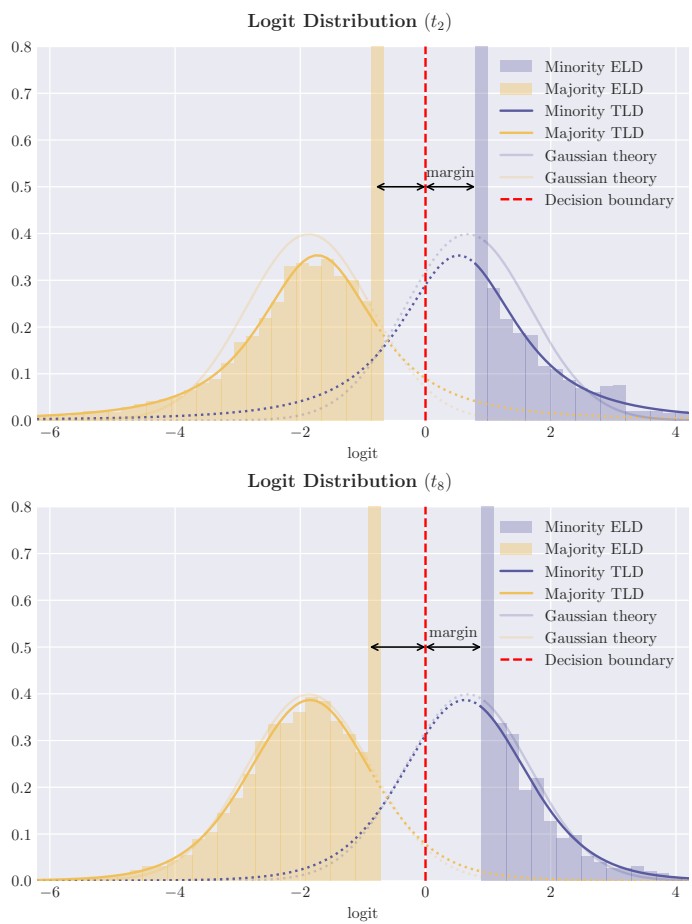

Figure 16: **Empirical logit distribution (ELD) and testing logit distribution (TLD) under a non-Gaussian setting.** We train a max-margin classifier $\widehat{f}$ on synthetic data sampled from a two-component mixture of multivariate $t$ distributions (with degree of freedom 2 and 8). Colors denote labels $y_i$, and the $x$-axis shows the logits $\widehat{f}(\boldsymbol{x}_i)$. **ELD:** truncated $t$ distributions (histograms). **TLD:** $t$ distributions (curves). Transparent lines indicate the theoretical TLD derived under the Gaussian assumption (Theorem 2.1).

To examine how our theory adapts to non-Gaussian data, we repeat the experiment in Fig. 1 but replace the Gaussian mixture with a mixture of multivariate $t$ distributions with a chosen degree of freedom, while keeping all other parameters fixed. The key observations from Fig. 16 are as follows:

- We find that the TLDs follow $t$ distributions, whereas the ELDs follow the same $t$ distributions but *truncated* at the margin. This shows that even when the data are non-Gaussian, the effect of overfitting is still characterized by truncation.

- We observe that the intersection of the actual minority and majority TLD curves nearly coincides with the intersection predicted by the Gaussian-based theory Theorem 2.1, as illustrated by the two transparent curves. This demonstrates the *robustness* of margin rebalancing technique, suggesting that the hyperparameter $\tau^{\mathrm{opt}}$ derived in Theorem 3.1 may still perform well under non-Gaussian data.

## L.2 PHASE TRANSITION IN HIGH IMBALANCE REGIME

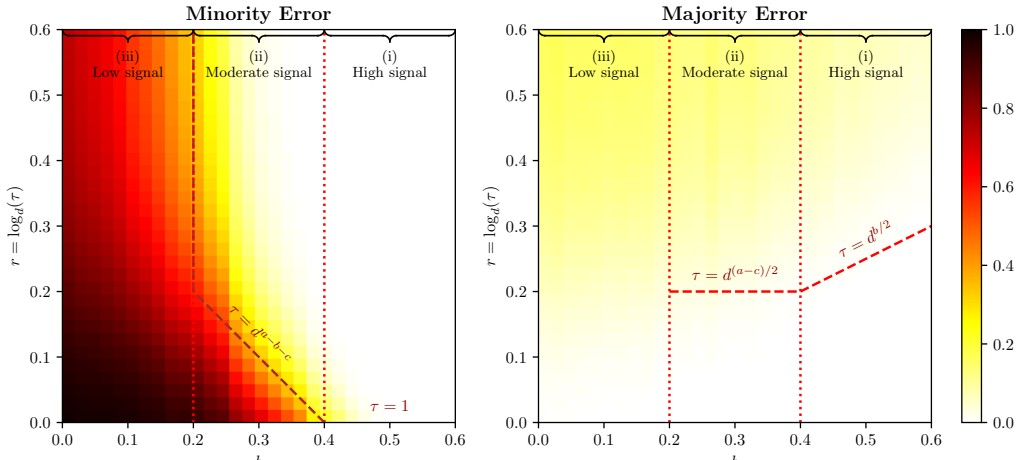

Figure 17: **Phase transition in high imbalance regime**. Minority/majority errors under different settings of parameters $b$ (signal) and $\tau = d^r$. **Left:** minority accuracy is (i) high for any $\tau$ under high signal, (ii) high for $\tau \gg d^{a-b-c}$ under moderate signal, but (iii) low for any $\tau$ under low signal. **Right:** majority accuracy is close to 1 under high and moderate signal as long as $\tau$ is not too large.

To illustrate how the acceptable range of $\tau$ varies with the signal strength $b$ (recall that $\|\boldsymbol{\mu}\|_2^2 \propto d^b$), we repeat the experiment in Fig. 5 but modify the setting by fixing $a = 0.5$ and $c = 0.1$ while varying $b$ and $r$. As shown in Fig. 17, the same three phases in majority/minority errors given by Theorem 3.2 emerge. The plot also reveals the upper and lower bounds on the admissible range of $\tau$ as the signal strength changes.

## L.3 Logit distribution for CIFAR-10 image dataset

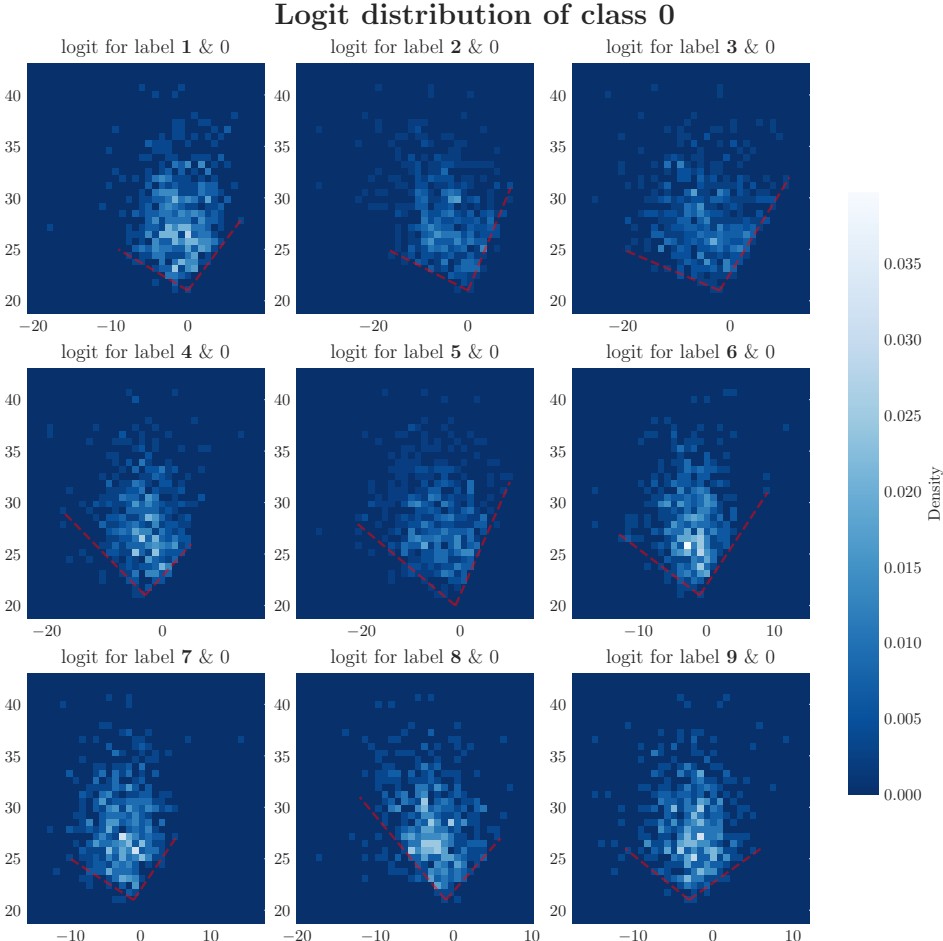

Figure 18: **Joint empirical logit distributions of multinomial logistic regression for CIFAR-10.** The heatmaps show the empirical joint logits $\left(\widehat{f}_0(\boldsymbol{x}_i), \widehat{f}_k(\boldsymbol{x}_i)\right)$ for features $\boldsymbol{x}_i$ from class 0, where $k = 1, 2, \ldots, 9$. The vertical axis is $\widehat{f}_0(\boldsymbol{x}_i)$ for all subplots, and the horizontal axis is $\widehat{f}_k(\boldsymbol{x}_i)$. The truncation boundaries are indicated by the red dashed lines.

We extend our experiment in Fig. 7 to the full 10-class setting. We first simulate an imbalanced CIFAR-10 dataset, where class 0 has sample size 500, class 9 has 100, and the sizes of the remaining classes follow an exponential decay. We then train a linear probe (multinomial logistic regression) on the ResNet-18 features extracted from this dataset. In Fig. 18, we visualize the logits for data whose true label is 0. Again, we observe *truncation* in two directions of the pairwise logits $(\widehat{f}_0(\boldsymbol{x}), \widehat{f}_k(\boldsymbol{x}))$, where $\widehat{f}_k(\cdot)$ denotes the logit for label $k$.

