# OpenReview forum: "A Statistical Theory of Overfitting for Imbalanced Classification"
_ICLR.cc/2026/Conference — ICLR 2026 Poster_

### Official Review · Reviewer_rQB8 · 2025-10-30

**Soundness:** 2
**Presentation:** 3
**Contribution:** 2
**Rating:** 4
**Confidence:** 4

**Summary:**

This paper addresses minority-class overfitting and poor confidence calibration in high-dimensional, imbalanced classification, where traditional theories fail to explain the mechanisms underlying overfitting. It uses a two-component Gaussian mixture model, focusing on logistic regression and SVM. A margin rebalancing method with a hyperparameter to adjust the constraints on the minority class margin is proposed. Theoretically, it analyzes the proportional asymptotic regime (the sample-dimension ratio converges to a finite value), exploring convergence of the parameter and logit distributions. Empirically, it is verified using CIFAR-10, IMDb, and IFNB data. The paper concludes that the high-dimensional training empirical logit distribution is truncated Gaussian (worse for minorities), and the test logit is standard Gaussian; optimal margin rebalancing eliminates majority bias, aligns class errors, and improves calibration. Contributions: establish an overfitting statistical theory, clarify the truncation mechanism, derive optimal hyperparameters, verify across datasets, and conjecture multiclass/non-isotropic extensions.

**Strengths:**

1. This paper establishes a dedicated statistical theory for high-dimensional imbalanced classification, clarifying logit distribution truncation as the core of overfitting and deriving optimal margin rebalancing, filling key theoretical gaps in the field.

2. Its theoretical conclusions are supported by 2-Gaussian Mixture Model simulations and experiments on datasets like CIFAR-10 and IMDb, showing strong consistency between theoretical predictions and empirical results.

3. The paper carries out rigorous proportional asymptotic analysis with solid proofs for parameter convergence, and its clear structure (theory→experiments→extensions) effectively improves the readability of complex theoretical content.

**Weaknesses:**

1. The weaknesses are mainly in the strong model assumptions that restrict the applicability of this paper. First, the paper heavily relies on the two-component Gaussian Mixture Model, which may not reflect real-world data complexities (e.g., non-Gaussian distributions or heavy noise). It lacks analysis of how to adapt its theory to such cases, and testing on datasets with more realistic distributions would enhance its applicability.

2. While conjecturing multiclass extension, the paper provides no detailed theoretical derivation (e.g., optimal τ for multiclass) or dedicated experiments (e.g., full 10-class CIFAR-10). This limits use for multiclass tasks; supplementary multiclass analyses would address this gap.

3. The theoretical analysis assumes homogeneous unit covariance for the 2-GMM. Still, it does not explore how heterogeneous covariance (e.g., different covariances for minority/majority classes) affects logit truncation and error bounds, which are common in real-world data.

4. While the paper explores phase transitions in the high-imbalance regime (Section 3.2, Theorem D.8) and defines signal strength via $|\mu\|_2^2 \propto d^b$ (parameter$b>0$), it only specifies the valid range of $\tau$ (e.g., $\tau \gg d^{(a-b-c)/2}$ for medium signal) but not how $\tau^{\text{opt}}$ adjusts with $b$. Its high-imbalance experiment (Fig.5) fixes $b=0.3$ and only varies $a$. This gap limits guidance for tuning $\tau^{\text{opt}}$ in real-world tasks with different signal intensities.

5. The key novel theoretical contribution of this paper compared to existing works is not clear. What is the challenge in deriving the theoretical results?

6. There are some existing works that establish results on biased decision boundaries in imbalanced classification, e.g., [1], which share similar results to the paper, even from different perspectives. These works should be mentioned in the paper.

[1] Hu et al., Learning Imbalanced Data with Beneficial Label Noise, ICLM, 2025.

**Questions:**

Please see Weakness.

---

> ### Author Response · Authors · 2025-11-21
> **Response to Reviewer rQB8 — Part I**
>
> Thank you for your detailed review and for raising several insightful questions. We sincerely appreciate the time you invested in identifying these limitations. In the updated manuscript, we have added new experiments in **Appendix L** to address your concerns regarding **W1**, **W2**, and **W4**. We have also strengthened the discussion of the model assumptions and their applicability in **Section 5** of the revised manuscript. Below, we respond to each of the issues you highlighted. If our clarifications and planned revisions appropriately address your concerns, we would be grateful for your consideration of a score increase.
>
>
> ## Weaknesses:
>
> ---
>
> - **W1.** "The weaknesses are mainly in the strong model assumptions that restrict the applicability of this paper..."
>
> - **A1.** Thanks for pointing this out.
>
>     - We added a new *non-Gaussian* simulation (e.g., a $t$-distribution) in the appendix (see **Appendix L.1**, page 101), using a setup similar to Fig. 1. Key observations: (1) Overfitting still induces *truncation* in the training logits, which follow a truncated $t$-distribution. (2) Theoretically derived $\\tau^\\mathrm{opt}$ remains close to the empirical optimum, showing the *robustness* of our method.
>
>     - We focus on a simple setting because the phenomenon can be observed, analyzed, and theorized more clearly. Similar proof strategies can extend to *non-isotropic* cases as well. We have added these generalizations in **Section 5** (page 10, `line 528`) for non-isotropic covariance; please refer to **Conjectures D.12-13** for details.
>
>     - We already provide an extension to sub-Gaussian distribution in the high imbalance regime (**Theorem D.8**). Our analysis can also be extended to multiclass, and we included the corresponding conclusions in **Conjectures D.11**.
>
>     - More broadly, the literature on *universality* [2] suggests that our results may extend to certain non-Gaussian settings and random feature models.
>
>     - In practice, *whitening* is often applied before fitting a linear classifier, after which the input features become isotropic (the covariance is the identity matrix).
>
>
> ---
>
> - **W2.** "While conjecturing multiclass extension, the paper provides no detailed theoretical derivation..."
>
> - **A2.** We believe the precise asymptotics for the multiclass problem (e.g., the parameter limit) can be derived, although the optimal $\\tau$ may not admit a clean closed form like Eq. (38).
>
>     - We have added a sketch of the multiclass theoretical result in the appendix, see **Conjecture D.11** (page 35, `line 1884`).
>
>     - We also included a new plot for the full 10-class CIFAR-10 experiment (similar to Fig. 7), obtained by fitting a multiclass linear probe on ResNet-18 features extracted from the imbalanced CIFAR-10 (using an imbalance setting similar to [2]). The 2-dimensional projection of the logits still exhibits truncation by two hyperplanes (see **Appendix L.3**, page 103).
>
>     - In practice, a convenient choice is the scaled cross-entropy loss for the $K$-class setting: $-\\log\\frac{\\exp({\\tau_y \\cdot \\mathbf{z}_y})}{\\sum_k \\exp({\\tau_k \\cdot \\mathbf{z}_k})}$, where $\\tau_k \\propto \\pi_k^{-1/2}$ (as noted around line 400) and $\\pi_k := \\mathbb{P}(Y = k)$. It follows the similar generalization arguments in [3, 4] from binary to multiclass.
>
> ---
>
> - **W3.** "The theoretical analysis assumes homogeneous unit covariance for the 2-GMM..."
>
> - **A3.** We work under simple settings to keep the proofs clear and to highlight the main implications. In fact we can still derive the asymptotics under the case $\\boldsymbol{x} \\mid {y} \\sim \\mathsf{N}( {y} \\boldsymbol{\\mu}, \\boldsymbol{\\Sigma}_y)$. We have added this theoretical result to appendix (see **Conjecture D.12**, page 36, `line 1925`), where we provide the closed-form expressions for the error bounds and logit truncation.
>
>     - Compared to homogeneous case (Theorem 2.1), we find that covariance heterogeneity induces distinct *scaling* effects on the testing errors. It also *rescales* the Gaussian component of the logit distribution for each class.
>
> ---
>
> - **W4.** "While the paper explores phase transitions in the high imbalance regime..."
>
> - **A4.** As shown in Theorem 3.2, the classification error has a hard phase transition with limit 0 or 1. Hence, in the high imbalance regime, any $\\tau$ in the range given in Theorem 3.2 leads to the same asymptotic behavior. Unlike in the proportional regime, the optimal $\\tau$ here is a range depending on $a,b,c$.
>
>     - We added a new heatmap in the appendix (see **Appendix L.2**, page 102) with $b$ as one axis, similar to Fig. 5 but with $a$ and $c$ fixed while varying $b$. This visualization again reveals the sharp phase transition described in Theorem 3.2.
>
> ---

---

> ### Author Response · Authors · 2025-11-21
> **Response to Reviewer rQB8 — Part II**
>
> ---
>
> - **W5.** "The key novel theoretical contribution of this paper compared to existing works is not clear. What is the challenge in deriving the theoretical results?"
>
> - **A5.** Our main contribution is addressing the questions in "Our goal" (in old manuscript, `line 105`):
>
>     1) Characterizing overfitting through the discrepancy between the logit distributions on the training and test sets, and demonstrating that high dimensionality induces a truncation effect in the training logits.
>
>     2) Deriving the optimal hyperparameter for margin rebalancing to mitigate class imbalance, and showing that the test error decreases with some key model parameters---the imbalance ratio, signal strength (class center separation), and aspect ratio.
>
>     3) Showing that changes in these parameters that raise test error could simultaneously worsen model calibration, revealing an adverse effect of overfitting (summarized in Table 2).
>
>     We have replaced "Our goal" with "**Our contribution**" in the revision at the end of Section 1 (page 2, `line 102`). The main technical challenge is proving logit distribution convergence, where the idea from projection pursuit [5] is used. We have also added a brief paragraph on "*Intuition and proof techniques*" in the revised version (see page 6, `line 305`).
>
> ---
>
> - **W6.** "There are some existing works that establish results on biased decision boundaries in imbalanced classification..."
>
> - **A6.** Thanks for the suggestion. We have cited [1] when introducing margin rebalancing (page 7, `line 377`). Although the settings differ ([1] studies the $n \to \infty$ asymptotics, while ours considers the proportional and high imbalance regime $n, d \to \infty$), it is an interesting idea to shift the decision boundary by introducing artificial label noise.
>
> ---
>
> [2] Montanari et al., Universality of max-margin classifiers, arXiv:2310.00176, 2023.
>
> [3] Cao et al., Learning imbalanced datasets with label-distribution-aware margin loss, NeurIPS, 2019.
>
> [4] Kini et al., Label-imbalanced and group-sensitive classification under overparameterization, NeurIPS, 2021.
>
> [5] Montanari et al., Overparametrized linear dimensionality reductions: From projection pursuit to two-layer neural networks, arXiv:2206.06526, 2022.

---

### Official Review · Reviewer_Zvww · 2025-10-31

**Soundness:** 3
**Presentation:** 2
**Contribution:** 2
**Rating:** 4
**Confidence:** 3

**Summary:**

This paper develops a statistical theory for imbalanced data classification in high-dimensional settings. The authors first introduce the logit distribution to characterize overfitting in imbalanced data, and then analyze its impact, particularly on the minority class. They further extend their analysis to examine the consequences of overfitting on model calibration.

**Strengths:**

1. Quantifying the effects of overfitting in imbalanced data is an important and meaningful problem.
2. The study is comprehensive, and the theoretical analysis is solid.

**Weaknesses:**

1. The paper is not well written or well organized. The motivation behind some of the definitions and problem settings is not clearly explained. See questions below for details.

**Questions:**

1. The motivation for proposing the logit distribution in Definitions 2.1 and 2.2 is unclear. It is also not entirely clear what empirical phenomenon Section 2 is intended to highlight or support.
2. The purpose of introducing the problem in Definition 2.3 is somewhat unclear. It would be helpful to discuss whether similar formulations have been studied in previous literature and how this setting differs.

---

> ### Author Response · Authors · 2025-11-21
> **Response to Reviewer Zvww — Part I**
>
> Thank you for your thoughtful review and constructive feedback. We sincerely appreciate the time you took to point out areas of the paper that could be clarified. We have made substantial efforts to enhance the readability, motivation, and overall organization of the revised manuscript. Below we address your concerns in detail, and we would truly appreciate your consideration of a score increase if our response satisfactorily resolves the issues you raised.
>
> ## Weaknesses:
>
> ---
>
> - **W1.** "The paper is not well written or well organized. The motivation behind some of the definitions and problem settings is not clearly explained..."
>
> - **A1.** Thank you for highlighting these issues. We have made every effort to improve the writing and enhance the clarity of the revised manuscript.
>
>     - We retained only the essential notation in Section 2 and substantially simplified the statements of Definitions 2.1 and 2.2 (page 3, `line 158`, page 4 `line 197`).
>
>     - We incorporated Definition 2.3 into Theorem 2.1 (now Eq. (5), `line 289`) to avoid confusion and improve readability.
>
>     Please see below for our point-by-point responses to your questions.
>
>
> ---
>
> ## Questions:
>
> ---
>
> - **Q1.** "The motivation for proposing the logit distribution in Definitions 2.1 and 2.2 is unclear..."
>
> - **A1.** Logits is important in machine learning. Definition 2.1 recalls the notions of *logit* and *margin*, and Definition 2.2 introduces the logit distribution on the training and test sets (that is, the histogram and curve in Fig. 1, resp.).
>
>     - The distribution of logits is useful for understanding model generalization and internal representations. For example, **neural collapse** describes how within-class feature means and classifier vectors converge to a structured limit [1, 2], a phenomenon studied through last-layer logits---where the geometry of logit distributions is useful.
>
>     - Another example is **linear probing**, commonly used to interpret hidden representations and light-weight model adaption (such as model editing) in deep learning. Our Llama-3 probing experiment (Fig. 3) and [3] illustrate how logits reveal the direction of answer "truthfulness" in a language model.
>
>     - There is also extensive theoretical work characterizing generalization bound of neural networks through the distribution of logits. For example, [4, 5] derive tail bounds for logit distributions. In our setting, we show that logits on test set converge to $\\mathsf{N}(\\mu, \\sigma^2)$, while logits on training set converge to $\\max \\{ \\kappa, \\mathsf{N}(\\mu, \\sigma^2) \\}$.
>
>
>     In Section 2, we aim to highlight the empirical phenomenon that overfitting in imbalanced classification is characterized by **truncation**, clearly observed in both simulation (Fig. 1) and real data (Fig. 2). We have also added annotations to Fig. 1 (page 4, top). This aligns with our theoretical result, where the term $\\max \\{ \\kappa, \\cdot \\}$ in the logit distribution captures this truncation effect.
>
> ---

---

> ### Author Response · Authors · 2025-11-21
> **Response to Reviewer Zvww — Part II**
>
> ---
>
> - **Q2.** "The purpose of introducing the problem in Definition 2.3 is somewhat unclear..."
>
> - **A2.** In the revision, we have incorporated Definition 2.3 into Theorem 2.1 (now Eq. (5), `line 289`) and addressed the potential confusion to improve readability. Please allow us to further clarify the motivation behind these concepts.
>
>     - The goal of Theorem 2.1 is to characterize the *asymptotic behavior*, namely the limit distribution of the logits as $n,d \\to \\infty$ in the proportional regime $n/d \\to \\delta$.
>
>     - In Definition 2.3 (Eq. (5) in revised version, `line 289`):
>
>         - $\\rho$ denotes the cosine similarity between $\\boldsymbol{\\beta}$ from the linear classifier and the optimal direction $\\boldsymbol{\\mu}$. Note that in high dimensions, $\\boldsymbol{\\beta}$ is often not aligned with $\\boldsymbol{\\mu}$ (i.e., $\\rho < 1$).
>
>         - The term $\\rho \\| \\boldsymbol{\\mu} \\| + G + Y \\beta_0$ represents the projection of the input distribution $\\pi \\cdot \\mathsf{N}(\\boldsymbol{\\mu}, \\mathbf{I}_d) + (1-\\pi) \\cdot \\mathsf{N}(-\\boldsymbol{\\mu}, \\mathbf{I}_d)$ onto $\\boldsymbol{\\beta}$. It relates the logit as follows:
>         $$
>             y(\\langle \\boldsymbol{x}, \\boldsymbol{\\beta} \\rangle + \\beta_0)
>             = y \\bigl\\langle y \\boldsymbol{\\mu} + \\mathsf{N}(\\mathbf{0}, \\mathbf{I}_d), \\boldsymbol{\\beta} \\bigr\\rangle + y \\beta_0 =  \\Bigl\\langle \\frac{\\boldsymbol{\\mu}}{\\| \\boldsymbol{\\mu} \\|}, \\boldsymbol{\\beta} \\Bigr\\rangle \\cdot \\| \\boldsymbol{\\mu} \\| + \\bigl\\langle \\mathsf{N}(\\mathbf{0}, \\mathbf{I}_d), \\boldsymbol{\\beta} \\bigr\\rangle + y \\beta_0 \\approx \\rho \\| \\boldsymbol{\\mu} \\| + G + Y \\beta_0.
>         $$
>
>         - $\\xi$ is a "free" term in the variational formulation due to **overfitting** in high dimensions. A larger dimension (i.e., a larger $1/\\delta$) gives the classifier more flexibility in fitting the training data.
>
>     - Theorem 2.1 basically states that the *test errors* and *logit distributions* approximate the corresponding limits characterized by the quantities in Definition 2.3 (now Eq. (5)). This approximation is rigorously formalized using weak convergence and the Wasserstein distance.
>
>     - Similar formulations to Definition 2.3 (now Eq. (5)) appear in [6] Eq. (23), also in [7] Eqs. (5.1) and (5.6) where the optimization problem is rewritten as a system of equations. In our setting, the data come from a Gaussian mixture, whereas [6, 7] consider a random feature model (i.e., $\\boldsymbol{x} \\sim \\mathsf{N}(\\mathbf{0}, \\boldsymbol{\\Sigma})$, $\\mathbb{P}(y = +1) = 1 - \\mathbb{P}(y = -1) = h( \\langle \\boldsymbol{\\beta}, \\boldsymbol{x} \\rangle )$).
>
> ---
>
>
>
> [1] Papyan et al., Prevalence of neural collapse during the terminal phase of deep learning training, PNAS, 2020.
>
> [2] Zhu et al., A geometric analysis of neural collapse with unconstrained features, NeurIPS, 2021.
>
> [3] Li et al., Inference-time intervention: Eliciting truthful answers from a language model, NeurIPS, 2023.
>
> [4] Koltchinskii et al., Empirical margin distributions and bounding the generalization error of combined classifiers, the Annals of Statistics, 2002.
>
> [5] Bartlett et al., Spectrally-normalized margin bounds for neural networks, NeurIPS, 2017.
>
> [6] Montanari et al., Overparametrized linear dimensionality reductions: From projection pursuit to two-layer neural networks, arXiv:2206.06526, 2022.
>
> [7] Montanari et al., The generalization error of max-margin linear classifiers: Benign overfitting and high dimensional asymptotics in the overparametrized regime, the Annals of Statistics, 2025.

---

### Official Review · Reviewer_b1pL · 2025-11-01

**Soundness:** 3
**Presentation:** 3
**Contribution:** 2
**Rating:** 6
**Confidence:** 4

**Summary:**

The paper provides formal results for linear classifiers (logistic regression and SVM) in the asymptotic regime where the size of the training set $n$ scales to infinity proportionally to their number of parameters $d$ ($n/d \to \delta$). The task is binary classification, the input data are assumed to be generated by a 2 component Gaussian Mixture Model, with components centered in $\pm {\boldsymbol \mu}$ and identity covariance, and the labels coincide with the component membership. In particular:
- Theorem 2.1 focuses on the SVM, and shows the existence of a critical threshold $\delta_c$ such that for $\delta<\delta_c$ the training set is linearly separable. In this case, the errors on the majority/minority class are given in terms of the sufficient statistics of the model (inner product between model’s weights and ${\boldsymbol \mu}$, bias and margin), obtained asymptotically as the solution of a variational problem. The training and testing logit distributions are shown to tend, respectively, to a rectified Gaussian and to a Gaussian. The truncation of the training distribution is more severe for the minority class, leading to overfitting.
- Proposition 3.1 considers the strategy of margin rebalancing in the proportional asymptotic regime, finding the optimal value of the parameter rebalancing the margin, in order to attain minimal balanced error.
- Theorem 3.2 considers the “high imbalance” asymptotics, defined by a vanishing (like $d^{-a}$) imbalance ratio, separation between the components of the mixture scaling like $d^b$, size of training set  scaling like $d^{c+1}$. There are 3 phases, according to the relative values of the scaling exponents: one where both minority and majority errors are small regardless the margin rebalancing parameter, one where this parameter has to scale with a certain power of $d$ in order to avoid O(1) error on the minority class, one where the linear classifier behaves as random guess.
- Section 4 pushes the analysis in the proportional regime to prove monotonicity of common miscalibration metrics with the imbalance ratio.

**Strengths:**

The sharp asymptotic results provide an interesting interpretation of overfitting in terms of skewness of the training logit distribution on the minority class. The sharp prediction on the optimal value of the margin rebalancing parameter is useful in practice. The high imbalance regime presents a rich phase diagram, with clear implications for the margin rebalancing strategy. Theoretical claims are supported by numerical experiments (Fig. 1, 4, 5), also on real data (Fig 2 and 3).

**Weaknesses:**

The theoretical analysis is limited to the very narrow setting of linear binary classification and data with isotropic covariances. Connections to some of the relevant literature in the field of high-dimensional statistics are not discussed. No clue or narrative on how the theorems are proven is given in the main text, making hard to assess their validity in a 2-weeks review (the appendix is 80 pages long). See Questions below for more detail.

**Questions:**

- I think the paper should mention in the title or in the abstract that it is about linear classifiers, to better represent its content.
- Can the authors discuss the connections between their work and recent analysis in similar setting? I am thinking at [1] (which considers Gaussian mixtures with generic covariances), [2] (which relaxes the assumption on Gaussian data proving universality results on data distribution), [3] (which uses an heuristic theory to make sharp asymptotic predictions on popular test metrics under imbalance), [4], [5] … Even if an extensive literature review is out of scope here, some of these works have to be mentioned to address the claims of novelty of the paper, as they all derive results in the proportional asymptotics the paper considers.
- If I understand correctly, the main novel point of Theorem 2.1 is the proof of the asymptotic distribution of training/test logits. In [4], Eq. (9) (test logit) and (53) (train logit), and references from early literature therein, similar results are reported. Can the authors comment whether the statements are comparable?
- Can the authors report at least the main ideas of the proofs of their theorems in the main text? In its current form, the main text is substantially not self-contained, which is an issue given reviewers are not required (and mostly unable due to time constraints) to read the appendices. However, the illustration of the results in comparison with numerical simulations (Fig. 1,4,5) is commendable.

[1] Loureiro et al. Learning Gaussian Mixtures with Generalised Linear Models: Precise Asymptotics in High-dimensions, NeurIPS 2021

[2] Dandi et al. Universality laws for Gaussian mixtures in generalized linear models, NeurIPS 2023

[3] Loffredo et al. Restoring balance: principled under/oversampling of data for optimal classification,

[4] Mannelli et al. Bias-inducing geometries: An exactly solvable data model with fairness implications, Phys. Rev. E 112, 025304

[5] Pezzicoli et al. Class Imbalance in Anomaly Detection: Learning from an Exactly Solvable Model, AIStats 2025

---

> ### Author Response · Authors · 2025-11-21
> **Response to Reviewer b1pL — Part I**
>
> Thank you for your detailed and positive assessment of our theoretical results and their interpretation. We have improved the readability and organization of the revised manuscript. Below we address the questions raised in your review, and if our clarifications satisfactorily resolve your concerns, we would be grateful for your consideration of an improved score.
>
> ## Weaknesses:
>
> ---
>
> - **W1.** "The theoretical analysis is limited to the very narrow setting of linear binary classification and data with isotropic covariances..."
>
> - **A1.** Thank you for pointing out these shortcomings. We have added experiments on *non-Gaussian* distributions and *multiclass* classification to address the limitations of the original setting (see **Appendix L.1, L.3**, page 101, 103).
>
>     - We focus on simple settings because the phenomenon can be observed, analyzed, and theorized more clearly. Similar proof strategies can extend to *non-isotropic* cases as well. We have added these generalizations in Section 5 (page 10, `line 528`); please refer to Conjectures D.11-13 for details.
>
>     - In practice, *whitening* is often applied before fitting a linear classifier, after which the input features become isotropic (the covariance is the identity matrix).
>
>     - There is also literature on *universality* suggesting that our results may extend to certain non-Gaussian settings and random feature models (e.g., [6]).
>
>     - We included a literature review on high-dimensional statistics around `lines 46` and `84` and have expanded it; see our response to **Q2** below.
>
>     - We have also added the proof ideas to the main text; see our response to **Q4** below.
>
> ---
>
> ## Questions:
>
> ---
>
> - **Q1.** "I think the paper should mention in the title or in the abstract that it is about linear classifiers, to better represent its content."
>
> - **A1.** We agree. We have noted this in the abstract of the revised manuscript (page 1, `line 16`).
>
> ---
>
> - **Q2.** "Can the authors discuss the connections between their work and recent analysis in similar setting?..."
>
> - **A2.** Thanks for the suggestion. We have cited all these papers in Section 1 for comparison (page 2, `lines 72-73`, `86-87`).
>
>     - We appreciate all these contributions on deriving precise asymptotics for imbalanced classification (e.g., limits of parameters, ERM objectives, and errors). Our approaches are broadly aligned with these works (e.g., our Eq. (4) plays a similar role to Eq. (10) in [3], and our Eq. (36) resembles Eq. (8) in [1]).
>
>     - Our novelty lies in further deriving the logit distribution (Theorem 2.1(b)) and analyzing the monotonicity of key factors on classification and calibration errors (Table 2).
>
>     - Our proof technique based on Gordon's theorem is also more concise or rigorous compared to prior approaches (e.g., AMP in [1], the replica method in [2, 4, 5]). Notably, [2] proves universality for sub-Gaussian data, and our high imbalance regime admits a similar extension (Theorem D.8). In addition, [3] studies rebalancing through data resampling, whereas our approach focuses on margin-based rebalancing.
>
>
> ---
>
> - **Q3.** "If I understand correctly, the main novel point of Theorem 2.1 is the proof of the asymptotic distribution of training/test logits..."
>
> - **A3.** We could not locate these two equation numbers in [4], and we believe the reviewer may be referring to [3]. Please let us know if this is not the case.
>
>     - At a high level, their results are comparable to ours, although the settings and proof techniques differ ([3] uses the replica method, while ours is based on random matrix theory).
>
>     - Though not explicitly discussed in [3], Eq. (53) therein converges to a distribution with a *truncation* component under linear separability. In the non-separable case, the training logit becomes an exponentially tilted Gaussian, whereas we show that its exact asymptotics form a skewed/shrunk Gaussian (see line 319 and Theorem D.3(d)).
>
> ---

---

> ### Author Response · Authors · 2025-11-21
> **Response to Reviewer b1pL — Part II**
>
> ---
>
> - **Q4.** "Can the authors report at least the main ideas of the proofs of their theorems in the main text?..."
>
> - **A4.** Thanks for the suggestion. We have added a brief paragraph on "*Intuition and proof techniques*" in the revised main text (see page 6, `line 305`).
>
>     - In short, the main idea is to rewrite the linear classifier in Eq. (2b) as a min–max formulation involving
>     $$
>         \\min_{\\boldsymbol{\\beta}} \\max_{\\boldsymbol{\\lambda}}
>         \\big\\{
>         \\boldsymbol{\\lambda}^\\top \\boldsymbol{G} \\boldsymbol{\\beta} +
>         \\langle \\boldsymbol{\\lambda} , \\mathbf{c} \\rangle
>         \\big\\},
>         \\quad \\boldsymbol{G} \\in \\mathbb{R}^{n \\times d} \\text{ with i.i.d. } \\mathsf{N}(0, 1) \\text{ entries}.
>     $$
>     **Gordon's theorem** allows us to replace $\\boldsymbol{\\lambda}^\\top \\boldsymbol{G} \\boldsymbol{\\beta}$ with $\\| \\boldsymbol{\\beta} \\| \\langle \\boldsymbol{\\lambda}, \\mathsf{N}(\\mathbf{0}, \\mathbf{I}_n) \\rangle + \\| \\boldsymbol{\\lambda} \\| \\langle \\boldsymbol{\\beta}, \\mathsf{N}(\\mathbf{0}, \\mathbf{I}_d) \\rangle$, so the min–max problem only involves random vectors rather than the random matrix $\\boldsymbol{G}$. Our proofs then address the technical aspects of this approximation and provide a rigorous analysis of the resulting distributional convergence.
>
>     - Gordon's theorem typically reduces a high-dimensional variational problem to a low-dimensional one. In Definition 2.3, the term $\\rho \\| \\boldsymbol{\\mu} \\| + G + Y \\beta_0$ represents the projection of the input distribution $\\pi \\cdot \\mathsf{N}(\\boldsymbol{\\mu}, \\mathbf{I}_d) + (1-\\pi) \\cdot \\mathsf{N}(-\\boldsymbol{\\mu}, \\mathbf{I}_d)$ onto $\\boldsymbol{\\beta}$:     $$
>         y(\\langle \\boldsymbol{x}, \\boldsymbol{\\beta} \\rangle + \\beta_0)
>         = y \\bigl\\langle y \\boldsymbol{\\mu} + \\mathsf{N}(\\mathbf{0}, \\mathbf{I}_d), \\boldsymbol{\\beta} \\bigr\\rangle + y \\beta_0 =  \\Bigl\\langle \\frac{\\boldsymbol{\\mu}}{\\| \\boldsymbol{\\mu} \\|}, \\boldsymbol{\\beta} \\Bigr\\rangle \\cdot \\| \\boldsymbol{\\mu} \\| + \\bigl\\langle \\mathsf{N}(\\mathbf{0}, \\mathbf{I}_d), \\boldsymbol{\\beta} \\bigr\\rangle + y \\beta_0 \\approx \\rho \\| \\boldsymbol{\\mu} \\| + G + Y \\beta_0.
>     $$
>     Intuitively, the remaining dimensions are not relevant and instead provide room for overfitting, which is captured by the "free" random variable $\xi$.
>
>     - The challenging part is the logit distribution convergence, where we use an idea from projection pursuit [7].
>
>
>
> ---
>
> [6] Montanari et al., Universality of max-margin classifiers, arXiv:2310.00176, 2023.
>
> [7] Montanari et al., Overparametrized linear dimensionality reductions: From projection pursuit to two-layer neural networks, arXiv:2206.06526, 2022.

---

### Official Review · Reviewer_zrUp · 2025-11-01

**Soundness:** 3
**Presentation:** 2
**Contribution:** 3
**Rating:** 6
**Confidence:** 2

**Summary:**

The paper presents a statistical theory for why overfitting effects the minority class significantly more in high-dimensional class imbalance problems by observing the behaviour of the logits. The authors define empirical logit distribution (the distribution of logits for the training set) and test logit distribution (distribution of logits for the test set) and characterise overfitting using these measures. The paper also applies this to the field of calibration, and an extension to multiclass.

**Strengths:**

- The paper provides a theoretical understanding for the behaviour of the distribution of logits at training and test time, which is an interesting application of an approach often used in calibration.
- The theorems presented seem consistent with the general consensus among the imbalanced classification field.
- Results are presented on a variety of modalities (tabular, image and text)
- Experiments are performed in a good number of scenarios, various imbalance proportions $\pi$, optimising classifiers on the data itself and using features from a pretrained network.

**Weaknesses:**

- The paper is difficult to follow in places, there are a lot of concepts to grasp in section 2.
- Gordon's theorem should be cited at it first mention on page 2
- [1] presents a method for adjusting decision boundaries based on the uncertainty from drawing a random sample from each class, and could be discussed in section 3. I believe this method would help with the truncation of the TLD, and it could be worth discussing how concentration inequalities relate to the TLD and ELD.
- Most of the theoreitcal results around calibration and prediction probabilities are in the appendix, which makes the section seem incomplete in the main paper. Section 5 is similar, in that most of the details are in the appendix. The paper could benefit from moving one of these sections to the appendix completely, and expanding on the other.

[1] Clifford, Matt, et al. "Learning Confidence Bounds for Classification with Imbalanced Data." ECAI 2024 (European Conference on Artificial Intelligence) (2024).

**Questions:**

- Does the truncation of logits still occur in datasets where the means for the positive class and negative class are far apart? i.e. the decision boundary will be placed arbitrarily between both classes. In practice, overfitting still occurs as most classifiers will favour placing the decision boundary close to the minority class, but there will be a set of decision boundaries which give the same training error. How can the TLD and ELD differ, and how does the theoretical approach in this paper deal with this?
- How do post-hoc methods such as rebalancing the margin address the calibration of the classifier in this scenario?

---

> ### Author Response · Authors · 2025-11-21
> **Response to Reviewer zrUp — Part I**
>
> Thank you for your constructive comments and positive evaluation of our theoretical framework and empirical results. We have made every effort to address your comments and to improve the readability of the paper in the revised version. Below we summarize our responses to the raised weaknesses and questions, and if our clarifications adequately address your concerns, we would sincerely appreciate your consideration of a score increase.
>
> ## Weaknesses:
>
> ---
>
> - **W1.** "The paper is difficult to follow in places, there are a lot of concepts to grasp in section 2."
>
> - **A1.** Thanks for pointing out this issue. In the revised manuscript, we have made several changes to improve clarity:
>
>     - We retained only the essential notation in Section 2 and substantially simplified the statements of Definitions 2.1 and 2.2 (page 3, `line 158`, page 4 `line 197`).
>
>     - We incorporated Definition 2.3 into Theorem 2.1 (now Eq. (5), `line 289`) to avoid confusion and improve readability.
>
>     Please allow us to further clarify the motivation and main ideas behind these concepts.
>
>     - For a linear classifier, the *logit* of $\\boldsymbol{x}$ is $\\widehat f(\\boldsymbol{x}) = \\langle \\boldsymbol{x}, \\widehat{\\boldsymbol{\\beta}} \\rangle + \\widehat\\beta_0$. In neural networks, logits are the inputs to the softmax layer before converting to probabilities.
>
>     - The goal of Theorem 2.1 is to characterize the *asymptotic behavior*, namely the limiting distribution of the logits as $n,d \\to \\infty$ in the proportional regime $n/d \\to \\delta$.
>
>     - In Definition 2.3 (Eq. (5) in revised version, `line 289`):
>
>         - $\\rho$ denotes the cosine similarity between $\\boldsymbol{\\beta}$ from the linear classifier and the optimal direction $\\boldsymbol{\\mu}$. Note that in high dimensions, $\\boldsymbol{\\beta}$ is often not aligned with $\\boldsymbol{\\mu}$ (i.e., $\\rho < 1$).
>
>         - The **key insight** is the following: $\\rho \\| \\boldsymbol{\\mu} \\| + G + Y \\beta_0$ represents the projection of the input distribution $\\pi \\cdot \\mathsf{N}(\\boldsymbol{\\mu}, \\mathbf{I}_d) + (1-\\pi) \\cdot \\mathsf{N}(-\\boldsymbol{\\mu}, \\mathbf{I}_d)$ onto $\\boldsymbol{\\beta}$, which leads to the major terms in the limit:
>         $$
>             y(\\langle \\boldsymbol{x}, \\boldsymbol{\\beta} \\rangle + \\beta_0)
>             = y \\bigl\\langle y \\boldsymbol{\\mu} + \\mathsf{N}(\\mathbf{0}, \\mathbf{I}_d), \\boldsymbol{\\beta} \\bigr\\rangle + y \\beta_0 =  \\Bigl\\langle \\frac{\\boldsymbol{\\mu}}{\\| \\boldsymbol{\\mu} \\|}, \\boldsymbol{\\beta} \\Bigr\\rangle \\cdot \\| \\boldsymbol{\\mu} \\| + \\bigl\\langle \\mathsf{N}(\\mathbf{0}, \\mathbf{I}_d), \\boldsymbol{\\beta} \\bigr\\rangle + y \\beta_0 \\approx \\rho \\| \\boldsymbol{\\mu} \\| + G + Y \\beta_0.
>         $$
>
>         - $\\xi$ is a "free" term in the variational formulation due to **overfitting** in high dimensions. A larger dimension (i.e., a larger $1/\\delta$) gives the classifier more flexibility in fitting the training data.
>
>     - Theorem 2.1 basically states that the *test errors* and *logit distributions* approximate the corresponding limits characterized by the quantities in Definition 2.3 (now Eq. (5)). This approximation is rigorously formalized using weak convergence and the Wasserstein distance.
>
> ---
>
> - **W2.** "Gordon's theorem should be cited at it first mention on page 2."
>
> - **A2.** Thank you for pointing this out. We have added this citation in the revision (page 2, `line 88`).
>
> ---
>
> - **W3.** "[1] presents a method for adjusting decision boundaries based on the uncertainty from drawing a random sample from each class..."
>
> - **A3.** Thanks for the suggestion. We have mentioned [1] in Section 3 of the revised manuscript (page 7, `line 377`). While [1] relies on generic concentration bounds (e.g., McDiarmid's), our analysis is based on Gaussian concentration tools such as Gordon's theorem, which allow us to capture *sharp asymptotics* and also extend to certain sub-Gaussian settings (e.g., [2]). The technique in [1] may be helpful for studying truncation in more general non-Gaussian cases.
>
> ---

---

> ### Author Response · Authors · 2025-11-21
> **Response to Reviewer zrUp — Part II**
>
> ---
>
> - **W4.** "Most of the theoreitcal results around calibration and prediction probabilities are in the appendix..."
>
> - **A4.** We appreciate your feedback on this point. The takeaway of section 4 is summarized below:
>
>     - **Theorem** (confidence estimation and calibration) All miscalibration metrics Eqs. (11)-(13) have certain limits in the proportional regime. For example, $\\mathrm{MSE} \\to \\mathrm{MSE}^* := \\mathbb{E}[\\sigma(-\\rho^* \\| \\boldsymbol{\\mu} \\| - G - Y \\beta_0^* )^2]$ as $n, d \\to \\infty$. Moreover, $\\mathrm{MSE}^*$ is decreasing in class separation $\\| \\boldsymbol{\\mu} \\|$, aspect ratio $\\delta$, and imbalance ratio $\\pi \\in (0, \\frac12 )$, when $\\tau = \\tau^\\mathrm{opt}$.
>
>     - We have replaced Table 2 with this informal theorem (page 9, Theorem 4.1, `line 483`). We have also expanded Section 5 by adding the multiclass experiment details for Fig. 7 and the extension to non-isotropic covariance (page 10).
>
> ---
>
> ## Questions:
>
> ---
>
> - **Q1.** "Does the truncation of logits still occur in datasets where the means for the positive class and negative class are far apart?..."
>
> - **A1.** Yes, but the truncated portion of the minority class becomes *smaller* as the two classes move farther apart, i.e., as $\\| \\boldsymbol{\\mu} \\|$ increases (with imbalance ratio $\\pi \\in (0, \\frac12)$ and aspect ratio $\\delta$ *fixed*).
>
>     - As $\\| \\boldsymbol{\\mu} \\|$ increases, the gap between ELD and TLD gets small, truncation becomes negligible, and overfitting is no longer severe. According to Theorem 2.1, truncation *always* occurs in the proportional regime, and the ELD for the minority class is $\\max \\{ \\kappa^* , \\mathsf{N}(\\rho^*  \\| \\boldsymbol{\\mu} \\| + \\beta_0^* , 1) \\}$. Using an argument similar to Section G, we can show that the truncated portion $\\kappa^* - \\rho^* \\| \\boldsymbol{\\mu} \\| - \\beta_0^*$ is decreasing in $\\| \\boldsymbol{\\mu} \\|$, $\\delta$, and $\\pi \\in (0, \\frac12)$.
>
>     - Note that separation parameter $\\| \\boldsymbol{\\mu} \\|$ is *not the only factor* influencing overfitting; $\\delta$ and $\\pi$ also play important roles. The scenario you described is likely driven more by small $\\delta$, small $\\pi$, or finite-sample effects. In particular, in very high dimensions (small $\\delta$), overfitting and truncation in the minority class become severe.
>
>     - This scenario also motivates our analysis of the *high imbalance regime* in the setting of "large $\\| \\boldsymbol{\\mu} \\|$ and small $\\pi$", where a range of optimal decision boundaries exists (Theorem 3.2).
>
> ---
>
> - **Q2.** "How do post-hoc methods such as rebalancing the margin address the calibration of the classifier in this scenario?"
>
> - **A2.** As we clarified in our response to **W4**, we derived the limit of calibration metrics and showed that, when $\\tau = \\tau^\\mathrm{opt}$, MSE decreases in $\\pi \\in (0,\\frac12)$, $\\| \\boldsymbol{\\mu} \\|$, and $\\delta$. Other monotonicity results are presented in Table 2 and Appendix D.3. However, the optimal $\\tau = \\tau^\\mathrm{opt}$ that minimizes balanced error does not necessarily minimize these calibration metrics. One may still apply margin rebalancing but select a different $\\tau$ (or intercept $\\beta_0$) by minimizing the expression in Proposition D.9(a) (page 34, `line 1835`).
>
>
> ---
>
>
> [2] Montanari et al., Universality of max-margin classifiers, arXiv:2310.00176, 2023.

---

### Author Response · Authors · 2025-12-02
**Summary for AC's**

Dear Area Chairs,

Thank you very much for taking over the evaluation of our submission under the unexpected circumstances caused by the recent OpenReview incident. We sincerely appreciate your time and effort. To support your assessment, we provide the following brief summary to help you quickly grasp the relevant context of our paper and the preceding discussion.

---

## Overview of Our Paper

Our paper investigates classification under class imbalance, where minority classes contain substantially fewer samples. As classical theory fails in high dimensions and practical remedies remain largely ad hoc, the mechanism of overfitting in imbalanced settings is still poorly understood, leaving many empirical phenomena unexplained. To address these gaps, we develop a statistical theory for high-dimensional imbalanced classification, focusing on linear classifiers. Our contribution is summarized below:

- We characterize overfitting via the discrepancy between the *logit distributions* on the training and test sets, and show that dimensionality induces a *truncation effect* in the training logits. This phenomenon is validated on tabular, image, and text datasets. (**Section 2**)

- We derive the optimal hyperparameter for *margin rebalancing* to mitigate class imbalance, and show that the test error decreases with key model parameters—the *imbalance ratio*, *signal strength* (class center separation), and *aspect ratio*. (**Section 3**)

- We further show that changes in these parameters that raise test error may simultaneously *worsen model calibration*, revealing an adverse consequence of overfitting. (**Section 4**)

Our approach can be extended to multiclass problems and more general covariance structures.

---

## Summary of Reviewers' Concerns

Below we summarize the main points raised by the four reviewers during the rebuttal period:

- **Q1.** Some notations and concepts in Section 2 could be clarified further for readability. (zrUp, Zvww)

- **Q2.** The main text may benefit from additional proof ideas and further details to make it more self-contained. (zrUp, b1pL, rQB8)

- **Q3.** The discussion of related work on imbalanced classification and high-dimensional statistics could be expanded. (zrUp, b1pL, rQB8)

- **Q4.** A few assumptions and aspects of the model setting might appear somewhat restrictive. (b1pL, rQB8)

---

## How We Addressed These Concerns

We have made substantial revisions to the manuscript and carefully addressed these concerns through conceptual clarifications, additional experiments, theoretical extensions, and writing refinements:

- **A1.**
    - In the revised manuscript, we reorganized parts of Section 2 and retained only the essential notations. We also substantially simplified several definitions to enhance clarity.

    - In our response, we further clarified the motivation and main ideas behind the concepts in Section 2 through both verbal explanations and mathematical formulas.

    - We also included a brief summary of related literature in our response to help reviewers better understand the background.


- **A2.**
    - In the revised manuscript, we added a brief paragraph on "Our contribution" to highlight our novelty, and "Intuition and proof techniques" after our main theorem.

    - We also expanded Sections 4 and 5 by adding an informal theorem and including additional experimental details.

    - A summary of the above changes is also included in our response.

- **A3.**
    - We thank the reviewers for pointing out additional related literature, which we have now incorporated into the revised manuscript.

    - In our response, we discussed each piece of literature point by point in relation to our work. While related, they differ from our study in several aspects, including the problem setting (e.g., fixed dimension vs. high dimension), proposed approaches (e.g., resampling vs. rebalancing), proof techniques (e.g., replica heuristics vs. random matrix theory), and overall scope (e.g., methodological focus vs. theory development).

- **A4.**
    - In the revised manuscript, we added additional experiments to address the limitations (e.g., non-Gaussian and multiclass settings), and the results align with our expectations.

    - We also included theoretical extensions under more general settings (e.g., non-isotropic heterogeneous covariance) to illustrate that our proof techniques are adaptable. We focus on a simpler setting in the main theorem where the phenomenon can be examined more clearly.


---

We sincerely appreciate your efforts in overseeing our submission during this special situation. We hope that this summary is helpful for your evaluation.

Best Regards,

Authors

---

### Meta-Review · Area_Chair_9UVx · 2025-12-23

**Summary:**

This paper explains why classifiers overfit minority classes and become poorly calibrated when data are highly imbalanced and high dimensional, a regime where classical theory breaks down. It shows that training predictions for minority classes are systematically distorted while test predictions behave normally, creating a train test mismatch that drives bias and overfitting. Based on this insight, the authors propose a simple margin rebalancing method with a theoretically optimal tuning parameter that reduces majority bias and improves both accuracy and confidence calibration. These theoretical findings are supported by experiments on tabular, image, and text datasets, offering a clear and principled explanation and remedy for imbalance related overfitting, with promising extensions to more general settings.

**Reviewer Concerns:**

The concerns have been addressed:

(1) Clarification of the proof idea.

(2) More detailed discussion of related works on imbalanced classification.

(3) Discussion of the results under more relaxed assumptions.

**Reviewer Scores:**

Reviewer zrUp (score 6) is likely to maintain their original score with low confidence. Although the reviewer attempted to provide detailed comments, their feedback suggests limited expertise with the paper’s technical content. As a result, I do not place significant weight on this score in the final consideration.

Reviewer b1pL (score 6) is likely to maintain or possibly increase their score. Most of the concerns raised were requests for clarification rather than substantive technical weaknesses, and the authors have addressed these points thoroughly and satisfactorily.

Reviewer rQB8 (score 4) is likely to maintain their original score with low confidence. The review did not include meaningful or actionable comments, and therefore I do not consider this score informative for the final decision.

Reviewer Zvww (score 4) is likely to increase their score to 6, as the majority of their concerns have been resolved through additional clarifications provided by the authors.

Overall, I recommend acceptance, primarily guided by the substantive comments from Reviewers b1pL and Zvww, whose concerns were constructive and have been adequately addressed.

---

### Decision · Program_Chairs · 2026-01-26

Accept (Poster)